Analysis

# A systematic benchmark of Nanopore long-read RNA sequencing for transcript-level analysis in human cell lines

The human genome contains instructions to transcribe more than 200,000 RNAs. However, many RNA transcripts are generated from the same gene, resulting in alternative isoforms that are highly similar and that remain difficult to quantify. To evaluate the ability to study RNA transcript expression, we profiled seven human cell lines with five different RNA-sequencing protocols, including short-read cDNA, Nanopore long-read direct RNA, amplification-free direct cDNA and PCR-amplified cDNA sequencing, and PacBio IsoSeq, with multiple spike-in controls, and additional transcriptome-wide $N^6$-methyladenosine profiling data. We describe differences in read length, coverage, throughput and transcript expression, reporting that long-read RNA sequencing more robustly identifies major isoforms. We illustrate the value of the SG-NEx data to identify alternative isoforms, novel transcripts, fusion transcripts and $N^6$-methyladenosine RNA modifications. Together, the SG-NEx data provide a comprehensive resource enabling the development and benchmarking of computational methods for profiling complex transcriptional events at isoform-level resolution.

Gene expression is commonly estimated as the total amount of RNA transcripts that are generated from each gene. However, alternative promoters, exon skipping, intron retention, 3′-end sites and polyadenylation enable a single gene to generate a large number of alternative isoforms, each corresponding to a different RNA transcript. Such alternative transcripts can be differentially regulated, leading to differences in transcript expression across samples and conditions even when the overall gene expression levels are stable[1–5]. Due to the differences in their RNA sequence, alternative transcripts can have different functionality, and their regulation and expression play a key role during early development[6,7], in defining cellular identity[8,9] and in human diseases[10,11].

The comprehensive profiling and low cost of short-read RNA sequencing (RNA-seq) have made it one of the most widely used technologies to study molecular properties of cells and tissues[12]. The majority of short-read RNA-seq data are based on PCR-amplified sequencing of cDNA, which introduces biases that lead to different exons being sequenced at different coverage levels[13]. While short-read data generate robust estimates for gene expression, the presence of overlapping annotations and systematic biases limits the ability to uniquely assign reads to individual RNA transcripts[14,15]. To deal with the increased uncertainty, approaches have been developed that focus on specific splice-junction or exon usage[1,5,16,17]. However, more complex transcriptional events that involve multiple exons are often not fully captured[18–21].

Long-read sequencing promises to overcome some of the main limitations of current short-read RNA-seq protocols for transcriptome profiling[22–24] at a cost per gigabase comparable with current short-read technologies (Supplementary Fig. 1a), potentially enabling the widespread adoption of long-read RNA-seq. Long-read RNA-seq can be generated using the PacBio sequencing platform (IsoSeq), or the Oxford Nanopore sequencing platform, which offers three different protocols. PCR-amplified cDNA sequencing requires the least amount of input RNA and generates the highest throughput. When sufficient RNA is available, the PCR step can be omitted using the direct cDNA protocol.

✉e-mail: chen_ying@gis.a-star.edu.sg; gokej@gis.a-star.edu.sg

The direct RNA-seq protocol enables sequencing of native RNA, thereby avoiding the reverse transcription and amplification steps, as well as providing information about possible RNA modifications[25,26]. While several long-read RNA-seq datasets have been described, they are low throughput[23–25], lack replicates[27–30] or cover single conditions[24,31] or individual protocols[25]; thus, this limits the ability to comprehensively compare and evaluate the different RNA-seq protocols.

Here we present the results from the Singapore Nanopore Expression (SG-NEx) project, a comprehensive benchmark dataset and systematic comparison of five different RNA-seq protocols. The SG-NEx core data consist of seven human cell lines that were sequenced in multiple replicates using short-read RNA-seq, Nanopore long-read direct RNA, amplification-free direct cDNA and PCR-amplified cDNA sequencing, as well as PacBio IsoSeq. The core dataset includes six different spike-in RNAs with known concentrations[32–34] and includes a transcriptome-wide reference map of $N^6$-methyladenosine (m$^6$A). To provide a broader coverage of the human transcriptome and facilitate new biological discoveries, the core dataset is extended with long-read RNA-seq from seven additional cell lines and tissues. We further provide a community-curated nf-core pipeline that simplifies data processing, method evaluation and biological discoveries. We compare the different RNA-seq protocols, investigate the impact of fragmentation and short-read sequencing on transcript quantification, and illustrate how long-read RNA-seq facilitates analysis of full-length fusion transcripts, alternative isoforms and RNA modifications. The inclusion of multiple cell lines, all major RNA-seq platforms and spike-in controls, and each with multiple replicates, makes the SG-NEx data a unique resource that will be invaluable for the benchmarking and development of computational methods for differential expression analysis, transcript discovery and quantification, fusion gene detection and identification of RNA modifications from long-read RNA-seq data.

## Results

### A comprehensive resource for long-read RNA-seq
The core SG-NEx data resource consists of seven commonly used cell lines for colon cancer (HCT116), liver cancer (HepG2), lung cancer (A549), breast cancer (MCF7), leukemia (K562), ovarian cancer (HEYA8) and the H9 human embryonic stem cell line. Each cell line was sequenced with at least three high-quality replicates using the direct RNA protocol (direct RNA), the amplification-free cDNA protocol (direct cDNA), the PCR cDNA protocol (cDNA) and paired-end, 150-bp short-read Illumina cDNA sequencing (Fig. 1a). For a subset of sequencing runs, we included Sequin (V1, V2), ERCC, spike-in RNA variants (SIRVs; E0, E2) and long SIRV spike-in RNAs with known concentrations[32–34], PacBio IsoSeq data and transcriptome-wide m$^6$A profiling (m6ACE-seq) to evaluate the ability to detect RNA modifications from direct RNA-seq data. This core SG-NEx dataset was extended with additional samples from two stomach cancer cell lines (NCC24, IM95), a head and neck cancer cell line (HN1-NPC7), the HEK293T cell line and three samples from individuals with multiple myeloma. These additional samples were sequenced with selected protocols to cover a broad spectrum of the human transcriptome and its variation across cell lines and individuals. In total, we have sequenced 139 libraries for 14 cell lines and tissues, with an average sequencing depth of 100.7 million long reads for the core cell lines (Fig. 1b,c, Supplementary Tables 1 and 2 and Supplementary Fig. 2). The SG-NEx provides a community resource through GitHub via https://github.com/GoekeLab/sg-nex-data/.

### nf-core/nanoseq: a community-curated pipeline
To facilitate the streamlined processing and analysis of long-read RNA-seq data, we developed the nanoseq pipeline (Fig. 1d). Nanoseq performs quality control, alignment[35,36], transcript discovery and quantification[37–39], differential expression analysis[17,40–42], RNA fusion detection[43] and RNA modification detection[44,45]. Each module provides the option to use different existing methods that can be seamlessly integrated with the other

modules. The pipeline is dynamically tested on a full-sized dataset, it allows data processing through Docker and Singularity, and it can be executed on the cloud. Nanoseq is implemented in the Nextflow language[46] and maintained as a community-curated pipeline on nf-core[47].

### A comparison of five RNA-seq protocols
Using the SG-NEx core data, we compared the five different RNA-seq protocols in terms of throughput, read length, transcript coverage, potential library preparation biases and gene and transcript expression estimates. Among the long-read RNA-seq protocols, PCR-amplified cDNA sequencing consistently generated the highest throughput per sample, with the most recent sequencing data matching short-read RNA-seq (Fig. 2a and Supplementary Figs. 1b,c and 3a,c).

PacBio IsoSeq generated the longest reads on average, followed by the direct RNA-seq protocol (Fig. 2b and Supplementary Fig. 3b). The long-read protocols showed higher coverage at the 5′ and 3′ ends of transcripts when compared to short-read RNA-seq, possibly reflecting short-read-specific limitations due to RNA fragmentation (Fig. 2c and Supplementary Figs. 1e and 3e). The direct RNA-seq protocol starts the sequencing process at the poly(A) tail, which is reflected in a higher coverage at the 3′ end compared to the 5′ end (Fig. 2c and Supplementary Figs. 1e and 3e). The PCR-amplified cDNA sequencing and the PacBio IsoSeq data showed the most uniform coverage across the transcript length and the highest proportion of reads spanning all exon junctions ('full-splice-match reads', Fig. 2c,d and Supplementary Figs. 1e–g and 3e,f), whereas short-read RNA-seq had the highest fraction of reads that were assigned to multiple transcripts (Fig. 2d and Supplementary Fig. 1f). However, not all transcripts appear to be equally amplified by all protocols. In particular, transcripts from the 1,000 genes with the highest gene expression estimate accounted for a significantly larger proportion of the overall transcript expression in the PCR-amplified cDNA sequencing protocol compared to the PCR-free Nanopore RNA-seq (Fig. 2e; two-sided $t$-test, $P = 0.00041$; Supplementary Fig. 1g), and the PacBio IsoSeq data showed a significant depletion of shorter transcripts (Supplementary Fig. 1h,i; two-sided $t$-test, $P < 0.00001$). Finally, we observed that transcripts from some genes were incompletely amplified and sequenced in the PCR cDNA protocol across all cell lines when compared to direct sequencing of the same RNA sample (Fig. 2f, Supplementary Fig. 1j,k and Supplementary Table 3). While the biases that are introduced in the fragmentation step in short-read RNA-seq are avoided with long reads, these results show that the library preparation method still introduces differences in read length, coverage and transcript diversity when using long-read RNA-seq.

### Gene expression is robustly estimated across protocols
To avoid uncertainty of individual transcript expression estimates, the overall gene expression estimate, corresponding to the sum of all transcripts for each gene, is commonly used. To compare the ability of the different long-read and short-read RNA-seq protocols to quantify gene expression, we first analyzed estimates for spike-in data using ERCC, long SIRV, SIRV E0, SIRV E2 and two different Sequin mixtures. The direct RNA-seq protocol was removed from this comparison due to its incompatibility with existing spike-in RNA libraries (Supplementary Fig. 4a). Gene expression estimates from Nanopore long-read RNA-seq data showed the lowest estimation error overall and a higher correlation with the expected concentrations (Fig. 3a, Supplementary Fig. 4a and Supplementary Table 4), which was consistent when alternative computational quantification methods were used (Supplementary Fig. 4a,d,e). Among all RNA-seq protocols, the largest difference was observed in the PacBio IsoSeq data, which was possibly due to an uneven representation of transcript lengths, with low coverage for short transcripts (<1 kb; Supplementary Figs. 1h,i and 4a,d,e and Supplementary Text Fig. 13).

Gene expression estimates were highly correlated between short-read and Nanopore long-read RNA-seq both on spike-in RNAs and non-spike-in RNAs, with the highest correlation observed for

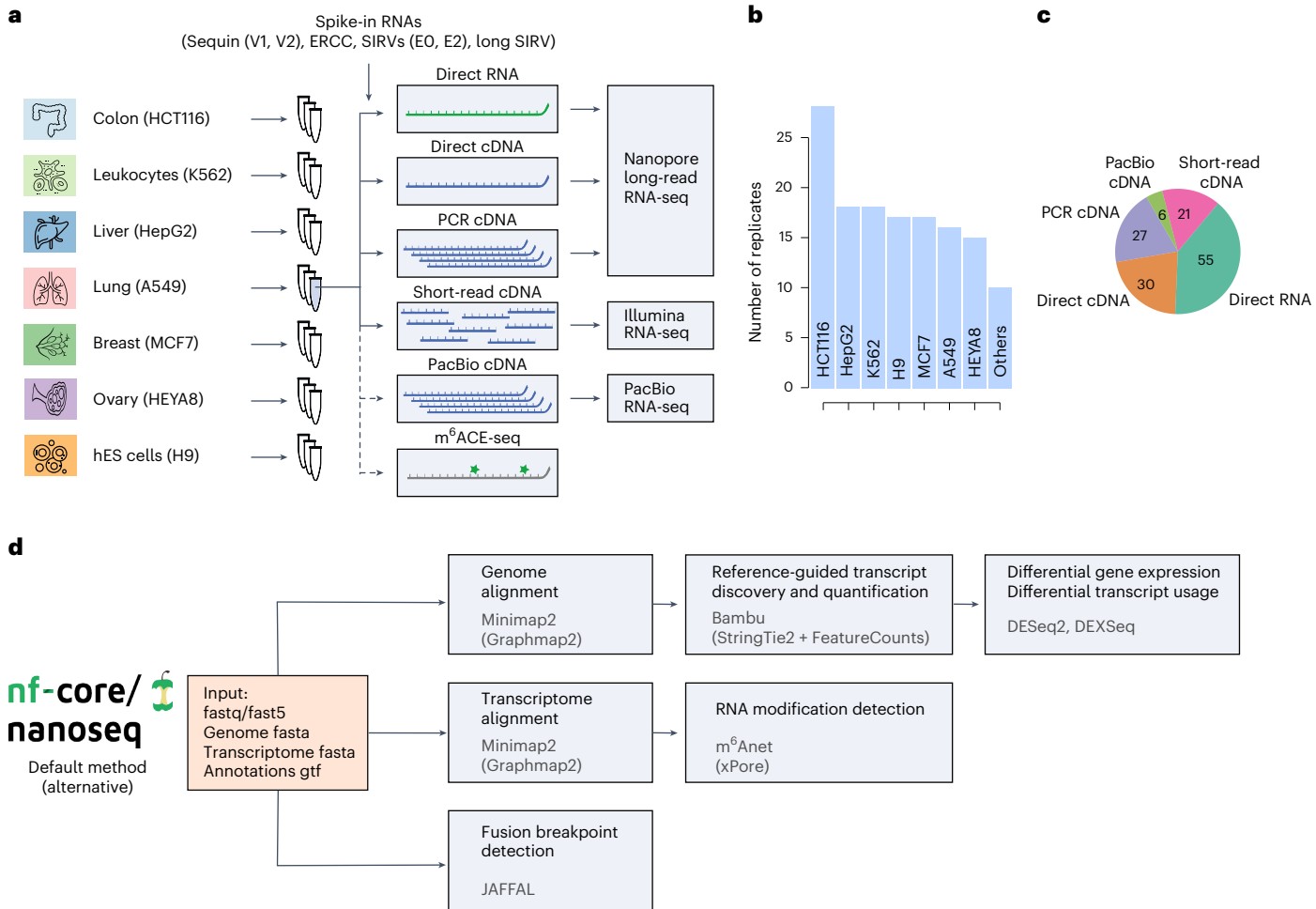

**Fig. 1 | Overview of the SG-NEx datasets and processing pipeline. a**, Seven human cell lines were sequenced with multiple replicates using different RNA-seq protocols. Short-read cDNA was sequenced with 150-bp paired-end reads. hES cells, human embryonic stem cells. Icons from Noun Project under a Creative Commons license CC BY 3.0: colon, Mungang Kim; leukocytes, ProSymbols; liver, Prettycons; lung, Mahmure Alp; breast, Karina; ovary, Amethyst Studio; hES cells, DailyPM. **b**, Number of sequencing runs generated for each SG-NEx core cell line. **c**, Number of sequencing runs for each of the RNA-seq technologies. **d**, Illustration of the nf-core Nextflow pipeline (nanoseq) for streamlined processing of Nanopore long-read RNA-seq data.

protein-coding genes (Fig. 3b–e and Supplementary Fig. 4b,c,f–i). A hierarchical clustering of the different samples resulted in the same cell lines being clustered together even when different RNA-seq protocols were used (Fig. 3f). While technology-specific differences in gene expression can be observed (Supplementary Fig. 4j–m and Supplementary Text Figs. 6–8), they can be modeled and removed during differential gene expression analysis when long-read and short-read RNA-seq data are available across conditions of interest, as is the case for the SG-NEx core data (Supplementary Text Fig. 9). These results suggest that aggregated gene-level profiling remained consistent despite the observed differences across sequencing protocols, indicating that Nanopore RNA-seq data can be integrated and combined with short-read RNA-seq data for gene expression analysis.

**Major isoforms are more robustly estimated with long reads**

While gene expression analysis is well established, the estimation of transcript expression abundance is more challenging as distinct transcripts from the same genes are often largely similar[19–21]. Similarly to gene expression on spike-in RNAs, we observed that the Nanopore long-read sequencing protocols showed higher correlation between the estimated and expected transcript expression levels compared to short-read RNA-seq on the Sequin RNAs (Fig. 4a), and SIRV spike-ins showed higher absolute errors with short-read RNA-seq, including SIRVs (E0) and long SIRVs (Supplementary Fig. 5a,b,d,e).

To compare differences in transcript expression on RNA from human cell lines, we first identified the most abundantly expressed transcript for each gene in each cell line ('major isoform') using all replicates of the short-read and Nanopore long-read RNA-seq. On average, long-read and short-read RNA-seq identified the same major isoform for 13,481 (±2,021) genes per cell line, whereas they disagreed in the major isoform for 7,389 (±523) genes per cell line, resulting in more than 28,000 short-read-specific and long-read-specific major isoforms across all cell lines. These numbers are further reflected when transcript expression estimates are compared between replicates from the same cell line (same RNA extract) that were profiled with different sequencing technologies. While gene expression estimates from human cell lines were highly correlated, transcript expression estimates showed higher variation across different protocols, possibly related to differences in library preparation, with the strongest differences observed between long-read and short-read protocols (Fig. 4b,c and Supplementary Fig. 5c,f–k). The highest agreement in abundance estimates was observed for major isoforms that are shared between long-read and short-read RNA-seq data (Fig. 4b,c and Supplementary Fig. 5f–j). For genes where long-read and short-read RNA-seq identified different major isoforms, long-read-specific major isoforms showed significantly higher correlation of expression estimates between both sequencing technologies (two-sided Mann-Whitney $U$-test $P < 0.0001$; Fig. 4b,c and Supplementary Fig. 5g,h). The most frequently observed

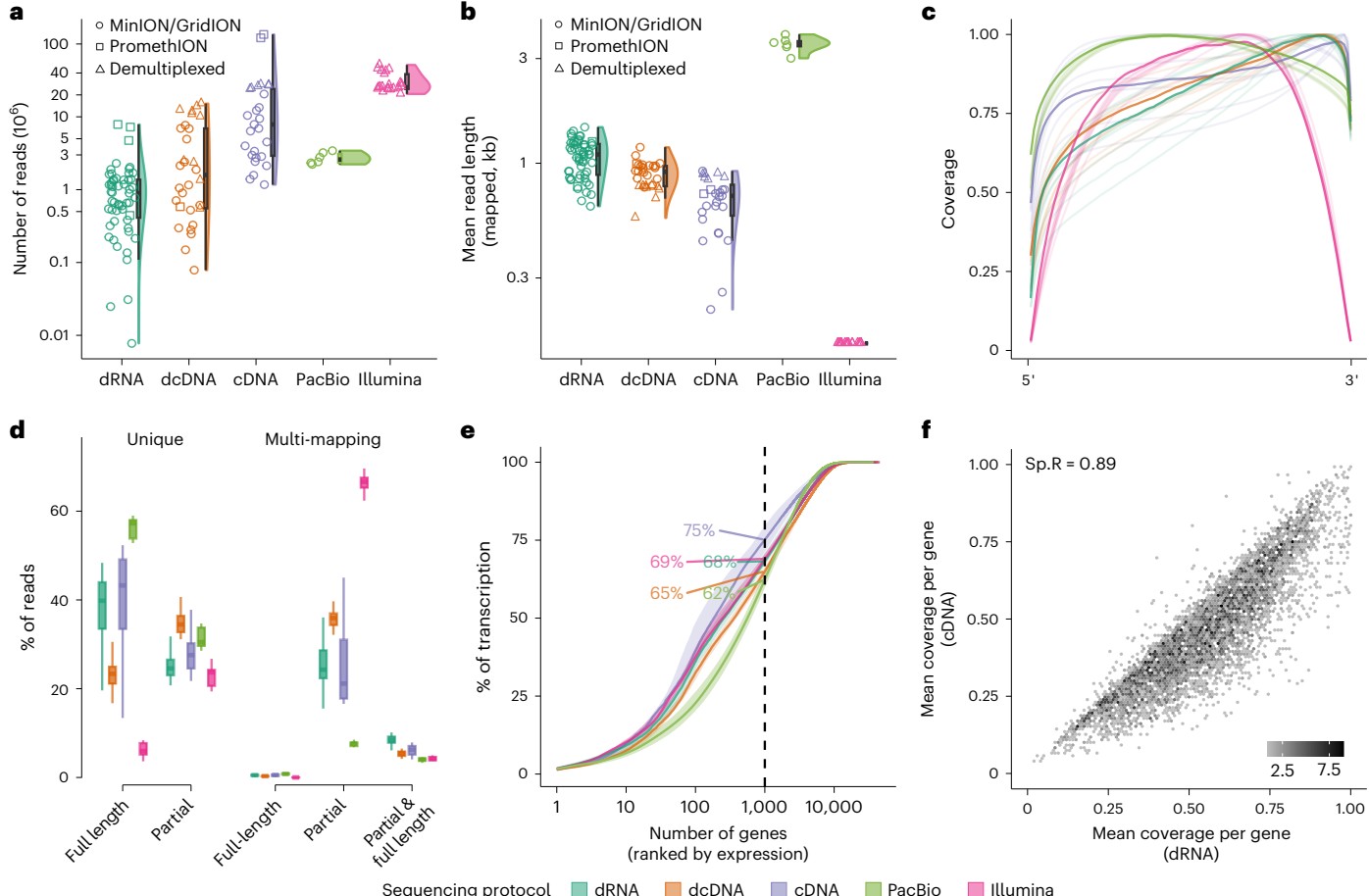

**Fig. 2 | Comparison of RNA-seq protocols. a**, Violin plot showing the median, upper and lower quartiles and 1.5 times the interquartile ranges of the sequencing throughput of RNA (direct RNA, $n = 55$), cDNA (direct cDNA, $n = 30$), PCR (cDNA, $n = 27$), PacBio IsoSeq ($n = 6$) and Illumina ($n = 21$) protocols. Circles represent MinION or GridION experimental runs without multiplexing, squares represent PromethION and non-demultiplexed experimental runs, and triangles represent demultiplexed experimental runs. **b**, Violin plot showing the median, upper and lower quartiles and 1.5 times the interquartile ranges of the average read length per sample of RNA (direct RNA, $n = 55$), cDNA (direct cDNA, $n = 30$), PCR (cDNA, $n = 27$), PacBio IsoSeq ($n = 6$) and Illumina ($n = 21$) protocols. Each point represents an experimental run, squares represent PromethION and non-demultiplexed experimental runs, and triangles represent demultiplexed experimental runs. **c**, Coverage along the normalized transcript length for RNA (direct RNA), cDNA (direct cDNA), PCR (cDNA), PacBio IsoSeq and Illumina

protocols. Each light shaded line represents the average across one cell line, and the darker shaded line represents the average across all cell lines for each protocol. **d**, Box plots showing the median, upper and lower quartiles, and 1.5 times the interquartile ranges of the percentage of reads being uniquely or multi-mapped to transcripts, and whether the read is full-splice-junction matched to the transcript or not (full-splice-match versus partial) for all five protocols ($n = 55, 30, 27, 6$ and 21 for direct RNA, direct cDNA, cDNA, PacBio and Illumina, respectively). **e**, Transcription diversity depicted by the percentage of reads attributed to the number of genes ranked by expression levels from highest to lowest for the five protocols. The dashed line represents the top 1,000 expressed genes, and colored numbers indicate the percentage of reads accounted for them. **f**, Mean read coverage of genes generated using the direct RNA and the PCR cDNA protocol. Each point is colored by the density of genes. Sp.R, Spearman correlation.

---

difference between long-read-specific and short-read-specific major isoforms was the use of alternative first and last exons. Specifically, major isoforms identified by short-read RNA-seq significantly more often used internal exons from the long-read-specific major isoform as first or last exons, typically corresponding to shorter versions of the long-read-specific major isoforms (internal first/last exons; Fig. 4d, two-sided $z$-test, Bonferroni-corrected $P < 0.0001$; Supplementary Fig. 5n). Furthermore, short-read-specific major isoforms were frequently estimated to be lowly expressed by long-read data (counts per million (CPM) < 1; Fig. 4e).

While transcripts from genes with only a few annotated alternative isoforms show close agreement between short-read and long-read data (Supplementary Fig. 5l), variation increased for transcripts from genes with a large number of alternative isoforms (Supplementary Fig. 5l). A comparison of all library preparation methods showed that long reads cover substantially more junctions (Fig. 4f), with substantially more reads being uniquely assigned to a transcript (Fig. 4g). Furthermore,

major isoforms identified by long-read RNA-seq showed substantially higher support by full-splice-match reads (Supplementary Fig. 5m). These data indicate that ambiguity in read-to-transcript assignments related to shorter read length may contribute to the observed difference in transcript expression between short-read and long-read RNA-seq data.

### Read fragmentation impacts transcript expression estimates

To evaluate the impact of shorter read length and fragmentation on transcript abundance estimation, we generated in silico fragmented long-read RNA-seq data to simulate short-read RNA-seq. For this simulation, we used long reads that were aligned to the transcriptome, performed error correction using the reference transcriptome sequence and then sampled 150-bp error-corrected, short (fragmented) reads from these long reads (Fig. 5a). Using this procedure, we obtained matched long-read, fragmented long-read (simulated short-read) and Illumina short-read data for 67 SG-NEx samples, which we then used to estimate the impact of fragmentation on transcript abundance estimation.

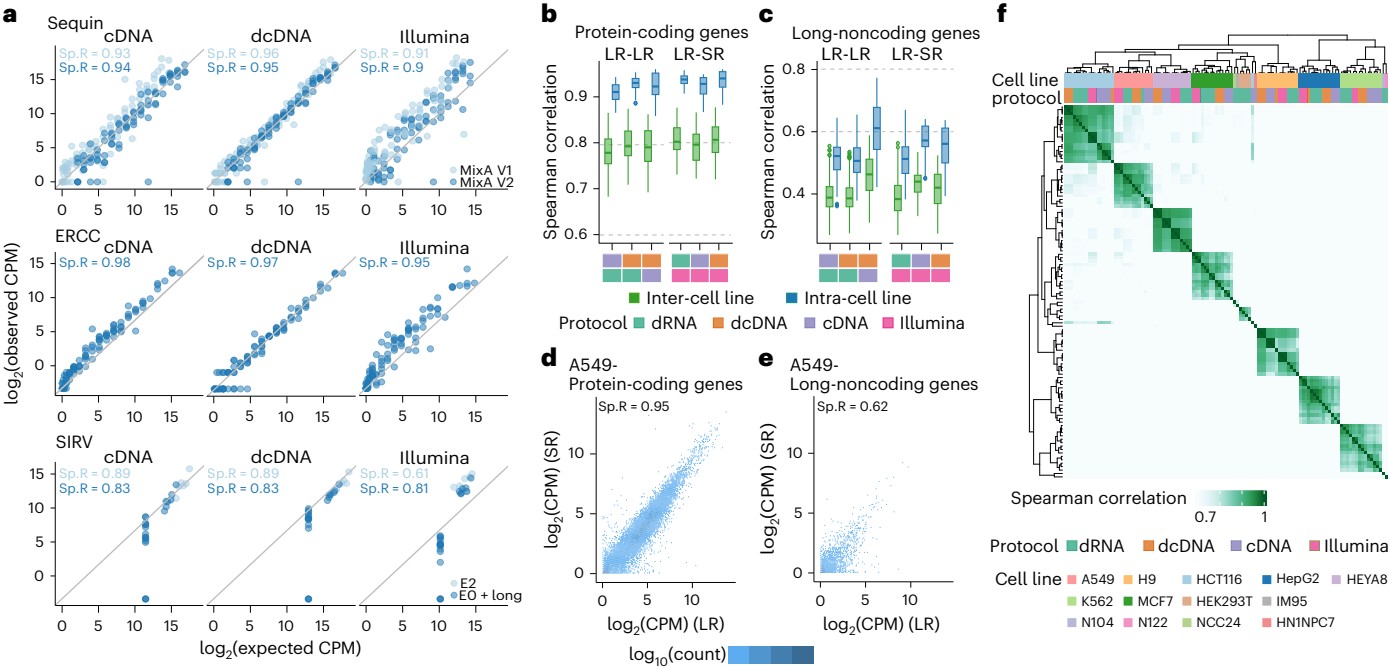

**Fig. 3 | Long-read RNA-seq shows consistency in gene expression quantification with short-read RNA-seq data. a**, Scatterplots of spike-in gene log$_2$-transformed CPM values obtained from long-read direct cDNA and PCR cDNA RNA-seq (using Salmon), and short-read RNA-seq (using Salmon), compared with expected log$_2$-transformed spike-in CPM for five different spike-in RNAs. Light blue points represent Sequin Mix A version 1 and SIRV E2; dark blue points represent Sequin Mix A version 2, ERCC and SIRV E0 + long SIRV RNAs. **b**, Box plots showing the median, upper and lower quartiles, and 1.5 times the interquartile ranges of the Spearman correlation between log$_2$-transformed CPMs (using Salmon) for protein-coding genes from replicates generated by different protocols. Light green represents replicates from different cell lines (inter-cell line: $n$ = 667, 617, 534, 514, 447 and 411 for dRNA versus cDNA, dRNA versus dcDNA, cDNA versus dcDNA, dRNA versus Illumina, cDNA versus Illumina, and dcDNA versus Illumina, respectively) and light blue represents replicates from the same cell line (intra-cell line: $n$ = 113, 103, 90, 86, 73 and 69. **c**, Box plots showing the median, upper and lower quartiles, and 1.5 times the interquartile

ranges of the Spearman correlation between log$_2$-transformed CPMs (using Salmon) for long-noncoding RNA genes from replicates generated by different protocols. Light green represents replicates from different cell lines (inter-cell line: $n$ = 667, 617, 534, 514, 447 and 411, for dRNA versus cDNA, dRNA versus dcDNA, cDNA versus dcDNA, dRNA versus Illumina, cDNA versus Illumina, and dcDNA versus Illumina, respectively). Light blue represents replicates from the same cell line (intra-cell line: $n$ = 113, 103, 90, 86, 73 and 69). **d**, Scatterplot of log$_2$-transformed CPMs from protein-coding genes obtained from long-read direct cDNA (using Salmon) compared with those obtained from short-read RNA-seq (using Salmon) in the A549 cell line. **e**, Scatterplot of log$_2$-transformed CPMs from long-noncoding genes obtained from long-read direct cDNA (using Salmon) compared with those obtained from short-read RNA-seq (using Salmon) in the A549 cell line. **f**, Heatmap showing the correlation of gene log$_2$-transformed CPM estimates across the SG-NEx samples generated using PCR cDNA, direct cDNA, direct RNA and short-read protocols.

While major isoforms identified by long-read RNA-seq were more robust against read fragmentation (Spearman correlation coefficient: major long + short read, $\rho$ = 0.9; long-read-specific, $\rho$ = 0.7), short-read-specific major isoforms were most strongly impacted ($\rho$ = 0.5; Fig. 5b), reflecting our observations from the comparison of long-read RNA-seq with Illumina short-read RNA-seq data. Furthermore, major isoforms that were only identified by Illumina short-read RNA-seq data showed significantly higher transcript expression estimates in the fragmented long-read RNA-seq data when compared with the expected transcript expression from the original long-read data (Fig. 5b,c). A comparison of the fragmented long-read RNA-seq data with the Illumina short-read RNA-seq data generated from the same RNA samples further confirmed that the simulated read fragmentation significantly increased the correlation (before fragmentation, $\rho$ = 0.38; after fragmentation, $\rho$ = 0.61; two-sided pairwise $t$-test, $P$ < 0.0001; Fig. 5d), demonstrating that read fragmentation partially explains the observed difference in transcript expression estimates between short-read and long-read RNA-seq data (Supplementary Fig. 5o). Together, these results demonstrate that transcript quantification is sensitive to RNA fragmentation and reduced read length. In particular, read fragmentation increases transcript expression estimates for some isoforms that appear as the major isoforms only in Illumina short-read RNA-seq data.

## Experimental validation confirms long-read-specific major isoforms

To further confirm our findings, we shortlisted 13 highly expressed genes with discordant major isoforms between long-read and short-read RNA-seq data in the MCF7 cell line, and performed quantitative PCR (qPCR) and digital PCR (dPCR) on the unique sequences specific to each of the long-read-specific and short-read-specific major isoforms. In some cases, where the short-read-specific major isoform is a subset of the long-read-specific major isoform (that is, the splice junctions are completely contained within the long-read-specific major isoform), we compared the unique sequence from the long-read-specific major isoform and the common sequence shared between the long-read-specific and short-read-specific major isoforms. Firstly, we observed that the concentration levels estimated (relative fluorescence units, RFUs) from dPCR experiments agree well with the abundance estimates obtained from long-read RNA-seq, but not those from short-read RNA-seq (Pearson correlation coefficient: ($r$) 0.97 (long-read RNA-seq) versus −0.6 (short-read RNA-seq); Fig. 5e,f). Furthermore, the data confirm that for all 13 genes, the long-read-specific major isoform was the most abundant isoform (Fig. 5e–n and Supplementary Text Fig. 22). qPCR results also confirmed that long-read-specific major isoforms are detectable at high concentration levels, while short-read-specific major isoforms were detected at much lower concentrations (Fig. 5h,l, Supplementary

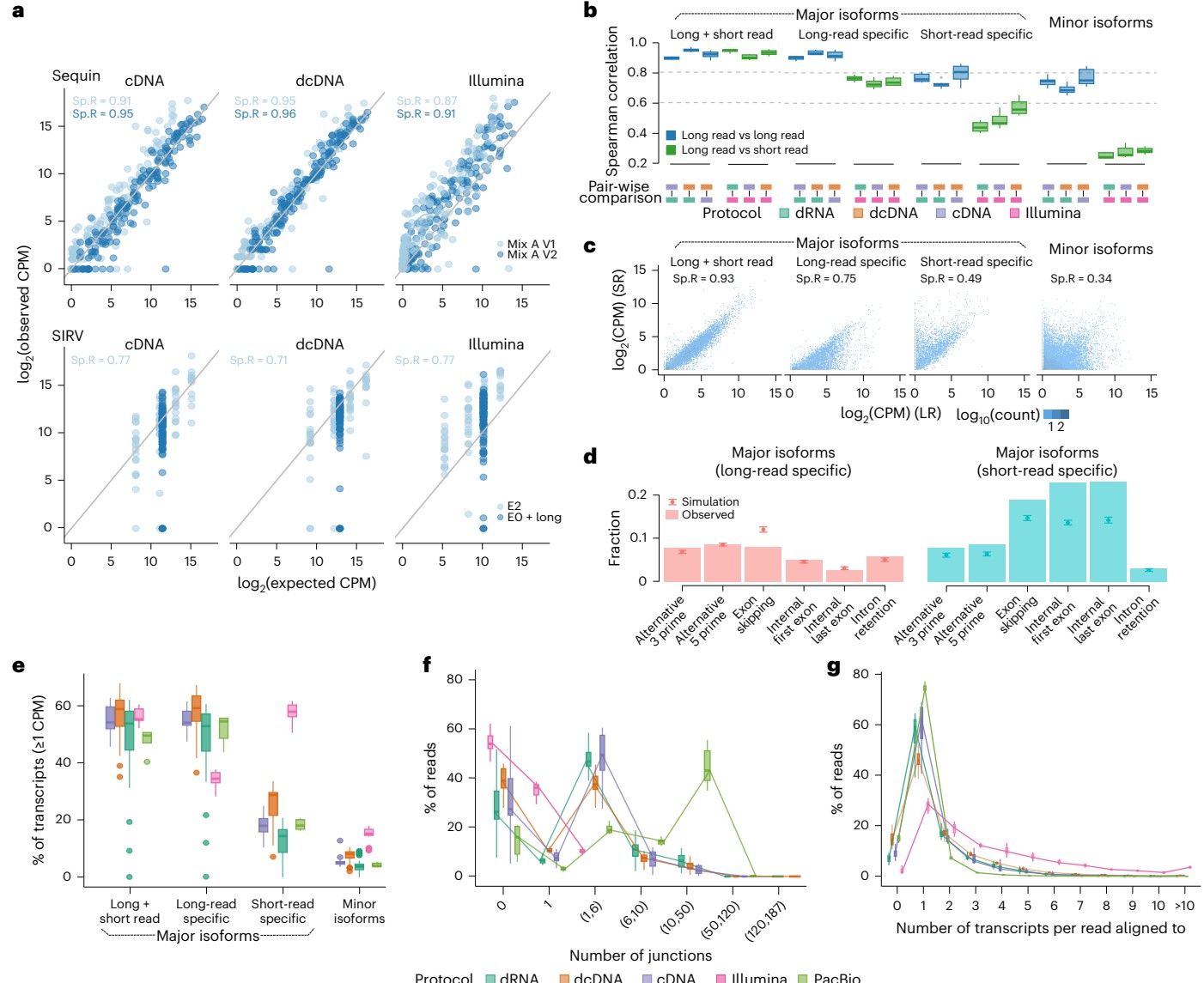

**Fig. 4 | Long-read RNA-seq data improves read-to-transcript assignment and transcript abundance estimation compared to short-read RNA-seq data. a**, Scatterplots of log₂-transformed CPM values obtained from long-read direct cDNA and PCR cDNA, and short-read RNA-seq, compared with expected log₂-transformed CPMs for spike-in transcripts of four different spike-in RNAs. Light blue points represent Sequin Mix A version 1 and SIRV E2; dark blue points represent Sequin Mix A version 2, and SIRV E0 + long SIRV RNAs. **b**, Box plots showing the median, upper and lower quartiles, and 1.5 times the interquartile ranges of the Spearman correlation coefficient for mean log₂-transformed CPM estimates for dominant-status-categorized protein-coding gene isoforms between different RNA-seq protocols for each cell line ($n = 7$). Dark blue indicates comparison between long-read RNA-seq protocols; light blue indicates comparison between long-read and short-read protocols. **c**, Scatterplot of log₂-transformed CPM for dominant-status-categorized protein-coding gene isoforms obtained from long-read direct cDNA RNA-seq compared with those obtained from short-read RNA-seq in the A549 cell line. **d**, Fraction of alternative events identified when comparing major isoforms only in long-read (long-read-specific major isoform) and major isoforms only in short-read RNA-seq (short-read-specific major isoform). Background simulation distribution with mean ± s.d. represented by a point with an error bar ($n = 20$). **e–g**, Box plots showing the median, upper and lower quartiles, and 1.5 times the interquartile ranges of the fraction of dominant-status-categorized protein-coding gene isoforms expressed with at least 1 CPM (**e**), the number of junctions covered per read (**f**) and the number of transcripts uniquely assigned per read for all experiments categorized by five RNA-seq protocols (**g**; $n = 55, 30, 27, 6$ and 21, for direct RNA, direct cDNA, cDNA, PacBio and Illumina, respectively).

Table 5 and Supplementary Text Fig. 22). These observations confirm that long-read RNA-seq data more robustly identify major isoforms compared to short-read RNA-seq data, suggesting that the observed difference in transcript expression between short-read and long-read RNA-seq data may partially be the consequence of inflated expression estimates in short-read RNA-seq due to fragmentation and reduced read length.

**Alternative isoform expression in the SG-NEx data**

The presence of full-splice-match reads in long-read RNA-seq data can unambiguously identify alternative isoforms that are expressed,

enabling the analysis of complex splicing events involving multiple exons[48–50]. Across all seven SG-NEx core cell lines, we observed that thousands of genes used multiple isoforms in each cell line that were supported by full-splice-match reads (Fig. 6a and Supplementary Fig. 6a). The most frequent difference between alternative isoforms and the major isoform is exon skipping (40.4%), followed by alternative promoters (21%) and alternative last exons (16.8%; Fig. 6a,b and Supplementary Fig. 6a). Some of the most complex genes used more than 20 distinct isoforms, often involving alternative promoters, termination sites and splicing (Supplementary Fig. 6b).

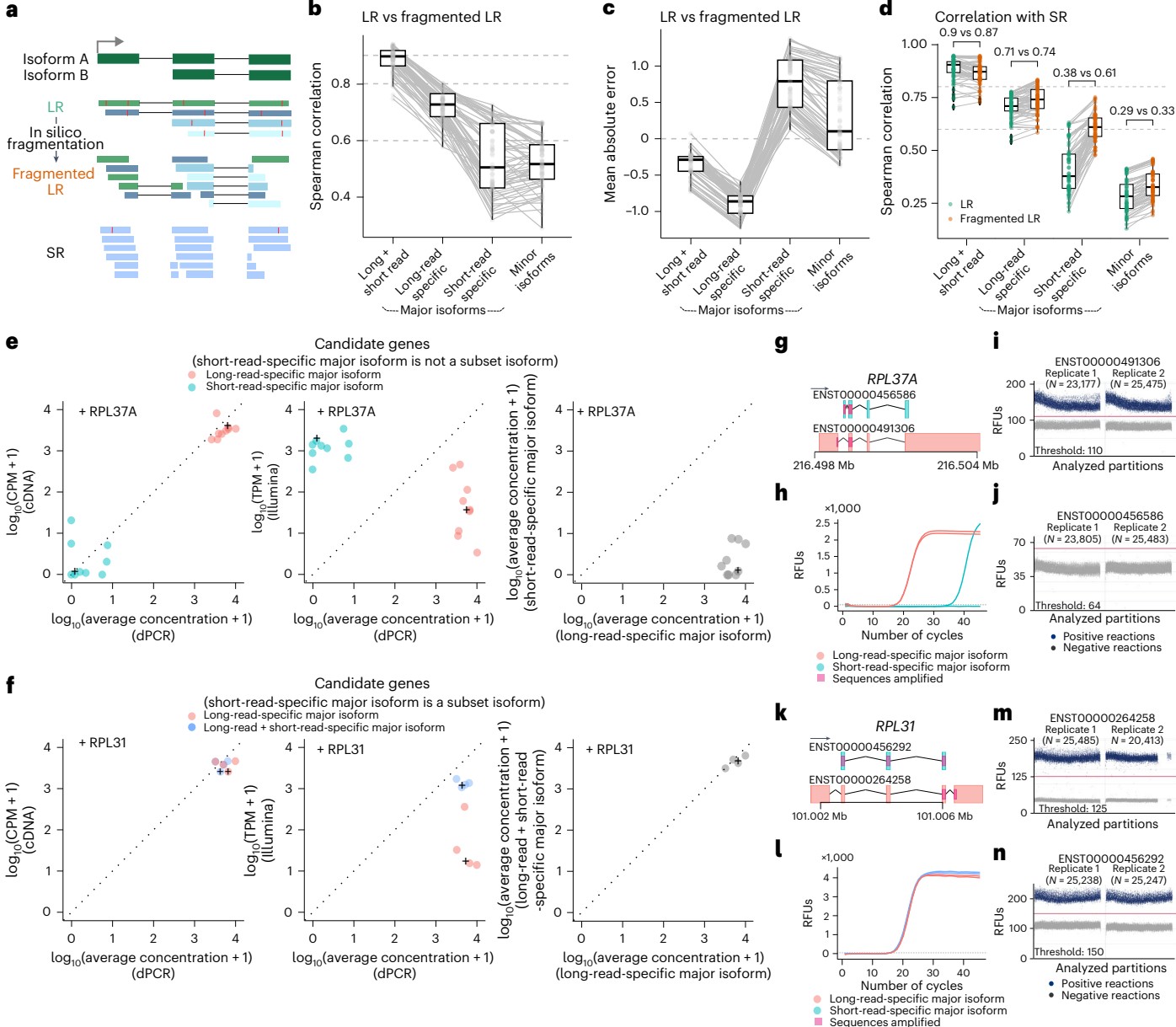

**Fig. 5 | Long-read-specific major isoform is more robust compared to short-read-specific major isoform. a**, Schematic of fragmentation simulation of short-read (SR) from long-read (LR) data. **b–d**, Box plots showing the median, upper and lower quartiles, and 1.5 times the interquartile range of the Spearman correlation (**b**) and mean absolute error (**c**) between LR and matched in silico-simulated short-read RNA-seq data (fragmented LR), and the Spearman correlation between SR and LR or fragmented LR (**d**), for Major isoforms, long-read-specific major isoforms, short-read-specific major isoforms and Minor isoforms. Light gray lines connect the metrics from the same sample pair (*n* = 67). **e,f**, From left to right, the scatterplots showing the log₁₀-transformed: average concentration (cop/µl, copies per microlitre) versus CPM estimates in cDNA long-read RNA-seq data (left); average concentration (cop/µl) versus transcripts per million (TPM) estimates in Illumina short-read RNA-seq data (middle); average concentration (cop/µl) for the long-read-specific major isoform versus that of the short-read-specific major isoform (right); **e**, candidate genes where the short-read-specific major isoform and the long-read-specific major

isoform can be uniquely identified; **f**, candidate genes where the short-read specific major isoform is a subset of the long-read-specific major isoform. **g,k**, Genomic annotations for the long-read-specific and short-read-specific major isoforms and the sequences amplified for each isoform in qPCR with reverse transcription (RT–qPCR) and dPCR experiments. For example, *RPL37A* (**g**), where short-read-specific major isoform is not a subset isoform, and *RPL31* (**k**), where short-read-specific major isoform is a subset isoform. **h,l**, Line plots showing the relationship between the number of PCR cycles and the RFUs in the RT–qPCR experiments, for the assays designed for the long-read-specific and short-read-specific major isoforms of *RPL37A* (**h**) and *RPL31* (**l**). The dotted gray line indicates the threshold defaulted at 50. **i,j,m,n**, Scatterplots showing RFUs in all analyzed partitions, for the assays designed for the long-read-specific (**i**) and short-read-specific (**j**) major isoforms of *RPL37A*, and the long-read-specific (**m**) and short-read-specific (**n**) major isoforms of *RPL31*. Dark blue indicates a positive reaction, and light gray indicates a negative reaction.

Here we found that isoform switching often involves multiple events (13%), illustrating how full-splice-match reads provide the possibility to investigate long-range associations of transcriptional regulation such as alternative promoters with alternative splicing

and alternative transcription end sites (Fig. 6b and Supplementary Fig. 6c).

Next, we identified cell-type-specific major isoform switching events that were supported by full-splice-match reads, highlighting

the variation in isoform expression and alternative splicing within and across human cell lines that is captured in the SG-NEx dataset (Fig. 6c, Supplementary Tables 6 and 7, Supplementary Fig. 6d and Methods).

### Novel transcripts are enriched in repetitive elements

A key advantage of long-read RNA-seq is the ability to identify novel transcripts that are missing in reference annotations[37,49,51]. Using a 10% novel discovery rate (NDR) threshold for transcript discovery in well-annotated genomes (NDR = 0.1), we identified 1,531 novel multi-exon transcript candidates across all samples in the SG-NEx data after filtering potential reverse transcription switch and intra-priming artifacts with SQANTI3 (ref. 52), 622 (40.6%) of which belonged to genomic loci that were not annotated (Supplementary Figs. 7a and 8a–c and Supplementary Table 8). New transcripts generally have lower expression, a lower number of exons and lower transcript length compared to reference transcripts (Supplementary Figs. 7b and 8d,e), particularly for new transcripts from novel genes, indicating that the human genome is comprehensively annotated (see Supplementary Text 7 for a comparison with different reference annotation versions and Supplementary Table 8 for additionally validated novel transcripts in this study). Compared to annotated transcripts, we observed a significant enrichment of repetitive elements in exons from novel transcripts (two-sided Mann–Whitney $U$-test, $P < 0.001$; Supplementary Fig. 7c). While most major isoforms are annotated, we noted that several novel transcripts were involved in cell line-specific isoform switching events (Fig. 6c and Supplementary Fig. 6d), illustrating the value of long reads for transcript discovery, even in well-annotated genomes.

### Discovery and quantification of fusion transcripts

Genomic rearrangements can introduce fusion genes that are associated with clinical characteristics in patients with cancer[53]. Using the SG-NEx long-read RNA-seq data, we searched for fusion genes in the six cancer cell lines from the core dataset. We firstly identified fusion genes and assigned them to genomic breakpoints using JAFFAL[43] and created fusion chromosomes. Next, we aligned reads against the predicted fusion chromosome regions, and then identified and quantified fusion transcripts using Bambu (Methods). Across all cell lines, we identified 106 fusion genes, 79 (74.5%) of which have been validated previously or observed in short-read data (Fig. 6d and Supplementary Table 10). To confirm these results, we selected 12 fusion genes identified in the MCF7 breast cancer cell line for validation with PCR, all of which were confirmed (Supplementary Table 11, Fig. 6d and Supplementary Fig. 9), indicating that long-read RNA-seq enables the robust identification of fusion genes. Interestingly, we additionally found full-splice-match read support for most of the 5′ and 3′ genes, showing that both the fusion genes and the unfused wild-type genes can be detected with long-read RNA-seq (Fig. 6d).

Unlike short-read RNA-seq data, which is limited to detecting the breakpoint, long-read RNA-seq data enable the reconstruction of complete fusion transcripts. On average, we identified two alternative isoforms per fusion gene that are supported by full-splice-match reads (Supplementary Table 12), illustrating how the SG-NEx data provide a unique resource for analyzing full-length fusion transcripts with long-read RNA-seq.

### m⁶A RNA modifications in the SG-NEx data

The ability to directly sequence RNA using the Nanopore technology facilitates the discovery of RNA modifications that otherwise requires dedicated experimental protocols[54]. Here, we used m6Anet[45] to obtain a set of candidate m⁶A positions (Fig. 6e and Supplementary Text Fig. 24). To provide an independent, transcriptome-wide map of m⁶A, we also generated m6ACE-seq data for the HCT116 cell line. Across all seven SG-NEx core cell lines, we found 6,337 positions that were predicted to be modified in at least one cell line (Supplementary Table 13), with 59% of m⁶A sites in the HCT116 cell line being confirmed by m6ACE-seq (Supplementary Fig. 10a). Globally, we note that m⁶A sites can be either cell-type specific or modified across cell types (constitutive sites; Fig. 6f,g and Supplementary Fig. 10e). While constitutively expressed transcripts also showed cell-type-specific modifications, the majority of cell-type-specific m⁶A sites originated from cell-type-specific transcripts (Fig. 6g). Among the most heavily modified genes across all SG-NEx cell lines was the oncogene *MYC*, which is shown to be regulated by m⁶A in cancer[55], illustrating how direct RNA-seq can simultaneously profile RNA expression and modifications (Fig. 6h and Supplementary Fig. 10f,g).

## Discussion

Here we present the results from the SG-NEx project, a systematic resource and benchmark dataset of current RNA-seq technologies. Our study highlights differences between the alternative library preparation and sequencing methods and illustrates the value of the SG-NEx data as a rich resource to study numerous aspects of transcriptome profiling, such as the identification of alternative isoform expression using full-splice-match reads, transcript discovery, fusion transcript quantification and the identification of m⁶A modifications.

The ability to quantify transcript expression has been previously compared between long-read and short-read RNA-seq data using spike-in RNAs[56–60] and cell lines from human[28,58–60], mouse[56,57,59] and other species[59,61,62], consistently showing a high level of variation in transcript expression estimates[23,56,59] (Supplementary Table 2). However, the limited complexity of spike-in RNAs, the lack of a ground truth for biological samples, low sequencing depth, limited replicates or a focus on single protocols or cell lines have made it difficult to quantify the differences in the estimation accuracy when comparing different sequencing technologies[23,57,59,62]. Here we compare differences in transcript expression between short-read and long-read RNA-seq data on six spike-in RNAs, seven human cell lines with multiple replicates, matched simulated short-read data and experimental validation by digital and quantitative PCR. Our analysis shows that the observed variation is partially explained by the fragmentation step, resulting in an overestimation of major isoforms in the short-read RNA-seq data. While short-read RNA-seq generates abundance estimates that are highly correlated with replicates generated using the same technology, our results provide evidence that a benefit of using long-read RNA-seq

---

**Fig. 6 | Profiling of complex transcriptional events, novel transcript, full-length fusion transcript and m⁶A modification in seven human cell lines.** **a**, Bar plots of different isoform switching-type events in the seven human cell lines. **b**, Upset plot of isoform switching event combinations. Top, number of isoforms for each combination. **c**, Heatmap showing the expression levels of 325 isoforms showing significant dominant isoform switching events across the seven human cell lines. The type of events associated with the isoform is indicated at the bottom. Expression is shown for the cell-type-specific isoforms. **d**, Heatmap of fusion gene candidates detected using long-read RNA-seq data, showing the status of validations in this study and in the literature (top), number and class of breakpoints (middle) and full-splice-match read support for the 5′ gene, 3′ gene and the fusion gene (bottom). **e**, Workflow for identifying m⁶A positions from direct RNA-seq data. **f**, Heatmap showing the clustering of direct RNA-seq samples based on the similarity of their m⁶A profile. The similarity was estimated using a two-sided Fisher's test based on the number of common m⁶A sites among all sites that were tested for m⁶A in each pairwise comparison. The odds ratio was then used as enrichment score across sample replicates from the seven cell lines. **g**, Bar plots showing the number of m⁶A sites that were found across the SG-NEx cell lines, for predicted m⁶A sites at genes that are expressed across all cell lines (blue, top), and predicted m⁶A positions at genes that are expressed in at least one cell line (green, bottom). **h**, The *MYC* gene with m6ACE-seq-detected m⁶A positions (green bars) and m6Anet-detected m⁶A probability inferred from direct RNA-seq data (blue bars). The direct RNA-seq coverage is shown in light blue for each cell line.

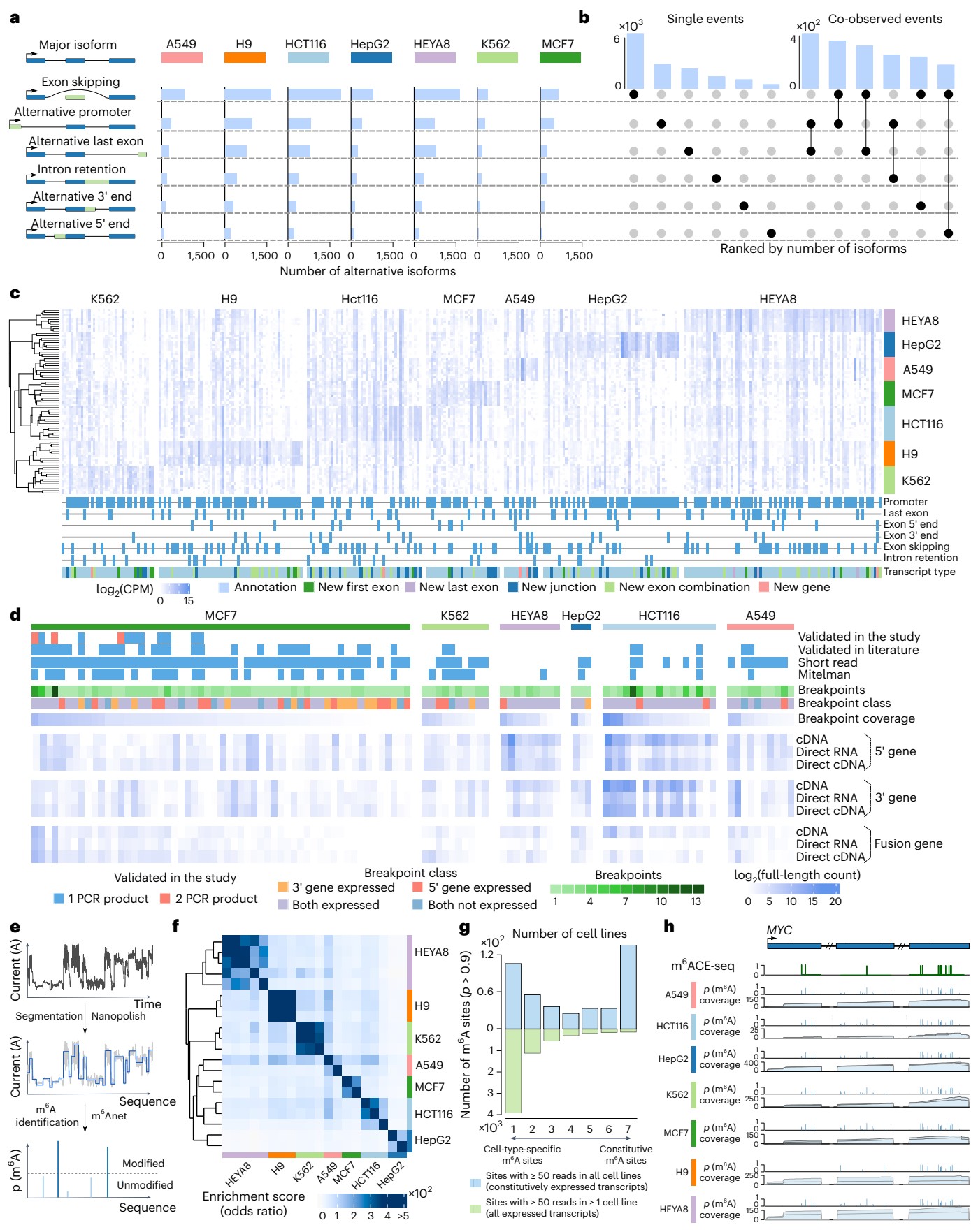

is a more robust and technology-independent identification and estimation of major isoforms compared to short-read data.

Highly repetitive transposable elements have been shown to be expressed in embryonic development[63], and adult tissues[16]. They have also been associated with diseases[64], and remain among the most difficult to study[65,66]. Our data show that newly discovered transcripts in the SG-NEx data are enriched for such repetitive elements compared to annotated transcripts. While the expression pattern, transcript structure and relevance or function of these transcripts remains to be validated, transcript discovery with long-read RNA-seq data may be of particular relevance for tissues that show high expression of repeats, such as cells from early embryos or samples from species that are known to be particularly repeat-rich[67,68].

Compared to short-read and PacBio RNA-seq data, Nanopore RNA-seq has a higher sequencing error rate[24], which affects the precision of read alignments in particular for splice junctions. Therefore, assignment of Nanopore reads to transcripts requires approximate matching, providing a barrier for using tools designed for short-read RNA-seq data. Error correction[69] or splice alignment correction as used in Bambu[37] and Flair[51] or NanoSplicer[70] can alleviate this limitation. Here we observe that the direct RNA-seq data, which have the highest error rate[23,24], generate transcript abundance estimates that are consistent with the direct cDNA data. This indicates that sequencing or alignment errors at splice junctions can effectively be dealt with for transcript quantification.

One of the main advantages of short-read RNA-seq data is the availability of a large amount of public data for benchmarking and discovery[14,71]. Even though the long-read RNA-seq technology is widely available for bulk[27,62,72,73], single-cell[57,74–76] and spatial[77,78] profiling, and despite new methods being actively developed[37,51,79], the lack of a comprehensive data resource prevents widespread use of long-read RNA-seq for routine profiling of the transcriptome[80]. Here we present the SG-NEx dataset and use it to compare the different sequencing protocols as well as comparative benchmarking of computational methods. By including multiple cell lines with multiple replicates sequenced on five different RNA-seq protocols, the data enable the development and evaluation of methods for transcript quantification and discovery, differential expression analysis and fusion gene detection. Furthermore, the SG-NEx data contain the raw current signal from direct RNA-seq for seven core cell lines, each profiled with multiple replicates with partially matched m6A profiling, providing a unique resource to develop and evaluate methods for identification of RNA modifications. Together, we provide a systematic benchmark dataset for short-read and long-read cDNA and direct RNA-seq, a comprehensive overview of transcription in human cell lines and a systematic evaluation of transcriptomics protocols that highlights the benefit of long reads to studying complex transcriptional phenotypes at the resolution of individual transcripts.

## Online content

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

Ying Chen [1,30] ✉, Nadia M. Davidson [2,3,4,30], Yuk Kei Wan [1,30], Fei Yao[1,30], Yan Su [1], Hasindu Gamaarachchi [5,6], Andre Sim [1], Harshil Patel[7], Hwee Meng Low[1], Christopher Hendra[1,8], Laura Wratten [1], Christopher Hakkaart [7], Chelsea Sawyer [9], Viktoriia Iakovleva [1,10], Puay Leng Lee[1], Lixia Xin[1,11], Hui En Vanessa Ng [12], Jia Min Loo [1], Xuewen Ong[13], Hui Qi Amanda Ng[1], Jiaxu Wang [1], Wei Qian Casslynn Koh[1], Suk Yeah Polly Poon[1], Dominik Stanojevic [1,14], Hoang-Dai Tran [1], Kok Hao Edwin Lim[1], Shen Yon Toh [15], Philip Andrew Ewels [7], Huck-Hui Ng[1], N. Gopalakrishna Iyer [15,16], Alexandre Thiery [17], Wee Joo Chng[12,18,19], Leilei Chen[12,20], Ramanuj DasGupta [1], Mile Sikic [1,14], Yun-Shen Chan[1], Boon Ooi Patrick Tan [1,12,13], Yue Wan [1], Wai Leong Tam [1,12,21], Qiang Yu[1], Chiea Chuan Khor [1,16,22], Torsten Wüstefeld [1,15,23], Alexander Lezhava [1], Ploy N. Pratanwanich [1,24,25], Michael I. Love[26,27], Wee Siong Sho Goh[1,28], Sarah B. Ng[1], Alicia Oshlack [4,29], SG-NEx consortium & Jonathan Göke [1,17] ✉

[1]Genome Institute of Singapore (GIS), Agency for Science, Technology and Research (A*STAR), Singapore, Republic of Singapore. [2]The Walter and Eliza Hall Institute of Medical Research, Parkville, Victoria, Australia. [3]Department of Medical Biology, Faculty of Medicine, Dentistry and Health Sciences, The University of Melbourne, Parkville, Victoria, Australia. [4]Peter MacCallum Cancer Centre, Melbourne, Victoria, Australia. [5]School of Computer Science and Engineering, UNSW Sydney, Sydney, New South Wales, Australia. [6]Kinghorn Centre for Clinical Genomics, Garvan Institute of Medical Research, Sydney, New South Wales, Australia. [7]Seqera, Barcelona, Spain. [8]Institute of Data Science, National University of Singapore, Singapore, Singapore. [9]Bioinformatics and Biostatistics, The Francis Crick Institute, London, UK. [10]Division of Gastroenterology and Hepatology, Weill Cornell Medicine, New York, NY, USA. [11]Cardiovascular and Metabolic Disorders Program, Duke-NUS Medical School, Singapore, Singapore. [12]Cancer Science Institute of Singapore, National University of Singapore, Singapore, Singapore. [13]Cancer and Stem Cell Biology Program, Duke-NUS Medical School, Singapore, Singapore. [14]Department of Electronic Systems and Information Processing, Faculty of Electrical Engineering and Computing, University of Zagreb, Zagreb, Croatia. [15]National Cancer Centre Singapore, Singapore, Singapore. [16]Duke-NUS Medical School, Singapore, Singapore. [17]Department of Statistics and Applied Probability, National University of Singapore, Singapore, Singapore. [18]Department of Hematology-Oncology, National University Cancer Institute of Singapore, National University Health System, Singapore, Singapore. [19]Department of Medicine, Yong Loo Lin School of Medicine, National University of Singapore, Singapore, Singapore. [20]Department of Anatomy, Yong Loo Lin School of Medicine, National University of Singapore, Singapore, Singapore. [21]Department of Biochemistry, Yong Loo Lin School of Medicine, National University of Singapore, Singapore, Singapore. [22]Singapore Eye Research Institute, Singapore, Singapore. [23]School of Biological Sciences, Nanyang Technological University, Singapore, Singapore. [24]Department of Mathematics and Computer Science, Faculty of Science, Chulalongkorn University, Bangkok, Thailand. [25]Chula Intelligent and Complex Systems Research Unit, Chulalongkorn University, Bangkok, Thailand. [26]Department of Biostatistics, University of North Carolina at Chapel Hill, Chapel Hill, NC, USA. [27]Department of Genetics, University of North Carolina at Chapel Hill, Chapel Hill, NC, USA. [28]Institute of Molecular Physiology, Shenzhen Bay Laboratory, Shenzhen, China. [29]School of Mathematics and Statistics, University of Melbourne, Parkville, Victoria, Australia. [30]These authors contributed equally: Ying Chen, Nadia M. Davidson, Yuk Kei Wan, Fei Yao. ✉e-mail: chen_ying@gis.a-star.edu.sg; gokej@gis.a-star.edu.sg

**SG-NEx consortium**

Ying Chen[1,30], Nadia M. Davidson[2,3,4,30], Yuk Kei Wan[1,30], Fei Yao[1,30], Yan Su[1], Hasindu Gamaarachchi[5,6], Andre Sim[1], Harshil Patel[7], Hwee Meng Low[1], Christopher Hendra[1,8], Laura Wratten[1], Christopher Hakkaart[7], Chelsea Sawyer[9], Viktoriia Iakovleva[1,10], Puay Leng Lee[1], Lixia Xin[1,11], Hui En Vanessa Ng[12], Jia Min Loo[1], Xuewen Ong[13], Hui Qi Amanda Ng[1], Jiaxu Wang[1], Wei Qian Casslynn Koh[1], Suk Yeah Polly Poon[1], Dominik Stanojevic[1,14], Hoang-Dai Tran[1], Kok Hao Edwin Lim[1], Shen Yon Toh[15], Philip Andrew Ewels[7], Huck-Hui Ng[1], N. Gopalakrishna Iyer[15,16], Alexandre Thiery[17], Wee Joo Chng[12,18,19], Leilei Chen[12,20], Ramanuj DasGupta[1], Mile Sikic[1,14], Yun-Shen Chan[1], Boon Ooi Patrick Tan[1,12,13], Yue Wan[1], Wai Leong Tam[1,12,21], Qiang Yu[1], Chiea Chuan Khor[1,16,22], Torsten Wüstefeld[1,15,23], Alexander Lezhava[1], Ploy N. Pratanwanich[1,24,25], Michael I. Love[26,27], Wee Siong Sho Goh[1,28], Sarah B. Ng[1], Alicia Oshlack[4,29] & Jonathan Göke[1,17]

## Methods

### Data generation and processing

**Cell growth.** Cell growth protocols are described in Supplementary Table 1.

**RNA extraction.** RNA extraction protocols are described in Supplementary Table 1.

**Library preparation.** Sequencing libraries were prepared using the Nanopore direct RNA, direct cDNA and PCR cDNA kits, PacBio-SMRTcell, and short-read sequencing (Illumina 150-bp paired-end reads). m6ACE-seq was generated as described in ref. [81]. Details and deviations are described in Supplementary Table 1.

**Sequencing.** Sequencing runs were performed using MinION/GridION (FLO-MIN106/106D/107), or PromethION (FLO-PRO001/002; Oxford Nanopore Technologies), PacBio Sequel II and Illumina HiSeq 4000 (Supplementary Table 1).

**Reference genome and annotation.** We used the GRCh38 Ensembl annotations release (version 91)[82] and the primary assembly fasta sequence as the reference genome sequence. For transcriptome alignment, we combined the coding and noncoding RNA reference fasta files, selected transcript IDs that matched the reference annotations and removed other transcripts from the transcriptome fasta file. For repeat elements, we used the matched release version of RepeatMasker sequences obtained from the UCSC genome browser[83]. For the spike-in dataset, we used the Sequin annotation and the SIRV-Set 4 annotations, which include SIRVs, ERCC and long SIRV annotations. All reference files can be downloaded from https://github.com/GoekeLab/sg-nex-data/.

**Basecalling.** The raw Nanopore RNA-seq data (fast5) were first converted to BLOW5 files[84] and then basecalled using Guppy (version 6.4.2)[85].

**Read alignment.** The resulting fastq files were aligned using the long-read aligner Minimap2 version 2.22 with parameters '--ax splice --junc-bed' for alignments against the genome, using the junction bed file to correct splicing junctions; and with parameters '--ax map-ont' for alignments against the transcriptome. For direct RNA-seq runs, the additional parameters '--k14' and '--uf' were used as recommended. For short reads, we performed STAR[86] alignment with parameters '--outMultimapperOrder Random --outSAMattributes NH HI NM MD AS nM jM jI XS' to obtain the genome alignment (which was then used to calculate the junction counts). For PacBio samples, we first converted the provided bam files to fastq files and then performed alignment using Minimap2 version 2.22 with parameters '--ax splice:hq -uf' for alignments against the genome and with parameters '--ax map-hifi -uf' for alignments against the transcriptome.

**Transcript abundance estimation.** For long-read data, we applied Bambu, NanoCount[87] and Salmon[88] to estimate the gene and transcript expression levels. For Bambu, we used Bambu version 3.2.4 on genomic alignments obtained from Minimap2 with degradation model off and recommended NDR (only annotated transcripts were used for this analysis, unless otherwise specified). For NanoCount, we first aligned fastq files to the transcriptome with recommended alignment parameters '--ax map-ont -p 0 -N 10' and then applied NanoCount version 1.0.0.post3. For Salmon, we first aligned fastq files to the transcriptome with the recommended alignment parameters '--ax map-ont -p 1.0 -N 100' and then applied Salmon version 1.9.0 with parameters '--ont -q -l U'. For short-read data, we applied Salmon and RSEM[89] to estimate the transcript expression. We used Salmon version 1.9.0 with paired-end fastq files using the quasi mapping-based mode, with parameters '--validateMappings --seqBias --gcBias --posBias' to allow correcting for

sequence bias, fragment-level GC bias and the nonuniform coverage biases. For RSEM, we used RSEM version 1.3.3.

**Gene expression estimation.** Here we define gene expression as the sum of transcript expression from all transcripts that belong to each gene based on reference genome annotations. For Bambu, we used the function transcriptToGeneExpression() to estimate gene expression. For other methods, we calculated the sum of CPM values for all transcripts from each gene to obtain the gene expression estimate.

**Spike-in data.** For the analysis of spike-in data, we first extracted spike-in reads from each sample before combining spike-in reads of the same spike-in type and protocol into one sample ('spike-in only data').

### Data analysis

**Comparison of RNA-seq protocols.** We compared the different read RNA-seq protocols: direct RNA, direct cDNA, cDNA, PacBio cDNA and Illumina short-read RNA-seq protocols in terms of sequencing depth, read length, transcript diversity, coverage and error rate. For sequencing depth (Fig. 2a), we compared the total number of reads generated from each sequencing run. To compare the read length (Fig. 2b), we calculated the mean read length of aligned reads for each sequencing run and each protocol. To evaluate the read coverage between protocols (Fig. 2c), we used the coverage function from GenomicAlignments[90] to obtain the coverage for each position along the transcript. Here we included regions corresponding to D(deletion) in the CIGAR string for the coverage calculation. We then divided each transcript into 100 equal bins and took the average of the coverage read count within each bin, which was then normalized by the maximum for each transcript to obtain the relative transcript coverage. We then averaged the normalized relative coverage across samples within each protocol and each cell line for each bin.

To compare the percentage of full-splice-match reads (Fig. 2d), we looked at the splice-junction match between read alignments (after junction alignment correction with Bambu) and annotation, and defined a read as a full-splice-match read upon an exact splice-junction match to any of the annotated transcripts, or as a partial read upon the splice junction of the read alignments being only partially contained within the transcript. For fair comparison, data generated from all RNA-seq protocols were processed by Bambu to calculate the splice-junction match.

To compare the transcript diversity between protocols (Fig. 2e), we first ranked the genes by the average expression within each protocol and each cell line and then calculated the cumulative read count for genes ranked from top expressed to low expressed for each sequencing run. Due to the existence of similar read counts, there were missing points in the gene ranks; therefore, we performed interpolation to estimate the cumulative read frequency for such points. To summarize the transcript diversity for each protocol, we then took the median of the cumulative read frequency across samples for each protocol. We also compared the transcript diversity between protocols when ranking genes by their gene length to understand how genes with different lengths were being sequenced by each protocol. For this analysis, we ranked genes by their gene length (defined as the maximum isoform length) and then calculated the cumulative read count for genes ranked from shortest to longest for each sequencing run.

To compare the gene coverage (Fig. 2f), we aggregated the read count from all read classes aligned to each gene as the expected read count if the read classes were full-splice-match reads with respect to the gene. We then calculated the relative length ratio of each read class where the width of each read class was divided by the maximum transcript length within the gene, and then computed a relative length-adjusted total read count for each gene as the observed read counts. By taking the ratio between the length-adjusted and original total read count, we obtained an approximate measure for gene

coverage. Similarly, we averaged the gene coverage ratios across samples generated from the same protocol for each gene to compare between protocols. We only included genes that were expressed in all protocols with an average expression level being above 30 CPM. To identify the genes that show significant differences in coverage, we tested the proportions for each gene using a two-sided $z$-test without assuming the equal variance between two protocols.

In addition to sequencing depth, we also compared the total mapped bases filtered by cigar string and error rate reported for each sequencing run by running 'samtools stats'.

To show the distribution of full-splice-match coverage, we calculated the coverage for each read based on the transcriptome alignment and calculated the ratio against the transcript length, normalizing the coverage ratio to a number over a thousand. We then calculated the mean number of transcripts for each coverage ratio ranging from 0 to 1 for each protocol and the cumulative distribution of the coverage ratios.

All analyses described above were also conducted for spike-in data.

**Comparison of gene and transcript expression.** For the analysis reported in the study, we compared the estimates obtained from Salmon for both long-read and short-read data ('Transcript abundance estimation'). Results generated from other methods are included in Supplementary Figs. 4 and 5. This analysis was based on annotated isoforms; novel isoforms were not included in the comparison. All long-read transcript expression estimates used CPM values as estimated by the different computational methods. Short-read RNA-seq estimates used TPM values.

*Definition of major and minor isoforms.* Here we defined major isoforms for each gene as the isoform that has the highest transcript expression estimate. For this analysis, major isoforms were calculated for each cell line using the average transcript expression from all Nanopore long-read RNA-seq replicates (long-read major isoforms) and the average from all short-read RNA-seq replicates (short-read major isoforms). We then compared long-read and short-read major isoforms to identify major isoforms identified by both sequencing technologies, long-read-specific major isoforms, short-read-specific major isoforms and isoforms that were never estimated to be the most highly expressed (minor isoform).

*Comparison of spike-in RNAs.* For the spike-in data analysis, we used the spike-in-only data that were combined for each protocol and each spike-in type. To obtain the expected CPM for the combined data for each spike-in type and protocol, we first multiplied the total number of reads by the spike-in percentage for the total expected spike-in reads, and then multiplied this number by the relative concentration of each spike-in transcript in the total spike-in set for each sample to obtain the true expected CPM. To benchmark different RNA-seq protocols in profiling gene and transcript expression using spike-in data, we compared the $\log_2$-transformed estimated CPM against the $\log_2$-transformed expected CPM using the Spearman correlation, mean absolute error, mean relative absolute difference, mean relative difference, root mean squared error and R-squared value, for both gene and transcript expression. The metrics were calculated for all RNA-seq protocols and all methods. To compare across protocols, we also computed the Spearman correlation between the $\log_2$-transformed estimated CPM between each pair of protocols.

*Comparison of human cell line RNA.* For the human chromosome gene and transcript estimation evaluation between different RNA-seq protocols, we matched the annotations to include only genes with all transcripts present in both genomic annotation and transcriptomic annotations, removing 1,066 (2%) genes in the process. As outlined in the study, we focused on protein-coding genes, antisense

RNAs, long intergenic noncoding RNAs, noncoding RNAs and macro long-noncoding RNAs, which further removed 25,441 (43.6%) genes (pseudogenes and short RNAs) from consideration. After filtering, a total of 32,861 genes were included, with 169,254 associated transcripts.

For each sequencing run, we identified the cross-cell line replicates (same replicate number but from different cell lines, different protocols) and the within-cell line replicates (same replicate number, same cell line, different protocols). For gene expression, we computed the Spearman correlation and mean relative absolute differences between each replicate pair for protein-coding genes, long-noncoding genes and other genes, respectively. For transcript expression, we computed the Spearman correlation and mean absolute error between each replicate pair for major isoform, major isoform-long read (LR), major isoform-short read (SR) and minor isoform, respectively.

*Comparison of splicing events/alternative promoters for short-read-specific and long-read-specific major isoforms.* To determine if there were systematic differences between the major isoforms identified between long reads and short reads for each gene, we used the function compareTranscripts in Bambu to calculate how often a long-read-specific major isoform (1) used an internal exon from the short-read-specific major isoform as first or last exon; (2) underwent an exon skipping event or intron retention event; and (3) underwent an alternative splicing event at both 3′ and 5′ ends against the short-read-specific major isoform of the same gene. To simulate the (expected) background distribution, we randomly chose one isoform that was not the short-read-specific major isoform from the same gene and compared it against the short-read-specific major isoform for the same alternative events, repeating it for 20 times. Similarly, we compared the short-read-specific major isoform against the long-read-specific major isoform from the same gene and conducted the simulation by randomly choosing one isoform from the non-long-read-specific major isoforms of the same gene and compared against the long-read-specific major isoform.

*Clustering of samples using gene expression estimates.* We clustered all SG-NEx samples using gene-level correlation computed for the genes with the top-ranked variation across all samples using the heatmap, dendrogram and PCA functions in R.

*Analysis if junction read coverage.* We determined the number of junctions covered by each read using the GenomicAlignments package[90]. To determine the number of reads that can be uniquely assigned, we processed short-read data using Bambu without transcript discovery to obtain the read class assignment. We then compared the distribution of the number of transcripts that can be assigned by each read class.

*In silico read fragmentation.* To simulate fragmented, short-read data from long-read data, we identified the transcripts that each read aligned to in the primary alignments. For each long read, we extracted the coordinates in the corresponding transcript, and randomly simulated the sequencing start position uniformly from the start position of the read alignment to 150 bp before the end position of the read alignment. To mimic the fragmentation process, the process was randomly repeated $N$ times for each read according to the read length ($N \leq$ mapped read length/150 bp, in silico fragmentation). After simulating the sequencing starts, we then extracted the 150-bp transcript sequences based on the annotation for each simulated read. Here we only used reads with mapped read lengths of at least 150 bp. We then compared the $\log_2$-transformed estimated CPM for major isoforms, long-read-specific major isoforms, short-read-specific major isoforms and minor isoforms between fragmented long-read data (simulated SR) with the long-read data and calculated the Spearman correlation and mean absolute error between the same sequencing run. We also calculated the Spearman correlation between fragmented long-read

and short-read data using the same replicates from the same cell line, to understand the impact of fragmentation in explaining the differences between long-read and short-read data.

We also performed this fragmentation simulation using short-read data (Supplementary Fig. 5p). To systematically perform the simulation, we fragmented the short-read data from the 5′ end to single-end read of different lengths ranging from 150 bp to 50 bp at a 25-bp difference for all short-read RNA-seq samples, using seqtk (https://github.com/lh3/seqtk/) with parameters 'trimfq -L'.

*RT–qPCR and dPCR validation of the long-read-specific versus short-read-specific major isoforms.* To further confirm whether long-read-specific or short-read-specific major isoforms are the true dominant isoform among the genes with discordant major isoforms between long and short-read RNA-seq data, we shortlisted 13 candidate genes from the MCF7 cell line, each with either long-read-specific or short-read-specific major isoforms being the most highly expressed. Around 1 μg of total RNA from MCF7 breast cancer cell lines was reverse transcribed using the SuperScript IV First-Strand cDNA Synthesis System (Invitrogen) following the manufacturer's instructions. For the 13 shortlisted candidate genes, we designed and selected 26 pairs of primers to amplify the unique transcript sequences from each isoform. In the case where the short-read-specific major isoform is a subset of the long-read-specific major isoform (that is, the splice junctions from the short-read-specific major isoform are completely contained within the long-read-specific major isoform), we amplified the common sequence shared between the long-read-specific and short-read-specific major isoforms instead for the short-read-specific major isoform. RT–qPCR was performed using Luna Universal qPCR Master Mix (NEB) with the following thermocycling conditions: 95 °C for 2 min, then 95 °C for 15 s, and 60 °C for 30 s for 45 cycles. Each reaction well contained 1 μl of 10× diluted cDNA and 19 μl of Master Mix, totaling 20 μl, in a Bio-Rad Hard-Shell 96-well PCR plate. dPCR was conducted using 3× EvaGreen PCR Master Mix (Qiagen) with the following thermocycling conditions: 95 °C for 2 min, then 95 °C for 15 s, 60 °C for 15 s, and 72 °C for 15 s for 40 cycles. Imaging conditions included an exposure duration of 300 ms and a gain of 4. Each reaction well contained 5 μl of 500× diluted cDNA and 35 μl of Master Mix, totaling 40 μl, in a 26k 24-well QIAcuity Nanoplate.

**Analysis of alternative isoform expression.** Details for the analysis of alternative isoform expression can be found in Supplementary Note 1.

**Novel transcripts and repeat enrichment analysis.** Details for novel transcripts and repeat enrichment analysis can be found in Supplementary Note 2.

**Fusion gene analysis.** Details for fusion gene analysis can be found in Supplementary Note 3.

**m⁶A modification analysis.** Details for $m^6A$ modification analysis can be found in Supplementary Note 4.

#### Reporting summary
Further information on research design is available in the Nature Portfolio Reporting Summary linked to this article.

#### Data availability
The SG-NEx data are available at https://github.com/GoekeLab/sg-nex-data/. All data are deposited in the European Nucleotide Archive under accession code PRJEB44348. Fast5 (BLOW5), fastq, bam and additional processed data are available for fast compute-optimized access on S3 through the AWS open data sponsorship (https://registry.opendata.aws/sgnex/). We also downloaded the datasets from the ENCODE portal[91] with the following identifiers: ENCSR000CON,

ENCSR000CWM, ENCSR000CPE, ENCSR000AEM and ENCSR000CPT (see Supplementary Text Table 1 for detailed download links).

#### Code availability
nf-core/nanoseq is a streamlined, community-curated pipeline for Nanopore sequencing data processing and analysis (https://nf-co.re/nanoseq/). The code used for this paper is deposited in GitHub via https://github.com/GoekeLab/sg-nex-data/tree/master/manuscript/.

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

#### Acknowledgements
This work is funded by the Agency for Science, Technology and Research (A∗STAR), Singapore. J.G. is supported by grants from the National Medical Research Council (OFIRG20nov-0108 and OFIRG16nov019). N.M.D. is supported by an NHMRC Investigator Grant (GNT2016547). L.C. is supported by Singapore Ministry of Education's Tier 3 Grant (MOE-MOET32023-0002) and a National Research Foundation (NRF) Competitive Research Programme Grant (NRF-CRP26-2021-0001). W.L.T. is supported by NRF-NRFIO8-2022-0007. A.O. is supported by NHMRC (GNT1196256). The funders had no role in study design, data collection and analysis, decision to publish or preparation of the manuscript. We thank AWS for making the SG-NEx data available through the AWS Open Data Sponsorship Program. We thank S. Prabhakar and N. Nagarajan for help with data generation. We thank M. Clark, and J. Gleeson for contributing tutorials on using NanoCount, R. Herwig and Y. Bi for contributing tutorials on using IsoTools, and A. Brooks and C. Felton for contributing tutorials on using FLAIR, to the SG-NEx GitHub repository.

#### Author contributions
Y.C. and J.G. conceived the project, designed the study and experiments. Y.C., N.M.D., Y.K.W., F.Y. and J.G. analyzed data. Y.S. performed qPCR and dPCR experiments. H.G. performed SLOW5 to convert fast5 to BLOW5 files, and basecalled BLOW5 files. A.S. and H.P. contributed to data analysis. H.M.L. contributed to library preparation and sequencing. C. Hendra contributed to RNA

modification data analysis. L.W., C. Hakkaart and C.S. contributed to the nanoseq pipeline. V.I., P.L.L., L.X., H.E.V.N., J.M.L., X.O., H.Q.A.N., J.W., W.Q.C.K., S.Y.P.P., D.S., H.-D.T., K.H.E.L., S.Y.T., P.A.E., H.-H.N., N.G.I., A.T., W.J.C., L.C., R.D., M.S., Y.-S.C., B.O.P.T., Y.W., W.L.T., Q.Y., C.C.K. and T.W. contributed to growing cell lines and extracting RNAs. A.L., P.N.P., M.I.L., W.S.S.G., S.B.N. and A.O. contributed to the analysis interpretation. Y.C., N.M.D., Y.K.W., F.Y. and J.G. organized and wrote the paper with contributions from all authors.

## Competing interests

J.G. received travel and accommodation expenses to speak at the Oxford Nanopore Community Meeting 2018. N.M.D. has previously received travel and accommodation expenses from Oxford Nanopore Technologies. H.G. has previously received travel and accommodation expenses from Oxford Nanopore Technologies. M.S. has been jointly funded by Oxford Nanopore Technologies and AI Singapore for the project AI-driven De Novo Diploid Assembler and has received travel funds to speak at events hosted by Oxford Nanopore Technologies.

W.S.S.G. owns shares in Oxford Nanopore Technologies. The other authors declare no competing interests.

## Additional information

**Correspondence and requests for materials** should be addressed to Ying Chen or Jonathan Göke.

# Reporting Summary

Nature Research wishes to improve the reproducibility of the work that we publish. This form provides structure for consistency and transparency in reporting. For further information on Nature Research policies, see our Editorial Policies and the Editorial Policy Checklist.

## Statistics

For all statistical analyses, confirm that the following items are present in the figure legend, table legend, main text, or Methods section.

| n/a | Confirmed | |
|---|---|---|
| ☐ | ☒ | The exact sample size (*n*) for each experimental group/condition, given as a discrete number and unit of measurement |
| ☐ | ☒ | A statement on whether measurements were taken from distinct samples or whether the same sample was measured repeatedly |
| ☐ | ☒ | The statistical test(s) used AND whether they are one- or two-sided *Only common tests should be described solely by name; describe more complex techniques in the Methods section.* |
| ☒ | ☐ | A description of all covariates tested |
| ☐ | ☒ | A description of any assumptions or corrections, such as tests of normality and adjustment for multiple comparisons |
| ☐ | ☒ | A full description of the statistical parameters including central tendency (e.g. means) or other basic estimates (e.g. regression coefficient) AND variation (e.g. standard deviation) or associated estimates of uncertainty (e.g. confidence intervals) |
| ☐ | ☒ | For null hypothesis testing, the test statistic (e.g. *F*, *t*, *r*) with confidence intervals, effect sizes, degrees of freedom and *P* value noted *Give P values as exact values whenever suitable.* |
| ☒ | ☐ | For Bayesian analysis, information on the choice of priors and Markov chain Monte Carlo settings |
| ☒ | ☐ | For hierarchical and complex designs, identification of the appropriate level for tests and full reporting of outcomes |
| ☐ | ☒ | Estimates of effect sizes (e.g. Cohen's *d*, Pearson's *r*), indicating how they were calculated |

*Our web collection on statistics for biologists contains articles on many of the points above.*

## Software and code

Policy information about availability of computer code

| Data collection | Cell growth and RNA extraction protocols are described in Supplementary Table 1. |
|---|---|
| Data analysis | Sequencing libraries were prepared using the Nanopore direct RNA, direct cDNA, and PCR cDNA kits, short read sequencing (Illumina paired end 150 bp), and PacBio long read SMRTcell. Details and deviations are described in Supplementary Table 1. Sequencing runs were performed using MinION/GridION (FLO-MIN106/106D/107), or PromethION (FLO-PRO001/002) (Oxford Nanopore Technologies), Illumina HiSeq 4000 system, and PacBio Sequel II system(Supplementary Table 1). nf-core/nanoseq is a streamlined, community curated pipeline for Nanopore Sequencing data processing and analysis: https://nf-co.re/nanoseq The code used for this manuscript is deposited in github repository and can be accessed here: https://github.com/GoekeLab/sg-nex-data/tree/master/manuscript. |

For manuscripts utilizing custom algorithms or software that are central to the research but not yet described in published literature, software must be made available to editors and reviewers. We strongly encourage code deposition in a community repository (e.g. GitHub). See the Nature Research guidelines for submitting code & software for further information.

## Data

Policy information about availability of data

All manuscripts must include a data availability statement. This statement should provide the following information, where applicable:
- Accession codes, unique identifiers, or web links for publicly available datasets
- A list of figures that have associated raw data
- A description of any restrictions on data availability

The SG-NEx data is available at: https://github.com/GoekeLab/sg-nex-data. All data is deposited in ENA (https://www.ebi.ac.uk/ena/browser/view/PRJEB44348). Fast5 (BLOW5), fastq, bam, and additional processed data is made available for fast compute optimised access on S3 through the AWS open data sponsorship (http://sg-nex-data.s3-website-ap-southeast-1.amazonaws.com/ and http://sg-nex-data-blow5.s3-website-ap-southeast-1.amazonaws.com/) . We also downloaded

the datasets from the ENCODE portal (Sloan et al. 2016) (https://www.encodeproject.org/) with the following identifiers: ENCSR000CON, ENCSR000CWM, ENCSR000CPE, ENCSR000AEM, ENCSR000CPT (see Supplementary Text Table 1 for detailed download links).

# Field-specific reporting

Please select the one below that is the best fit for your research. If you are not sure, read the appropriate sections before making your selection.

☒ Life sciences    ☐ Behavioural & social sciences    ☐ Ecological, evolutionary & environmental sciences

For a reference copy of the document with all sections, see nature.com/documents/nr-reporting-summary-flat.pdf

# Life sciences study design

All studies must disclose on these points even when the disclosure is negative.

| | |
|---|---|
| Sample size | To perform RNA-Seq differential expression analysis, we acquired a minimum number of at least 3 biological replicates for each protocol run from each cell line, for differential expression analysis (Schurch et al, 2016). |
| Data exclusions | Sequencing runs with less than 400,000 reads were excluded as these runs generally show low quality compared to the other runs and may produce unreliable results. |
| Replication | RT-PCR experiments to confirm novel transcripts were conducted once, with external validation from recent Gencode/Ensembl release. RT-qPCR experiment results to confirm long-read specific major isoforms were confirmed with dPCR experiments. |
| Randomization | Batch effects adjustment and covariate adjusted analysis is performed to control for potential confounding effects of sequencing platforms and batches. |
| Blinding | Blinding is not relevant to the study as no allocation is needed for data during data collection and analysis, as they are all predefined by the sequencing technologies and cell line types. |

# Reporting for specific materials, systems and methods

We require information from authors about some types of materials, experimental systems and methods used in many studies. Here, indicate whether each material, system or method listed is relevant to your study. If you are not sure if a list item applies to your research, read the appropriate section before selecting a response.

## Materials & experimental systems

| n/a | Involved in the study |
|---|---|
| ☒ | ☐ Antibodies |
| ☐ | ☒ Eukaryotic cell lines |
| ☒ | ☐ Palaeontology and archaeology |
| ☒ | ☐ Animals and other organisms |
| ☒ | ☐ Human research participants |
| ☒ | ☐ Clinical data |
| ☒ | ☐ Dual use research of concern |

## Methods

| n/a | Involved in the study |
|---|---|
| ☒ | ☐ ChIP-seq |
| ☒ | ☐ Flow cytometry |
| ☒ | ☐ MRI-based neuroimaging |

# Eukaryotic cell lines

Policy information about cell lines

| | |
|---|---|
| Cell line source(s) | K562, A549, MCF7, HCT116, HepG2, and Hek293T are from American Type Culture Collection (ATCC). H9 cell line replicate 1 is obtained from WiCell and all other replicates are kindly gifted from Lawrence W. STANTON's lab. The HEYA8 cell line was obtained from collaborators. The NCC24 and SNU719 cell lines are from the Korean Cell Line Bank. The IM95 cell line is from Japan Health Science Research Resource Bank. |
| Authentication | Of all the cell lines, HEYA8, NCC, SNU and IM95 cell lines are authenticated by STR profiling. No authentication is done for K562, A549, MCF7, HCT116 , HepG2, and Hek293T cell lines as they are authenticated by ATCC. No authentication is done for the H9 cell line. |
| Mycoplasma contamination | All cell lines tested negative for Mycoplasma contamination. |
| Commonly misidentified lines (See ICLAC register) | None |

