## [Peer Review File · Nature Methods]

A systematic benchmark of Nanopore long read RNA sequencing for transcript level analysis in human cell lines

Corresponding Author: Dr Jonathan Goeke

Version 1:

Decision Letter:

2nd Jan 2022

Dear Jonathan,

Happy holidays! Your Resource entitled "A systematic benchmark of Nanopore long read RNA sequencing for transcript level analysis in human cell lines" has now been seen by 3 reviewers, whose comments are attached. In the light of their advice, I'm sorry to say that we cannot offer to publish your manuscript in Nature Methods.

You will see that, while they find your work of some potential interest, the reviewers raise serious novelty concern about the results presented in this paper as well as concerns about the analysis approaches used in this paper. We think that these criticisms are sufficiently important as to prevent publication of your work in Nature Methods.

I am very sorry that we cannot be more positive on this occasion but hope that you find the reviewers' comments helpful when preparing your paper for submission elsewhere.

Best regards,
Lei

Lei Tang, Ph.D.
Senior Editor
Nature Methods

Reviewer Comments:

Reviewer #1 (Remarks to the Author):

A. Summary of the key results

The manuscript presents the comprehensive sequencing of the transcriptome from 7 cell lines using Short-read (Illumina cDNA) and long-read (Nanopore direct RNA, direct cDNA, PCR-based cDNA) technologies. The manuscript describes a set of various analyses performed on these datasets: a comparison between the different technologies, abundance estimates, detection of alternative transcript isoforms, detection of transcripts on repeat elements, detection of fusions, and detection of m6A. The results provided are used to argue for the benefit of using Nanopore sequencing technology for these transcript analyses. However, the results presented are not novel, many of the analyses are poorly performed, and the paper does not highlight any novelty provided by their cell line dataset. All these issues are described below.

B. Originality and significance: if not novel, please include reference

Overall, the article is quite descriptive and does not present anything that has not been already described elsewhere:

The association between alternative transcription start sites and splicing using single-molecule long-read sequencing has

been already documented, see e.g.:
<https://pubmed.ncbi.nlm.nih.gov/29598823/>
<https://pubmed.ncbi.nlm.nih.gov/30320766/>
But these articles have not been cited.

Isoform diversity from long-read sequencing has already been documented multiple times (e.g. <https://pubmed.ncbi.nlm.nih.gov/31676752/>, <https://pubmed.ncbi.nlm.nih.gov/30320766/>) but none of these has been cited. Moreover, although this manuscript and many articles before have reported novel transcripts from long-read sequencing, whether these are real or not remains to be confirmed. This manuscript does not provide any novelty regarding this question.

Quantification with Nanopore Long-read sequencing (cDNA and RNA) and comparison with short reads and with spike-ins have been done before:
<https://pubmed.ncbi.nlm.nih.gov/31624302/>
<https://pubmed.ncbi.nlm.nih.gov/28722025/>
But these articles have not been cited.

The result related to the ability of Nanopore direct RNA sequencing to identify m6A RNA modifications has already been shown before multiple times (e.g. <https://pubmed.ncbi.nlm.nih.gov/31501426/>, not cited)

The result about the conservation of the mRNA methylome across cell types has been reported before multiple times (e.g. <https://pubmed.ncbi.nlm.nih.gov/22575960/>, not cited)

C. Data & methodology: validity of approach, quality of data, quality of presentation

Some of the tools used and analyses performed have not been validated. E.g. nanoseq is described as a pipeline to perform alignment; transcript discovery and quantification; and differential expression analysis. Nanoseq uses already published tools, but some of its steps include new software that has not been validated and/or compared with other tools.

Some of the plots show the results using one of these tools, bambu, in comparison with spike-in data and across technologies. So this is a sort of benchmarking of bambu, but integrated with other things that have nothing to do with bambu. This leaves the reader confused about what the main aim of the paper is. The benchmarking is using bambu, which is not published but also not described in the manuscript; but the manuscript is presented to be a description of a comprehensive dataset.

The results for m6A use m6Anet and include the two cell lines that have already been analysed in the m6Anet preprint: <https://www.biorxiv.org/content/10.1101/2021.09.20.461055v1?rss=1>
This makes some of the data presented somewhat redundant.

Similarly, the analysis of fusions uses JAFFA and JAFFAL. But the analyses of these cell lines with the same two tools have already been presented in another preprint: <https://www.biorxiv.org/content/10.1101/2021.04.26.441398v2>
So again, this part of the manuscript is redundant and has been already described elsewhere.

D. Appropriate use of statistics and treatment of uncertainties

Some of the benchmarking presented is not new in relation to the data itself. For instance, the comparisons of gene and transcript expression have been performed before using other tools and datasets. By themselves, they do not say anything relevant about the datasets. A similar thing happens with the analyses of m6A, fusions, and repeats.

In the case of repeats, their analyses do not support their conclusions. The authors say: "Compared to annotated transcripts, we observed a significant enrichment of repetitive elements in novel transcripts"... "suggesting that one of the major advantages of long read RNA-Seq data is an improved ability to reconstruct RNAs in repeat-rich regions." However, this is misleading. The analysis is described in methods as performed at the exon level, looking at exons not included in the annotation. However, many novel exons have been found before overlapping repeat elements using short-read sequencing (see e.g. <https://pubmed.ncbi.nlm.nih.gov/25801031/>). The right analysis to arrive to the conclusion written by the authors would be to compare the ability of identifying these repeat-overlapping exons using long and short-read sequencing.

The article lacks any robust analyses and is mainly describing suggestive findings. For instance, the analysis of repeats is not supported by any statistical analysis about the uncertainties and reliability of the mappings. Some of the repeats observed may occur multiple times in an intron. The authors do not describe any test to confirm that what they observed is not an ambiguity in the mapping process. In the case of the novel genes overlapping repeats, it is known that some repeats are expressed, so mapping long reads to the genome is bound to find them.

The level of variation in reads due to errors might be comparable to the variation between repeat copies. This begs the question of whether the detected repeats are the right ones and what is the level of uncertainty in such mappings. This is not described in the manuscript.

E. Conclusions: robustness, validity, reliability

The results do not provide any new insights that have not been already described in the prior literature for nanopore long-read sequencing data and do not clearly show the advantage or uniqueness of this data beyond these known facts.

The only apparent novelty is the large amount of data produced for each cell line. This would be more appropriate for a publication in Scientific Data or similar.

F. Suggested improvements: experiments, data for possible revision

The authors argue that their datasets are valuable for benchmarking to help the development of tools for long-read sequencing. However, the manuscript does not really show this. The authors describe results produced by methods that are not explained or published anywhere else, present analyses that they have already described in at least two preprints or that have been shown before in the literature or show results that are merely anecdotal and do not prove anything that is specific or relevant to their datasets.

However, the authors present results obtained with bambu, a method that is not fully described. And the text and figures appear as if that part of the paper is the benchmarking bambu, rather than a description of their datasets. Perhaps the authors should convert this article into the bambu article.

G. References: appropriate credit to previous work?

The manuscript fails to cite many articles that already presented the results described in the manuscript. Some of these examples are mentioned above.

H. Clarity and context: lucidity of abstract/summary, appropriateness of abstract, introduction and conclusions

The text fails to cite prior results and be clear about the methodologies used. Additionally, the actual aim of the paper is not clear. Although the abstract and introduction appear to have a clear direction, the main text is a mix of results from tools that have not been described, results that are partly already published or discussed elsewhere, or results that do not provide any novelty. As a consequence, the value of the datasets and the analyses performed remain unclear.

Reviewer #2 (Remarks to the Author):

In this manuscript, Chen & colleagues describe a set of nanopore cDNA, direct RNA and Illumina cDNA sequencing datasets, SG-NEX, on which they benchmark aspects of gene expression. The dataset is comprehensive and will undoubtedly be very useful to nanopore researchers and tool developers. Other groups have previously published similar benchmarking datasets & technological analyses (Schulz et al., 2021; Sessegolo et al., 2019; Sonesson et al., 2019; Workman et al., 2019). The strength of the SG-NEX dataset over these previous datasets is the range of different cell lines that are sequenced, although I feel that this element is underutilised in the manuscript, which instead mostly focuses on comparing the different technologies. Many of the technological findings have been previously shown by other benchmarking papers. The manuscript could be strengthened by the addition of biological insights gained from the dataset.

- The gene and transcript level annotation, and quantification of gene expression was performed using a tool called bambu, which is developed by the Goeke lab. The tool is not yet published and I could not find a description of the methodology. How does bambu perform transcript discovery and quantification? A more detailed description should be provided in the materials and methods.
- The authors indicate that transcript level counts are less well correlated with short read transcript level counts. This was previously described by Sonesson et al, benchmarked quantification methodologies for nanopore cDNA and direct RNA datasets and showed that minimap2 alignment followed by using Salmon in alignment-based mode performed best in terms of correlation of ONT transcript level counts with Illumina counts. How much does the authors finding depend upon quantification method? How does minimap2+salmon quantification perform on their data?
- In the section on linked splicing events, the authors state that 12.8% of splicing changes involve multiple events. This does not necessarily imply that these events are linked, i.e. occur in a co-dependent or mutually exclusive manner. To take the example of alternative promoters and exon skips that the authors use, there may be transcripts where alternative promoters and exon skipping occur together, but this may not occur more frequently than expected by chance given the background level of both splicing events. Have the authors used any statistical tests to verify these co-dependent or mutually exclusive splicing events?
- The authors identified novel genes in repetitive element of the human genome. It is my experience that the error rate of nanopore sequencing is high enough that it is often difficult to unambiguously assign reads to genes when they are in repetitive regions which contain whole gene duplication events (e.g. when gene copies have diverged at a lower rate than nanopore error rate). Is the rate of multimapping higher for the author's novel genes than for annotated genes? How confident can the authors be that these are transcribed genes and not pseudogene copies with mismatched reads?
- The section on fusion transcripts is to me the most interesting and novel element of the study. 25.5% of the fusion transcripts that the authors discover are novel. Can the authors validate any of these fusion transcripts by other means? Are any of the fusion transcripts found in known oncogenes?

Schulz L, Torres-Diz M, Cortés-López M, Hayer KE, Asnani M, Tasian SK, Barash Y, Sotillo E, Zarnack K, König J, Thomas-Tikhonenko A. 2021. Direct long-read RNA sequencing identifies a subset of questionable exons likely arising from reverse

transcription artifacts. *Genome Biol* 22:190.

Sessegolo C, Cruaud C, Da Silva C, Cologne A, Dubarry M, Derrien T, Lacroix V, Aury J-M. 2019. Transcriptome profiling of mouse samples using nanopore sequencing of cDNA and RNA molecules. *Sci Rep* 9:14908.

Soneson C, Yao Y, Bratus-Neuenschwander A, Patrignani A, Robinson MD, Hussain S. 2019. A comprehensive examination of Nanopore native RNA sequencing for characterization of complex transcriptomes. *Nat Commun* 10:3359.

Workman RE, Tang AD, Tang PS, Jain M, Tyson JR, Razaghi R, Zuzarte PC, Gilpatrick T, Payne A, Quick J, Others. 2019. Nanopore native RNA sequencing of a human poly (A) transcriptome. *Nat Methods* 16:1297–1305.

Reviewer #3 (Remarks to the Author):

The manuscript describes a comprehensive long-read dataset for several cancer cell lines, generated with several RNA-seq protocols and matching short-read RNA-seq. The Authors claim ONT technology can replace short-read sequencing for RNA-seq analyses; however, I am not convinced by the Authors' arguments. I believe this paper will benefit from an unbiased assessment of ONT shortcomings and a clear demonstration of its advantages.

1. The cost is crucial for a fair comparison between technologies. The best comparisons are made for sequencing done at the same price point. If this is impossible (since ONT requires a higher sequencing cost), then the price difference should be clearly stated, and it needs to be discussed whether the benefits of the technology outweigh the cost increase

2. Both short and long RNA-seq were done in multiple replicates; however, the replicates are not compared. Comparing the gene expression between replicates for long and short reads provides a good metric for the reproducibility of the technologies.

3. Looking at the Supp. Table 1, it seems that ONT runs have widely varying numbers of sequenced reads. This is a known disadvantage of ONT and should be discussed in the text.

Were all the replicates/runs combined for the expression analyses?

Also, it makes it tricky for a fair estimation of the price point. For instance, among the A549 libraries, one run contains 12M reads, which is 8 times larger than the average run (~1.5M reads). It looks like sequencing depths for short RNA-seq are missing from Supp. Table 1.

4. One of the major questions is whether the sequenced reads represent the full-length transcripts, as this is one of the central promises of long-read technologies. The answer to this question is contained in Fig.2f, and it's disappointing: the median transcript coverage is only 40-60%, i.e., only half of the reads have coverage > 40-60%. Additionally, only a tiny % of reads (<10%) represent full-length transcripts. This severe shortcoming is not discussed in the text. Several questions arise here, e.g.:

How does this result compare with the previous ONT-based RNA-seq studies?

For PCR and cDNA protocols, is this due to incomplete RT or incomplete sequencing of the reads (this can probably be answered by checking the RT/PCR adapter sequences at the ends of the reads)?

What % of reads are cut short at 3', 5', and both 3'/5'?

5. The units are not given for expression scatter plots Fig.3a-f, presumably it's log₂(TPM)? While the scatterplots provide a general idea about correlations, it's also essential to quantify the change in expression. This can be done, for instance, with a histogram for relative deviation. It seems that there will be a large number of genes with significantly different expressions in long vs. short methods, and so the gene expression estimates between ONT and short reads cannot be called "highly comparable."

Also, in addition to Spearman R, the Pearson of log₂(TPM) would be informative.

5. Fig.2a: tSNE does not quantitatively measure the distance between different libraries. Please show PCA and hierarchical clustering.

6. Fig.2d compares read lengths among different protocols. While it's clear that the read length is significantly shorter in cDNA and PCR protocols compared to direct-RNA, it would be great to make this clear by showing the reduction of the read length for matched samples.

7. The short-read gene and transcript quantification is performed with Salmon. The most widely used and trusted quantification tool is RSEM, and getting a second opinion on short-read quantification is needed to increase confidence in the comparative analyses.

8. One of the most important promises of long-read RNA-seq is that "Long read data improves over short-read data for transcript abundance estimation." However, no direct evidence to support this claim is given in this section. It may not be easy to prove that long reads provide not just different but better transcript quantification than short reads. One suggestion is to calculate the correlation between replicates and show that it's higher for long reads than short reads.

9. The only circumstantial for the above claim evidence is: "a similar level of variation is seen when 150bp paired-end short read RNA-Seq data is compared to the identical data trimmed to 75bp single-end reads (Supplementary Figure 3b) indicating that shorter read length is the main reason for the observed variation for transcript level estimates (Supplementary Figure 3c,d)."

With only two points (75b and 2x150b), it's hard to judge the dependence of transcript quantification accuracy on the read length. This investigation needs to be extended to a series of trimmed lengths, e.g., 2x125, 2x100, 2x75, 2x50.

10. The justification for this claim is also weak: "of the major advantages of long read RNA-Seq data is an improved ability to reconstruct RNAs in repeat-rich regions." The transcript/gene annotations are based on long, high-quality cDNA sequences, and thus (unlike short reads) should not be affected by repeatedness of the sequence. On the other hand, the alternative explanation could be that the high error rate of ONT reads results in erroneous mapping to regions with high repeat content.

11. Possibly related to the point above, in Fig.Supp.1b, we see a surprisingly large expression of pseudogenes: ~10% of the reads (just among the top 1000 expressed genes) map to pseudogenes. Then it's stated: "pseudogenes which have higher read counts with long read data, possibly due to improved resolution at repetitive regions or higher error rates of long reads." It is vital to figure out whether this is an issue with ONT reads or the high expression of pseudogenes is a true observation.

10. Another interesting observation is hidden in Fig.4h: ~15% of long reads cannot be assigned to a transcript, compared to <3% for short reads. Again, do the Authors believe that this is a true observation, or is this also caused by the high error rate of ONT reads?

11. The number of isoform switching events (Fig 5) is only a few hundred. Across all samples, only 2,202 novel transcripts are identified. Is this number limited by the methodology used, where only full-length reads are used, which constitute a small proportion of all reads? How does this number compare to the novel splice junctions and "exons" identified in short-read sequencing?

12. I believe several claims in the Discussion session appear to be not well justified:
"Our data suggests that long reads improve most tasks in transcriptome profiling..."
There is no evidence that long reads improve gene expression analysis.
Fusion detection was not done with short reads so that no comparison could be made.
Isoform switching analysis was not compared with short-read analysis.

13. "...identification of novel isoforms associated with well-known fusion genes..."
Only one example of such isoform was discovered, with no further validation.
How many of such isoforms are discovered? An orthogonal experimental validation would be required to make it a strong claim here.

14. "Our data suggests that one of the main advantages of long reads is an improved resolution to identify highly repetitive transcripts"
As discussed above, the data suggests that ONT reads often map to highly repetitive regions. However, such alignments can be unreliable because of their high error rate. This claim requires orthogonal experimental validation.

Although we cannot publish your paper, it may be appropriate for another journal in the Nature Portfolio. If you wish to explore the journals and transfer your manuscript please use our manuscript transfer portal. If you transfer to Nature journals or the Communications journals, you will not have to re-supply manuscript metadata and files. This link can only be used once and remains active until used.

All Nature Portfolio journals are editorially independent, and the decision on your manuscript will be taken by their editors. For more information, please see our [manuscript transfer FAQ](http://www.nature.com/authors/author_resources/transfer_manuscripts.html?WT.mc_id=EMI_NPG_1511_AUTHORTRANSF&WT.ec_id=AUTHOR) page.

Note that any decision to opt in to In Review at the original journal is not sent to the receiving journal on transfer. You can opt in to [In Review](https://www.nature.com/nature-research/for-authors/in-review) at receiving journals that support this service by choosing to modify your manuscript on transfer. In Review is available for primary research manuscript types only.

** For Nature Research Group general information and news for authors, see <http://npg.nature.com/authors>.

Version 2:

Decision Letter:

29th Jan 2024

Dear Dr Goeke,

I sincerely apologize for the long review process. Your revised Resource, "A systematic benchmark of Nanopore long read RNA sequencing for transcript level analysis in human cell lines", has now been seen by 3 reviewers. As you will see from their comments below, although the reviewers find your work of considerable potential interest, they have raised a number of concerns. We are interested in the possibility of publishing your paper in Nature Methods, but would like to consider your

response to these concerns before we reach a final decision on publication.

We therefore invite you to revise your manuscript to address these concerns.

Link Redacted

We hope to receive your revised paper within 8 weeks. If you cannot send it within this time, please let us know. In this event, we will still be happy to reconsider your paper at a later date so long as nothing similar has been accepted for publication at Nature Methods or published elsewhere.

OPEN SCIENCE REQUIREMENTS

REPORTING SUMMARY AND EDITORIAL POLICY CHECKLISTS

DATA AVAILABILITY

All novel DNA and RNA sequencing data, protein sequences, genetic polymorphisms, linked genotype and phenotype data, gene expression data, macromolecular structures, and proteomics data must be deposited in a publicly accessible database, and accession codes and associated hyperlinks must be provided in the "Data Availability" section.

To further increase transparency, we encourage you to provide, in tabular form, the data underlying the graphical representations used in your figures. This is in addition to our data-deposition policy for specific types of experiments and large datasets. For readers, the source data will be made accessible directly from the figure legend. Spreadsheets can be submitted

in .xls, .xlsx or .csv formats. Only one (1) file per figure is permitted: thus if there is a multi-paneled figure the source data for each panel should be clearly labeled in the csv/Excel file; alternately the data for a figure can be included in multiple, clearly labeled sheets in an Excel file. File sizes of up to 30 MB are permitted. When submitting source data files with your manuscript please select the Source Data file type and use the Title field in the File Description tab to indicate which figure the source data pertains to.

CODE AVAILABILITY

Please include a "Code Availability" subsection in the Online Methods which details how your custom code is made available. Only in rare cases (where code is not central to the main conclusions of the paper) is the statement "available upon request" allowed (and reasons should be specified).

ORCID

Sincerely,
Lei

Lei Tang, Ph.D.
Senior Editor
Nature Methods

Reviewers' Comments:

Reviewer #1:

Remarks to the Author:

The authors have put together in this manuscript a collection of analyses to illustrate what one might do with their SG-NEx comprehensive datasets. Some of the analyses provide an interesting benchmark of the technologies. Others are not compelling and would require additional work. The analyses are also not clearly described, and often they lack enough details or these details are entirely missing.

While the authors motivated this manuscript as a showcase their SG-NEx dataset, the manuscript ends up being something quite different. The authors initially describe SG-NEx project as a systematic resource and benchmark data set, but the benchmarking aspect is only illustrated for gene and transcript expression. After the initial part, the authors motivation to compare the different technologies and discuss how the SG-NEx facilitates novel analyses seems to fade. The authors recurrently fall to the temptation to make biological claims, often without enough support, while at the same time failing to properly highlight how the unique depth and breadth of their data set specifically made possible those analyses. In this sense, they contradict their own intention, and fail to accomplish a convincing manuscript.

The manuscript needs overall proofreading. There is a general discordance between present and past tenses, e.g. "We searched We identify.... "... "We then applied... we find...", etc. The manuscript also present frequent confusion between genes and transcripts, with often a discordance between the text using one and the figures using the other.

Overall, the details of the analyses performed are poorly described or incomplete. There are multiple discrepancies and missing information, and the article does not present a good balance between the different applications. More importantly, the article does not emphasise enough the advantages or uniqueness of the SG-NEx dataset.

More detailed questions and suggestions are provided below:

--- description of the datasets ---

The description of the datasets is confusing. The authors describe using 7 cell lines, but then mention 8 additional cell lines. However, these additional 8 cell lines do not seem to be described further in the manuscript. What are the platforms used for these additional cell lines? This is not clearly explained.

The authors also mention 8 cancer types, but they actually sequenced cell lines, and the data is shown for 7 cell lines. They should probably not refer to them as cancer types and be more clear about the data presented in the manuscript versus the data not included.

--- gene and transcript expression ---

These comparisons are interesting and highlight some of the differences between the technologies. However, the analyses quickly become quite convoluted.

The statement “highly-expressed genes were over-represented in the PCR-cDNA” It is a bit confusing, given that the high or low expression is defined by the same sequencing reads. In this case, the authors probably mean that genes at the top of the ranked expression estimates account for a larger proportion of the overall abundance.

The authors use “gene expression” without explaining how this is calculated. For all technologies expression is estimated in the manuscript at the transcript level. However, gene expression estimates vary quite a lot depending on the technology and methodology (see e.g. <https://f1000research.com/articles/4-1521>). The authors should clearly define their method to calculate gene expression. Is this the most commonly used in the literature?

The authors show that samples cluster together independently of the technology (Figure 3f) and use this fact to say that “Nanopore RNA-Seq data can be integrated with existing short read RNA-Seq data for gene expression analysis”. This is a big statement that would need further support. They used Spearman correlation, which will test ranks rather than values. Moreover, gene expression estimates just by themselves are rare, they most often involve differential gene expression analysis. It is not shown whether this would also be equivalent.

The authors illustrate the discrepancies in transcript expression estimates in Figure 4b. However, this figure is not easy to interpret, and the caption does not provide sufficient information to understand what is shown in the plot.

The authors then focus on the major isoform, which they define as the “most active isoform of each gene”. The authors should use consistent language and perhaps call it the “most abundant.....”.

Also, it is not clarified whether this is defined per cell line or not. The definition of “major isoform” should be clearly stated.

Here is where the text gets more difficult to follow. For instance, “Isoforms that were identified to be major isoforms only in the long read RNA-Seq data (major-long read) showed significantly higher correlation compared to isoforms that were identified only by short read data as major (major-short read)...”. It is not clear what correlation it refers to. Also, the authors should specify how many cases this refers to. Is this a major effect? Or just marginal?

The rest of the paragraph remains confusing. For instance “A comparison of these isoforms showed that short read RNA-Seq identified significantly more often isoforms as major that correspond to truncated versions of the long read major isoforms”. The authors refer to Figure 4d, but these are alternative splicing events rather than truncations. One expects that most of the differences between isoforms can be described as alternative splicing events. Also, it is not clear how the length difference (somehow implicit in the word “truncation” is described in this plot).

The last sentence says: “Furthermore, short read-specific major isoforms were frequently estimated to be lowly Expressed (1 <CPM)”. This should probably be “CPM < 1” or “expression < 1 CPM” depending whether CPM is used to represent expression or whether CPM is used as units of measure. Also, this probably means that they must be at the level of noise? Can these transcripts and genes considered “expressed” at all?

The authors here miss an opportunity to provide more useful insights into the differences between technologies. Are these results just true for nanopore long-read sequencing? Which protocol? Do all long-read technologies give the same result?

The authors indicate “Compared to short read data, ... major isoforms identified by long read RNA-Seq show significantly higher full-length read support”. This result is not easy to see in Supp Fig 5m. Also, it is not clearly defined what the authors mean by “full-length read support”.

To study the differences between long and short read-based expression estimation, the simulated short reads from long reads through fragmentation. However, they simulated single-end reads, whereas their short-read data is paired-end reads. It is not possible to know whether the results they obtained are due to this difference or not. A proper comparison should use the same type of short reads for both the experimental and the simulated data.

The analysis of the simulation from fragmented reads seem to be only relevant in the case of major isoforms defined only by short-reads. The authors indicate that the correlation is impacted and the expression estimation is increased. However, these were described above to have low expression (less than 1 CPM) in the short-read data, and moreover, they are not called as major by long reads, so they may have also a lower long-read support in general. Given the low counts expected in these transcripts, the results are not very surprising. Also, it is not clear how relevant this result is. The authors do not explain whether this affects many transcripts per cell line. So one is left with the impression that the authors are trying to make a big story about something that is purely anecdotal and irrelevant.

For the analysis of transcripts, the authors simple quote results related to “long reads”. But it is not clear whether all long reads were pooled together, or it is meant that the results were similar across technologies. Are they describing only results from nanopore reads? If so, which protocol? This is not explained at all.

This fact defeats the purpose and the motivation of their dataset and manuscript. The authors motivate this manuscript on the advantages of having multiple technologies. But all long-read technologies seem to be described as one single type.

The authors described “cell type-specific major isoform switching event which were supported by full length reads”. How are these switches calculated? Are they defined in terms of a statistical test that evaluates the significance of the switch? Are these switches consistent across long-read technologies? Here there is also a missed opportunity to highlight the advantage of SG-NEEx in providing multiple technologies.

--- Overlap with repeats ---

Overall, this part is intriguing and worth exploring further, but in its current state, it is only partially supported and not very compelling.

The authors should provide information on the samples that were used for this analysis. This is not described in the text.

The authors should describe what is the read and technology support for the novel transcripts: number of reads? Expression values? Are they found across replicates? Are they found between different technologies? Without these details, it is not possible to know whether these novel transcripts are reliable or accumulated noise as a result of using many samples together.

What is the expected validation rate of those novel transcripts? Based on the various properties mentioned above like read support, reproducibility, etc... the authors should be able to estimate the expected proportion of those transcripts that are likely to be real. It is possible that novel transcripts appear as a consequence of the fact that so many reads from various technologies are being analysed, but their expression level is comparable to experimental noise. Alternatively, could they validate some of them?

What is the minimum number of reads, technologies, and replicates that are generally needed to identify a transcript, known and novel?

About these novel transcripts, the authors indicate that they have “lower sequence complexity” compared to reference transcripts. However, “sequence complexity” is not defined anywhere and the result is not shown. The authors should be clearer about what they mean, the description of the analysis and the results, and show the data.

The authors use “unannotated genes”. However, this is incorrect. They should use instead “unannotated loci”, “unannotated genomic loci” or “novel genes”.

The authors describe “Compared to annotated transcripts, we observed a significant enrichment of repetitive elements such as LTRs, LINES, and SINES”. However, they do not explain anywhere what repeat annotations were used. Is this the RepeatMasker annotation from UCSC or Ensembl? Did it include the simple repeats? Or only complex repeats? The authors should clarify.

How the repeat analysis was performed is not explained. The Methods section only indicates keeping only cases with 80% overlap, however, Figure 6c shows a wide range of overlaps. The authors should describe in more detail how this analysis was performed.

The authors should clearly explain how the overlap was calculated. Was it between the repeat annotation and the exonic regions of the transcript only? Or did they include the intronic regions as well?

If the authors included the intronic regions, it might not be surprising that there is a high overlap with repeats. Introns are enriched in repetitive elements, both simple repeats and complex repeats. The authors should describe what are the repeat types that the novel transcripts in Figure 6c overlap with. This is only indicated in Figure 6d for a subset of the transcripts. This plot also shows an overlap with “others”, but the authors do not

explain what they are, i.e. simple repeats?

The results are overall not very clearly described. The authors indicate “Compared to annotated transcripts, we observed a significant enrichment of repetitive elements such as LTRs, LINES, and SINEs in novel transcripts (two-sided t-test, $p < 0.001$, Figure 6c,d).” The test probably refers to the distributions of Figure 6c, but these include all novel transcripts, not just those from Figure 6d with 80% overlap. Is that correct? So this statement is misleading, as the overlaps in Figure 6c do not seem to be just 80% and they probably include many other repeats not shown.

Furthermore, there seem to be lots of annotated transcripts with >80% overlap. Are the numbers comparable?

They additionally say: “The enrichment of these highly repetitive elements was observed for both novel transcripts from annotated genes and for novel gene candidates, ...” However, they only show one test p-value. Did they test both distributions? The authors should show the p-value for both distributions to be able to make this claim.

The authors also say “... and was confirmed when only uniquely mapped reads are used for transcript discovery (Supplementary Figure 7f-i).” However, this figure does not include the statistical test. Is the difference significant? The authors should include a p-value.

Regarding multimapping reads, the authors should indicate how many of their detected novel transcripts in unannotated loci are due to multimapping reads. An alternative possible explanation of their results is that most, if not all, of the novel transcripts in novel loci, especially those shown in Figure 6d, are actually due to multimappers caused by the ambiguity of the repeats.

In fact, according to their Figure 6d, novel transcripts in annotated genes mostly overlap with “others”, which might be low-complexity repeats (which is expected), and LTRs, which is also expected, as LTRs often overlap with promoters and regulatory regions. All the LINE, SINE, and DNA elements might be intronic. This point should be clarified.

--- fusion analysis ---

No details are given about which datasets were used to calculate the fusions. All long read datasets together? The manuscript is motivated on the advantages of having multiple technologies, but this crucial point is missed here again.

Not much information is provided in terms of reproducibility of the fusion discovery across replicates or technologies, what's the read support? What are their expression values? These are questions where a dataset like SG-NEx could be useful for.

The reader is left with no information about whether there is anything in the SG-NEx data that is specially unique or useful for that. For instance, is it the number of reads? The combination of sequencing technologies? Etc. The authors should provide more details on how the SG-NEx data is particularly useful or unique for this type of analysis.

The authors mention “we additionally find full length read support for most of the 5' and 3' genes, suggesting that fusion genes are present together with their unfused wildtype gene.” This probably refers to the fact that fusions can be heterozygous or homozygous? The authors describe this as a remarkable result. Is that novel result? Or is it actually common? The authors should clarify.

The authors conclude this section with “... illustrating how long read RNA-Seq enables the discovery and quantification of full-length fusion transcripts in cancer.” The authors should clarify that these are not cancer samples, but cell lines. The material and sequencing efficiency might differ drastically.

Additionally, this statement might not be true in general. The authors may have identified those full-length fusion transcripts only because they sequenced so many replicates and used so many technologies. So the statement is misleading.

Fusion discovery might be possible under certain conditions regarding sequencing depth and technologies. This is a question where the depth and breadth of SG-NEx data could provide useful insights, but this opportunity seems to be missed again.

--- m6A analysis ---

In the abstract, the authors mention various analysis indicating “..., quantify full-length fusion isoforms and simultaneously profile m6A RNA modifications when RNA is...”. This is misleading as the authors do not combine in the manuscript the m6A analysis with any of the other analyses.

The m6A analysis is carried out by aligning the read signals to the “reference nucleotide sequences”. The authors should be more specific about what they did. Are these genomic or transcriptomic sequences? The Methods section is not clear about this either.

The authors describe that the m6A sites recovered follow an expected DRACH motif. But this is misleading. m6Anet only detects m6A on DRACH motifs. While they show that non-modified sites have a different logo, these are all DRACH motifs as well. They should rephrase or explain it better.

The authors also mention an enrichment in the 3' UTR. If the signals were aligned to the genome, the 3'UTR enrichment might be due to biases. 3' UTRs are generally longer, and read signals are easier to align to long sequences. In contrast, exons tend to be shorter than 150nt, and signals might be harder to align. The authors should control for length and for the efficiency of the signal alignments.

Figure 8b. The figure caption indicates "Heatmap showing the two-sided Fisher's test odd ratio (enrichment score) across sample replicates from the seven cell lines.". However, it is not explained here or in the text what this test is about and what is meant for. The analysis described by this figure is not clear.

Figure 8c: the figure says genes, but the text mentions transcripts "While these transcripts also show cell type-specific modifications, the majority of cell type-specific m6A sites originate from cell type-specific transcripts (Figure 8c)." Cell specific genes or transcripts are two very different things. The authors should use consistently genes or transcripts in text and corresponding figures.

The authors have included m6ACE-seq data in the manuscript, but this data (whether it is a new experiment or extracted from the literature) is not described. The authors should describe where this data comes from.

The authors claim that the SG-NEx data is useful as a benchmarking dataset, the authors should then compare the m6ACE-seq data with the nanopore-based m6A predictions to illustrate this aspect.

Reviewer #2:

None

Reviewer #3:

Remarks to the Author:

The authors addressed my questions and concerns in a thorough and satisfactory manner.

Reviewer #4:

Remarks to the Author:

The updated manuscript by Chen et al. offers a thorough comparison of long-read RNA sequencing techniques utilizing various cell lines and replicates, enriched by a diverse collection of spike-ins. The state the superiority of long-read sequencing methods over Illumina for transcript detection and quantification. The data compilation is significant, and its availability via the Nextflow pipeline is beneficial. Although certain conclusions of the benchmarking may not be novel, some findings contradict previous works. Nonetheless, certain aspects of the analysis warrant further scrutiny. My detailed comments are as follows:

1 The claim regarding the dataset's uniqueness for benchmarking purposes should be tempered. The dataset includes analyses and cell types potentially overlooked in other studies, yet the diversity of the data is not unprecedented. For instance, LRGASP also incorporates multiple library preparation methods, sequencing platforms, cell lines, spike-ins, and it enables benchmarking for transcript models, quantification, RNA modifications, and fusion transcript detection. This acknowledgment does not diminish the value of the presented data collection, but the claim of uniqueness should be qualified. Specifically:

- The statement "The benchmark with the highest number of protocols" is inaccurate. LRGASP employed seven different protocols. This is wrong in Figure R.R1. (Supplementary Figure 2?) and should be corrected.
- The description "The most comprehensive resource with available ground-truth" should be rephrased in the context of spike-ins. LRGASP utilized 270 manually curated transcripts by GENCODE annotators as ground truth and also included orthogonal data sources where ground truth was established.

2. The manuscript should clarify the quality control steps in the nanoseq pipeline. Specifically, it should detail the measures taken to prevent artifacts like intrapriming and RT-switching events, which are relevant to the claim of enhanced detection of repeat-containing transcripts, predominantly found in intergenic regions. In my experience, such intergenic detections often diminish with stringent QC filters, so it is crucial to disclose which QC measures were employed to ensure the authenticity of these transcripts.

3.To address the concerns by reviewer one on the possible role of mapping errors for novel transcript detections, the authors evaluate the differences in novel transcript calls when using mappers that allowed multiple or single mappings. They found that using reads that map only uniquely reduces the number of novel transcripts. As this reduces the number of novel transcripts, they claim this is controlling possible false mappings. However, to be sure this is specific for the novel transcripts, they should also evaluate the overall transcript detection when they discard an X number of reads, regardless of their mapping ambiguity.

4.The manuscript asserts that the benchmarking conclusions, particularly regarding library preparation methods, are not algorithm-dependent. However, this claim conflicts with some LRGASP findings. For instance, Figure R1.4.a shows significant differences in LongSIRV analysis outcomes between methods. This should be discussed. Also, for figure R1.3 a, Supplementary Figure 4?) why Long SRIVs have not been evaluated in terms of Spearman Correlation and R2? Moreover, R1.3 b) is hard to read. Possible other choice of colors or splitting in different figures will make it more readable.

5. Certain terminology used in the paper, such as "full-length read," requires explicit definition. While "full-length read" has a specific connotation in PacBio technology, it seems the authors refer to "full-splice-junction", equivalent "full-splice-match" in the SQANTI terminology. A clear definition should be included to avoid ambiguity.

6. Figure 3 illustrates the consistency of gene expression quantification, highlighting the superior performance of long-read RNA-seq methods compared to Illumina, especially with spike-ins. However, I feel this analysis needs further elaboration. Spike-ins are relatively highly expressed genes in comparison to a normal sample and correlation with expected values might not be as good at the lower expression range (results with lncRNA analysis reveals this), which may represent a large fraction of the transcriptome. How are correlation values, for example, at CPM < 2.5? Possibly a better breakdown of correlation by expression value with indication of the transcriptome fraction it encompasses, would be more informative. Moreover, the lack of correlation observed for the E0 SIRV dataset across all methods demands an explanation.

7. Figure 4 demonstrates the accuracy of long-read methods in isoform detection. Yet, it is unclear how isoforms were matched across different technologies, and whether newly detected isoforms were considered. If the analysis is limited to annotated transcripts, this limitation should be clearly stated, as it is not evident in the methods section.

8. Figure 5 presents alternative splicing patterns uncovered by long-read RNA-seq. It should be specified whether these patterns are consistent across various methods and algorithms or if they stem from a single analysis approach. The LRGASP identified significant variability in isoform identification contingent on the analysis tool, which could influence the results. Furthermore, the data supporting the isoforms presented in panels a and b should be clarified—are these isoforms detected by a single read in at least one sample? This could represent transcriptional noise. Although the observation is still interesting, this should be clarified.

Version 3:

Decision Letter:

3rd May 2024

Dear Dr Goeke,

Please accept my sincere apologies for the length of time that it has taken to get the decision on your manuscript entitled "A systematic benchmark of Nanopore long read RNA sequencing for transcript level analysis in human cell lines". As you will see, the manuscript has now been seen by 2 reviewers, whose comments are attached. While they find your work of potential interest, they remain concerned about the analysis quality. We have had an editorial discussion about this revised manuscript and the review report. We think these technical concerns are significant to us and unfortunately preclude the publication of the work in Nature Methods, at least in its present form.

We don't think the concern about novelty should preclude the publication, so we will overrule these concerns. Should further experimental data allow you to address the technical criticisms we would be willing to look at a revised manuscript again.

If you are interested in revising this manuscript for submission to Nature Methods again, please email me about your appeal before making any revisions. Otherwise, we hope that you find the reviewers' comments helpful when preparing your paper for submission elsewhere.

Again, I'm very sorry about the long review process.

Sincerely,
Lei

Lei Tang, Ph.D.
Senior Editor
Nature Methods

Reviewers' Comments:

Reviewer #1:
Remarks to the Author:

While the data is useful and interesting, the analyses performed are still not novel and significant enough and of sufficient quality for what I would consider to be an outstanding and exciting manuscript worth publishing in Nature Methods. The results are anecdotal or repeat what has already been published elsewhere. The manuscript reads as a collection of rather shallow results without any clear biological relevance. Additionally, the analyses still do not exploit the depth and breadth of the datasets fully. It is not clear how such analyses could be acceptable for publication in Nature Methods.

The datasets could be published and showcased in a more data-focused journal, such as Scientific Data. Some of the analyses, those more promising regarding the preliminary results, could be developed more thoroughly and with the appropriate controls as separate research article.

The introduction and sections (1)-(4) provide now a clearer description of the datasets and an interesting comparison between the different sequencing protocols.

From section (5) things become less clear. The authors report a difference of ~7000 genes per cell line, where the major isoform described by long or short reads is different. Some of the results related to this are a bit confusing, eg: "For genes where long and short read RNA-Seq identified different major isoforms, long read-specific major isoforms showed significantly higher correlation of expression estimates between both sequencing technologies". This result and its interpretation is not very clear.

The authors also indicate "major isoforms identified by short read RNA-Seq significantly more often used internal exons from the long read major isoform as first or last exons, typically corresponding to shorter versions...". This should be clarified. It could be an issue with the mapping and reconstruction algorithm. In the Methods section, the authors indicate that they use STAR to map illumina reads to the genome. However, they only mention Salmon and RSEM in relation to transcript quantification, and they use a direct mapping against the annotation. They present an intriguing result but it may be related to a technical issue of the methodology rather than the data. There might be a methodology that could be used that would provide a better read alignment and transcript reconstruction strategy with the Illumina reads such that this result is not true anymore. This analysis is not of sufficient quality – it would require more in-depth analysis to more clearly show what is happening.

For section (6), the text does not describe the simulated paired-end read data that the authors claim to have added in the response to reviews. This is rather unusual.

It is also not very clear the relevance of the results in this section: "read fragmentation increases transcript expression estimates for isoforms that are only identified in Illumina short read RNA-Seq data as major isoforms". It is not clear how this result is generalisable. This fails to grasp the opportunity of this rich dataset to provide a useful guide to the community.

Their final conclusion for this section could be all based on an artifact of their methodology to map/reconstruct/analyse short reads: "These observations suggest that the difference in transcript expression between short and long read RNA-Seq data may partially be the consequence of inflated expression estimates in short read RNA-Seq due to fragmentation and reduced read length." Additionally, it remains unclear from the text how relevant and how widespread this issue is, whether it occurs in genes functionally relevant or genes of unknown function, genes that are key for cell identity, or house-keeping genes. The authors do not provide enough controls to support such a general claim.

Section (7) describes the analysis of alternative splicing events and isoform switches. Comparisons related to alternative splicing between long and short reads have been done before (eg <https://pubmed.ncbi.nlm.nih.gov/35012468/>), long-range coupling of transcript variations has also been described before (eg <https://pubmed.ncbi.nlm.nih.gov/29598823/>, <https://pubmed.ncbi.nlm.nih.gov/25985263/>), comparison between technologies has also been performed before, albeit at a lower scale (<https://pubmed.ncbi.nlm.nih.gov/35301264/>).

It is strange that the authors do not explore more thoroughly the differences in alternative splicing and isoform switches between nanopore and pacbio, or between nanopore methods.

The result of this section is underwhelming, they simply report the number of isoform switches detected across the cell lines. It is not clear the relevance of this result or whether it is simply a result of the method used, in this case DEXSeq. It could be that the results would be completely different using other analysis methods.

Analysis of repeats

The authors explain that the NDR corresponds to the False Discovery Rate. However, they use a value of 0.1, which is a 10% false discovery rate. This is by no means a strict cutoff. From all the transcripts detected, 10% are expected to be false.

The authors have now clarified that they used the RepeatMasker annotations downloaded from the UCSC browser. However, this analysis remains problematic and shallow. First, the authors included the low-complexity repeats, whose implications are completely different from other repeat types. There seems to be also many overlaps identified with RNA pseudogene families. The implications of this is also quite different, but it is not investigated. The authors are misleading in their explanations: "Compared to annotated transcripts, we observed a significant enrichment of repetitive elements in exons from novel transcripts (two-sided t-test, $p < 0.001$, Figure 6c), among others from LTRs, LINES, and SINEs (Figure 6d, Supplementary Text Figure 17, see methods). The enrichment of these highly repetitive elements was observed for both novel transcripts from annotated genes and for novel gene candidates, and was confirmed when only uniquely mapped reads are used for transcript discovery".

It is important to note that "These highly repetitive elements" also include Low complexity repeats (no relation to mobile elements or repeat families whatsoever) and RNA pseudogenes (also not a clear relation with repeat families). The text is misleading as "These" may be interpreted as those mentioned explicitly: LTRs, LINES, SINEs. There is no clarity about what repeats are exactly overlapping the predicted and known transcripts.

In this regard, it is important to note that the bulk of the predicted novel transcripts have < 80% overlap (Fig 6c), so the Fig 6d may not be representative of this result. Here the authors chain a series of results and build a narrative that does not seem well-supported upon scrutiny.

The authors tested the distribution of the % overlap (what % of the exons overlap a repeat) with a t-test (Fig. 6d). This test might not be appropriate given that the distributions present such skewness large number of outliers. Also, this does not test whether repeats are significantly associated with the predicted novel transcripts or not.

In Fig. 6d, there is no clear enrichment in the association of repeats with predicted novel transcripts. Relative to the number of novel or annotated shown, the proportions might be the same. The rate of overlap of known or novel transcripts with repeats could be the same.

Since the predicted novel transcripts are shorter (they can be almost half the length), just by chance the fraction of their length covered by repeats, even if they overlap at the same rate, could be higher.

An additional problem with this analysis, is that all the plots show a large number of outliers, especially from the annotated transcripts. It could even be that there are many more annotated transcripts with a large overlap at a given cell line.

Figure 6d is also misleading, since the plots the information for all cell lines, and more annotated transcripts with repeat overlaps seem to be present across most of the cell lines, unlike the predicted novel transcripts, which often seem cell-type specific. This requires a more in-depth analysis, especially in relation to the association with repeats besides the overlap. It is well known that LTRs overlap frequently with promoter/first-exon regions, and that ALUs and other SINE, as well as LINE, elements overlap with alternative exons in annotated transcripts. The measured % overlap differences might be due to the data but not related to actual biology.

The authors tried to address the potential confounding factor due to multimapping reads. They indicate that they used two strategies, 1) using only the primary alignment, or 2) only keeping uniquely aligned reads without any secondary alignment. The authors indicate: "The results indicate that the majority of transcripts can be found with both alignment strategies (91.7%, Supplementary Figure 2a), suggesting that novel transcripts are not due to multiple alignments from reads due to repetitive genomic sequences."

However, this analysis fails to deliver a convincing explanation. It is not clear whether "found" means that they would have sufficient expression and be considered reliable using just the unique reads. Also, if ~90% of transcripts are found by both strategies, wouldn't this mean that most of them may emerge from multimapping reads? Here the relative proportion of reads from one or the other type in each transcript, as well as their lengths, are important elements to provide reliable support to this claim.

In their response, the authors indicate Supp. Fig. 2, but this seems to be about something else.

The authors in the manuscript to Supp. Fig. 7 regarding the overlap with repeats "...was confirmed when only uniquely mapped reads are used for transcript discovery". From this figure or their text, it is not clear whether the predicted novel transcripts were recalculated using only uniquely mapping reads, or they just plotted the results for the subset of transcripts that had uniquely mapped reads assigned to them.

The claim that repeats are significantly associated with novel transcripts requires more thorough scrutiny. Also, it is not clear how that represents an important biological finding by itself. The work remains unclear and of insufficient quality to feel confident that it deserves publication in a journal like Nature Methods.

Fusion analysis

That authors explain that "The SG-NEx data was made publicly available as a community resource very early on, and it was already used for benchmarking of 3 different long read fusion detection methods (FLAIRfusion, JAFFAL, FusionSeekerLR), making it the most widely used resource for long read fusion detection". Given that this is the case, it is the authors' burden to provide an analysis that has not already been published elsewhere. It is not clear otherwise, what is the purpose of this section.

The authors indicate that fusion transcripts were identified using individual samples (not combined samples). But this is misleading, as they probably used all the replicates and sequencing runs for each condition. So again, the number of reads used in the detection of fusions is relevant and should be characterised.

The authors did not attempt to identify the sequencing depth required to identify fusion transcripts. However, they concede that this is "an interesting question that could possibly be answered using the SG-NEx data resource." It is again not clear why this question, which is important for the problem of fusion detection and can be answered with this dataset and tools produced by the authors, is not addressed in the manuscript that the authors are putting forward to illustrate the advantage of the very same dataset for the study of fusions.

M6A modification analysis

The prediction of m6A from nanopore sequencing and comparison with other methods has already been published before, eg <https://pubmed.ncbi.nlm.nih.gov/37019930/>. The landscape of m6A across different sample types has already been published before, eg across cell types <https://pubmed.ncbi.nlm.nih.gov/22575960/> or across tissues, including tissue specificity: <https://pubmed.ncbi.nlm.nih.gov/31676230/>. So it remains unclear what novelty this manuscript provides in relation to m6A.

The m6A analyses of the manuscript remain unconvincing and they do not provide any novel insights given what has already been published about m6A.

Supp. Fig 8a seems to indicate that m6Anet precision is low, and that high precision is only achieved at a very low recall, so it is not clear how relevant the results could be. The authors identified only ~6000 sites in all cell lines, ~2700 in the cell line where they performed m6ACE-seq, where only ~58% of them are validated by m6ACE-seq. This seems quite low given that they are using a high probability and coverage cutoff. There might be now methods with better precision and recall than m6Anet that may provide a more accurate picture of m6A across the cell lines analysed.

Supp. Fig. 8b shows a metagene plot. This seems to indicate that no m6A sites are found in the 5' UTRs. However, m6A is also abundant in the 5' UTR as shown by others previously. The density shown in the y-axis is perhaps not properly calculated?

Supp. Fig. 8e: These correlations are actually quite low. Most of the correlations between the replicate samples appear to show low correlation values, very similar to the correlations between samples from different sources. The authors should indicate the correlation values and associated p-values for the pairs of replicates. Also, the authors do not describe what value is being used for the correlation. This might be due to a limitation of m6Anet rather than a property of the data.

Supp. Fig. 8g. The authors show a heatmap based on the m6A probability. It is not clear what this plot is supposed to illustrate or whether the finding has any biological relevance.

The m6A analysis remains anecdotal and suboptimal and does not showcase the possible advantages that such large dataset could provide. It is not clear how such analyses could be acceptable for publication in Nature Methods.

Reviewer #4:

Remarks to the Author:

I thank the authors for making an effort to address my points of review. While some of them have been satisfactorily resolved, there are still aspects that require attention.

1. My concerns about quality control of the data remain unaddressed. While the authors indicate that QC programs are included in their pipeline, they acknowledge that this leads to no quality filtering beyond removing samples with few reads. Therefore, library preparation artifacts such as RT and intra-priming may still bias their conclusions about the repeat-containing transcripts. The authors indicate that these biases can be removed by SQANTI but they were unable to run the tool, which is surprising as this is a widely used method applied in many long-read transcriptomics papers. As an alternative, they claim PCR amplification of 5 novel transcripts, but this does not seem to include those with repeated regions. Therefore, doubts about the soundness of these conclusions remain.
2. I am still puzzled by the results in Figures 3a and 4b. In Figure 3a, results for comparison to ground truth at the gene level are given, while Figure 4a reports results at the transcript level. However, for some of the spike-ins, such as the ERCC, transcript and gene levels are the same, as these transcripts contain only one transcript per gene. I am not sure about Sequins, as I could not find information about alternative splicing in this set. Moreover, SIRVs E0 indicate significant differences between expected and observed expression estimates, which are not commented on the manuscript. The authors indicate that no Spearman correlation can be computed for this dataset, but do not discuss the apparent difference between expected and observed expression values.
3. Moreover, the statement in the last sentence of Page 7, indicating that library preparation introduces biases in read length, coverage, and transcript diversity, seems to be in contradiction with the next sentence stating that "Gene expression is robustly estimated across different RNA-seq protocols". This contradiction is not well addressed in the manuscript and is not mentioned in the Discussion. If the conclusion is that gene expression estimates are robust but not transcript expression estimates, this needs to be discussed further, including how the mentioned biases affect estimates.
4. The authors indicate that "among all RNA-Seq protocols, the PacBio IsoSeq data showed the highest estimation error, largely due to the uneven representation of transcript lengths, with low coverage for short transcripts". While this might be true, there is not a formal demonstration of this, therefore the statement should be rephrased as "possibly", as other causes, such as sequencing depth may contribute. On the other hand, this statement is misleading as it focuses the problem on the poorer capture of Pacbio of shorter transcripts and not on the limitation of Nanopore for quantifying longer transcripts, which might be an issue according to Figure 2b.
5. I strongly suggest re-wording full-length-read to complete-junction-set reads or full-splice-match reads, as their definition does not correspond to the concept of full-length.

6. I could not find the Supplementary Text referred to in the rebuttal.
7. A few improvements in the writing are suggested.
 - a. Page 3. Line 4. Change "alternative gene isoforms" to "alternative isoforms"
 - b. Page 3. Line 5. Change "can be independently regulated" to "can be differentially regulated"
 - c. Page 3. Line 8. Change "their regulation and expression plays" to "their regulation and expression play"
 - d. Page 3. The first sentence of paragraph 3 starting with "transcriptome profiling that uses" is awkward.
 - e. Page 4. The sentence "our study provides a data-driven guide for experimental design of transcriptomics and epitranscriptomics experiments" should be removed. Experimental design questions (statistical power, number of replicates, sequencing depth, batch effects, etc) are not addressed in this work.
 - f. Page 7. Line 3. Change "some genes generated" to "some genes had"

** For Nature Portfolio general information and news for authors, see <http://npg.nature.com/authors>.

Version 4:

Decision Letter:

Our ref: NMETH-AS45852D-Z

10th Oct 2024

Dear Dr. Goeke,

Thank you for submitting your revised manuscript "A systematic benchmark of Nanopore long read RNA sequencing for transcript level analysis in human cell lines" (NMETH-AS45852D-Z). It has now been seen by the original referees and their comments are below. The reviewers find that the paper has improved in revision, and therefore we'll be happy in principle to publish it in Nature Methods, pending minor revisions to satisfy the referees' final requests and to comply with our editorial and formatting guidelines. We suggest moving the content about highly repetitive genes into a supplementary file. Please let us know if you have any additional concerns.

TRANSPARENT PEER REVIEW

Please note: we allow redactions to authors' rebuttal and reviewer comments in the interest of confidentiality. If you are concerned about the release of confidential data, please let us know specifically what information you would like to have removed. Please note that we cannot incorporate redactions for any other reasons. Reviewer names will be published in the peer review files if the reviewer signed the comments to authors, or if reviewers explicitly agree to release their name. For more information, please refer to our <https://www.nature.com/documents/nr-transparent-peer-review.pdf> target="new">FAQ page.

ORCID

IMPORTANT: Non-corresponding authors do NOT have to link their ORCIDs but are encouraged to do so. Please note that it will not be possible to add/modify ORCIDs at proof. Thus, please let your co-authors know that if they wish to have their ORCID added to the paper they must follow the procedure described in the following link prior to acceptance: <https://www.springernature.com/gp/researchers/orcid/orcid-for-nature-research>

Sincerely,
Lei

Lei Tang, Ph.D.
Senior Editor

Reviewer #1 (Remarks to the Author):

As previously mentioned, while the data is useful and the general analyses presented in the first part of the manuscript are interesting, the remaining analyses are still not novel for what one would consider to be an outstanding and exciting manuscript worth publishing in Nature Methods. Also, these additional analyses bear no relevance or connection with the presented datasets and the manuscript title "A systematic benchmark of Nanopore long read RNA sequencing for transcript level analysis in human cell lines". No such systematic benchmarking is presented, but rather a collection of disconnected results that mostly repeat what has already been published elsewhere. They are also analyses that could have been done with any other datasets. In this sense, there is no connection with the comprehensive data presented. These additional analyses are preliminary and could be developed more thoroughly and with the appropriate controls as separate research articles.

Overlap with repeat elements

While the authors have added some additional details and filters, the results remain unclear. The initial set of novel transcripts includes ~40% of novel genes. However, from Fig. 6d, it seems that most of the largest overlaps are for novel transcripts that are within known genes and have one exon difference (first exon, internal exon, ...) with respect to the annotation. Some complex repeat families can be quite long, some LINEs can be >4kb. Figure 6 does not yet unequivocally show that these repeats are significantly associated with these novel transcripts. They may be the same repeats that also overlap the annotated transcripts. So the overlap is linked to the locus, not necessarily to the novel isoform. The authors do not describe how many isoforms overlap repeats in each subset and whether these are different repeats or the same repeats when the isoforms are in the same gene.

Also, the authors use annotated transcripts "with at least one full-splice-match read support". It is not clear whether this may bias the comparison against long annotated genes, which may be more likely to overlap with repeats. Transcription initiation has been seen to overlap with retrotransposons (<https://pubmed.ncbi.nlm.nih.gov/19377475/>). It is possible that by restricting the annotated dataset, many overlaps are not considered, in particular the longest ones, which are more likely to overlap with repeats.

As described by the authors "New transcripts generally have lower expression, number of exons, and transcript length compared to reference transcripts". It is possible that the enrichment observed is confounded by the differences in length. This could explain why after filtering for 80% overlap, figure 6d shows a low number of annotated transcripts, whereas thousands of annotated transcripts are known to overlap repeats (e.g. <https://www.ncbi.nlm.nih.gov/pmc/articles/PMC186627/>). So Figure 6d remains misleading, as it may be simply an artifact of the length difference.

The authors overlap the annotated and novel parts of the isoforms with repeats. But they talk about enrichment of novel isoforms. This is also misleading. Do they mean enrichment of novel parts? Considering that most of the overlaps are with isoforms that are within annotated genes, are they actually finding an enrichment of repeats in novel "exons"? This links to the previous question. Are these mostly new alternative first exons? Figure 6d is just a subset of the cases and do not represent the global behaviour.

The authors also use the term "these highly repetitive novel transcripts" (supp text, page 25). This is misleading and "highly repetitive" should be quantified properly. The authors do not show that these transcripts are repetitive, or that the repeats that overlap with novel transcripts are multiple copies of the same repeat class and family.

Transposable elements have already been described to contribute to lncRNAs (<https://pubmed.ncbi.nlm.nih.gov/23637635/>), which have similar properties to the novel transcripts described in the manuscript. Is it still possible that those novel transcripts are lncRNAs? A clearer definition of novel transcript is necessary in this context. How did the authors confirm that these transcripts are novel?

As indicated before, the claim that repeats are significantly associated with novel transcripts requires unequivocal evidence, a clearer description, and a more thorough analysis. It is intriguing and interesting, but it may deserve a separate focused publication with more information. At the moment, it is unclear how these results represent an important biological finding by itself and whether the datasets used play any relevant role or not.

Fusion transcripts

The authors indicate "we implement a novel approach to align reads to a genome that contains the fusion breakpoints". However, this approach is not novel. Other methods have used before the genomic breakpoints to identify the fusion transcripts, (e.g. <https://pubmed.ncbi.nlm.nih.gov/21478487/>, <https://pubmed.ncbi.nlm.nih.gov/31728961/>). The analysis produced by the authors is correct, but it is incremental compared with prior results accomplished in other papers. Also, as previously mentioned, the authors do not make any effort to show how their SG-NEx data resource was particularly unique or relevant to this analysis.

m6A analysis

The authors did not directly address the raised concerns. They have added additional datasets and comparisons that do not add much novelty to the paper. They simply illustrate the capacity of different methods to predict m6A from nanopore data and the general low precision of nanopore-based methods to recover m6A sites predicted by orthogonal methods. The results also show that in DRACH sites, dorado performs better than m6Anet. However, this is not mentioned in the main manuscript. It is not clear what this adds to the overall manuscript.

These additional results are correct but incremental compared with prior results reported in the literature, and do not add any novelty or significance to the main manuscript. As mentioned before, the prediction of m6A from nanopore sequencing and comparison with other methods has already been published before (e.g. <https://pubmed.ncbi.nlm.nih.gov/37019930/>), the landscape of m6A across different sample types has already been published before (e.g. across cell types <https://pubmed.ncbi.nlm.nih.gov/22575960/>, and across tissues, including tissue specificity: <https://pubmed.ncbi.nlm.nih.gov/31676230/>). So it remains unclear what novelty this manuscript provides in relation to m6A.

Reviewer #4 (Remarks to the Author):

Concerns have been satisfactorily addressed. I thank the authors for their efforts.

Version 5:

Decision Letter:

4th Feb 2025

Dear Dr Goeke,

I am pleased to inform you that your Analysis, "A systematic benchmark of Nanopore long read RNA sequencing for transcript level analysis in human cell lines", has now been accepted for publication in Nature Methods. The received and accepted dates will be 18th Oct 2021 and 4th Feb 2025. This note is intended to let you know what to expect from us over the next month or so, and to let you know where to address any further questions.

Over the next few weeks, your paper will be copyedited to ensure that it conforms to Nature Methods style. Once your paper is typeset, you will receive an email with a link to choose the appropriate publishing options for your paper and our Author Services team will be in touch regarding any additional information that may be required. It is extremely important that you let us know now whether you will be difficult to contact over the next month. If this is the case, we ask that you send us the contact information (email, phone and fax) of someone who will be able to check the proofs and deal with any last-minute problems.

If you are active on Twitter/X, please e-mail me your and your coauthors' handles so that we may tag you when the paper is published.

Best regards,
Lei

Lei Tang, Ph.D.
Senior Editor
Nature Methods

** Visit the Springer Nature Editorial and Publishing website at http://editorial-jobs.springernature.com?utm_source=ejP_NMeth_email&utm_medium=ejP_NMeth_email&utm_campaign=ejp_Nmeth >www.springernature.com/editorial-and-publishing-jobs for more information about our career opportunities. If you have any questions please click [here](mailto:editorial.publishing.jobs@springernature.com) .**

Open Access This Peer Review File is licensed under a Creative Commons Attribution 4.0 International License, which permits use, sharing, adaptation, distribution and reproduction in any medium or format, as long as you give appropriate credit to the original author(s) and the source, provide a link to the Creative Commons license, and indicate if changes were made. In cases where reviewers are anonymous, credit should be given to 'Anonymous Referee' and the source.

Summary of Response to Reviewers:

In our submitted manuscript we describe the data resource that was developed from the Singapore Nanopore Expression Project (SG-NEx). We very much appreciate the detailed comments and suggestions from all reviewers related to the analysis and description of the SG-NEx resource in our original manuscript. To address these concerns, we have made major changes to the manuscript. In particular, we now more strongly emphasise the unique value of the SG-NEx data, we included new data and analyses, we improved the robustness of the analysis and rewrote the manuscript to more precisely describe our findings:

Data resource:

- We now include additional PacBio IsoSeq data for 6 cell lines. The SG-NEx resource now includes data from all 5 major RNA-Seq protocols (ONT direct RNA-Seq, PCR-cDNA-Seq, direct cDNA-Seq, PacBio IsoSeq, Illumina short read RNA-Seq)
- We compare short read and the different long read RNA-Seq protocols using 6 major spike-in RNA sets (Sequin MixA V1, Sequin MixA V2, SIRV E0, SIRV E2, Long SIRV, ERCCs), making the SG-NEx data the most complete resource for benchmarking when ground truth transcript expression is required
- The SG-NEx data set is now available through the AWS open data registry which allows the download of individual samples in seconds (in addition to being hosted on ENA)
- The SG-NEx data is fully documented, with tutorials on data access and data analysis that are open to community contributions as well (current tutorials include FLAIR, IsoTools, Bambu, NanoCount, m6Anet, xPore, and BLOW5 tools).

Robustness of analysis:

- We now generated results with 5 different transcript quantification methods (Salmon - short read; Salmon -long read; RSEM; Bambu; NanoCount), and we quantified results using multiple different metrics (spearman correlation, mean absolute error (MAE), root mean squared error (RMSE), mean absolute relative difference (MARD), mean relative difference (MRD), and coefficient of determination/R²)
- To control for variation in error rate, we have re-basecalled the entire raw nanopore sequencing data (20TB, including recently added BLOW5 files) with an identical, recent version of Guppy (v6.4.2)
- To evaluate the impact of sequencing and alignment error we generated results with two different alignment strategies, firstly focusing only on primary alignments, and secondly using only reads that do not have any secondary alignment
- We quantified our results using matched replicates where the same RNA was used for sequencing with different library preparation methods
- We further quantified findings across all SG-NEx cell lines to ensure that the results are not specific to any single cell line that is used

Technology comparison:

- Instead of comparing the different ONT protocols to each other (original manuscript), we now compare all 5 different RNA-Seq protocols in the technology comparison,

thereby providing a more systematic and comprehensive overview of current RNA-Seq library preparation and sequencing methods

- We now include a comparison of sequencing throughput and cost for all different technologies. Based on the available price information at this time, the cost per throughput is comparable between short read and long read RNA-Seq
- We show that the different library preparation and sequencing protocols show differences in terms of throughput, read length, biases, and quantification, highlighting the need for a single resource that contains matched data from these different protocols so that computational methods can be developed and benchmarked without being optimised to just a single technology

New analysis:

- In order to understand the large observed differences in transcript expression between short and long read RNA-Seq data (which has previously been reported), we now included an in-silico fragmentation framework, in which we simulated short read data from 112 long read RNA-Seq samples in the SG-NEx resource. The results indicate that read fragmentation partially explains the observed differences, in particular for major isoforms that are only identified by Illumina short read RNA-Seq data. These short-read specific isoforms show significantly higher expression when long reads are fragmented to 150 bp read length.
- We have selected 12 fusion genes, including several with multiple fusion gene isoforms from the MCF7 cell line for experimental validation.

Manuscript:

- In addition, we have made changes to the manuscript to increase the focus, clarity, and more precisely describe the analysis and findings in the context of existing literature. Furthermore, we now more strongly emphasise the value of the SG-NEx data as a comprehensive and unique resource for the community in the abstract, introduction, results, and discussion.

The SG-NEx resource provides 139 RNA-Seq samples, covering 14 cell lines, 5 RNA-Seq protocols, 6 spike-in RNA sets, and matched replicates where the same RNA extract was sequenced with different library preparation methods. Together this makes the SG-NEx data a single resource for all aspects of RNA-Seq analysis and methods benchmarking. The combination of this breadth of data with easy accessibility is already reflected in high numbers of downloads and citations (the pre-print is cited 55 times since publication, “compared to other publications in the same field, this publication is extremely highly cited and has received approximately 18 times more citations than average” (<https://badge.dimensions.ai/details/id/pub.1137410720>)). We very much believe that the major effort that went into the revision of this manuscript over the last 22 months will further increase the value of the data set and the impact of this study, which we believe will be a defining resource for RNA-Seq in the next years.

We thank all reviewers for their comments and suggestions, and hope that our effort in addressing these comments is reflected in an improved, revised manuscript that will be acceptable for publication in *Nature Methods*. Please find our detailed response below.

Reviewer Comments:

Reviewer #1 (Remarks to the Author):

A. Summary of the key results

The manuscript presents the comprehensive sequencing of the transcriptome from 7 cell lines using Short-read (Illumina cDNA) and long-read (Nanopore direct RNA, direct cDNA, PCR-based cDNA) technologies. The manuscript describes a set of various analyses performed on these datasets: a comparison between the different technologies, abundance estimates, detection of alternative transcript isoforms, detection of transcripts on repeat elements, detection of fusions, and detection of m6A. The results provided are used to argue for the benefit of using Nanopore sequencing technology for these transcript analyses. However, the results presented are not novel, many of the analyses are poorly performed, and the paper does not highlight any novelty provided by their cell line dataset. All these issues are described below.

Response:

We thank Reviewer #1 for the detailed assessment of our manuscript describing the SG-NEX Project and data resource. Reviewer #1 highlights 3 key points, in particular: (1) the novelty of the data set is not sufficiently demonstrated, (2) concerns about the computational analysis, and (3) that some results have been reported previously.

The SG-NEX resource provides 139 RNA-Seq samples in total, covering 14 cell lines and 5 RNA-Seq protocols (nanopore direct RNA-Seq, direct cDNA-Seq, PCR-cDNA-Seq, short read RNA-Seq, and PacBio Iso Seq, which we have included in the revised manuscript). Most of these cell lines were sequenced with all protocols and multiple replicates, which makes this resource useful as a benchmark data set for most applications of RNA-Seq. Furthermore, unlike any other currently available resource, the SG-NEX data set is available through the AWS open data registry which allows the download of individual samples in seconds, it is described with a detailed documentation on data access, and the documentation includes several tutorials on data analysis, which are open to community contributions as well. Together, this has made the SG-NEX data a very popular resource, with 55 citations since the first release (<https://badge.dimensions.ai/details/id/pub.1137410720>).

We very much believe that the SG-NEX data is unique and we already see that it is of high value to many users, however, we appreciate the comments from Reviewer #1 that the original manuscript did not sufficiently convey the novelty of this work. In order to address these comments we have therefore made major changes to emphasise the key novelty (the data resource), while addressing the comments related to the findings and analysis. In line with these changes, we have also adjusted our claims that now emphasise the novelty and value of the data resource.

We hope that the revised manuscript will be much clearer, concise, and addresses the concerns regarding the data analysis. Please find our detailed response to each point below.

Reviewer #1:

B. Originality and significance: if not novel, please include reference

Overall, the article is quite descriptive and does not present anything that has not been already described elsewhere:

The association between alternative transcription start sites and splicing using single-molecule long-read sequencing has been already documented, see e.g.:

<https://pubmed.ncbi.nlm.nih.gov/29598823/>

<https://pubmed.ncbi.nlm.nih.gov/30320766/>

But these articles have not been cited.

Isoform diversity from long-read sequencing has already been documented multiple times (e.g.

<https://pubmed.ncbi.nlm.nih.gov/31676752/>, <https://pubmed.ncbi.nlm.nih.gov/30320766/>) but

none of these has been cited. Moreover, although this manuscript and many articles before have reported novel transcripts from long-read sequencing, whether these are real or not remains to be confirmed. This manuscript does not provide any novelty regarding this question.

Quantification with Nanopore Long-read sequencing (cDNA and RNA) and comparison with short reads and with spike-ins have been done before:

<https://pubmed.ncbi.nlm.nih.gov/31624302/>

<https://pubmed.ncbi.nlm.nih.gov/28722025/>

But these articles have not been cited.

The result related to the ability of Nanopore direct RNA sequencing to identify m6A RNA modifications has already been shown before multiple times (e.g.

<https://pubmed.ncbi.nlm.nih.gov/31501426/> , not cited)

The result about the conservation of the mRNA methylome across cell types has been reported before multiple times (e.g. <https://pubmed.ncbi.nlm.nih.gov/22575960/>, not cited)

Response:

In our manuscript we describe the SG-NEx data resource, which includes 14 cell lines, 7 of which are profiled with multiple replicates on 5 different RNA-Seq protocols, and it includes 6 different spike in RNA sets. Unlike previous studies, the breadth and comprehensiveness of the SG-NEx data makes it a unique resource that can be applied to many different tasks in transcriptome profiling. To illustrate this, we used the data set to compare the different library preparation methods and sequencing technologies, and we demonstrate insights from the

analysis of isoform expression, fusion detection, and RNA modification analysis of the SG-NEx data.

Reviewer #1 highlights that similar analyses and results have been reported previously by other studies using different data sets or different technologies. Furthermore, Reviewer #1 points out that the original manuscript did not cite these references.

The key novelty in our manuscript is the SG-NEx data resource. While other data sets have been used to describe alternative isoform expression, short read and long read RNA-Seq data, or the detection of m6A, the breadth of the SG-NEx data makes this a single, comprehensive resource and benchmark data set that we believe is of high value to many users (Figure R1.1, corresponding to Supplementary Figure 2, see Supplementary Table 2 for more details). We thank Reviewer #1 for pointing out that this novelty was not sufficiently emphasised in our original manuscript. To address this point, we now focus on demonstrating the unique value of the data resource in the revised manuscript. Among others we now provide a comparison of all 5 different sequencing protocols (instead of the original focus only on the Nanopore RNA-Seq data), and we quantify observations across replicates and cell lines. Furthermore, we rewrote the text to include the missing references.

We hope that the revised manuscript now more clearly highlights the novelty, value, and uniqueness of the data resource. Please find additional responses to the detailed suggestions below and in our response to Reviewer #1, point C.

(1) Alternative promoters and TSS/splicing

(a) To address the comment, we now cite these papers to highlight that similar analyses have been performed before. We still believe that our results, which quantify the number of alternative splicing events in the SG-NEx data provides a helpful overview and illustration of the variation in isoform expression in this data set, and we hope that the revised manuscript now more clearly emphasises that the data resource is the main novelty.

(2) Isoform diversity from long read RNA-Seq

(a) We now cite the suggested manuscripts. We agree that the functionality or relevance of novel transcripts is an open question. In the revised manuscript we now included this point in the discussion.

(3) Comparison of short read and nanopore long read RNA-Seq data

(a) We now cite these papers, along other key papers that have compared short read and nanopore long read RNA-Seq data. Unlike any of the previous studies, the SG-NEx data includes all nanopore RNA-Seq protocols and PacBio IsoSeq, and we include all 6 major spike-in RNA sets. In addition, all SG-NEx nanopore RNA-Seq data was re-basecalled in the revised manuscript using the latest version of Guppy to ensure the data quality reflects the current state of the technology, whereas other data sets are often outdated in terms of nanopore basecalling quality. Finally, since we have data from all protocols in multiple replicates across different cell lines, we can statistically quantify the observed differences across cell lines and replicates which is also a unique aspect of the data resource.

(b) Following a suggestion from Reviewer #3 we now simulate short read RNA-Seq data from the SG-NEx long read RNA-Seq data by artificially

fragmenting long reads (Figure R1.2a). We observe that the fragmented long read data now more closely resembles what is observed in Illumina short read RNA-Seq data (Figure R1.2b). Furthermore, read fragmentation leads to a significant over-estimation of transcript expression compared to the expected expression specifically for isoforms that were only identified by Illumina short read RNA-Seq as major isoforms (Figure R1.2c). These results demonstrate that read fragmentation introduces an over-estimation artefact that partially explains the observed differences between short read and long read RNA-Seq data. These results have not been reported and statistically quantified by previous studies (please refer to Reviewer #3, point [2], [9] and [10] with additional details). The simulated fragmentation was done for all 112 SG-NEx samples and was quantified across all matched replicates and cell lines, illustrating how the SG-NEx data provides a very rich resource to systematically compare RNA-Seq protocols.

(4) Detection of m6A and conservation of m6A across cell types

(a) As pointed out by Reviewer #1, the detection of m6A has been possible for many years with different technologies such as m6A-Seq, and recently also with nanopore direct RNA-Sequencing. The SG-NEx data provides the largest publicly available data set of nanopore direct RNA-Seq data, which makes it a highly useful resource to develop and evaluate computational methods that detect RNA modifications from direct RNA-Seq data. To illustrate the ability to simultaneously quantify transcript expression and RNA modifications using the SG-NEx data, we identified m6A sites and compared the landscape of m6A across multiple cell lines. To address the reviewers comment, we now cite the suggested papers and additional methods that detect RNA modifications to clarify that the novelty in our manuscript is the SG-NEx data resource and not a new method to detect m6A.

Figure R1.1: Data resource comparison

Barplots of number of reads, number of long reads, number of protocols per samples, number of cell lines (or tissues), number of spike-in sets, number of median replicates per samples, and for different long read data resource

Figure R1.2: Comparison of short read RNA-Seq data and fragmentation simulated short read RNA-seq data with long read RNA-Seq data

(a) Main Figure 4h Schematic showing the fragmentation simulation of short read from long read
 (b) Scatterplots showing fragmentation simulated short read or short read RNA-seq data against long read RNA-seq data

(c) Main Figure 4j Boxplots of mean absolute differences between long read RNA-Seq data and fragmentation simulated short read RNA-seq data for isoforms that are major in both long and short read RNA-seq data (Major isoforms), major isoforms that are only major in long read RNA-seq data (Major-LR isoforms), major isoforms that are only major in short read RNA-seq data (Major-SR isoforms), and minor isoforms.

Reviewer #1:

C. Data & methodology: validity of approach, quality of data, quality of presentation

Some of the tools used and analyses performed have not been validated.

E.g. nanoseq is described as a pipeline to perform alignment; transcript discovery and quantification; and differential expression analysis. Nanoseq uses already published tools, but some of its steps include new software that has not been validated and/or compared with other tools.

Response:

In addition to describing the SG-NEx data resource, our manuscript presents a nf-core nextflow pipeline which is designed to simplify the analysis of Nanopore RNA-Seq data and make the data accessible to more users. The nanoseq pipeline is using modules, which allows the users to perform only specific steps, and to choose different tools for each step. Since the submission of the original manuscript, all methods used by nanoseq have been peer reviewed and are published (see below for additional details regarding Bambu and m6Anet). The nanoseq pipeline follows the nf-core regulations and is open to contributions from the community, and we hope to maintain a pipeline that will be updated with new methods in future as well.

Reviewer #1:

[C.2] Some of the plots show the results using one of these tools, bambu, in comparison with spike-in data and across technologies. So this is a sort of benchmarking of bambu, but integrated with other things that have nothing to do with bambu. This leaves the reader confused about what the main aim of the paper is. The benchmarking is using bambu, which is not published but also not described in the manuscript; but the manuscript is presented to be a description of a comprehensive dataset.

Response:

Since the submission of the original manuscript, Bambu has been published (Chen et al. Nature Methods (2023)). However, to demonstrate that the results in this manuscript are independent from the computational method, we now use multiple different methods to quantify transcript expression (RSEM (short read); Salmon (short read); Salmon (long read); NanoCount (long read), Bambu (long read)). We find that the results are consistent when alternative methods are used on spike-in RNAs and human cell lines (Figure R1.3-1.5).

In the revised manuscript we now use Salmon for the comparison of short and long read RNA-Seq data as it is the only method that has an option to quantify expression for both data types, and we show the results with Bambu, RSEM, and NanoCount in Supplementary Figures 4-5. For the analysis of alternative isoform expression and fusion transcripts we have used Bambu in the main figures, as Bambu is the only method that provides full length

read counts to reduce possible false positives (results with Salmon, NanoCount, and StringTie2 are shown in Supplementary Figures 6-7).

We thank Reviewer #1 for this comment and hope that the revised manuscript now provides a clearer and more robust summary of the differences between short and long read RNA-Seq data.

Figure R1.4 Comparing log-2 transformed estimated CPM against log2-transformed expected CPM for spike-in transcripts

(a) Heatmaps of spearman correlation, mean absolute error (MAE), root mean squared error (RMSE), mean absolute relative difference (MARD), mean relative difference (MRD), and coefficient of determination (R^2), for ERCC, Sequin MixA V1, Sequin MixA V2, SIRV E2, SIRV E0 and Long SIRVs, when using Bambu, NanoCount and Salmon for cDNA, direct cDNA and PacBio data, and Salmon and RSEM for short read RNA-seq data

(b) Histograms of absolute errors (AE) for spike-in transcripts in ERCC, Sequin MixA V1, Sequin MixA V2, SIRV E2, SIRV E0 and Long SIRVs, when using Bambu, NanoCount and Salmon for cDNA, direct cDNA and PacBio data, and Salmon and RSEM for short read RNA-seq data

Figure 1.5 Comparison between sample replicates generated using different sequencing protocols

(a-b) Boxplots of (a) spearman correlation and (b) mean absolute relative differences between log₂-transformed CPM from replicates generated by different protocols, from the same cell line (intra- cell line) for protein-coding genes, long-noncoding RNA (lncRNA) genes, other genes and all genes, using Bambu and Salmon for long read data, RSEM and Salmon for short read data

(c-d) Boxplots of (c) spearman correlation and (d) mean absolute relative differences between log₂-transformed CPM for protein-coding gene isoforms that major in both long and short read (Major isoforms), isoforms major only in long read (Major-LR isoform), isoforms major only in short read (Major-SR isoforms) and isoforms that are not major isoforms (Minor isoforms) from replicates generated by long read protocols (LR) and short read protocol (SR) from the same cell line obtained, using Bambu and Salmon for long read data, RSEM and Salmon for short read data

Reviewer #1:

[C.3] The results for m6A use m6Anet and include the two cell lines that have already been analysed in the m6Anet preprint:

<https://www.biorxiv.org/content/10.1101/2021.09.20.461055v1?rss=1>

This makes some of the data presented somewhat redundant.

Similarly, the analysis of fusions uses JAFFA and JAFFAL. But the analyses of these cell lines with the same two tools have already been presented in another preprint:

<https://www.biorxiv.org/content/10.1101/2021.04.26.441398v2>

So again, this part of the manuscript is redundant and has been already described elsewhere.

Response:

Reviewer #1 points out that the SG-NEx data was used in other studies that present new computational methods, and which were applied to the SG-NEx data. In addition to the manuscripts describing m6Anet and JAFFAL, the SG-NEx data was also used for example in manuscripts describing FusionSeeker (Chen et al. 2023), IsoTV (Annaldasula, Gajos, and Mayer 2021) , FLAIR-fusion (Felton et al. 2023) or LocusMasterTE (Lee et al. 2023).

We have made the SG-NEx data available very early after data generation (even before the preprint was posted) to allow many people to use the resource. We think the use of these data for evaluation and benchmarking of computational methods is an excellent use case, and we believe these manuscripts are a demonstration of the value of the data resource.

To address the concern that the results presented in this manuscript are redundant with results presented by other manuscripts that use the SG-NEx data, we now include additional experimental validation of the fusion genes from the MCF7 cell line which have not yet been validated (see Response to Reviewer #3 for additional details, Figure R3.26, and Supplementary Table 9), and removed results that showed the signal difference between m6a and non-m6A sites and which have been reported several times in previous work.

We hope that the revised manuscript does not include any redundant results, and that it now more clearly emphasises the value of the data set for developing, benchmarking and applying new computational methods by providing a rich resource of short and long read RNA-Seq data.

Reviewer #1:

D. Appropriate use of statistics and treatment of uncertainties

Some of the benchmarking presented is not new in relation to the data itself. For instance, the comparisons of gene and transcript expression have been performed before using other tools and datasets. By themselves, they do not say anything relevant about the datasets. A similar thing happens with the analyses of m6A, fusions, and repeats.

Response:

To address the Reviewers comment, we now include additional data and analyses in the revised manuscript. Furthermore, we rewrote the manuscript to emphasise the unique value of the SG-NEx data, which we believe is a key novelty and resource for the community. Specifically, we made the following changes in response to the Reviewer's comment:

- 1) Gene and transcript expression comparison has been performed before using other tools and datasets
 - a) In the original manuscript we have compared nanopore long read RNA-Seq data with short read RNA-Seq data using Bambu on sequin spike-in RNAs and a single cell line from the SG-NEx data. To address the reviewers comments, we have included additional data and performed new analysis to highlight the value of the SG-NEx data beyond previous studies:
 - i) We now include PacBio IsoSeq data in addition to Nanopore RNA-Seq data. Together, the SG-NEx data includes RNA-Seq from 5 different library preparation methods, which is not available in any other previous data resource
 - ii) We now analyse data from 6 different spike-in RNAs that are all available as part of the SG-NEx resource (Sequin MixA V1, Sequin MixA V2, Long SIRV, SIRV E0, SIRV E2, ERCC). The SG-NEx data is the only resource with such a complete set of spike-in RNAs for long read and short read data from matched samples.
 - iii) Instead of interpreting results using a single quantification method, we have analysed the data using 5 different computational methods (Salmon short read, RSEM short read, Salmon long read, NanoCount, Bambu). We find that the results are reproducible when different quantification methods and evaluation metrics are used, highlighting the robustness of our analysis (see also response to point C.2 above).
 - iv) In contrast to previous studies, the SG-NEx data includes RNA-Seq data from multiple replicates, multiple cell lines, and multiple RNA-Seq protocols, including matched replicates sequenced from the same RNA sample. In the revised manuscript we now use these aspects to quantify differences in gene and transcript expression across cell lines and RNA-Seq protocols. Using these data, we show that the high variation in transcript expression between short read and long read RNA-Seq data mostly impacts major isoforms that identified only by short read RNA-Seq, whereas major isoforms identified by long read RNA-Seq data are more robustly estimated (Figure R1.6).

Figure 1.6 Transcript expression profiling between long and short-read RNA-seq data
(a) Main Figure 4b Boxplots of spearman correlation between log₂-transformed CPM for protein-coding gene isoforms that major in both long and short read (Major isoforms), isoforms major only in long read (Major-LR isoform), isoforms major only in short read (Major-SR isoforms) and isoforms that are not major isoforms (Minor isoforms) from replicates generated by long read protocols (LR) and short read protocol (SR) from the same cell line obtained using Salmon
(b) Main Figure 4c Scatterplot of log₂-transformed CPM for protein coding gene isoforms that major in both long and short read (Major isoforms), isoforms major only in long read (Major-LR isoform), isoforms major only in short read (Major-SR isoforms) and isoforms that are not major isoforms (Minor isoforms) obtained from long read direct cDNA RNA-Seq (using Salmon) against that obtained from short read RNA-Seq (using Salmon) in the A549 cell line.

- v) Following a suggestion from Reviewer #3, we have expanded the framework to simulate read fragmentation from long read RNA-Seq data, thereby generating simulated short reads. The simulated fragmentation leads to a significantly higher similarity with the Illumina short read RNA-Seq data, in particular due to the overestimation of transcript expression for short read-specific major isoforms when compared against the expected expression. These results demonstrate that differences in transcript expression between long read and short read RNA-Seq data are partially explained by the fragmentation process, which leads to over-estimation of specific isoforms with short read RNA-Seq data. We quantified this observation using 112 samples from the SG-NEx data (please see Figure R1.2, above). We believe that these results provide a strong argument for long read RNA-Seq data that have not been reported before.

- vi) In the revised manuscript we now include a paragraph in the discussion to emphasise differences to previous studies in terms of data resource and technology comparison.
- 2) Fusion transcripts
- a) In the revised manuscript we included additional validation experiments to confirm fusion transcripts that were detected in the MCF7 cell line. We specifically selected 6 cancer cell lines for the SG-NEx data resource as we believe that fusion detection is one of the promising applications of long read RNA-Sequencing. The availability of these 6 cell lines with short read and long read data, including direct RNA, cDNA, and PacBio IsoSeq, spike in RNAs, and validated fusion transcripts makes this a unique resource to validate new fusion detection methods, and we believe that the results and new experimental data are useful as a demonstration of this aspect. We are very happy to see that the SG-NEx data was already used for validation of 3 different long read fusion detection methods (FLAIR fusion, JAFFAL, FusionSeekerLR), making it the most widely used resource for long read fusion detection.
- 3) m6A detection
- a) The possibility to profile RNA modifications using direct RNA-Seq data is one of the outstanding advantages of nanopore long read RNA-Seq, with many new findings and methods that are being developed. The SG-NEx data provides 55 sequencing runs of direct RNA-Seq, with 65.3 million reads in total (in comparison the second biggest resource of direct RNA-Seq (Workman et al. 2019) includes 30 samples with 13 million reads). Furthermore, the analysis of RNA modifications requires the download of raw nanopore signal data, which has a size of 4.3 TB for the SG-NEx direct RNA samples. While the data can be obtained through ENA, download would be extremely slow, and basecalling and read alignment would have to be done again. The SG-NEx data is all available through AWS as fast5 and blow5, including matched fastq and bam files, which makes the SG-NEx resource the only data set that can be used to analyse RNA modifications at high scale.
We agree with Reviewer #1 that m6A has been studied before using other technologies or different samples, but due to the promise and interest in direct RNA-Seq data for detecting RNA modifications, we believe the profiling of m6A on the SG-NEx samples is an important results that highlights the value of the SG-NEx resource for RNA modification detection.

Reviewer #1:

[D.2] In the case of repeats, their analyses do not support their conclusions. The authors say: “Compared to annotated transcripts, we observed a significant enrichment of repetitive elements in novel transcripts”.... “suggesting that one of the major advantages of long read RNA-Seq data is an improved ability to reconstruct RNAs in repeat-rich regions.” However, this is misleading. The analysis is described in methods as performed at the exon level, looking at exons not included in the annotation. However, many novel exons have been found before overlapping repeat elements using short-read sequencing (see e.g. <https://pubmed.ncbi.nlm.nih.gov/25801031/>). The right analysis to arrive to the conclusion

written by the authors would be to compare the ability of identifying these repeat-overlapping exons using long and short-read sequencing.

The article lacks any robust analyses and is mainly describing suggestive findings. For instance, the analysis of repeats is not supported by any statistical analysis about the uncertainties and reliability of the mappings. Some of the repeats observed may occur multiple times in an intron. The authors do not describe any test to confirm that what they observed is not an ambiguity in the mapping process. In the case of the novel genes overlapping repeats, it is known that some repeats are expressed, so mapping long reads to the genome is bound to find them.

The level of variation in reads due to errors might be comparable to the variation between repeat copies. This begs the question of whether the detected repeats are the right ones and what is the level of uncertainty in such mappings. This is not described in the manuscript.

Response:

In our original manuscript we have identified novel transcripts in the SG-NEx samples, and found that they are enriched in highly repetitive elements compared to transcripts found in reference annotations. Reviewer #1 raises two points:

Firstly, Reviewer #1 points out that the claim in the discussion section that long read RNA-Seq has an improved ability to identify novel transcripts in repeat-rich regions is can be misinterpreted as a comparison of long read RNA-Seq with short read RNA-Seq for transcript discovery, which was not done in this manuscript. We thank Reviewer #1 for this comment, with which we agree. We have only compared the enrichment of repetitive elements in novel transcripts found in the SG-NEx long read RNA-Seq data against annotations, which makes this claim in the original manuscript inaccurate and possibly misleading. We apologise for this out-of-context statement, in the revised manuscript we replaced the claim that *“one of the major advantages of long read RNA-Seq data is an improved ability to reconstruct RNAs in repeat-rich regions”* with the more accurate and specific statement that *“newly discovered transcripts in the SG-NEx data are enriched for repetitive elements compared to annotated transcripts.”*

Secondly, Reviewer #1 points out that the enrichment of repetitive elements could be related to uncertainty in the read alignment caused by reads which are mapped to multiple loci (multi-mapped reads) as a consequence of the sequencing or alignment errors. To limit the influence of the error rate, we have re-basecalled all data using an identical, recent version of Guppy (v6.4.2). We have then analysed the impact of multi-mapping reads on the identification of repeat-enriched transcripts from long read RNA-Seq data using two different alignment and transcript discovery strategies:

(1) *primary read alignment*: Using only the primary read alignment from the minimap2 default alignment setting (allowing up to 5 multiple alignments per reads, original manuscript);

(2) *uniquely aligned reads*: Only reads that have a single alignment are used for transcript discovery, all reads with multiple alignments are discarded

We then used Bambu for transcript discovery, which provides a confidence score for each transcript (the Novel Discovery Rate (NDR), which approximately corresponds to the False Discovery Rate). A NDR threshold of 1.0 corresponds to the complete set of novel transcript candidates with only minimal filters (at least 2 read counts in any sample and at least 5% contribution to the total expression for the respective gene). For the human genome, which is well annotated, a NDR threshold of 0.1 is recommended, which ensures that novel transcript candidates that have very low read support, or which could arise from sequencing or alignment artefacts are not considered as “valid” novel transcripts.

Using the most stringent alignment option (“unique alignments”) reduces the number of reads by 38% on average across all data sets. This results in a reduction of 25% of novel transcript candidates compared to the original alignment mode (NDR=1.0), indicating that the presence of multi-mapped reads leads to the identification of additional novel transcripts candidates (Figure R1.7a). However, using the recommended NDR threshold (NDR=0.1), most novel transcript candidates are removed, resulting in a comparable number of novel transcripts (Figure R1.7a, unique alignment vs all alignment: 1681 vs 1635 at NDR=0.1). A comparison of the novel transcripts that are identified by the different read filtering strategies shows that 86% are still identified even when only unique reads are considered (Figure R1.7b). Furthermore, even when only uniquely aligned reads are used for transcript discovery, we still observed a similar, significant enrichment of novel transcripts in repetitive elements ($p < 0.001$, Figure R1.7c, corresponding to Supplementary Figure 7g). Together, this indicates that multi-mapping reads have an influence on transcript discovery, but that the enrichment of repetitive elements in novel transcripts is still observed even when multi-mapped reads are removed.

We have included these additional results in the revised manuscript, and we now include the estimated NDR for each transcript from both alignment strategies as a supplementary table to provide an additional measure of confidence for repeat-derived novel transcripts (Supplementary Table 7). Furthermore, we now rewrote the discussion and results to more precisely describe our findings.

Finally, to address the concern about the robustness of the analysis, we have re-analysed the data using different computational methods (Salmon, RSEM, Bambu, NanoCount, see response to point C2. above), and we have quantified our observations from spike in RNAs and the comparison of human cell line RNAs for gene and transcript expression with different metrics (correlation, mean absolute error, mean absolute relative error, root mean squared error, and R-squared). The results show that our findings are confirmed when alternative computational methods and quantification metrics are used. We have included these new results in the revised manuscript, and hope that they demonstrate the robustness of our computational analysis.

We thank Reviewer #1 for this comment, and we hope that the revised manuscript now provides a more robust analysis of gene and transcript expression in the SG-NEx data.

Figure R1.7 Novel transcripts identification using primary and reads with unique alignments

(a) Barplots presenting the number of novel transcripts identified when primary alignments of all reads are used vs reads with unique alignments are used with varying transcript discovery threshold ranging from 0.1 to 1, at an increasing step of 0.1 Blue bars represent the number of novel isoforms of annotated genes discovered, green represents the number of novel isoforms of novel genes discovered. The grey points represent the total number of novel transcripts.

(b) Barplots representing the predicted NDR when using reads with unique alignments for novel transcripts identified when using primary alignments with an NDR threshold of 0.1: light blue represents a predicted NDR < 0.1, blue represents a predicted NDR of ≥ 0.1 and < 0.2, light green represents a predicted NDR ≥ 0.2 , and green represents not identified

(c) Boxplots of percentage overlapping with repeat elements for annotated transcripts with at least 1 full-length read support across samples (Annotated ≥ 1 FL), novel isoforms, and novel gene isoforms using primary alignments for all reads and reads with unique alignments

Reviewer #1:

E. Conclusions: robustness, validity, reliability

The results do not provide any new insights that have not been already described in the prior literature for nanopore long-read sequencing data and do not clearly show the advantage or uniqueness of this data beyond these known facts.

The only apparent novelty is the large amount of data produced for each cell line. This would be more appropriate for a publication in *Scientific Data* or similar.

Response:

Following the suggestions of all Reviewers, we have rewritten the manuscript to better emphasise the novelty of the data resource and findings. In addition, we also included additional data, performed new analysis, and generated validation experiments to further distinguish this study from previous work and increase the value of the data resource. In particular, we believe that the following points are unique to the SG-NEx data resource and present key new findings that have not been reported previously, and which we believe will be of interest to the readership of *Nature Methods*:

Data resource:

- 1) 5 different short and long read RNA-Seq protocols
 - a) The SG-NEx resource includes RNA-Seq data for 5 different RNA-Seq protocols: PacBio IsoSeq, Nanopore direct cDNA, Nanopore direct RNA, Nanopore PCR cDNA, and Illumina short read RNA-Seq. We show that the alternative library preparation and sequencing protocols show differences in throughput, read length, biases, and quantification. Therefore, we believe it is essential to have a single resource that contains matched data from these different protocols so that computational methods can be developed and benchmarked without being optimised to just a single technology.
- 2) Multiple cell lines
 - a) Gene and transcript expression can differ widely between cell lines. The SG-NEx data includes 14 different cell lines and tissues in total, 7 of which were sequenced with long read and short read technologies. These 7 cell lines represent 6 different tissues of origin and an embryonic state (H9 hESCs). The analysis of fusion transcripts provides an illustration of this value, as the analysis of a single cancer cell line would only provide a single snapshot that depends highly on the frequency of fusion transcripts. Furthermore, the comparison of gene and transcript expression results is also confirmed across all cell lines, enabling a more robust analysis that is not limited to just a single cell line. By including this variety of cell lines, the SG-NEx data provides the ability to confirm the validity of findings beyond a single context, which we think is essential to test new hypotheses and validate computational methods.
- 3) Multiple, matched, replicates

- a) Each cell line and library preparation protocol was sequenced in multiple replicates (with the exception of the PacBio IsoSeq data, which was added during the revision). For the core data, we have at least 3 replicates (often many more to include sufficient sequencing depth). Furthermore, we sequenced the same RNA extract with different library preparation methods and sequencing technologies, providing matched replicates that only differ in the sequencing technology but which otherwise used the same RNA as input. Both the availability of matched replicates, as well as the minimum requirement of three replicates for all core cell lines and protocols makes the SG-NEx data resource unique compared to other data sets.
- 4) 6 spike in data sets
 - a) Spike in RNAs provide a way to quantify the accuracy of transcript discovery and quantification with a ground truth, making them essential for benchmarking computational methods and comparison of sequencing technologies. The SG-NEx data includes all major spike-in RNAs that are currently available and used (Sequin V1, Sequin V2, long SIRV, SIRV E0, SIRV E2, ERCC). While individual spike-in RNAs are used in other studies, the inclusion of all spike-in RNAs across different RNA-Seq protocols makes the SG-NEx data a unique resource that addresses all requirements for any spike-in based benchmark, making this a gold standard data set for evaluation of computational methods when ground truth data is needed.
- 5) RNA modifications
 - a) The SG-NEx resource is the largest currently available data set for direct RNA-Sequencing, making this a unique resource to develop and apply computational methods that detect and quantify RNA modifications. To provide additional value as a benchmark data set, SG-NEx resource also includes m6ACE-Seq data to provide a map of m6A for one of the core cell lines (Hct116).
- 6) Fusion transcript validation experiments
 - a) In the revised manuscript we have validated novel fusion transcript candidates that were identified in the SG-NEx cell lines. We provide the list of validated fusion genes, and we hope that this further establishes the SG-NEx data as a benchmark data set for new fusion detection methods.
- 7) Throughput
 - a) The SG-Nex data contains the largest amount of long read data compared to previous studies. We agree with Reviewer #1 that the amount of data does not indicate the usefulness of the data (which is supported by the points above). However, we still believe that the high number of samples and reads per cell line are useful to maintain the relevance of the data resource in the long term.
- 8) Available through AWS Open Data Registry
 - a) While data from other studies can be downloaded through public data repositories such as GEO or ENA, this can be probability slow in practice, in particular for raw sequencing data to detect RNA modifications. In contrast, the SG-NEx data is available through the AWS open data registry, providing very fast access and download speeds for the raw and processed data (>20TB in total). Furthermore, all data can be browsed interactively using the genome browser as the processed raw data tracks are hosted on AWS as

well. In addition, all SG-NEx data is available through ENA as well to ensure long term archival and access.

- 9) Detailed documentation including tutorials on data use
 - a) To make the SG-NEx data a useful resource, we have developed a detailed data access documentation. Furthermore, we provide multiple tutorials to illustrate how to analyse long read RNA-Seq data. These tutorials can be run on Google Colab without any local installation, making it very easy to analyse these data by anyone even without computational expertise. The SG-NEx repository is open for community contributions as well, and hosts tutorials for FLAIR, NanoCount, IsoTools, and BLOW5 tools, and we hope to host more such tutorials in the future.

In addition to highlighting the unique value of the SG-NEx data, we also present new findings that have not been previously reported, and which we believe provide key insights into long read RNA-Sequencing. Among others, we show protocol-specific differences in terms of read length, 3' bias, and transcript coverage, we demonstrate that read fragmentation partially explains the observed difference between transcript expression estimates in short read and long read data, and we demonstrate that long read RNA-Seq data can be used to identify and quantify individual fusion transcripts, which we validated (see responses above for additional details).

Together we hope that the revised manuscript now more clearly emphasises the value and novelty of the SG-NEx data resource, and we believe that both the data and results will be of great interest to the readership of *Nature Methods*.

Reviewer #1:

F. Suggested improvements: experiments, data for possible revision

The authors argue that their datasets are valuable for benchmarking to help the development of tools for long-read sequencing. However, the manuscript does not really show this. The authors describe results produced by methods that are not explained or published anywhere else, present analyses that they have already described in at least two preprints or that have been shown before in the literature or show results that are merely anecdotal and do not prove anything that is specific or relevant to their datasets.

However, the authors present results obtained with bambu, a method that is not fully described. And the text and figures appear as if that part of the paper is the benchmarking bambu, rather than a description of their datasets.

Perhaps the authors should convert this article into the bambu article.

Response:

We thank Reviewer #1 for pointing out that the use of unpublished computational methods has confounded the message of the manuscript. Since the first submission, all methods used in the manuscript have been peer reviewed and published. Furthermore, we have analysed the data using alternative computational methods to demonstrate the robustness of our analysis. We believe that the data set is very helpful for benchmarking and development

of new computational methods, and several studies have already used the data resource in such a way. However, we did not perform a benchmark or quantitative comparison of computational analyses methods, as we believe that this is beyond the scope of this project. In the revised manuscript we emphasise the unique value of the SG-NEx data, which we hope leads to a clearer and more focused message, and improved manuscript.

Reviewer #1:

G. References: appropriate credit to previous work?

The manuscript fails to cite many articles that already presented the results described in the manuscript. Some of these examples are mentioned above.

Response:

Originally we have focused on the Nanopore RNA-Seq data and mainly cited other findings based on Nanopore RNA-Seq data (the revised manuscript provides a more systematic comparison by including short read and PacBio IsoSeq in the technology comparison). We agree that findings using other technologies should still be cited as well. We thank Reviewer #1 for highlighting the examples of references that were missing, some of which using different technologies (eg PacBio or m6A-Seq), some of which using Nanopore RNA-Seq data. We have included these references and additional references that were missing in the original manuscript or which have been published since then. We hope that this addresses the reviewer's comment, but we very much appreciate any further advice if we have missed any additional key reference.

Reviewer #1:

H. Clarity and context: lucidity of abstract/summary, appropriateness of abstract, introduction and conclusions

The text fails to cite prior results and be clear about the methodologies used. Additionally, the actual aim of the paper is not clear. Although the abstract and introduction appear to have a clear direction, the main text is a mix of results from tools that have not been described, results that are partly already published or discussed elsewhere, or results that do not provide any novelty. As a consequence, the value of the datasets and the analyses performed remain unclear.

Response:

We again would like to thank Reviewer #1 for the detailed review about our manuscript.

The goal of this work was to highlight the SG-NEx data set, which we believe stands out in many ways. While the abstract and introduction already motivated the need for a comprehensive long read RNA-Seq data resource, the results and discussion heavily focused on describing the data set, but not the unique value and novelty of this resource. In response to comments from all Reviewers and in discussion with the editor we have made major changes to the revised manuscript to focus on the value of the data resource, include

additional data and results, and improve the analysis. We are grateful to all reviewers for their comments and suggestions, which we believe have led to a more focused, and greatly improved manuscript.

Reviewer #2 (Remarks to the Author):

In this manuscript, Chen & colleagues describe a set of nanopore cDNA, direct RNA and Illumina cDNA sequencing datasets, SG-NEX, on which they benchmark aspects of gene expression. The dataset is comprehensive and will undoubtedly be very useful to nanopore researchers and tool developers. Other groups have previously published similar benchmarking datasets & technological analyses (Schulz et al., 2021; Sessegolo et al., 2019; Soneson et al., 2019; Workman et al., 2019). The strength of the SG-NEX dataset over these previous datasets is the range of different cell lines that are sequenced, although I feel that this element is underutilised in the manuscript, which instead mostly focuses on comparing the different technologies. Many of the technological findings have been previously shown by other benchmarking papers. The manuscript could be strengthened by the addition of biological insights gained from the dataset.

Response:

We are very happy to read the positive comments from Reviewer #2 highlighting the value of the SG-NEx data sets. Following the comments from Reviewer #2, we have included new analysis, data, and made adjustments to the manuscript to better illustrate the unique value of the SG-NEx data and provide additional insights that go beyond previous studies.

To showcase the unique value of the SG-NEx data, we now compare all 5 major RNA-Seq protocols (ONT direct RNA-Seq, ONT direct cDNA-Seq, ONT PCR-cDNA-Seq, Illumina short read RNA-Seq, and PacBio IsoSeq which was included for this revision), we quantify results using matched replicates, which were generated from the same RNA extract that was used with the different short read and long read library preparation and sequencing methods; and we quantify results across 6 different spike-in RNA sets and across multiple SG-NEx cell lines to ensure that our findings are not just limited to observations from a single cell line or spike-in control. Furthermore, we have re-basecalled the entire nanopore sequencing data set using an identical, recent version of the basecalling software to keep the resource up-to-date with advances in data processing. Finally, all the data is available through the AWS Open Data Registry with detailed documentation and data analysis tutorials. All of these aspects are unique to the SG-NEx data, and we hope that the revised manuscript now better utilizes and emphasises the value of this comprehensive resource.

To provide additional insights that have not been previously reported, we have selected 12 fusion genes and transcripts for validation (as suggested by Reviewer #2, see below), some of which have not been previously described. Furthermore, we now included additional analyses to better understand the observed differences in transcript expression between short and long read RNA-Seq data. In the revised manuscript, we now show that

fragmentation of long read RNA-Seq recapitulates these differences, suggesting that short read RNA-Seq transcript expression estimates are partially over-estimated due to shorter read length.

Please find the detailed response to all points below. We again would like to thank Reviewer #2 for their positive comments and helpful suggestions, and we hope that these new analyses and findings are reflected in an improved manuscript and description of this comprehensive data resource.

Reviewer #2 (Point 1):

- The gene and transcript level annotation, and quantification of gene expression was performed using a tool called bambu, which is developed by the Goeke lab. The tool is not yet published and I could not find a description of the methodology. How does bambu perform transcript discovery and quantification? A more detailed description should be provided in the materials and methods.

Response:

This point was raised by other Reviewers as well. Since the original manuscript was submitted, Bambu was published (Chen et al. Nature Methods (2023)). However, to demonstrate the reproducibility of the results in this manuscript, we have generated results with Bambu (long read), Salmon (long read and short read), NanoCount (long read), and RSEM (short read), which demonstrate that similar results are obtained when alternative methods are used for gene and transcript expression quantification. Please refer to the Point 2 below with a detailed response.

Reviewer #2 (Point 2):

- The authors indicate that transcript level counts are less well correlated with short read transcript level counts. This was previously described by Sonesson et al, benchmarked quantification methodologies for nanopore cDNA and direct RNA datasets and showed that minimap2 alignment followed by using Salmon in alignment-based mode performed best in terms of correlation of ONT transcript level counts with Illumina counts. How much does the authors finding depend upon quantification method? How does minimap2+salmon quantification perform on their data?

Response:

In our original manuscript we have shown that transcript expression estimates show lower correlation between short read and long read RNA-Seq compared to gene expression estimates (using Bambu). As pointed out by Reviewer #2, this result could be influenced by the choice of the computational method that is used to estimate gene and transcript expression for short and long read RNA-Seq data.

Following the suggestion from Reviewer #2, we have performed the analysis to compare gene and transcript expression between short and long read RNA-Seq using Salmon (long

read option) + minimap2 (long read RNA-Seq) and Salmon (short read RNA-Seq). Additionally, we also used RSEM (short read), and NanoCount (long read) as alternative quantification methods to further evaluate the robustness of our findings when different quantification methods are used. In addition to quantifying the correlation, we have also quantified alternative metrics using the different computational methods (See Supplementary Figure 3-4).

The results show that the comparison of short and long read RNA-Seq data is similar when alternative quantification methods and metrics are used. In particular, we show that (1) nanopore long read RNA-Seq shows higher correlation with expected spike-in concentrations for gene and transcript expression compared to short read RNA-Seq data (Figure R2.1a,c vs Figure R2.1b,d, Supplementary Figure 4a,h,i, Supplementary Figure 5a,j,k), (2) correlation in gene expression between short and long read RNA-Seq being higher than correlation in transcript expression (Figures R2.1a-b vs Figure R2.1c-d), and (3) correlation in transcript expression between short and long red RNA-Seq being higher for major isoforms identified by long read RNA-Seq whereas major isoforms identified by short read RNA-Seq show lower correlation (Figure R2.1d).

These results confirm that transcript expression estimates between short and long read RNA-Seq are less correlated than gene expression estimates, regardless of which computational methods are used. Since major isoforms identified by short read RNA-Seq appear to be less robustly quantified across short and long read RNA-Seq, we investigated if read fragmentation could cause overestimation of transcript expression in short read RNA-Seq data. To test this, we simulated read fragmentation and short read RNA-Seq data from the original long read RNA-Seq data by sampling 150bp, error-corrected reads from the long reads that were aligned to the transcriptome (Figure R2.2a). Compared to the original transcript expression estimates, read fragmentation indeed leads to a significant over-estimation for short read-specific major isoforms (Figure R2.2b-c) and increased correlation with short read RNA-Seq data (Figure R2.2b, please also refer to Figure 4i-k and Supplementary Figures 5o,p for additional details). Together, these results indicate that the high variation in transcript expression estimates between short and long read RNA-Seq data are not caused by the choice of analysis methods, but that they are most likely due to differences in library preparation and sequencing technologies, with read fragmentation partially explaining the observed differences for major isoforms.

In the revised manuscript, we have now included results from all methods and alternative evaluation metrics. We have used Salmon for the comparison of short and long read RNA-Seq as it is the only method that has an option to quantify expression for both data types (results with Bambu and RSEM, NanoCount are found in Supplementary Figures 4-5). For the analysis of alternative isoform expression and fusion transcripts we have used Bambu in the main figures, as Bambu is the only method that provides full length read counts to reduce possible false positives (results with Salmon, RSEM, and NanoCount are shown in Supplementary Figures 5-7). Furthermore, we have included the new results showing that differences in transcript expression are partially explained by read fragmentation and differences in major isoform quantification, which has not yet been reported and quantified previously (please also refer to our response above for a comparison with previous studies). We thank Reviewer #2 for this comment and hope that the revised

manuscript now provides a more complete, robust, and detailed summary of the differences between short and long read RNA-Seq data.

Figure R2.2: Comparison of short read RNA-Seq data and fragmentation simulated short read RNA-seq data with long read RNA-Seq data

(a) Main Figure 4h Schematic showing the fragmentation simulation of short read from long read

(b) Scatterplots showing fragmentation simulated short read or short read RNA-seq data against long read RNA-seq data

(c) Main Figure 4j Boxplots of mean absolute differences between long read RNA-Seq data and fragmentation simulated short read RNA-seq data for isoforms that are major in both long and short read RNA-seq data (Major isoforms), major isoforms that are only major in long read RNA-seq data (Major-LR isoforms), major isoforms that are only major in short read RNA-seq data (Major-SR isoforms), and minor isoforms.

Reviewer #2 (Point 3):

- In the section on linked splicing events, the authors state that 12.8% of splicing changes involve multiple events. This does not necessarily imply that these events are linked, i.e. occur in a co-dependent or mutually exclusive manner. To take the example of alternative promoters and exon skips that the authors use, there may be transcripts where alternative promoters and exon skipping occur together, but this may not occur more frequently than expected by chance given the background level of both splicing events. Have the authors used any statistical tests to verify these co-dependent or mutually exclusive splicing events?

Response:

In our original manuscript we have reported that alternative isoforms frequently show changes involving multiple exons. Using the full length reads provides a way to study the association of such long range events, however, we have not used a statistical test to identify which of these events are dependent or mutually exclusive. While we find that this analysis would be of great interest, we were unsure about the possible test for independence that we could use, and we could not find any existing method that describes a similar test. As an approximation we have calculated the fraction of linked splicing events which occur only in this specific combination for each gene, providing a lower limit of the fraction of such splicing events that are co-dependent (Figure R2.3). Overall, we observe a high variation across the different combinations, partially explained by the frequency of the individual splicing events that are observed. Nevertheless, this analysis shows that there is a fraction of such events that only co-occurs in this specific combination, even though this analysis is insufficient to quantify the precise fraction of such sites. In the revised manuscript we have

now included these results as Supplementary Figure 6c. Furthermore, since our observations do not distinguish dependent from independent events, we now rewrote the text to avoid suggesting that the identification of co-occurring events indicates co-regulation or mutual dependency, and which we believe is an important research question for possible future studies (which could be done using the SG-NEEx data).

Figure R2.3 Co-occurring of alternative isoform events

Light blue represent number of genes with event co-occur, and darker blue represents for number of genes with event co-occur only, i.e., no independent event occurring for alternative event in this combination within each gene

Reviewer #2 (Point 4):

- The authors identified novel genes in repetitive element of the human genome. It is my experience that the error rate of nanopore sequencing is high enough that it is often difficult to unambiguously assign reads to genes when they are in repetitive regions which contain whole gene duplication events (e.g. when gene copies have diverged at a lower rate than nanopore error rate). Is the rate of multimapping higher for the author’s novel genes than for annotated genes? How confident can the authors be that these are transcribed genes and not pseudogene copies with mismapped reads?

Response:

In our original manuscript we have reported that transcripts which are identified using long read RNA-Seq data and which are not found in reference annotations (novel transcripts/novel genes) are enriched in repetitive elements. As pointed out by Reviewer #2, these transcripts could be artefacts from reads that are mapped to multiple locations in the reference genome (multi-mapped reads), in particular since the higher error rate in Nanopore RNA-Seq data allows higher alignment error rates.

Following this comment, we have analysed the impact of multi-mapping reads on the identification of repeat-enriched transcripts. To limit the influence of the error rate, we have re-basecalled all data using an identical, recent version of Guppy (version 6.4.2). We have then analysed the long read RNA-Seq data using two different alignment and transcript discovery strategies:

(1) *primary read alignment*: Using only the primary read alignment from the minimap2 default alignment setting (allowing up to 5 multiple alignments per reads, original manuscript);

(2) *uniquely aligned reads*: Only reads that have a single alignment are used for transcript discovery, all reads with multiple alignments are discarded

We then use Bambu for transcript discovery, which provides a confidence score for each transcript (the Novel Discovery Rate (NDR), which approximately corresponds to the False Discovery Rate). A NDR threshold of 1.0 corresponds to the complete set of novel transcript candidates with only minimal filters (at least 2 read counts in any sample and at least 5% contribution to the total expression for the respective gene). For the human genome, which is well annotated, a NDR threshold of 0.1 is recommended, which ensures that novel transcript candidates that have very low read support, or which could arise from sequencing or alignment artefacts are not considered as “valid” novel transcripts.

Using the most stringent alignment option (“unique alignments”) reduces the number of reads by 38% on average across all data sets. This results in a reduction of 25% of novel transcript candidates compared to the original alignment mode (NDR=1.0), indicating that the presence of multi-mapped reads leads to the identification of additional novel transcripts candidates (Figure R2.4a). However, using the recommended NDR threshold (NDR=0.1), most novel transcript candidates are removed, resulting in a comparable number of novel transcripts (Figure R2.4a, unique alignment vs all alignment: 1681 vs 1635 at NDR=0.1). A comparison of the novel transcripts that are identified by the different read filtering strategies shows that 86% are still identified even when only unique reads are considered (Figure R2.4b). Furthermore, even when only uniquely aligned reads are used for transcript discovery, we still observed a similar, significant enrichment of novel transcripts in repetitive elements ($p < 0.001$, Figure R2.4c, corresponding to Supplementary Figure 7g). Together, this indicates that multi-mapping reads have an influence on transcript discovery, but that the enrichment of repetitive elements in novel transcripts is still observed even when multi-mapped reads are removed.

We have included these additional results in the revised manuscript, and we now include the estimated NDR for each transcript from both alignment strategies as a supplementary table to provide an additional measure of confidence for repeat-derived novel transcripts (Supplementary Table 7^(Obj)). Furthermore, we now rewrote the discussion and results to more precisely describe our findings.

Pseudogenes were not included in the enrichment analysis of repeats, which focused on transposable elements and simple repeats as annotated by RepeatMasker (please also refer to Reviewer #3, point [12] for additional details regarding pseudogenes, many of which are annotated and not considered novel transcript candidates). We thank Reviewer #2 for highlighting this point, which was not clearly described in our original manuscript. In the

revised manuscript we now rewrote the results section to specifically describe that we analysed the enrichment of repeats such as transposable elements.

We thank Reviewer #2 for this comment, which we agree is a key concern regarding repetitive elements, and we hope that the revised manuscript now more robustly demonstrates the enrichment of transposable elements in newly discovered transcripts in the SG-NEx data.

Figure R2.4 Novel transcripts identification using primary and reads with unique alignments

(a) Barplots presenting the number of novel transcripts identified when primary alignments of all reads are used vs reads with unique alignments are used with varying transcript discovery threshold ranging from 0.1 to 1, at an increasing step of 0.1 Blue bars represent the number of novel isoforms of annotated genes discovered, green represents the number of novel isoforms of

novel genes discovered. The grey points represent the total number of novel transcripts.
(b) Barplots representing the predicted NDR when using reads with unique alignments for novel transcripts identified when using primary alignments with an NDR threshold of 0.1: light blue represents a predicted NDR < 0.1, blue represents a predicted NDR of ≥ 0.1 and < 0.2, light green represents a predicted NDR ≥ 0.2 , and green represents not identified
(c) Boxplots of percentage overlapping with repeat elements for annotated transcripts with at least 1 full-length read support across samples (Annotated ≥ 1 FL), novel isoforms, and novel gene isoforms using primary alignments for all reads and reads with unique alignments

Reviewer #2 (Point 5):

- The section on fusion transcripts is to me the most interesting and novel element of the study. 25.5% of the fusion transcripts that the authors discover are novel. Can the authors validate any of these fusion transcripts by other means? Are any of the fusion transcripts found in known oncogenes?

Response:

We are happy to hear the positive comment regarding the analysis of fusion transcripts in our manuscript. Following the suggestion from Reviewer #2, we have selected 12 fusion genes from the MCF7 cell line for validation, all of which could be confirmed by Sanger sequencing. Furthermore, we now provide the complete set of fusion transcripts discovered in the SG-NEx data as a supplementary table (Supplementary Table 8).

Following this comment, we have compared all fusion transcripts with cancer-associated genes from the Cancer Gene Census (Sondka et al. 2018) and the PCAWG cancer driver gene mutation list (ICGC/TCGA Pan-Cancer Analysis of Whole Genomes Consortium 2020). This analysis shows that several of the novel fusion transcripts occur at cancer-associated genes, such as *HMGA2:CHMP1A* where the 5' gene *HMGA2* is related to multiple cancers including lipoma, leiomyoma, and pleomorphic salivary gland adenoma; *HNRNPA3:HMGA1* where the 3' gene is related to microfollicular thyroid adenoma, and benign mesenchymal tumours; *WWTR1:THRAP3* where the 5' gene is related to epithelioid haemangi endothelioma and the 3' gene is related to aneurysmal bone cyst, according to the Cancer Gene Census; and lastly *KANSL1:ARL17A* where the 5' gene is in the PCWAG cancer driver gene mutation list.

The results are now included in the main text and as supplementary table, and will make the SG-NEx data an even more useful resource for analysing fusion transcripts and evaluation of computational methods to identify breakpoints and fusion genes.

Reviewer #2 (References):

Schulz L, Torres-Diz M, Cortés-López M, Hayer KE, Asnani M, Tasian SK, Barash Y, Sotillo E, Zarnack K, König J, Thomas-Tikhonenko A. 2021. Direct long-read RNA sequencing identifies a subset of questionable exons likely arising from reverse transcription artifacts. *Genome Biol* 22:190.

Sessegolo C, Cruaud C, Da Silva C, Cologne A, Dubarry M, Derrien T, Lacroix V, Aury J-M. 2019. Transcriptome profiling of mouse samples using nanopore sequencing of cDNA and RNA molecules. *Sci Rep* 9:14908.

Soneson C, Yao Y, Bratus-Neuenschwander A, Patrignani A, Robinson MD, Hussain S. 2019. A comprehensive examination of Nanopore native RNA sequencing for characterization of complex transcriptomes. *Nat Commun* 10:3359.

Workman RE, Tang AD, Tang PS, Jain M, Tyson JR, Razaghi R, Zuzarte PC, Gilpatrick T, Payne A, Quick J, Others. 2019. Nanopore native RNA sequencing of a human poly (A) transcriptome. *Nat Methods* 16:1297–1305.

Reviewer #3 (Remarks to the Author):

The manuscript describes a comprehensive long-read dataset for several cancer cell lines, generated with several RNA-seq protocols and matching short-read RNA-seq. The Authors claim ONT technology can replace short-read sequencing for RNA-seq analyses; however, I am not convinced by the Authors' arguments. I believe this paper will benefit from an unbiased assessment of ONT shortcomings and a clear demonstration of its advantages.

Response:

We thank Reviewer #3 for the detailed assessment, comments and suggestions regarding our manuscript. To address the points raised by Reviewer #3, we have made several changes with the aim to provide a more robust and unbiased assessment and comparison of current RNA-Seq library preparation and sequencing methods. In particular, we now include additional PacBio IsoSeq data, we compare short read and long read RNA-Seq using 6 different spike-in RNA sets, and we quantify the results using matched replicates where the same RNA was used for sequencing with different library preparation methods. To increase the robustness of our analysis, we now generated results with 5 different transcript quantification methods (Salmon - short read; Salmon -long read; RSEM; Bambu; NanoCount), and we generated results with two different alignment strategies (with and without multi-mapping reads). To further control for variation in error rates, we have re-basecalled the entire Oxford Nanopore data set using an identical, recent version of Guppy (6.4.2).

Following the suggestion to provide a more unbiased assessment of the shortcomings and advantages of long read RNA-Seq, we now include all 5 RNA-Seq protocols in the technology comparison (PacBio IsoSeq, ONT direct RNA-Seq, direct cDNA-Seq, PCR-cDNA-Seq, Illumina short read RNA-Seq). Furthermore, we made major changes to the comparison of short read and long read data to provide specific and quantifiable insights. Among others, we now compare the replicability within the same sequencing technology (higher for short read RNA-Seq), and across sequencing technologies (higher for long read RNA-Seq). To better understand this observation, we have expanded the framework to simulate short reads from long read RNA-Seq to specifically study the impact of fragmentation on transcript expression quantification. The comparison of long read RNA-Seq, fragmented long read RNA-Seq, and short read RNA-Seq highlights that fragmentation increases transcript expression estimates for isoforms that are otherwise only identified by Illumina short read RNA-Seq to be highly expressed. These results suggest that fragmentation partially explains the differences between short read and long read RNA-Seq data.

Based on the new data and analysis we have rewritten the manuscript and adjusted our claims to be more specific. In particular, instead of suggesting that long read RNA-Seq improves almost all aspects, we now specifically describe that long read RNA-Seq more robustly identifies major isoforms, whereas major isoforms that are only identified by short read RNA-Seq are partially explained by a quantification bias that is introduced in the fragmentation step.

Please find our detailed response to each point below. We hope that the revised manuscript presents a more systematic and unbiased comparison of the different RNA-Seq technologies, and better highlights the unique value of the SG-NEx data as a comprehensive resource and benchmark data set for all major RNA-Seq protocols.

Reviewer #3:

1. The cost is crucial for a fair comparison between technologies. The best comparisons are made for sequencing done at the same price point. If this is impossible (since ONT requires a higher sequencing cost), then the price difference should be clearly stated, and it needs to be discussed whether the benefits of the technology outweigh the cost increase

Response:

We agree that the cost of sequencing is a key consideration for designing experiments. In response to this comment, we have therefore estimated the cost per sample for Nanopore RNA-Sequencing in comparison to short read sequencing. Nanopore provides two sequencing platforms, the MinION/GridION (lower price compared to short read RNA-Seq) and the PromethION (high throughput, comparable price to short read RNA-Seq) (Figure R3.1a). Among the different Nanopore RNA-Seq protocols, the PCR-cDNA is most similar to short read sequencing as it uses both reverse transcription into cDNA and PCR amplification. The recent sequencing kit for the PCR-cDNA kit (SQK-PCS110) on PromethION generates a throughput that is comparable to short read RNA-Seq (Figure R3.1b), illustrating that Nanopore long read RNA-Seq does not require a cost increase at the same throughput compared to short read sequencing.

The direct RNA-Seq protocol generates less throughput while being substantially more costly as there is yet no multiplexing option available. The key benefit of direct RNA-Seq is the ability to profile RNA modifications, which is otherwise only possible with experimental protocols such as miCLIP or m6ACE-Seq, at a substantial extra cost compared to direct RNA-Seq.

The comparison of sequencing cost and throughput is now included in the revised manuscript (Supplementary Figure 1a-b).

Figure R3.1: Comparison of cost and expected throughput
(a) Cost of sequencing per sample when 1 to 12 samples are sequenced when using MinION/GridION, PromethION, and Illumina platforms. Cost calculations are based on charge from the service provider.
(b) Sequencing throughput was approximated based on the sequencing throughput of the most recent experiments multiplied by the mean read length for each of the protocols in SG-NEx samples.

Reviewer #3:

2. Both short and long RNA-seq were done in multiple replicates; however, the replicates are not compared. Comparing the gene expression between replicates for long and short reads provides a good metric for the reproducibility of the technologies.

Response:

Two of the unique aspects of the SG-NEx data are (1) the availability of multiple biological replicates for all sequencing protocols, and (2) that the same RNA samples were sequenced with multiple protocols (matched replicates). However, as pointed out by Reviewer #3, in our original manuscript, we have not used these aspects to compare the reproducibility of gene and transcript expression for the different sequencing technologies and library preparation protocols. Following this suggestion, we now compare biological replicates and matched replicates in the following scenarios:

(1) Replicability using the same platform and protocol

First, we compared gene and transcript expression estimates across biological replicates from each protocol to evaluate the replicability within each technology. Here, short read RNA-Seq showed higher correlation across replicates compared to the nanopore long read RNA-Seq protocols both for gene and transcript expression (Figure R3.2). When only protein coding genes are considered, replicability improves for all RNA-Seq protocols (Figure R3.2a). For transcript expression, a higher correlation across biological replicates was

observed for all protocols when only major isoforms were considered (Figure R3.2b). These results indicate that replicability within the same technology is higher for short read RNA-Seq compared to nanopore long read RNA-Seq, possibly reflecting a higher variation in throughput, among others.

Figure R3.2 Reproducibility between biological replicates
 Reproducibility between biological replicates for a) gene expression and b) transcript expression. For transcript expression, only transcripts from proteining coding genes are included.

(2) Replicability across different platforms and protocols

To study the replicability across different RNA-Seq protocols, we compared the correlation in transcript expression for isoforms which are identified to be the most highly expressed isoform for each gene (major isoform). To better understand differences in sequencing technologies, we analysed isoforms that were estimated to be most highly expressed in both short read and long RNA-Seq (major isoform), only in long read (major-LR isoform), only in short read (major-SR isoform), and the remaining set of minor isoforms. For this comparison, we used matched replicates from all cell lines.

Major isoforms identified in both short and long read RNA-Seq data showed the highest correlation between long and short read RNA-Seq data, as expected (Figure R3.3). Among the technology-specific major isoforms, those identified by long read RNA-seq showed significantly higher correlation between short and long read RNA-Seq data compared to short read-specific major isoforms (Figure R3.3, $p < 0.0001$). These results indicate that major isoforms identified by long read RNA-Seq show a higher replicability across the different sequencing technologies.

(3) Simulated read fragmentation recapitulates short read-specific transcript expression

To further investigate the source for the discrepancy in replicability within technologies (higher for short read) and across technologies (higher for long read), we followed the suggestion from Reviewer #3 (point [10]) to simulate short read fragmentation, thereby generating simulated, matched short read RNA-Seq samples for each long read RNA-Seq sample in the SG-NEx resource (please refer to our response below for additional details and analyses). When we fragment the long read RNA-Seq data that was aligned to the transcriptome to 150bp single end, error-corrected reads (Figure R3.4a-b) and process the data using the short read RNA-Seq quantification method, we observe that expression of short read-specific major isoforms is significantly overestimated compared to the identical (long read) data without the simulated fragmentation (Figure R3.4c), resulting in higher correlation with the Illumina short read RNA-Seq data (see Response to points [9] and [10] for additional details).

This suggests that differences between short read and long read RNA-Seq are partially explained by the fragmentation step, with long read RNA-Seq generating more robust transcript expression estimates that have a higher replicability across technologies.

Summary

Together, these results show that short read RNA-Seq has a higher level of replicability when the same technology is used, but that transcript expression estimates appear to be consistently over-estimated for short read-specific major isoforms due to read fragmentation. In contrast, long read RNA-Seq shows higher levels of variation among replicates, but transcript expression estimates are more reproducible across technologies. For a detailed comparison of short read and long read RNA-Seq data, please also refer to our response to Reviewer #3 points [9] and [10]. We have included these new results in the revised manuscript as Figures 4b-c and h-k and Supplementary Figure 5o and in the Discussion. We thank Reviewer #3 for this comment and suggestion, and we hope that these new analyses contribute to a more unbiased comparison of the different RNA-Seq technologies in our revised manuscript.

Reviewer #3:

3. Looking at the Supp. Table 1, it seems that ONT runs have widely varying numbers of sequenced reads. This is a known disadvantage of ONT and should be discussed in the text. Were all the replicates/runs combined for the expression analyses?

Also, it makes it tricky for a fair estimation of the price point. For instance, among the A549 libraries, one run contains 12M reads, which is 8 times larger than the average run (~1.5M reads). It looks like sequencing depths for short RNA-seq are missing from Supp. Table 1.

Response:

As pointed out by Reviewer #3, sequencing depth shows higher variation for the nanopore long read RNA-Seq data compared to the short read RNA-Seq data. The five main factors

that cause this variation are (1) that the different protocols (direct RNA-Seq, direct cDNA-Seq, PCR-cDNA-Seq) generate different read numbers, (2) that the sequencing devices (MinION/GridION vs PromethION) generate different throughput, (3) that multiplexing was used for some samples, whereas other samples were sequenced without multiplexing, (4) that some samples were sequenced multiple times and then combined (technical replicates), whereas other samples were sequenced only once, and (5) that improvements in sequencing chemistry have led to higher throughput in the more recent data sets (e.g SQK-PCS109 kit vs SQK-PCS110 kit). In the revised manuscript, we now show the throughput separated by protocol, platform, and multiplexing strategy (Figure R3.5). Furthermore, as suggested by Reviewer #3, we now included the short read data into Supplementary Table 1. Additionally, we have added PacBio data into Supplementary Table 1 as well. We also agree that the variation in throughput for nanopore sequencing is an important consideration, and we now included it as a section in the Supplementary Text.

For the analysis of spike-in RNAs, all samples were combined to account for variation in throughput across the different data sets. For the analysis of gene and transcript expression from human cell line RNA, only technical replicates that were sequenced with the identical platform, flow cell, and library preparation protocol were combined, all other samples were analysed individually. In the revised manuscript, we now show the distribution across all individual replicates for each protocol to illustrate the variation in gene and transcript expression and coverage across all samples.

For the comparison of sequencing cost, we show the data separately for the different protocols, platforms, and multiplexing options, using the estimate from the most recent sequencing runs and kits (H9 and HEYA8 cell lines) as a benchmark for throughput. We specifically indicate the kit version in this comparison for additional clarity.

We thank Reviewer #3 for this comment, which is an important improvement to the revised manuscript to provide a more unbiased and transparent comparison of the different sequencing technologies.

Figure R3.5 Total number of reads sequenced for each sample by protocols, sequencing platforms.

Main Figure 2a Circle points for MinION or GridION samples, square points for PromethION

samples, and triangle points for samples that are demultiplexed from a multiplexed sample.

Reviewer #3:

4. One of the major questions is whether the sequenced reads represent the full-length transcripts, as this is one of the central promises of long-read technologies. The answer to this question is contained in Fig.2f, and it's disappointing: the median transcript coverage is only 40-60%, i.e., only half of the reads have coverage > 40-60%. Additionally, only a tiny % of reads (<10%) represent full-length transcripts. This severe shortcoming is not discussed in the text. Several questions arise here, e.g.:

How does this result compare with the previous ONT-based RNA-seq studies?

For PCR and cDNA protocols, is this due to incomplete RT or incomplete sequencing of the reads (this can probably be answered by checking the RT/PCR adapter sequences at the ends of the reads)?

What % of reads are cut short at 3', 5', and both 3'/5'?

Response:

We agree with Reviewer #3 that the ability to sequence full length reads is a key argument to using long read RNA-Seq. In our original manuscript we have mentioned this aspect, but we did not quantify the fraction of full length reads. To address this point, we followed the reviewer's suggestions and quantified (1) the fraction of reads corresponding to full length transcripts (in comparison against reference annotations), (2) the fraction of completely sequenced reads (using adapter sequences), and (3) the fraction of reads that are cut short at the 3', 5' and both 3' and 5' ends.

(1) Reads corresponding to full length transcripts

Since transcription start and end sites are often not precisely annotated (Kodzius et al. 2006; Pardo-Palacios et al. 2021), we define any read that covers all splice junctions of a transcript as "full length" (corresponding to the commonly used definition of Full Splice Match/FSM (Pardo-Palacios et al. 2021; Tardaguila et al. 2018)(see paragraph below for additional analysis). Using this definition, we observe that the long read RNA-Seq data on average contains 29.2% (direct cDNA), 49.9% (direct RNA), 51.3% PCR-cDNA, and 62.1% (PacBio IsoSeq) full length reads. If we only consider the fraction of full length reads among all spliced reads at protein coding genes, this fraction further increases to 53.2% (direct cDNA), 64.8% (direct RNA), 71.3% (PCR-cDNA), and 76.7% (PacBio IsoSeq) (Supplementary Figure 1f). Following the suggestion from Reviewer #3 we compared these results against a recent study using the direct RNA-seq protocol for samples with different quality values (RIN) (Praver et al. 2023). We find that these samples provide comparable estimates of full length read counts, with higher quality samples having more full length reads (45.4% for all reads, and 59.8% for spliced reads at protein coding genes)

Figure R3.6 Percentage of full-length length reads in SG-NEx samples and publicly available data for different protocols

(a) Boxplots showing the percentage of reads being uniquely or multi-mapped to transcripts, and whether read is full-splice-junction matched to transcript or not (full-length vs partial) for all five protocols for spliced reads originating from protein coding genes in SG-NEx samples

(b) Boxplots showing the percentage of reads being uniquely or multi-mapped to transcripts, and whether read is full-splice-junction matched to transcript or not (full-length vs partial) for all reads and spliced reads originating from protein coding genes for five different degraded direct RNA samples in Praver et al study

Reviewer #3 refers to the average coverage plot (Figure 2f in the original manuscript) that suggests that only a small fraction of reads (<10%) cover the entire transcript annotation. The coverage plot is based on reference annotations, however, the annotated start and end coordinates of the first and last exons are frequently different from what is observed in the data, and are therefore not suitable to estimate full length read counts (Kodzius et al. 2006; Derti et al. 2012; Lin et al. 2012; Ye et al. 2018). To illustrate this, we have calculated the distance of direct RNA-Seq reads from the annotated 3' end of compatible transcripts. Direct RNA-Seq reads are sequenced from the polyA tail, which ensures that all reads except sequencing and alignment artefacts cover the complete 3' end (Niazi, Krause, and Valen 2021). However, the average distance between the observed 3' end and polyAdenylation site from the direct RNA-Seq data and the annotated 3' end is 330 bp (Figure R3.7), illustrating that start and end coordinates from annotations do not provide a reliable measure to identify full length reads. Therefore, while the coverage plot that is based on annotations can still be useful to show differences among the RNA-Seq protocols, it should not be interpreted as an estimate of full length read counts. In the revised manuscript we now clarify this in the figure legend, and we included these additional figures and analyses to provide a more accurate overview of the fraction of full length reads in the SG-NEx samples.

(2) Completely sequenced reads

Due to naturally occurring RNA degradation and the presence of novel genes and isoforms, some reads that do not match a full length transcripts (and which are therefore considered non-full length in the analysis) can still correspond to a completely sequenced reads. The PCR-cDNA protocol contains adapters at the 3' and 5' end, which can be used to identify reads that were completely sequenced. As suggested by Reviewer #3, we estimated the fraction of completely sequenced reads by searching for the presence of 3' and 5' adapters in the PCR-cDNA protocol using Pychopper (<https://github.com/epi2me-labs/pychopper>). In total 91% of all reads have both adapters and correspond to completely sequenced reads (Figure R3.8). These results suggest that incomplete sequencing only contributes to a small fraction of non-full length reads.

Figure R3.8 Percentage of reads contain both 5' and 3' primers for SGNex_Hct116_cDNA_replicate3_run5

Note:

Both primers found reads included reads containing strand-switching primer (SSP) and anchored oligo(dT) VN primer (VNP) in the correct orientation with default parameters in Pychopper

Rescued fused reads include reads containing multiple non-consecutive right configured primers' combinations ("GitHub - epi2me-Labs/pychopper: cDNA Read Preprocessing" n.d.)

Not usable reads include reads contain no right configured primers

(3) 5' and 3' end coverage

To further investigate non-full length reads, we followed the suggestion by Reviewer #3 and analysed the fraction of reads that cover the 5' exon and the 3' exon of transcripts for the different long read protocols. As expected, the direct RNA-Seq data has the highest fraction of reads covering the 3' exon (median across all SG-NEx samples: 91%, Figure R3.9). However, even for the direct cDNA and the PCR-cDNA protocols, most reads cover the 3' exon (81% and 86% respectively) (Figure R3.9). In contrast, the 5' exon is covered less often (72%, 64%, 80%), possibly indicating the presence of RNA degradation or other protocol-specific limitations (Figure R3.9).

Summary

By avoiding the RNA fragmentation step, long read RNA-Seq can generate reads which correspond to full length transcripts. However, due to naturally occurring RNA degradation and due to sequencing or library preparation artefacts, not all reads correspond to full length reads. Following the suggestions from Reviewer #3, we have now in depth analysed the fraction of full length and completely sequenced reads, their transcript coverage, and compared observations from the SG-NEx data with other recently published data. Our analysis indicates that more than 90% of reads are completely sequenced (PCR-cDNA), and that the fraction of full-length reads is 50% and higher for all long read protocols, with the exception of the direct cDNA protocol. However, if only spliced reads at protein coding genes are considered, the fraction of full length reaches 53% (direct cDNA) to 76% (PacBio IsoSeq). We now include these new results in the revised manuscript (Figure 4e and Supplementary Figure 1f Supplementary Text Figures 1-4). We agree with Reviewer #3 that this is a key promise of long read RNA-Seq and we hope that the revised manuscript now provides a more complete picture of full length reads observed in the SG-NEx data.

Reviewer #3:

5. The units are not given for expression scatter plots Fig.3a-f, presumably it's log₂(TPM)? While the scatterplots provide a general idea about correlations, it's also essential to quantify the change in expression. This can be done, for instance, with a histogram for relative deviation. It seems that there will be a large number of genes with significantly different expressions in long vs. short methods, and so the gene expression estimates between ONT and short reads cannot be called "highly comparable."

Also, in addition to Spearman R, the Pearson of $\log_2(\text{TPM})$ would be informative.

Response:

In our original manuscript we have provided scatterplots to compare gene expression between long read and short read RNA-Seq data for spike-in RNAs (original Figure 3a-c) and for RNA from human cell lines (original Figure 3d-e). While these scatter plots provide a helpful illustration, they only showed results from a single protocol (direct cDNA) and a single replicate from the A549 cell line (with data from additional cell lines and protocols, original Supplementary Figure 2c). We agree with Reviewer #3 that the similarity in gene expression between the different RNA-Seq protocols should not only be illustrated with example scatterplots, but that these results should be quantified.

A key consideration when designing the SG-NEx resource was the inclusion of multiple matched replicates from the same RNA extracted for each cell line in 4 different RNA-Seq protocols. To address the Reviewers' comment, we now quantified the difference in expression estimates between long read and short read sequencing by comparing data from these matched replicates across all protocols. For spike-in RNAs, the original manuscript only showed results for 1 of the 6 spike-in RNAs, and in the revised manuscript we show results from all spike in RNAs (Sequin MixA V1, Sequin MixA V2, SIRV E0, SIRV E2, Long SIRV, ERCCs). Furthermore, following the suggestion from Reviewer #3, we quantify these differences using the correlation (spearman and pearson, see Supplementary Table 4), relative deviation, and mean squared error. Finally, we also performed this analysis with 4 different computational methods (Salmon for both long and short read, Bambu and NanoCount for long read, RSEM for short read) to demonstrate that the results are consistent and robust when alternative analyses methods are used.

The results show that gene expression estimates both for short read and long read RNA-Seq show high correlation with expected concentrations (ERCC: cDNA-Salmon vs Illumina-Salmon, Spearman correlation $\rho=0.98$ vs $\rho=0.95$, Figure R3.10a). When short read and long read RNA-Seq data is compared, gene expression shows the highest correlation for protein coding genes (A549: $R=0.95$ Figure R3.10b-e). The exception is the PacBio IsoSeq data (which was included for this revision), and which shows lower correlation with short read RNA-Seq (median $\rho=0.79$ for spike-ins) and with Nanopore long read RNA-Seq (median $\rho=0.61$ for spike-ins), likely due to an under-representation of shorter transcripts (Figure 3.11). All results are consistent when alternative methods and evaluation metrics are used (Figure R3.12-3.13).

We have now included these new results and analysis in the revised manuscript (Figures 3, Supplementary Figure 1h-i and Supplementary Figure 3a,e,h,i). We have also included the appropriate axes labels ($\log_2(\text{TPM})$), and we have rewritten claims such that they are supported by an appropriate analysis and statistical test if needed. In particular, the claim that long read and short read gene expression estimates are "highly comparable" (compared to transcript expression) was replaced with "highly correlated" (compared to transcript expression).

We thank Reviewer #3 for this comment, which we believe has helped to improve robustness of the analysis and better highlight some of the unique aspects of the SG-NEX data resource such as the availability of multiple replicates, cell lines, and spike-in RNAs that were previously not described, but which are essential to statistically quantify differences in gene and transcript expression.

Figure R3.10 Main Figure 3a-e Long read RNA-Seq shows consistency in gene expression quantification with short read RNA-Seq data.
(a) Scatterplots of spike-in gene log₂-transformed CPM obtained from long read direct cDNA and PCR cDNA RNA-Seq (using Salmon), short read RNA-Seq (using Salmon), compared against expected log₂-transformed spike-in CPM for five different spike-in RNAs, where light blue points represent Sequin MixA version 1, SIRV E2, and dark blue points represent Sequin Mix A version 2, ERCC, and SIRV E0 + long SIRV RNAs **(b)** Boxplots of spearman correlation between protein-coding genes log₂-transformed CPM (using Salmon) from replicates generated by different protocols, light green representing replicate from different cell lines (inter- cell line) and light blue representing replicate from the same cell line (intra- cell line) **(c)** Boxplots of spearman correlation between long-noncoding RNA genes log₂-transformed CPM (using Salmon) for from replicates generated by different protocols, light green representing replicate from different cell lines (inter- cell line) and light blue representing replicate from the same cell line (intra- cell line) **(d)** Scatterplot of protein-coding genes log₂-transformed CPM obtained from long read direct cDNA (using Salmon) against that obtained from short read (using Salmon) in the A549 cell line. **(e)** Scatterplot of long-noncoding genes log₂-transformed CPM obtained from long read direct cDNA (using Salmon) against that obtained from short read (using Salmon) in the A549 cell line.

Figure R3.11 Transcription by gene length for different sequencing protocols
(a) Cumulative distribution plots for percentage of transcription when ranking genes by gene length for direct RNA-seq (dRNA), direct cDNA (dcDNA), PCR cDNA (cDNA), PacBio long read RNA-seq and Illumina short read RNA-seq
(b) Boxplots of fraction of transcription the top genes with gene length less than 1kb for direct RNA-seq (dRNA), direct cDNA (dcDNA), PCR cDNA (cDNA), PacBio long read RNA-seq and Illumina short read RNA-seq

Figure 3.13 Comparison between sample replicates generated using different sequencing protocols

(a-b) Boxplots of (a) spearman correlation and (b) mean absolute relative differences between log₂-transformed CPM from replicates generated by different protocols, from the same cell line (intra- cell line) for protein-coding genes, long-noncoding RNA (lncRNA) genes, other genes and all genes, using Bambu and Salmon for long read data, RSEM and Salmon for short read data

(c-d) Boxplots of (c) spearman correlation and (d) mean absolute relative differences between log₂-transformed CPM for protein-coding gene isoforms that major in both long and short read (Major isoforms), isoforms major only in long read (Major-LR isoform), isoforms major only in short read (Major-SR isoforms) and isoforms that are not major isoforms (Minor isoforms) from replicates generated by long read protocols (LR) and short read protocol (SR) from the same cell line obtained, using Bambu and Salmon for long read data, RSEM and Salmon for short read data

Reviewer #3:

[6] 5. Fig.2a: tSNE does not quantitatively measure the distance between different libraries. Please show PCA and hierarchical clustering.

Response:

In the revised manuscript we have replaced the tSNE plot with a hierarchical clustering and heatmap of sample to sample correlation (Figure R3.14a, original Figure 3f). We also included the PCA plot in the supplement (Figure R3.14b, Supplementary Figure 4j-m).

Reviewer #3:

[7] 6. Fig.2d compares read lengths among different protocols. While it's clear that the read length is significantly shorter in cDNA and PCR protocols compared to direct-RNA, it would be great to make this clear by showing the reduction of the read length for matched samples.

Response:

Following this suggestion, we have compared the average read length differences for matched samples, which confirms the differences in read length distribution between direct RNA-Seq and cDNA and PCR-cDNA Sequencing (Figure R3.15a). Furthermore, we have compared the difference in the mean read length per transcript using matched replicates from the most recent PCR-cDNA and direct RNA-Seq runs (Figure R3.15b-c). This comparison suggests that most transcripts have similar mean read coverage (mode = 0), while a subset of transcripts show reduced mean coverage (mean = 9% to 16%, Figure R3.15b). This confirms our previous observations when (non-matched) replicates are used (Figure 2b and 2f, Supplementary Table 3), which shows a high similarity for most transcripts, while a specific subset of transcripts appears to show lower coverage with the PCR cDNA protocol, possibly due to limitations in the library preparation protocol.

We have included these results in the revised manuscript as Figure 2b,f and Supplementary Figure 1j-k.

Figure R3.15 Read length distribution comparison

(a) Boxplots of mean read length for all matched samples that are sequenced in all cDNA, direct cDNA and direct RNA protocols

(b) Histograms of coverage differences using mean read length per transcript relative to transcript length between cDNA and direct RNA for all three HEYA8 replicates

(c) Scatterplots of mean read length per transcript between PCR cDNA and direct RNA samples for all three HEYA8 replicates

(d) Main Figure 2f Scatterplot for the mean coverage per gene in PCR-cDNA vs direct RNA across all samples

Reviewer #3:

[8] 7. The short-read gene and transcript quantification is performed with Salmon. The most widely used and trusted quantification tool is RSEM, and getting a second opinion on short-read quantification is needed to increase confidence in the comparative analyses.

Response:

Following the suggestion by Reviewer #3 we have now performed quantification of short read RNA-Seq data with RSEM in addition to using Salmon. The results are similar, independently of the method that is used for quantification and the metric that is used to summarise results, demonstrating the robustness of the analysis both for gene and transcript expression (Figure R3.13, corresponding to Figure 3b-c Supplementary Figure 4h-j, please see response to Reviewer #3 point [5] for more details). In the main figure we are now showing results from Salmon, as it was the only method to provide options for processing both long and short read data. We show results from RSEM in the Supplementary Figures 4h-j, Supplementary Figure 5j-k. In addition, we also show results from 3 different long read RNA-Seq quantification methods to demonstrate the robustness of our analysis (Bambu, Salmon, NanoCount, Figure R3.13, Supplementary Figure 4h-j, Supplementary Figure 5j-k).

Reviewer #3:

[9] 8. One of the most important promises of long-read RNA-seq is that “Long read data improves over short-read data for transcript abundance estimation.” However, no direct evidence to support this claim is given in this section. It may not be easy to prove that long reads provide not just different but better transcript quantification than short reads. One suggestion is to calculate the correlation between replicates and show that it’s higher for long reads than short reads.

[10] 9. The only circumstantial for the above claim evidence is: “a similar level of variation is seen when 150bp paired-end short read RNA-Seq data is compared to the identical data trimmed to 75bp single-end reads (Supplementary Figure 3b) indicating that shorter read length is the main reason for the observed variation for transcript level estimates (Supplementary Figure 3c,d).”

With only two points (75b and 2x150b), it’s hard to judge the dependence of transcript quantification accuracy on the read length. This investigation needs to be extended to a series of trimmed lengths, e.g., 2x125, 2x100, 2x75, 2x50.

Response to points [9] and [10]:

We agree with Reviewer #3 that the potential to improve transcript expression estimation over short read RNA-Seq is a key motivation to using long read RNA-Seq data. As pointed out by Reviewer #3, quantifying any such an improvement is challenging as the ground truth (“true”) transcript expression estimates are largely unknown. In our original manuscript we have provided indirect evidence (such as a higher number of reads that are uniquely assigned to transcript), however, we agree with Reviewer #3 that these analyses do not provide direct evidence that long read RNA-Seq improves over short read RNA-Seq for transcript quantification.

Following this comment, we have now compared transcript expression between short read and long read RNA-Seq data in greater detail to specifically quantify how transcript expression estimates differ between both technologies. Furthermore, we now expanded the short read simulations (as suggested by Reviewer #3) to provide direct evidence that these observed differences are partially a result of overestimation due to read fragmentation.

In particular, we have performed the following new analyses:

(1) Comparison using spike-in RNAs with known expression

Spike-in RNAs provide a way to evaluate the accuracy of transcript expression estimates when the ground truth is known. In our original manuscript, we have compared short read and long read RNA-Seq data using one set of sequin spike-in RNAs. In the revised manuscript, we now show the comparison of short and long read RNA-Seq data using 6 different spike-in sets (Sequin MixA V1, Sequin MixA V2, SIRV E0, SIRV E2, Long SIRV, ERCC). Furthermore, following the suggestion from Reviewer #3 (point [5]), in addition to estimating the correlation coefficient, we now estimate the mean absolute error (MAE), Root Mean Squared Error (RMSE), R-squared (R2), mean absolute relative difference (MARD), mean absolute difference (MRD), and we show the histogram of the absolute differences to quantify the accuracy of transcript expression estimates compared to the expected expression. Finally, we have performed this analysis on all data sets with 5 different computational methods to demonstrate the robustness (RSEM, Salmon (short read), Salmon (long read), Bambu, NanoCount).

When we compared the gene expression estimates to the expected expression, nanopore long read RNA-Seq showed a higher correlation and lower mean absolute error than short read RNA-Seq for all spike-in sets (Figure R3.17a, Table R3.1-3.2). When we compared transcript expression estimates, long read RNA-Seq showed a higher correlation and lower mean absolute error on all spike-in sets, except SIRV E2, where the correlation estimate was lower for the long read direct cDNA protocol (Figure R3.17b, Table R3.3-4). On all data sets, the PacBio long read RNA-Seq data showed higher deviation from the expected, which is most likely due to the enrichment for longer transcripts (Figure R3.11). These results were consistent when evaluated with other metrics and other computational methods (Figure R3.12, Figure R3.18).

Overall, these results suggest that on spike-in RNAs, nanopore long read RNA-Seq generates more accurate results compared to short read RNA-Seq data. However, while the correlation between short read and long read gene expression estimates on spike-in (median $\rho=0.76$ for directcDNA vs Illumina and $\rho=0.81$ for PCR cDNA vs Illumina) is still comparable to the correlation observed in cell lines (median $\rho=0.84$ for direct cDNA vs Illumina and median $\rho = 0.83$ for PCR cDNA vs Illumina), the correlation for transcript expression is vastly different (median $\rho=0.73$ for direct cDNA vs Illumina and median $\rho = 0.73$ for PCR cDNA vs Illumina (spike-in), vs mean $\rho = 0.48$ for direct cDNA vs Illumina and mean $\rho = 0.47$ for PCR cDNA vs Illumina (cell lines)) (see Figure R3.19), indicating that the spike-in RNAs do not sufficiently capture the complexity and challenges of transcript expression estimation for human cell lines. Therefore, while these results suggest that long read RNA-Seq provides an advantage for gene and transcript quantification on spike-in RNAs, the design of spike-ins is currently insufficient to explain the much larger observed difference in transcript expression on human cell line RNA between short and long read RNA-Seq.

Figure R3.17 Spike-in gene and transcript expression quantification

(a)Main Figure 3a Scatterplots of spike-in gene log₂-transformed CPM obtained from long read direct cDNA and PCR cDNA RNA-Seq (using Salmon), short read RNA-Seq (using Salmon), compared against expected log₂-transformed spike-in CPM for five different spike-in RNAs, where light blue points represent Sequin MixA version 1, SIRV E2, and dark blue points represent Sequin Mix A version 2, ERCC, and SIRV E0 + long SIRV RNAs

(b)Main Figure 4a Scatterplots of log₂-transformed CPM for spike-in transcripts obtained from long read direct cDNA and PCR cDNA RNA-Seq (using Salmon), and short read RNA-Seq (using Salmon with bias correction), compared against expected log₂-transformed CPM for for spike-in transcripts of five different spike-in RNAs, where light blue points represent Sequin MixA version 1, SIRV E2, and dark blue points represent Sequin Mix A version 2, ERCC, and SIRV E0 + long SIRV RNAs

Protocol	ERCC	Sequin MixA V1	Sequin MixA V2	SIRV E2	SIRV E0
cDNA	0.98	0.93	0.94	0.89	0.94
directcDNA	0.97	0.96	0.95	0.89	0.99
PacBio	0.88	NA	0.84	NA	0.41
Illumina	0.95	0.91	0.90	0.61	0.59

Table R3.1 Spearman correlation between spike-in gene expression estimation against expected expression levels across protocols, using salmon

NA indicates that spike ins were not used in these samples. The highest correlation is shown in green. Long SIRV are not included here as concentrations are constant and no correlation can be calculated.

Protocol	ERCC	Sequin MixA V1	Sequin MixA V2	SIRV E2	SIRV E0	Long SIRV
cDNA	1.27	1.66	1.19	0.64	0.57	3.58
directcDNA	0.72	0.97	0.82	0.57	0.76	3.42
PacBio	1.10	NA	2.20	NA	2.20	6.80
Illumina	1.64	2.71	2.00	3.42	2.66	3.81

Table R3.2 Mean absolute error between spike-in gene expression estimation against expected expression levels across protocols, using salmon

Protocol	ERCC	Sequin MixA V1	Sequin MixA V2	SIRV E2
cDNA	0.98	0.91	0.95	0.77
directcDNA	0.97	0.95	0.96	0.71
PacBio	0.88	NA	0.82	NA
Illumina	0.95	0.87	0.91	0.77

Table R3.3 Spearman correlation between spike-in transcript expression estimation against expected expression levels across protocols, using salmon

Protocol	ERCC	Sequin MixA V1	Sequin MixA V2	SIRV E2	SIRV E0	Long SIRV
cDNA	1.27	1.42	1.06	1.73	1.60	3.58
directcDNA	0.72	0.99	0.92	1.82	1.84	3.42
PacBio	1.10	NA	2.33	NA	3.76	6.80
Illumina	1.64	2.22	1.87	3.24	2.38	3.81

Table R3.4 Mean absolute error between spike-in transcript expression estimation against expected expression levels across protocols, using salmon

Figure R3.12 Comparing log-2 transformed estimated CPM against log2-transformed expected CPM for spike-in genes
(a) Heatmaps of spearman correlation, mean absolute error (MAE), root mean squared error (RMSE), mean absolute relative difference (MARD), mean relative difference (MRD), and coefficient of determination (R^2), for ERCC, Sequin MixA V1, Sequin MixA V2, SIRV E2, SIRV E0 and Long SIRVs, when using Bambu, NanoCount and Salmon for cDNA, direct cDNA and PacBio data, and Salmon and RSEM for short read RNA-seq data
(b) Histograms of absolute errors (AE) for spike-in genes in ERCC, Sequin MixA V1, Sequin MixA V2, SIRV E2, SIRV E0 and Long SIRVs, when using Bambu, NanoCount and Salmon for cDNA, direct cDNA and PacBio data, and Salmon and RSEM for short read RNA-seq data

Figure R3.19 Spearman correlation between long and short read samples (a-b)Spearman correlation between gene or transcript expression estimates from long read samples and short read samples for both spike-in and cell line samples. Please note that the expected transcript expression is constant for all transcripts in SIRV E0, and low correlation is expected.

(2) Comparison of correlation across sequencing technologies

In our original manuscript, we reported that the correlation for transcript expression estimates between short and long read RNA-Seq is highest for major isoforms, whereas minor isoforms showed a much higher variation. To better understand the differences between short and long read RNA-Seq estimates, we now analysed this in more detail by comparing major isoforms that were identified by both short and long read RNA-Seq (major - both, Major Isoform), major isoforms that were identified by long read only (major - long read, Major-LR isoform), and major isoforms which were identified by short read only (major-short read, Major-SR isoform). For each of these sets, we then compared transcript expression estimates between the different RNA-Seq protocols using matched replicates where the identical RNA sample was used for sequencing (please refer to Reviewer #3 point 2 for a comparison of replicability within the same technology). As expected, the correlation of transcript expression between short read and long read RNA-Seq is highest for major isoforms where both technologies agree (Figure R3.20). In contrast, among the technology-specific major isoforms, short-read specific major isoforms showed much higher variation and lower correlation between technologies ($\rho=0.49$, Figure R3.20). These results suggest that major isoforms which are identified by long read RNA-Seq data are more robustly estimated across different sequencing technologies compared to short read-specific major isoforms. The same results are obtained when alternative computational methods are used for short read RNA-Seq data (RSEM, Salmon) or for long read RNA-Seq data (Salmon, Bambu, NanoCount), or if alternative evaluation metrics are used (Figure R3.13c-d).

Figure 3.20 Transcript expression profiling between long and short-read RNA-seq data
(a) Main Figure 4b Boxplots of spearman correlation between log₂-transformed CPM for protein-coding gene isoforms that major in both long and short read (Major isoforms), isoforms major only in long read (Major-LR isoform), isoforms major only in short read (Major-SR isoforms) and isoforms that are not major isoforms (Minor isoforms) from replicates generated by long read protocols (LR) and short read protocol (SR) from the same cell line obtained using Salmon
(b) Main Figure 4c Scatterplot of log₂-transformed CPM for protein coding gene isoforms that major in both long and short read (Major isoforms), isoforms major only in long read (Major-LR isoform), isoforms major only in short read (Major-SR isoforms) and isoforms that are not major isoforms (Minor isoforms) obtained from long read direct cDNA RNA-Seq (using Salmon) against that obtained from short read RNA-Seq (using Salmon) in the A549 cell line.

Figure 3.13 Comparison between sample replicates generated using different sequencing protocols

(a-b) Boxplots of (a) spearman correlation and (b) mean absolute relative differences between log₂-transformed CPM from replicates generated by different protocols, from the same cell line (intra-cell line) for protein-coding genes, long-noncoding RNA (lncRNA) genes, other genes and all genes, using Bambu and Salmon for long read data, RSEM and Salmon for short read data

(c-d) Boxplots of (c) spearman correlation and (d) mean absolute relative differences between log₂-transformed CPM for protein-coding gene isoforms that major in both long and short read (Major isoforms), isoforms major only in long read (Major-LR isoform), isoforms major only in short read (Major-SR isoforms) and isoforms that are not major isoforms (Minor isoforms) from replicates generated by long read protocols (LR) and short read protocol (SR) from the same cell line obtained, using Bambu and Salmon for long read data, RSEM and Salmon for short read data

(3) Simulated read fragmentation from long read RNA-Seq data

To investigate the impact of read fragmentation on the estimation of transcript expression, we followed the suggestion from Reviewer #3 and expanded the short read simulation framework. In our original manuscript we have simulated short reads by trimming the original short read RNA-Seq data from 150 bp paired end to 75bp single end for a single sample. In the revised manuscript we now simulate read fragmentation using both long read and short read data for all samples in the SG-NEx resource.

3.1 Long read-based simulation:

First we used the long read RNA-Seq data to simulate read fragmentation and short read RNA-Seq by sampling 150 bp, error-corrected short reads from the original long reads that were aligned to the transcriptome. In total, we obtained 67 matched long and fragmented long read RNA-Seq samples. A comparison of the long read RNA-Seq data with the matched Illumina short read RNA-Seq data shows a significantly higher correlation after artificially introducing read fragmentation ($p < 0.0001$, $\rho=0.44$ vs $\rho= 0.49$, Figure R3.21a-b), demonstrating that the simulated fragmentation closely recapitulates the observations from real data (Figure R3.21c). The simulated read fragmentation has the strongest impact on short read-specific major isoforms, whereas major isoforms identified by the long read RNA-Seq data were more robust (Figure R3.21d), resembling the observations from the comparison of long read and Illumina short read RNA-Seq data (Figure R3.20a-b). Short read-specific major isoforms showed a significant over-estimation of transcript expression after read fragmentation when compared against the expected expression, whereas other major isoforms are under-estimated (Figure R3.21e).

Together, these results suggest that read fragmentation introduces a systematic estimation error that leads to over-estimation of transcript expression for short read-specific major isoforms and under-estimation for long read-specific isoforms, and which partially explains the observed differences between short and long read RNA-Seq data.

Figure R3.21 Read fragmentation from long-read RNA-seq data recapitulates the short read RNA-seq data

(a) Boxplots showing the median, upper and lower quartiles, and 1.5 x interquartile range of the spearman correlation between \log_2 -transformed CPM shown for Major isoforms, Major-LR isoforms, Major-SR isoforms, and Minor isoforms between short read RNA-seq data and long read (LR) or matched in-silico simulated short read RNA-seq data (simulated SR)

(b) Spearman correlation between \log_2 -transformed CPM of simulated reads and long reads vs original short read shown for all isoforms

(c) Scatterplots showing fragmented long read or short read RNA-seq data against long read RNA-seq data

(d) Original Figure 4i Boxplots showing the median, upper and lower quartiles, and 1.5 x interquartile range of the spearman correlation for Major isoforms, Major-LR isoforms, Major-SR isoforms, and Minor isoforms between long read (LR) and matched in-silico simulated short read RNA-seq data (simulated SR). Light gray lines connecting the spearman correlation from the same sample pair

(e) Boxplots showing the median, upper and lower quartiles, and 1.5 x interquartile range of the spearman correlation between \log_2 -transformed CPM for Major isoforms, Major-LR isoforms, Major-SR isoforms, and Minor isoforms between original paired-end short read (SR) and short read-based simulated single-end 150bp, 125bp, 100bp, 75bp, 50 bp short reads.

3.2 Short-read based simulation:

Finally, we also expanded the short read based simulation of fragmentation by generating matched samples with 150bp, 125bp, 100 bp, 75 bp, and 50 bp read length for all 21 short

read RNA-Seq samples. Similar to the observations in our original manuscript, we observe that major isoforms are more robust, while minor isoforms are more sensitive to shorter read length (Figure R3.21e). Interestingly, for simulated read length below 100 bp, short read-specific major isoforms do not show a significantly higher correlation with the original short read expression estimates compared to long read-specific major isoforms (75bp: $\rho = 0.81$ (LR) vs $\rho = 0.81$ (SR), $p = 0.67$; 50bp: $\rho = 0.71$ (LR) vs $\rho = 0.75$ (SR), $p = 0.18$). Therefore, even when short read data is used for the simulation, short read-specific major isoforms show sensitivity to reduced read length (Figure R3.21e).

Summary

Following the comment from Reviewer #3 we have systematically compared the differences between short read and long read RNA-Seq data using 6 spik- in RNA sets, matched replicates using identical RNA samples, and a comprehensive simulation framework which resembles the observations made on real data. Our results show that a key difference between short and long read data is the identification of technology-specific major isoforms. While isoforms identified by long read RNA-Seq are robustly estimated even with short read RNA-Seq data, short read-specific isoforms show significantly lower correlation across technologies. These differences can be simulated by artificially fragmenting long read RNA-Seq data, which leads to significant over-estimation of transcript expression for short read-specific isoforms. Together, these new results confirm that read fragmentation is partially responsible for the observed differences in transcript expression, which is avoided when using long read RNA-Seq data.

In the revised manuscript we have included these additional results as Figure 4b-c,h-k, Supplementary Figure 5o-p. Furthermore, we have adjusted our claims to be more specific, and we have quantified all results using statistical tests across replicates. We thank Reviewer #3 for this comment, which we believe has led to a better and more precise description and quantification of the differences between long and short read RNA-Seq data, and we hope that this is reflected in an improved manuscript.

Reviewer #3:

[11] 10. The justification for this claim is also weak: “of the major advantages of long read RNA-Seq data is an improved ability to reconstruct RNAs in repeat-rich regions.” The transcript/gene annotations are based on long, high-quality cDNA sequences, and thus (unlike short reads) should not be affected by repeatedness of the sequence. On the other hand, the alternative explanation could be that the high error rate of ONT reads results in erroneous mapping to regions with high repeat content.

Response:

In our original manuscript we have identified novel transcripts in the SG-NEx samples, and found that they are enriched in highly repetitive, transposable elements compared to transcripts found in reference annotations. Reviewer #3 raises two points:

Firstly, Reviewer #3 points out that the claim in the discussion section that long read RNA-Seq has an improved ability to identify novel transcripts in repeat-rich regions over annotations is not justified by our analysis.

We thank Reviewer #3 for this comment, with which we fully agree. In addition to the explanation provided by Reviewer #3, reference annotations also reflect the samples that were used and which differ from the SG-NEx cell lines, and they might further reflect specific design decisions regarding the exclusion of highly repetitive transcripts. Therefore, we agree that the observed enrichment of repetitive elements in the SG-NEx samples does not provide evidence for an improved ability to identify repeat-rich transcripts over existing annotations. We apologise for this misleading and out-of-context statement, which we have now rewritten in the revised manuscript (see response to comment 15 below as well).

Secondly, Reviewer #3 points out that the enrichment of repetitive elements could be related to reads which are mapped to multiple loci (multi-mapped reads) as a consequence of the higher error rate observed in nanopore sequencing data. To limit the influence of the error rate, we have re-basecalled all data using an identical, recent version of Guppy. We have then analysed the impact of multi-mapping reads on the identification of repeat-enriched transcripts from long read RNA-Seq data using two different alignment and transcript discovery strategies:

(1) primary read alignment:

Using only the primary read alignment from the minimap2 default alignment setting (allowing up to 5 multiple alignments per reads, original manuscript);

(2) uniquely aligned reads:

Only reads that have a single alignment are used for transcript discovery, all reads with multiple alignments are discarded

We then use Bambu for transcript discovery, which provides a confidence score for each transcript (the Novel Discovery Rate (NDR), which approximately corresponds to the False Discovery Rate). A NDR=1.0 threshold corresponds to the complete set of novel transcript candidates with only minimal filters (at least 2 read counts in any sample and at least 5% contribution to the total expression for the respective gene). For the human genome, which is well annotated, a NDR threshold of 0.1 is recommended, which ensures that novel transcript candidates that have very low read support, or which could arise from sequencing or alignment artefacts are not considered as “valid” novel transcripts.

Using the most stringent alignment option (“unique alignments”) reduces the number of reads by 38% on average. This results in a reduction of 25% of novel transcript candidates compared to the original alignment mode (NDR=1.0), indicating that the presence of multi-mapped reads leads to the identification of additional novel transcripts candidates (Figure R3.22a). However, using the recommended NDR threshold (NDR=0.1), most novel transcript candidates are removed, resulting in a comparable number of novel transcripts (Figure R3.22a, unique alignment vs all alignment: 1681 vs 1635). A comparison of the novel transcripts that are identified by the different read filterings strategies shows that 85% are still identified even when only unique reads are considered (Figure R3.22b). Furthermore, even when only uniquely aligned reads are used for transcript discovery, we still observed a

similar, significant enrichment of novel transcripts in repetitive elements ($p < 0.001$, Figure R3.22, corresponding to Supplementary Figure 7). Together this indicates that multi-mapping reads have an influence on transcript discovery, but that the enrichment of repetitive elements in novel transcripts is still observed even when multi-mapped reads are removed.

We have included these additional results in the revised manuscript, and we now include the estimated NDR for each transcript from both alignment strategies as a supplementary table to provide an additional measure of confidence for repeat-derived novel transcripts. Additionally, we rewrote the discussion and results sections to more precisely describe our findings. In particular, we replaced the claim that *“one of the major advantages of long read RNA-Seq data is an improved ability to reconstruct RNAs in repeat-rich regions”* with the more accurate and specific statement that *“one of the main differences between transcripts which are newly discovered in the SG-NEx data compared to annotated transcripts is an enrichment in such repetitive elements”*.

We thank Reviewer #3 for this comment, and we hope that the revised manuscript now more robustly demonstrates the enrichment of repetitive elements in newly discovered transcripts in the SG-NEx data.

Figure R3.22 Novel transcripts identification using primary and reads with unique alignments

(a) Barplots presenting the number of novel transcripts identified when primary alignments of all reads are used vs reads with unique alignments are used with varying transcript discovery threshold ranging from 0.1 to 1, at an increasing step of 0.1 Blue bars represent the number of novel isoforms of annotated genes discovered, green represents the number of novel isoforms of novel genes discovered. The grey points represent the total number of novel transcripts.

(b) Barplots representing the predicted NDR when using reads with unique alignments for novel transcripts identified when using primary alignments with an NDR threshold of 0.1: light blue represents a predicted NDR < 0.1, blue represents a predicted NDR of ≥ 0.1 and < 0.2, light green represents a predicted NDR ≥ 0.2 , and green represents not identified

(c) Boxplots of percentage overlapping with repeat elements for annotated transcripts with at least 1 full-length read support across samples (Annotated ≥ 1 FL), novel isoforms, and novel gene isoforms using primary alignments for all reads and reads with unique alignments

Reviewer #3:

[12] 11. Possibly related to the point above, in Fig.Supp.1b, we see a surprisingly large expression of pseudogenes: ~10% of the reads (just among the top 1000 expressed genes) map to pseudogenes. Then it's stated: "pseudogenes which have higher read counts with long read data, possibly due to improved resolution at repetitive regions or higher error rates of long reads." It is vital to figure out whether this is an issue with ONT reads or the high expression of pseudogenes is a true observation.

Response:

In our original manuscript we have described the observation that long read RNA-Seq data has a higher fraction of reads aligned to pseudogenes. To limit the influence of mis-alignments due to differences in error rates when older versions of the basecalling software are used, we have re-basecalled all data using an identical, recent version of Guppy (version 6.4.2). Furthermore, to provide a more complete picture, we have estimated the fraction of reads aligned to pseudogenes for each sample (instead of providing only the summarised mean for the top 1000 genes). Across all samples, we observe that 3.5% of primary read alignments are assigned to pseudogenes (Figure R3.23). When we discard all reads with multiple alignments, this fraction is reduced to 0.2% (Figure R3.23), reflecting that pseudogenes expression estimates largely originate from multi-mapping reads (Figure R3.23). To test if the sequencing or alignment error could explain the observed read count for pseudogenes, we compared the results to PacBio IsoSeq data, which has a lower error rate (and which we generated for 6 of the SG-NEx cell lines as part of the manuscript revision). The PacBio data shows a lower fraction of reads being aligned to pseudogenes compared to the Nanopore RNA-Seq data (0.1%, Figure R3.23), suggesting that alignment errors might partially explain the observed read count for pseudogenes.

In the revised manuscript we now include these results and a discussion of the error rate for Nanopore RNA-Seq data and the possible impact on alignment and transcript quantification.

In contrast to pseudogenes, which are highly similar to the original gene copy, transcripts which overlap repetitive elements such as LINE, SINE, and LTR transposons often have some unique sequence part, and we do not observe that multi-mapped reads have an influence on these results (see response to point [11] above). We would also like to note that pseudogenes are included in reference annotations and are therefore not considered novel transcripts, and they are also not included in the list of repetitive elements which is based on RepeatMasker. To better clarify this, we now rewrote the manuscript and describe these details in the methods section.

We very much thank Reviewer #3 for this comment, which we think highlighted an important insight into the differences between nanopore and other sequencing technologies.

Reviewer #3:

[13] 10. Another interesting observation is hidden in Fig.4h: ~15% of long reads cannot be assigned to a transcript, compared to <3% for short reads. Again, do the Authors believe that this is a true observation, or is this also caused by the high error rate of ONT reads?

Response:

Following this comment, we have investigated the cause for the larger fraction of reads that cannot be assigned to a transcript in long read RNA-Seq compared to short read RNA-Seq. Most of these reads use a combination of exons that is only observed by a single read (Median: 3.26%, IQR: 2.54%-6.8%, Range: 1.0%-16.5%, Figure R3.24a-c), which are supported by less than 5% of all reads which align to the respective gene (Mean: 1.0%, IQR: 0.5%-2.0%, Range: 0%-32%, Figure R3.24a-c), or which have Novel Discovery Rate (NDR)

above the specified threshold in Bambu (Median: 2.9%, IQR: 2.2%-4%, Range: 1.2%-26.9%, Figure R3.24a-c). Furthermore, we have analysed highly accurate PacBio IsoSeq data for the same cell lines and found a similar fraction of reads that are not assigned to any transcript (Median: 15%, IQR: 14%-16%), indicating this observations is not caused by the higher error rate in the nanopore sequencing data (Figure R3.24d).

Figure R3.24 Investigation for reads incompatible to transcripts
(a-c)Percentage of incompatible reads for each sample classified by the filtering conditions for cDNA, direct cDNA, and direct RNA samples
(d)Number of transcripts per read assigned to for different RNA-seq protocols, and fragmentation simulated reads

Next we tested if the fraction of unassigned reads is reduced when long read RNA-Seq data is fragmented to 150bp long reads (without error correction). Indeed, we find that the fragmentation reduces the fraction of unassigned reads to levels that are similar to short read RNA-Seq data, independently from the higher error rate (Figure R3.24d, Fragmented LR vs SR, 2.1% vs 1.8%).

Together, these results indicate that the difference in the fraction of unassigned reads between short and long read RNA-Seq data is not caused by a higher error rate, but due to read fragmentation which results in a higher number of compatible reads. Noteworthy, in case of the simulated read fragmentation, these reads will be wrongly assigned, suggesting that the low number of unassigned reads might be an artefact of short read RNA-Seq data.

We have included these results in the main text and figure, and we now provide the additional PacBio IsoSeq data as part of the SG-NEx data resource. We thank Reviewer #3 for this detailed observation and suggestion, which in our opinion has led to a very important new result, a more detailed understanding of the differences between short and long read RNA-Seq data, and hopefully a better manuscript.

Reviewer #3:

[14] 11. The number of isoform switching events (Fig 5) is only a few hundred. Across all samples, only 2,202 novel transcripts are identified. Is this number limited by the methodology used, where only full-length reads are used, which constitute a small proportion of all reads? How does this number compare to the novel splice junctions and “exons” identified in short-read sequencing?

Response:

The number of isoform switching events and the number of novel transcripts which are discovered rely on the thresholds that are used. In the original manuscript, we have performed transcript discovery using Bambu with a single recommended threshold, and we have analysed alternative isoforms with Bambu focusing only on major isoforms that are supported by full length reads. Following this comment, we now describe results from alternative thresholds, analyses choices and computational methods to provide more context and better interpretability of the numbers that we report. In particular we included the following additional analyses:

- We now analyse alternative isoforms (1) using Bambu, restricting the analysis to full length reads and major isoforms (original manuscript), (2) using Bambu, restricting the analysis to full length counts but including minor isoforms, (3) using Bambu without the restriction to full length counts, and (4) using NanoCount and Salmon (which do not provide full length counts). The results are referenced in the main text and included as Supplementary Figure 6a,d
- We now perform transcript discovery using Bambu when the transcript discovery parameter (NDR) is varied between 0 (no transcript discovery) to 1 (all novel transcripts are included) (Figure R3.22a, please refer to response to Reviewer # 3

point [11] for more details). While we still use the automatically estimated NDR that corresponds to a high precision and low number of false positives (NDR = 0.316), we hope that this additional analysis provides a more comprehensive overview of transcript discovery independent from the threshold that it uses. Since Bambu is currently the only method that allows transcript discovery with a continuous parameter, we have not compared the results with other methods and short read RNA-Seq data, however, such a comparison is presented among others in other benchmark studies (Pardo-Palacios et al. 2021; Dong et al. 2023; Su et al. 2023; Volden et al. 2023).

Figure R3.25 Alternative isoform detection using different filtering conditions and methods
(a) Barplots of different isoform switching type events in the 7 human cell lines using Salmon, NanoCount and Bambu without full-length filtering
(b) Heatmap showing the expression levels of 947 isoforms showing significant isoform switching events (including minor isoforms) across the 7 human cell lines, the type of events associated with the isoform is indicated at the bottom. Expression is shown for the cell-type-specific isoforms.

Reviewer #3:

[15a] 12. I believe several claims in the Discussion session appear to be not well justified:

“Our data suggests that long reads improve most tasks in transcriptome profiling...”
There is no evidence that long reads improve gene expression analysis.
Fusion detection was not done with short reads so that no comparison could be made.
Isoform switching analysis was not compared with short-read analysis.

[15b] 13. “...identification of novel isoforms associated with well-known fusion genes...”
Only one example of such isoform was discovered, with no further validation.
How many of such isoforms are discovered? An orthogonal experimental validation would be required to make it a strong claim here.

[15c] 14. “Our data suggests that one of the main advantages of long reads is an improved resolution to identify highly repetitive transcripts”
As discussed above, the data suggests that ONT reads often map to highly repetitive regions. However, such alignments can be unreliable because of their high error rate. This claim requires orthogonal experimental validation.

Response:

We thank Reviewer #3 for the comment and suggestions regarding claims that are made in the discussion of our original manuscript. To address these points, we have rewritten the discussion to better reflect the data and results, and we rewrote the abstract accordingly to focus on the value of the data set and the aspects in the technology comparison that we can statistically quantify.

Specifically we made the following changes in response to these comments:

- Reviewer #3: “Our data suggests that long reads improve most tasks in transcriptome profiling...” There is no evidence that long reads improve gene expression analysis. Fusion detection was not done with short reads so that no comparison could be made. Isoform switching analysis was not compared with short-read analysis.

In our manuscript we compare long read and short read RNA-Seq data in terms of throughput, read length, reproducibility, biases, and accuracy of gene and transcript expression estimates. We thank Reviewer #3 for highlighting this claim which we agree is not justified as we have not attempted to compare short read and long read RNA-Seq data for “most tasks in transcriptome profiling”. In the revised manuscript we have removed this claim. Furthermore, we have now included a new paragraph that summarises our findings from the comparison of short read and long reads in the context of previous literature, with an emphasis on the additional value of the SG-NEx data resource.

- Reviewer #3: “...identification of novel isoforms associated with well-known fusion genes...” Only one example of such isoform was discovered, with no further validation. How many of such isoforms are discovered? An orthogonal experimental validation would be required to make it a strong claim here.

In our original manuscript we have identified fusion breakpoints, and then identified and quantified fusion transcripts using full length reads from the long read RNA-Seq data. While the breakpoints can be identified using both short read and long read RNA-Seq data, the unique value of long reads is the ability to identify full length fusion transcripts. As an illustration of this, we presented an example with a fusion transcript that uses an alternative exon that is not observed in the annotated 3' or 5' fusion genes ("novel fusion transcript"). However, while we think this provides a good example to illustrate the value of identifying full length fusion transcripts, the main novelty is the ability to reconstruct and quantify individual fusion transcripts (novel and those using existing splice patterns) using full length reads.

To address the comment by Reviewer #3, we now rewrote the discussion to emphasise that the additional value of long reads is the ability to detect and quantify fusion transcripts (not just novel fusion transcripts). Furthermore, we now quantified the number of fusion transcripts that are detected for each fusion gene (2 on average), and we provide the complete list of fusion transcripts as a supplementary table. Finally, we have also selected 12 fusion genes from the MCF7 cell line including 3 with alternative fusion transcripts for experimental validation (Figure R3.26). All of the selected fusion transcripts could be confirmed (Supplementary Table 9[DOI]).

The SG-NEX data has been made public before publication of this manuscript, and we were very happy to see multiple new fusion detection methods already using this resource for evaluation (Davidson et al. 2022; Chen et al. 2023; Felton et al. 2023). We hope that the additional validation and the Supplementary Table, the addition of PacBio IsoSeq data, and the table with the set of fusion transcripts that we discovered will further increase the value of the SG-NEX resource for benchmarking new fusion detection methods using long read RNA-Seq data.

- Reviewer #3: "Our data suggests that one of the main advantages of long reads is an improved resolution to identify highly repetitive transcripts" As discussed above, the data suggests that ONT reads often map to highly repetitive regions. However, such alignments can be unreliable because of their high error rate. This claim requires orthogonal experimental validation.

In our original manuscript we have compared the fraction of repetitive elements that contribute to novel transcripts and annotated transcripts, reporting that one of the main differences is a significant enrichment of repeat-derived transcripts when performing transcript discovery. We agree with Reviewer #3 that this analysis does not support the claim that long reads have an "improved resolution", as we did not perform this analysis, and instead we now emphasise that our results only show differences between novel transcripts and annotations. Furthermore, following the suggestion of Reviewer #3, we have investigated the impact of reads that are not uniquely aligned and which could be due to the sequencing error rate (see response above for additional details). We also agree that an experimental validation is required to make claims about the validity or relevance of these novel transcripts, which we now emphasise in the discussion. In the revised manuscript, we rewrote this section as follows:

Discussion, 3rd paragraph: “Our data shows that newly discovered transcripts in the SG-NEx data are enriched for repetitive elements compared to annotated transcripts. While the expression pattern, transcript structure, and relevance or function of these transcripts remains to be validated, our results further suggest that there is a high level of transcription from repetitive elements that is not captured in annotations.”

Reviewer #3 (References):

- Annaldasula, Siddharth, Martyna Gajos, and Andreas Mayer. 2021. “IsoTV: Processing and Visualizing Functional Features of Translated Transcript Isoforms.” *Bioinformatics* 37 (18): 3070–72.
- Chen, Yu, Yiqing Wang, Weisheng Chen, Zhengzhi Tan, Yuwei Song, Human Genome Structural Variation Consortium, Herbert Chen, and Zechen Chong. 2023. “Gene Fusion Detection and Characterization in Long-Read Cancer Transcriptome Sequencing Data with FusionSeeker.” *Cancer Research* 83 (1): 28–33.
- Davidson, Nadia M., Ying Chen, Teresa Sadras, Georgina L. Ryland, Piers Blombery, Paul G. Ekert, Jonathan Göke, and Alicia Oshlack. 2022. “JAFFAL: Detecting Fusion Genes with Long-Read Transcriptome Sequencing.” *Genome Biology* 23 (1): 10.
- Derti, Adnan, Philip Garrett-Engele, Kenzie D. Macisaac, Richard C. Stevens, Shreedharan Sriram, Ronghua Chen, Carol A. Rohl, Jason M. Johnson, and Tomas Babak. 2012. “A Quantitative Atlas of Polyadenylation in Five Mammals.” *Genome Research* 22 (6): 1173–83.
- Dong, Xueyi, Mei R. M. Du, Quentin Gouil, Luyi Tian, Jafar S. Jabbari, Rory Bowden, Pedro L. Baldoni, et al. 2023. “Benchmarking Long-Read RNA-Sequencing Analysis Tools Using in Silico Mixtures.” *Nature Methods*, October. <https://doi.org/10.1038/s41592-023-02026-3>.
- Felton, Colette, Alison D. Tang, Binyamin A. Knisbacher, Catherine J. Wu, and Angela N. Brooks. 2023. “Detection of Alternative Isoforms of Gene Fusions from Long-Read RNA-Seq with FLAIR-Fusion.” *bioRxiv*. <https://doi.org/10.1101/2022.08.01.502364>.

- "GitHub - epi2me-Labs/pychopper: cDNA Read Preprocessing." n.d. GitHub. Accessed September 29, 2023. <https://github.com/epi2me-labs/pychopper>.
- ICGC/TCGA Pan-Cancer Analysis of Whole Genomes Consortium. 2020. "Pan-Cancer Analysis of Whole Genomes." *Nature* 578 (7793): 82–93.
- Kodzius, Rimantas, Miki Kojima, Hiromi Nishiyori, Mari Nakamura, Shiro Fukuda, Michihira Tagami, Daisuke Sasaki, et al. 2006. "CAGE: Cap Analysis of Gene Expression." *Nature Methods* 3 (3): 211–22.
- Lee, Sojung, Jayne A. Barbour, Yee Man Tam, Haocheng Yang, Yuanhua Huang, and Jason W. H. Wong. 2023. "Integrating Long-Read RNA Sequencing Improves Locus-Specific Quantification of Transposable Element Expression." *bioRxiv*. <https://doi.org/10.1101/2023.03.21.533716>.
- Lin, Yuefeng, Zhihua Li, Fatih Ozsolak, Sang Woo Kim, Gustavo Arango-Argoty, Teresa T. Liu, Scott A. Tenenbaum, et al. 2012. "An in-Depth Map of Polyadenylation Sites in Cancer." *Nucleic Acids Research* 40 (17): 8460–71.
- Niazi, Adnan M., Maximilian Krause, and Eivind Valen. 2021. "Transcript Isoform-Specific Estimation of Poly(A) Tail Length by Nanopore Sequencing of Native RNA." *Methods in Molecular Biology* 2284: 543–67.
- Pardo-Palacios, Francisco, Fairlie Reese, Silvia Carbonell-Sala, Mark Diekhans, Cindy Liang, Dingjie Wang, Brian Williams, et al. 2021. "Systematic Assessment of Long-Read RNA-Seq Methods for Transcript Identification and Quantification." *Research Square*. Research Square. <https://doi.org/10.21203/rs.3.rs-777702/v1>.
- Prawer, Yair D. J., Josie Gleeson, Ricardo De Paoli-Iseppi, and Michael B. Clark. 2023. "Pervasive Effects of RNA Degradation on Nanopore Direct RNA Sequencing." *NAR Genomics and Bioinformatics* 5 (2): lqad060.
- Sondka, Zbyslaw, Sally Bamford, Charlotte G. Cole, Sari A. Ward, Ian Dunham, and Simon A. Forbes. 2018. "The COSMIC Cancer Gene Census: Describing Genetic Dysfunction across All Human Cancers." *Nature Reviews. Cancer* 18 (11): 696–705.
- Su, Yaqi, Zhejian Yu, Siqian Jin, Zhipeng Ai, Ruihong Yuan, Xinyi Chen, Ziwei Xue, et al. 2023. "Comprehensive Assessment of Isoform Detection Methods for Third-Generation Sequencing Data." *bioRxiv*. <https://doi.org/10.1101/2023.08.03.551905>.
- Tardaguila, Manuel, Lorena de la Fuente, Cristina Marti, Cécile Pereira, Francisco Jose Pardo-Palacios, Hector Del Risco, Marc Ferrell, et al. 2018. "SQANTI: Extensive Characterization of Long-Read Transcript Sequences for Quality Control in Full-Length Transcriptome Identification and Quantification." *Genome Research* 28 (3): 396–411.
- Volden, Roger, Kayla D. Schimke, Ashley Byrne, Danilo Dubocanin, Matthew Adams, and Christopher Vollmers. 2023. "Identifying and Quantifying Isoforms from Accurate Full-Length Transcriptome Sequencing Reads with Mandalorion." *Genome Biology* 24 (1): 167.
- Workman, Rachael E., Alison D. Tang, Paul S. Tang, Miten Jain, John R. Tyson, Roham Razaghi, Philip C. Zuzarte, et al. 2019. "Nanopore Native RNA Sequencing of a Human poly(A) Transcriptome." *Nature Methods* 16 (12): 1297–1305.
- Ye, Congting, Yuqi Long, Guoli Ji, Qingshun Quinn Li, and Xiaohui Wu. 2018. "APATrap: Identification and Quantification of Alternative Polyadenylation Sites from RNA-Seq Data." *Bioinformatics* 34 (11): 1841–49.

Reviewers' Comments:

Reviewer #1:

Remarks to the Author:

The authors have put together in this manuscript a collection of analyses to illustrate what one might do with their SG-NEx comprehensive datasets. Some of the analyses provide an interesting benchmark of the technologies. Others are not compelling and would require additional work. The analyses are also not clearly described, and often they lack enough details or these details are entirely missing.

While the authors motivated this manuscript as a showcase their SG-NEx dataset, the manuscript ends up being something quite different. The authors initially describe SG-NEx project as a systematic resource and benchmark data set, but the benchmarking aspect is only illustrated for gene and transcript expression. After the initial part, the authors motivation to compare the different technologies and discuss how the SG-NEx facilitates novel analyses seems to fade. The authors recurrently fall to the temptation to make biological claims, often without enough support, while at the same time failing to properly highlight how the unique depth and breadth of their data set specifically made possible those analyses. In this sense, they contradict their own intention, and fail to accomplish a convincing manuscript.

The manuscript needs overall proofreading. There is a general discordance between present and past tenses, e.g. “We searched We identify.... “... “We then applied... we find...”, etc. The manuscript also present frequent confusion between genes and transcripts, with often a discordance between the text using one and the figures using the other.

Q1 Response:

We thank Reviewer #1 for carefully reading the manuscript and for the suggested improvements. We have now used a professional manuscript editing service to proofread and edit the revised manuscript. Furthermore, we now went through the text to be consistent and more precise when using scientific terminology. Please also refer to our response below regarding the definition of transcripts, alternative isoforms, and genes. We hope that the revised manuscript has addressed these concerns.

Reviewer #1:

Overall, the details of the analyses performed are poorly described or incomplete. There are multiple discrepancies and missing information, and the article does not present a good balance between the different applications. More importantly, the article does not emphasise enough the advantages or uniqueness of the SG-NEx dataset.

Q2 Response:

Reviewer #1 suggests to improve the description of the analysis, include additional details that were missing in our original manuscript, and better emphasise the advantageous and uniqueness of the SG-NEx data set. To address this point, we (1) rewrote the methods section for more depth and improved structure and clarity, and (2) we now further emphasise the unique value and use-cases for the SG-NEx data set. Finally, we have addressed the

suggestions related to the analyses that were raised by Reviewer #1 in their comments. Please see our response below for additional details.

Reviewer #1:

More detailed questions and suggestions are provided below:

--- description of the datasets ---

The description of the datasets is confusing. The authors describe using 7 cell lines, but then mention 8 additional cell lines. However, these additional 8 cell lines do not seem to be described further in the manuscript. What are the platforms used for these additional cell lines? This is not clearly explained.

The authors also mention 8 cancer types, but they actually sequenced cell lines, and the data is shown for 7 cell lines. They should probably not refer to them as cancer types and be more clear about the data presented in the manuscript versus the data not included.

Q3 Response:

We thank Reviewer #1 for the suggestion to improve the description of the data set, which is an essential component of this manuscript. The SG-NEx data consists of two different data sets:

(1) the SG-NEx core data

The core data set consists of seven human cell lines (colon cancer (Hct116), liver cancer (HepG2), lung cancer (A549), breast cancer (MCF7), Leukaemia (K562), ovary cancer (HEYA8) and the H9 human embryonic stem cell line), which were sequenced with at least three high quality replicates on multiple long read RNA-Seq protocols, with matched short read RNA-Seq data and spike-in RNA controls. This data set was used for all analysis in the manuscript.

(2) the extended data set

This core SG-NEx data set was extended with additional samples from two stomach cancer cell lines (NCC24, IM95), a head and neck cancer cell line (HN1-NPC7), the Hek293T cell line, and three samples from multiple myeloma cancer patients. The extended data set was only sequenced with a single long read RNA-Seq and without the requirement for multiple replicates, with the aim to provide a broader coverage of the human transcriptome. This data set was used together with the core data set to identify novel transcripts.

In the revised manuscript, we now more clearly describe the core SG-NEx data and the extended SG-NEx data, and we specifically describe in each results section which data set was used (the core SG-NEx data, or the extended SG-NEx data).

Following the suggestion of Reviewer #1, we also now refer to these cell lines as cancer cell lines, not as cancer types.

We thank Reviewer #1 for this comment, and we hope that the revised manuscript now more clearly and consistently describes the SG-NEx data resource.

Reviewer #1:

--- gene and transcript expression ---

These comparisons are interesting and highlight some of the differences between the technologies. However, the analyses quickly become quite convoluted.

Response:

We are happy to read the overall positive evaluation by Reviewer #1 about our analysis that compared gene and transcript expression between the different technologies, and we would like to thank Reviewer #1 for the detailed suggestions to improve the presentation of our findings. We hope that the revised manuscript now more clearly describes the results from the comparison of different RNA-Seq technologies. Please find our detailed response to each point below.

Reviewer #1:

The statement “highly-expressed genes were over-represented in the PCR-cDNA ...” It is a bit confusing, given that the high or low expression is defined by the same sequencing reads. In this case, the authors probably mean that genes at the top of the ranked expression estimates account for a larger proportion of the overall abundance.

Q4 Response:

We have rephrased this sentence following the suggestion from Reviewer #1, which we agree more precisely describes this observation:

Main text, section “(3) A comparison of five RNA-Seq protocols”

“In particular, transcripts from the 1,000 genes with the highest gene expression estimate accounted for a significantly larger proportion of the overall transcript expression in the PCR-cDNA protocol compared to the PCR free nanopore RNA-Seq (Figure 2e, two-sided t-test, $p=0.00041$, Supplementary Figure 1g)[...]”

Reviewer #1:

The authors use “gene expression” without explaining how this is calculated. For all technologies expression is estimated in the manuscript at the transcript level. However, gene expression estimates vary quite a lot depending on the technology and methodology (see e.g. <https://f1000research.com/articles/4-1521>). The authors should clearly define their method to calculate gene expression. Is this the most commonly used in the literature?

Q5 Response:

In our manuscript, we define a *gene* as a region/locus in the genome which is transcribed into RNA. We define *transcript* as a RNA product that was generated from a gene through transcription. Due to alternative start and end sites and alternative splicing, genes can generate different transcripts. When a gene uses multiple, different transcripts, we refer to them as *alternative transcripts*, or *alternative isoforms*.

As pointed out by Reviewer #1, all methods in this manuscript estimate transcript expression levels, which corresponds to the relative number of RNA transcripts in a sample. For all long read RNA-Seq methods, we use counts per million (CPM) as an estimate of transcript expression, as no length normalisation is required. For short read RNA-Seq, we use transcripts per million (TPM) to account for fragmentation and transcript length correction. We then followed the same definition to calculate gene expression as Sonesson et al. (2016), with gene expression being estimated as the sum of transcript expression for all transcripts that are associated with the same gene based on reference annotations (“Gene-level TPM estimates, representing the overall transcriptional output of each gene, were obtained by summing the corresponding transcript-level TPM estimates”(Sonesson et al. 2016)). For Bambu, gene expression estimates included reads that could be assigned to a gene, but which were otherwise incompatible with any transcript, and which are represented as an “unknown transcript” CPM estimate.

We now rewrote the text to more clearly introduce genes, transcripts, as well as gene and transcript expression. Furthermore, we now included a section in the methods that specifically described the definition of gene and transcript expression. We thank Reviewer #1 for this comment, and hope that the revised manuscript is more precise and consistent in the use of gene and transcript expression.

Reviewer #1:

The authors show that samples cluster together independently of the technology (Figure 3f) and use this fact to say that “Nanopore RNA-Seq data can be integrated with existing short read RNA-Seq data for gene expression analysis”. This is a big statement that would need further support. They used Spearman correlation, which will test ranks rather than values. Moreover, gene expression estimates just by themselves are rare, they most often involve differential gene expression analysis. It is not shown whether this would also be equivalent.

Q6 Response:

In our manuscript we compare gene and transcript expression between short read and long read RNA-Seq data. While transcript expression estimates show large differences, gene expression estimates are more similar. To quantify this, we have estimated the Spearman correlation (Supplementary Figure 4h), as well as the mean absolute difference (Supplementary Figure 4i). We have then used hierarchical clustering using protein coding gene expression estimates, which shows that short and long read gene expression estimates are clustered by cell line and not by technology. We agree with Reviewer #1, that this clustering does not demonstrate that long and short read RNA-Seq can be integrated.

To further explore the differences between long and short read RNA-Seq gene expression estimates, we have investigated the batch effect associated with each RNA-Seq protocol. Removing the technology batch effect results in a similar heatmap, however, the observed clustering by technology within each cell type can be removed in this way (Supplementary Text Figure 7). To visualise the technology-specific batch effect, we have removed the cell-line specific expression (treating the cell line as batch). The resulting heatmap highlights the remaining, technology-specific batch effect that might impact an analysis when short read and long read RNA-Seq is jointly analysed (Supplementary Text Figure 6). While the

batch effect associated with short and long read RNA-Seq is minor when gene expression estimates are used compared to transcript expression estimates, it is still visible, indicating that this still might impact differential expression estimates.

To analyse the impact of technology on differential gene expression analysis when short and long read RNA-Seq data are combined, we have used DESeq2 to identify differentially expressed genes across all pairs of the SG-NEx core cell lines for two different scenarios: (1) differential gene expression analysis using a single technology (2) combined-technology differential gene expression analysis (long + short read). To account for differences in read counts due to read fragmentation, we have converted CPM (long read) and TPM (short read) estimates to read count estimates, which were then used with DESeq2 (Love, Sonesson, and Patro 2018; Love et al. 2015).

A comparison of the resulting log-fold changes shows a high level of correlation between short and long read RNA-Seq (Sp.R=0.82 to 0.93, Figure R1.1a), with the majority of differentially expressed genes being detected in both technologies (orange points). However, technology-specific differentially expressed genes were still detected (blue points).

Since the SG-NEx cell lines were sequenced with multiple replicates of different RNA-Seq protocols for each cell line, we could perform a combined analysis where the technology was specified as a batch effect in DESeq2. This combined analysis increased the correlation, leading to differential expression results that are highly similar compared to using only long read RNA-Seq data (0.94-0.97), indicating that the technology-specific effect can be moderated in a combined analysis (Figure R1.1b).

Together, our analysis shows that gene expression estimates are highly correlated between long read and short read RNA-Seq data (in particular in comparison to transcript expression estimates). Technology-specific differences still exist, which can lead to differences when short and long read RNA-seq data is compared. These differences become minimal when long read and short read RNA-Seq data are jointly analysed, which allows the modelling of technology-specific effects during the differential expression analysis.

In the revised manuscript we now included additional details in the Supplementary Text Section 4 (“Differential gene expression using long and short read RNA-seq data”), and we rewrote this statement in the main text to better reflect these results. We hope that the revised manuscript provides a more detailed picture of the integration of long and short read gene expression estimates.

Main text, section “(4) Gene expression is robustly estimated across different RNA-Seq protocols”

“A hierarchical clustering of the different samples resulted in the same cell lines being clustered together even when different RNA-Seq protocols were used (Figure 3f). While technology-specific differences in gene expression can be observed (Supplementary Figure 4j-m, Supplementary Text Figures 6-8), they can be modelled and removed during differential gene expression analysis when long and short read RNA-Seq data are available across conditions of interest, as is the case for the SG-NEx core data (Supplementary Text Figure 9). These results indicate that Nanopore RNA-Seq data can be integrated and combined with short read RNA-Seq data for gene expression analysis.”

Reviewer #1:

The authors illustrate the discrepancies in transcript expression estimates in Figure 4b. However, this figure is not easy to interpret, and the caption does not provide sufficient information to understand what is shown in the plot.

Q7 Response:

To compare transcript expression across different RNA-Seq protocols, we have calculated the Spearman correlation coefficient for transcript expression estimates from sequencing experiments that used the same RNA extract, but a different RNA-Seq protocol. To better understand which gene isoforms show the strongest differences between long and short read RNA-Seq, we have separated this analysis into gene isoforms that were identified as

the most active isoform (major isoform) by short and long read RNA-Seq data, which were identified only by long read RNA-Seq data as major isoform, which were identified to be the major isoform only in short read RNA-Seq data, and all remaining isoforms that were identified as minor isoforms (see Figure R1.2, corresponding to Supplementary Figure 5j for a simplified summary of our results).

Figure R1.3b (corresponding to Figure 4c) illustrates this for a comparison of transcript expression estimates between RNA from the A549 cell line that was sequenced with the PCR-cDNA protocol and Illumina short read RNA-Seq data. To quantify the observed differences, we have estimated the correlation for all pairs in the SG-NEx data, and summarised the results as boxplots for all possible pairs of RNA-Seq protocol combinations (Figure R1.3a, corresponding to Figure 4b). The analysis shows that major isoforms that are only identified by short read RNA-Seq have significantly lower correlation compared to major isoforms identified by long read RNA-Seq data, indicating that these have the lowest reproducibility across technologies. To better understand this observation, we have simulated fragmented reads and demonstrated that read fragmentation is responsible for these differences.

We agree that the original Figure 4b was not sufficiently clear from the figure legend, caption, or from the main text. To address this point, we have simplified the figure, and improved the description, and included improved figure legends. We hope that the revised figure now more clearly presents the comparison of transcript expression between short and long read RNA-Seq data.

Figure R1.2 First revision response figure R3.3 Boxplots of transcript expression correlation generated using different RNA-seq protocols (direct cDNA, cDNA, direct RNA, Illumina) for all pairwise matched replicates from 7 cell lines
 Note t-test is performed to test the significance between different transcript groups.
 non-significance ($p > 0.05$) is indicated with ns, and significance of $p < 0.0001$ is indicated with ***

Figure R1.3 Transcript expression estimates comparison between long read and short read RNA-seq protocols

(a) Boxplots showing the Spearman correlation coefficient for transcript expression estimates from sequencing experiments that used the same RNA extract but different RNA sequencing protocols. The protocol comparison pair is indicated by colours below each boxplot, boxplots in dark blue indicate a comparison between long read RNA-Seq protocols, light blue indicates a comparison of long and short read protocols. The correlation was calculated separately for transcripts which were identified as major isoform by long read and short read RNA-Seq, as major isoform only in long read or short read RNA-Seq, and as minor isoforms by long read and short read RNA-Seq. Correlation was calculated using log₂-transformed CPM estimates for protein-coding gene isoforms obtained using Salmon. Boxplots show the median, upper and lower quartile, and 1.5 x interquartile range. **(b)** Scatterplot of log₂-transformed CPM for protein coding gene isoforms that major in both long and short read (Major isoforms), isoforms major only in long read (Major-LR isoform), isoforms major only in short read (Major-SR isoforms) and isoforms that are not major isoforms (Minor isoforms) obtained from long read direct cDNA RNA-Seq (using Salmon) against that obtained from short read RNA-Seq (using Salmon with bias correction) in the A549 cell line.

Reviewer #1:

The authors then focus on the major isoform, which they define as the “most active isoform of each gene”. The authors should use consistent language and perhaps call it the “most abundant.....”.

Also, it is not clarified whether this is defined per cell line or not. The definition of "major isoform" should be clearly stated.

Here is where the text gets more difficult to follow. For instance, “Isoforms that were identified to be major isoforms only in the long read RNA-Seq data (major-long read) showed significantly higher correlation compared to isoforms that were identified only by short read data as major (major-short read)...”. It is not clear what correlation it refers to. Also, the authors should specify how many cases this refers to. Is this a major effect? Or just marginal?

Q8 Response:

In our manuscript, we refer to major isoforms as those isoforms that are estimated to be most abundantly expressed, and we refer to minor isoforms as all other isoforms that are expressed (CPM>1). Since transcript abundance estimates differ between cell lines, we define major and minor isoforms for each cell line, combining all replicates from all long read protocols and all replicates from short read RNA-Seq to obtain long read and short read specific major and minor isoforms for each cell type.

We thank Reviewer #1 for highlighting that the definition was not sufficiently explained in our original manuscript. In the revised manuscript we now specifically define major isoform and minor isoform in the main text. Furthermore, we now include the numbers of such events in the main text (more than 14,000 genes have a different major isoform identified by short and long read RNA-Seq in at least one cell line, making this a major effect), and we included a dedicated paragraph in the methods section to specify how major and minor isoforms are calculated from transcript abundance estimates. We hope that the revised manuscript is more clear and consistent in the description of major and minor isoforms, and better emphasises the substantial impact that technology-specific major isoforms have on transcript expression estimates.

Main text, section “(5) Major isoform expression is more robustly estimated with long read RNA-Seq compared to short read RNA-Seq”:

“To compare differences in transcript expression on RNA from human cell lines, we first identified the most abundantly expressed transcript for each gene in each cell line (“major isoform”) using all replicates of the short read and nanopore long read RNA-Seq. On average, long read and short read RNA-Seq identified the same major isoform for 13,481 (\pm 2021) genes per cell line, whereas they disagree in the major isoform for 7,389 (\pm 523) genes per cell line, resulting in more than 28,000 short read- and long read-specific major isoforms across all cell lines. These numbers are further reflected when transcript expression estimates are compared between replicates from the same cell line (same RNA extract) that were profiled with different sequencing technologies. While gene expression estimates from human cell lines were highly correlated between short read and nanopore long read sequencing, transcript expression estimates showed systematic differences (Figures 4b,c, Supplementary Figure 5c,f-k). The highest agreement in abundance estimates was observed for major isoforms that are shared between long and short read RNA-seq data (Figure 4b,c Supplementary Figure 5f-j). For genes where long and short read RNA-Seq identified different major isoforms, long read-specific major isoforms showed significantly higher correlation of expression estimates between both sequencing technologies ($p < 0.0001$, Figures 4b,c, Supplementary Figure 5g,h).”

Reviewer #1:

The rest of the paragraph remains confusing. For instance “A comparison of these isoforms showed that short read RNA-Seq identified significantly more often isoforms as major that

correspond to truncated versions of the long read major isoforms". The authors refer to Figure 4d, but these are alternative splicing events rather than truncations. One expects that most of the differences between isoforms can be described as alternative splicing events. Also, it is not clear how the length difference (somehow implicit in the word "truncation" is described in this plot).

Q9 Response:

We thank Reviewer #1 for highlighting that this result was not clearly described. For this analysis, we compared the major isoform identified by short read RNA-Seq with the major isoform identified by long read RNA-Seq for each gene. As a control, we compared the long read and short read-specific major isoforms against a randomly sampled isoform for each gene. The results show that the most frequent difference between major isoforms is the use of alternative start and end sites, where the short read-specific major isoforms use first and last exons that correspond to internal exons of the long read specific major isoform. This observation indicates that short read RNA-Seq identifies isoforms as major which are shorter versions of the long read major isoform, even though additional alternative splicing events might be present as well. In the revised manuscript we have rewritten this section to provide more context and details.

Main text, section "(5) Major isoform expression is more robustly estimated with long read RNA-Seq compared to short read RNA-Seq "

"The most frequently observed difference between long- and short read-specific major isoforms was the use of alternative first and last exons. Specifically, major isoforms identified by short read RNA-Seq significantly more often used internal exons from the long read major isoform as first or last exons, typically corresponding to shorter versions of the long read major isoforms (internal first/last exons, Figure 4d, two sided z-test with Bonferroni correction for multiple testing, $p < 0.0001$, Supplementary Figure 5n)."

Reviewer #1:

The last sentence says: "Furthermore, short read-specific major isoforms were frequently estimated to be lowly Expressed ($1 < \text{CPM}$)". This should probably be " $\text{CPM} < 1$ " or "expression < 1 CPM" depending whether CPM is used to represent expression or whether CPM is used as units of measure. Also, this probably means that they must be at the level of noise? Can these transcripts and genes considered "expressed" at all?

Q10 Response:

We thank Reviewer #1 for this comment, the correct sentence should be "Furthermore, short read-specific major isoforms were frequently estimated to be lowly expressed by long read data ($\text{CPM} < 1$, Figure 4e)."

As pointed out by Reviewer #1, the very low expression estimates in long read RNA-Seq data could indicate that they might not be expressed at all, or only at levels comparable to noise. However, these isoforms are still estimated to be highly expressed in short read RNA-Seq data, illustrating that these isoforms represent the most dramatic differences

between long read and short read RNA-Seq data. The simulated read fragmentation experiment suggests that this might be an artefact of short read RNA-Seq data that leads to overestimation, illustrating that long read RNA-Seq generates more robust transcript expression estimates impacting thousands of genes and transcripts in each cell line.

Reviewer #1:

The authors here miss an opportunity to provide more useful insights into the differences between technologies. Are these results just true for nanopore long-read sequencing? Which protocol? Do all long-read technologies give the same result?

Q11 Response:

We apologise that the comparison across different technologies has not been sufficiently clear. We have performed this comparison for all nanopore long read RNA-Seq protocols with consistent results (direct RNA-Seq, direct cDNA-Seq, pCR-cDNA-Seq). The PacBio IsoSeq data was excluded from the results presented in the main text and figures as we observed substantial differences in gene and transcript expression estimates that resulted in unreliable estimates, most likely due to a bias in transcript length (“Among all RNA-Seq protocols, the PacBio IsoSeq data showed the highest estimation error, largely due to the uneven representation of transcript lengths, with low coverage for short transcripts (Supplementary Figure 4a,d,e, Supplementary Figure 1h-i).”). Results from PacBio IsoSeq are shown in Supplementary Figures 4 and 5. Please also refer to our response above for additional details.

Reviewer #1:

The authors indicate “Compared to short read data, ... major isoforms identified by long read RNA-Seq show significantly higher full-length read support”. This result is not easy to see in Supp Fig 5m. Also, it is not clearly defined what the authors mean by “full-length read support”.

Q12 Response:

To address the Reviewer’s comment, we have updated the Figure legend for Supplementary Figure 5m for more clarity.

We define full-length reads as reads that span all exon junctions (please refer to the Supplementary Text Section 1 for a detailed analysis on this topic). In the revised manuscript, we now specifically included this definition in the main text and methods section.

Main text, section “(5) Major isoform expression is more robustly estimated with long read RNA-Seq compared to short read RNA-Seq ”

“[...]and major isoforms identified by long read RNA-Seq showing significantly higher support by reads that span all exon junctions (full-length read support, Supplementary Figure 5m).”

Reviewer #1:

To study the differences between long and short read-based expression estimation, the simulated short reads from long reads through fragmentation. However, they simulated single-end reads, whereas their short-read data is paired-end reads. It is not possible to know whether the results they obtained are due to this difference or not. A proper comparison should use the same type of short reads for both the experimental and the simulated data.

Q13 Response:

The SG-NEx short read data consists of paired end, 150bp Illumina RNA-Seq. The main reason to simulate single end instead of paired end data is the variation in insert size, which is non-trivial to simulate as not all possible insert sizes are compatible with every transcript or long read (Figure R1.4a). To address the reviewer’s comment, we have now generated a simulated, paired end short read data set by sampling the insert size length from the empirical distribution, while limiting the insert size to be smaller than the observed long read length. The results indeed show an even higher similarity with the Illumina paired end RNA-Seq data, confirming our observations from the single end read simulation (Figure R1.4b-c). Since the simulated single end short read data already shows a high similarity with the Illumina short read RNA-Seq data, we have not attempted to generate paired end simulations for all samples, even though we would expect that this might result in minor improvements.

We have included these results in the Supplementary Text Section 5.

Figure R1.4 Paired end simulation results for one sample

- (a) Estimated insert sizes based on “samtools stats” reported IS values on one short read sample
- (b) Scatterplots showing fragmentation simulated single-end short read against paired-end short read RNA-seq data or long read RNA-seq data
- (c) Scatterplots showing fragmentation simulated paired-end short read against paired-end short read RNA-seq data or long read RNA-seq data

Reviewer #1:

The analysis of the simulation from fragmented reads seem to be only relevant in the case of major isoforms defined only by short-reads. The authors indicate that the correlation is impacted and the expression estimation is increased. However, these were described above to have low expression (less than 1 CPM) in the short-read data, and moreover, they are not called as major by long reads, so they may have also a lower long-read support in general.

Given the low counts expected in these transcripts, the results are not very surprising. Also, it is not clear how relevant this result is. The authors do not explain whether this affects many transcripts per cell line. So one is left with the impression that the authors are trying to make a big story about something that is purely anecdotal and irrelevant.

Q14 Response:

These isoforms have very low read support in long read RNA-Seq data (CPM<1), however, they are among the most highly abundant based on short read RNA-Seq data (Figure 4e). Therefore, these isoforms that are identified to be major isoforms only in short read RNA-Seq data represent the most dramatic difference between long and short read transcript abundance estimates. This observation impacts more than 14,000 genes across all cell lines, with more than 28,000 isoforms being only identified by either long or short read as the major isoforms, illustrating the scale that this observation has on transcript expression estimation.

We have included these numbers and additional details in the main text to clarify that this observation has a major impact on interpreting transcript expression estimates from short read RNA-Seq data, please also refer to our response above.

Reviewer #1:

For the analysis of transcripts, the authors simple quote results related to “long reads”. But it is not clear whether all long reads were pooled together, or it is meant that the results were similar across technologies. Are they describing only results from nanopore reads? If so, which protocol? This is not explained at all.

This fact defeats the purpose and the motivation of their dataset and manuscript. The authors motivate this manuscript on the advantages of having multiple technologies. But all long-read technologies seem to be described as one single type.

Q15 Response:

In the revised manuscript we now specify which RNA-Seq protocols are used in the text and in the figures, and we included a supplementary table (Supplementary Table 13) that lists all samples for each analysis (Table R1.1).

Due to the length selection bias that is observed in the PacBio data, these samples were not used for quantification (please refer to our response/Q11 for additional details). For all other protocols, results are shown separately in the section that compares the different technologies. The differences in transcript expression estimates between different long read protocols are much smaller compared to the difference between long and short read RNA-Seq data, therefore we have not explicitly described these results in the main text, however, we have improved the figure to more clearly highlight this aspect (please also see our response above, Q7)

Section name	Dataset	Number of samples	Cell lines/Patient samples	Protocol	NDR
--------------	---------	-------------------	----------------------------	----------	-----

Deep transcriptome profiling of human cell lines using Nanopore RNA-Sequencing	All SG-NEx samples	139	A549, HepG2, HEYA8, HCT116. H9, K562, MCF7, NCC24, N104, N122, HINIPC7, Hek293T, IM95	cDNA, directRNA, directcDNA, Illumina, PacBio	NA
A comparison of five RNA-Seq protocols	All SG-NEx samples	139	A549, HepG2, HEYA8, HCT116. H9, K562, MCF7, NCC24, N104, N122, HINIPC7, Hek293T, IM95	cDNA, directRNA, directcDNA, Illumina, PacBio	NA
Gene expression is robustly expressed across difference RNA-Seq protocols	Core SG-NEx samples with sequencing depth ≥ 400000	80	A549, HepG2, HEYA8, HCT116. H9, K562, MCF7	cDNA, directRNA, directcDNA, Illumina	0.316
Major isoform expression is more robustly estimated with long read RNA-Seq compared to short read RNA-Seq	Core SG-NEx samples with sequencing depth ≥ 400000	80	A549, HepG2, HEYA8, HCT116. H9, K562, MCF7	cDNA, directRNA, directcDNA, Illumina	0.316
Read fragmentation results in over-estimation of expression of major isoforms that are identified in short read RNA-Seq data	SG-NEx samples that can find a replicate in Illumina samples	67	A549, HepG2, HEYA8, HCT116. H9, K562, MCF7	cDNA, directRNA, directcDNA, Illumina	NA
Full-length reads enable the analysis of individual isoform expression	Core SG-NEx samples with sequencing depth ≥ 400000	80	A549, HepG2, HEYA8, HCT116. H9, K562, MCF7	cDNA, directRNA, directcDNA, Illumina	0.316
Novel transcripts are enriched in transposable elements	All SG-NEx samples	111	A549, HepG2, HEYA8, HCT116. H9, K562, MCF7, NCC24, N104, N122, HINIPC7, Hek293T, IM95	cDNA, directRNA, directcDNA	0.1
Detection and quantification of full-length fusion transcripts	Core SG-NEx samples	96	A549, HepG2, HEYA8, HCT116. H9, K562, MCF7	cDNA, directRNA, directcDNA, Illumina	1
The landscape of m6A RNA modifications in the SG-NEx cell lines	Core SG-NEx samples	23	A549, HepG2, HEYA8, HCT116. H9,	directRNA	NA

			K562, MCF7		
--	--	--	------------	--	--

Table R1.1 Dataset usage summary by section

Reviewer #1:

The authors described “cell type-specific major isoform switching event which were supported by full length reads”. How are these switches calculated? Are they defined in terms of a statistical test that evaluates the significance of the switch? Are these switches consistent across long-read technologies? Here there is also a missed opportunity to highlight the advantage of SG-NEx in providing multiple technologies.

Q16 Response:

We thank Reviewer #1 for pointing out that the description of our methodology to identify major isoform switches in the SG-NEx data was not sufficiently clear. In short, we have first estimated transcript expression for each isoform in each nanopore long read RNA-Seq sample from the core SG-NEx data set, and then identified for each gene the isoform that is most abundantly expressed in each cell type (major isoform, see above for a detailed response on the definition of major isoforms and corresponding changes in the text and methods). We have then calculated differential transcript usage using DRIMSeq and DEXSeq, adjusting for the protocol as a covariate in the model (which will ensure that results are consistent across protocols), and tested for DTU using a two-stage testing (DEXSeq and stageR) (Love, Sonesson, and Patro 2018). To remove lowly expressed transcripts we have applied several filtering steps, which are outlined in the methods section. Cell type-specific major isoform switching events are then defined as isoforms which are significantly different across cell types, and which are major isoforms only in one of them. In the revised manuscript, we have edited the methods section to include additional details and provide a clearer description. Furthermore, we have included additional details in the main text to clarify that we focus on significantly differentially expressed transcripts.

We have performed this analysis using Bambu (main figures). We also repeated the analysis with estimates obtained from NanoCount and Salmon as control to demonstrate the reproducibility of this analysis (Supplementary Figure 6). We have decided to combine all long read RNA-Seq protocols to highlight the presence of cell type specific major isoform events that are supported by full-length reads, and which illustrate a key application of the SG-NEx data (differential transcript expression). We have compared the different protocols in Figures 2-4, illustrating differences in gene and transcript expression that are specific to each protocol (e.g. Figure 2).

We hope that the revised manuscript now provides a more complete description of the analysis of cell type specific major isoforms, and that the additional edits more clearly emphasise protocol-specific differences in gene and transcript expression throughout the manuscript.

Reviewer #1:

--- Overlap with repeats ---

Overall, this part is intriguing and worth exploring further, but in its current state, it is only partially supported and not very compelling.

The authors should provide information on the samples that were used for this analysis. This is not described in the text.

Q17 Response:

For the identification of novel transcripts, all samples were used, including the extended SG-NEx cell lines to provide a broader coverage of the human transcriptome. We thank Reviewer #1 for pointing out that this was not clear in the original manuscript. In the revised manuscript, we now specify which samples are used. Furthermore, we now provide an additional supplementary table (Supplementary Table 13) that lists which samples were used in each analysis.

Main text, section “(8) Novel transcripts are enriched in transposable elements”

“A key advantage of long read RNA-Seq is the ability to identify novel transcripts that are missing in reference annotations (Tang et al. 2020; Chen et al. 2023; Wang et al. 2019; Gupta et al. 2018; Joglekar et al. 2021; Uapinyoying et al. 2020). Here we included the extended SG-NEx cell lines to provide a broader coverage of the human transcriptome (Supplementary Table 1). Using a stringent threshold for transcript discovery in well annotated genomes (NDR=0.1), we identified 1635 novel multi-exon transcript candidates across all samples in the SG-NEx data[...].”

Reviewer #1:

The authors should describe what is the read and technology support for the novel transcripts: number of reads? Expression values? Are they found across replicates? Are they found between different technologies? Without these details, it is not possible to know whether these novel transcripts are reliable or accumulated noise as a result of using many samples together.

Q18 Response:

The primary measure of confidence for novel transcript candidates is the Novel Discovery Rate (NDR), which is estimated for each transcript and which can be interpreted as a False Discovery Rate (see below for more details regarding the NDR/ expected false positives). To address the reviewer’s comment, we have analysed the number of samples, cell lines, and different RNA-Seq protocols in which novel transcripts are detected (Figure R1.5). We observe that the vast majority of novel transcript candidates are found in multiple samples (Figure R1.5a) and across all three nanopore RNA-Seq protocols (Figure R1.5c), and that many novel transcripts can be detected in multiple cell lines (Figure R1.5b). The expression level is lower compared to annotated genes, but comparable across cell lines and protocols (Figure R1.5d). We have included these new results in the Supplementary Text Section 7, and we provide information such as NDR and read counts for each transcript in Supplementary Table 7. We hope that the revised manuscript provides a more in-depth overview of the novel transcript candidates found in the SG-NEx data set.

These data show that novel transcripts are consistently found across multiple samples. For a detailed analysis on the expected false discovery rate and validation please see our response below.

Reviewer #1:

What is the expected validation rate of those novel transcripts? Based on the various properties mentioned above like read support, reproducibility, etc... the authors should be able to estimate the expected proportion of those transcripts that are likely to be real. It is possible that novel transcripts appear as a consequence of the fact that so many reads from various technologies are being analysed, but their expression level is comparable to experimental noise. Alternatively, could they validate some of them?

Q19 Response:

For the analysis of novel transcripts we have used Bambu, which infers a Novel Discovery Rate (NDR) for each candidate transcripts. The NDR corresponds to the fraction of novel transcripts among all transcripts with a similar or higher transcript discovery score. In the case of well annotated genomes such as the human genome it was shown that the NDR corresponds to the expected False Discovery Rate (FDR) (Chen et al. 2023). Here we used a NDR of 0.1, resulting in n=1,635 novel transcript candidates with an expected false discovery rate of 10% (i.e. 90% of novel transcripts are expected to be real). However, we provide the NDR for each individual novel transcript, thereby providing a measure of confidence for each novel transcript candidate. Among those transcripts that have a NDR which is smaller than 0.1, we expect that many more than 90% are likely to be real.

To investigate if these novel transcript candidates would be considered valid transcripts for genome annotations, we compared them with the most recent release of Gencode (GENCODE release 44, ENSEMBL release 110, released on July 2023) and RefSeq (Release 222, on Jan 16 2024). We found that 128 of novel transcript candidates were

included in the most recent version of both human genome annotations and additional 495 were included in either of these two annotations, indicating that they are likely to be valid transcripts. Among these, 54 transcripts have a repeat content of 80% or higher in the previously unannotated exons sequence, and additional 7 transcripts have an overall repeat content of 80% or higher. Additionally, we further confirmed 5 novel transcripts (1 already included in RefSeq Release 222, and 4 not included in any of the recent annotations yet) using PCR and Sanger sequencing validation (Figure R1.6, Supplementary Table 8).

In summary, we have aimed to identify novel transcripts with a stringent threshold to control the NDR/FDR, and we provide the estimate of confidence for each individual candidate. Since the data was analysed, several of these novel transcripts have been included in annotations, confirming that they are valid transcript candidates. We have included these new results and analyses as a specific section in the Supplementary Text Section 6, and we have referenced these results in the main text.

Reviewer #1:

What is the minimum number of reads, technologies, and replicates that are generally needed to identify a transcript, known and novel?

Q20 Response:

Instead of relying on thresholds in terms of read count or replicate number, we use the inferred NDR from BamBU for transcript discovery as a measure of confidence. As a result, it is possible that a novel transcript is only identified in a single sample or replicate, if the NDR

is below the stringent threshold of NDR=0.1. This design enables the inclusion of highly confident novel transcripts which are detected in samples without replicates (such as the extended SG-NEx data). However, the vast majority of novel transcripts are discovered across multiple samples (n=1583 out of 1635 in total).

For known transcripts, we don't have a detection threshold, therefore a single read that can be assigned to a transcript would result in a CPM>0. However, for downstream analysis, we use an expression threshold to filter out transcripts without 2 full-length read support or minimal expression of 2 CPM and 5% of total gene expression. Please also refer to our response above with a more detailed analysis and the corresponding changes in the manuscript.

Reviewer #1:

About these novel transcripts, the authors indicate that they have “lower sequence complexity” compared to reference transcripts. However, “sequence complexity” is not defined anywhere and the result is not shown. The authors should be clearer about what they mean, the description of the analysis and the results, and show the data.

Q21 Response:

In our original manuscript we have calculated the number of exons for novel transcripts, which we referred to as sequence complexity. We thank the Reviewer for highlighting that this was not clear from the context. We have now rephrased that sentence for clarity:

Main text, section “(8) Novel transcripts are enriched in transposable elements”

*“New transcripts generally have lower expression, **number of exons**, and transcript length compared to reference transcripts (Figure 6b, Supplementary Figure 7d,e)[...]”*

Reviewer #1:

The authors use “unannotated genes”. However, this is incorrect. They should use instead “unannotated loci”, “unannotated genomic loci” or “novel genes”.

Q22 Response:

We have changed the manuscript to refer to these genes either as novel genes or unannotated genomic loci.

Reviewer #1:

The authors describe “Compared to annotated transcripts, we observed a significant enrichment of repetitive elements such as LTRs, LINES, and SINEs”. However, they do not explain anywhere what repeat annotations were used. Is this the RepeatMasker annotation from UCSC or Ensembl? Did it include the simple repeats? Or only complex repeats? The authors should clarify.

Q23 Response:

We have used RepeatMasker annotations which are matched to the reference genome version used in this manuscript (Grch38, downloaded from UCSC genome browser). All repetitive elements were used for this analysis.

In the revised manuscript we have included a detailed description of the repeat analysis and we specify the RepeatMasker annotations in the methods section. Furthermore, we specifically refer to the methods section in the main text.

Reviewer #1:

How the repeat analysis was performed is not explained. The Methods section only indicates keeping only cases with 80% overlap, however, Figure 6c shows a wide range of overlaps. The authors should describe in more detail how this analysis was performed.

The authors should clearly explain how the overlap was calculated. Was it between the repeat annotation and the exonic regions of the transcript only? Or did they include the intronic regions as well?

Q24 Response:

We apologise for not being sufficiently clear in our description of the analysis of repetitive elements. The overlap of transcripts with repetitive elements was calculated for each exon using RepeatMasker annotations. The percentage overlap was then calculated for the entire isoform; introns were not considered.

The 80% threshold was only used for the visualisation as we wanted to show novel transcripts that have a particularly high repeat content in novel exons. However all other analyses were done based on the complete set of novel transcript candidates. We have specified this in the figure legend and methods section now.

Reviewer #1:

If the authors included the intronic regions, it might not be surprising that there is a high overlap with repeats. Introns are enriched in repetitive elements, both simple repeats and complex repeats.

The authors should describe what are the repeat types that the novel transcripts in Figure 6c overlap with. This is only indicated in Figure 6d for a subset of the transcripts. This plot also shows an overlap with “others”, but the authors do not explain what they are, i.e. simple repeats?

Q25 Response:

The overlap of transcripts with repetitive elements only considers exons, not introns. In the revised manuscript, we now specify that only exon sequences were considered to estimate the overlap with repetitive elements, please also refer to our response above.

In our manuscript we have specifically mentioned LINEs, SINEs, and LTRs as examples of repetitive elements, however, as pointed out by Reviewer #1, the complete set of repetitive elements that was analysed was not described. In the revised manuscript, we now include an overview that shows the overlap of all different repetitive elements (from RepeatMasker, Figure R1.7) which are used in our analyses for all novel transcripts, and for novel transcripts with a minimum of 80% overlap with repeats (see also response below).

Figure R1.7 Overview of the repeat family in expressed transcripts (a-b) Overview with repeat family in (a) transcripts at least 80% overlap with repeat elements expressed with ≥ 20 CPM in any samples and (b) all transcripts expressed with ≥ 20 CPM in any samples.

Reviewer #1:

The results are overall not very clearly described. The authors indicate “Compared to annotated transcripts, we observed a significant enrichment of repetitive elements such as LTRs, LINES, and SINEs in novel transcripts (two-sided t-test, $p < 0.001$, Figure 6c,d).” The test probably refers to the distributions of Figure 6c, but these include all novel transcripts, not just those from Figure 6d with 80% overlap. Is that correct? So this statement is misleading, as the overlaps in Figure 6c do not seem to be just 80% and they probably include many other repeats not shown.

Furthermore, there seem to be lots of annotated transcripts with $>80\%$ overlap. Are the numbers comparable?

Q26 Response:

We have rewritten this statement to clarify that LTR, LINE, and SINE elements are just examples of repetitive elements used in this analysis. Furthermore, we now clarify that the 80% threshold was only used for the heatmap (Figure 6d) to simplify the visualisation of elements that have the highest overlap with repetitive elements. Finally, we also show the fraction of repeats for all repeat classes, which shows that LTR elements are consistently the largest group (Figure R1.8). In the revised manuscript we have included these new results and updated the main text and figure legends for more clarity.

Reviewer #1:

They additionally say: “The enrichment of these highly repetitive elements was observed for both novel transcripts from annotated genes and for novel gene candidates, ...” However, they only show one test p-value. Did they test both distributions? The authors should show the p-value for both distributions to be able to make this claim.

The authors also say “ ... and was confirmed when only uniquely mapped reads are used for transcript discovery (Supplementary Figure 7f-i).” However, this figure does not include the statistical test. Is the difference significant? The authors should include a p-value.

Q27 Response:

We thank Reviewer #1 for highlighting the missing significance tests. We have now included the p-values in the revised manuscript in Figure 6c and Supplementary Figure 7f (See Figure R1.9)

Figure R1.9 Overlapping with repeat elements by transcript type (a-b)Boxplots showing the median, upper and lower quartile, and 1.5 x interquartile ranges of the percentage of exon sequence overlapping with repeat elements for annotated transcripts with at least 1 full-length read support across samples (Annotated >=1 FL), novel isoforms, and novel gene isoforms, with Bonferroni corrected two-sided t-test p-values reported for the pairwise mean differences, asterisks indicating p < 0.0001, using (a) primary alignments for all reads and (b) reads with unique alignments

Reviewer #1:

Regarding multimapping reads, the authors should indicate how many of their detected novel transcripts in unannotated loci are due to multimapping reads. An alternative possible explanation of their results is that most, if not all, of the novel transcripts in novel loci, especially those shown in Figure 6d, are actually due to multimappers caused by the ambiguity of the repeats.

Q28 Response:

To evaluate the impact of multi-mapping reads on the discovery of novel transcripts which overlap with repetitive elements, we have performed all analyses using two different alignment strategies: (1) using only the primary alignment (“primary alignments”, default), and (2) only keeping uniquely aligned reads that do not have any secondary alignment (“unique alignments”). The results indicate that the majority of transcripts can be found with both alignment strategies (91.7%, Supplementary Figure 2a), suggesting that novel transcripts are not due to multiple alignments from reads due to repetitive genomic sequences.

The full analysis comparing primary and unique alignments is described in Figure R1.10 (corresponding to Supplementary Figures 7a-i), and the NDR for each transcript is included in Supplementary Table 7 for the primary alignment and unique alignment strategy.

Figure R1.10, corresponding to Supplementary Figure 7: Novel transcripts identified with primary alignments and reads with only unique alignments

(a) Barplots presenting the number of novel transcripts identified when primary alignments of all reads are used vs reads with unique alignments are used with varying transcript discovery threshold ranging from 0.1 to 1, at an increasing step of 0.1 Blue bars represent the number of novel isoforms of annotated genes discovered, green represents the number of novel isoforms of novel genes discovered. The grey points represent the total number of novel transcripts.

(b) Barplots representing the predicted NDR when using reads with unique alignments for novel transcripts identified when using primary alignments with an NDR threshold of 0.1: light blue represents a predicted NDR < 0.1 , blue represents a predicted NDR of ≥ 0.1 and < 0.2 , light green represents a predicted NDR ≥ 0.2 , and green represents not identified

(c) Barplots presenting the number of novel transcripts identified when using PacBio data, StringTie2 method with and without unspliced novel transcripts included

(d-e) Boxplots of (d) number of exons and (e) isoform length for annotated transcripts with at least 1 full-length read support across samples (Annotated ≥ 1 FL), novel isoforms, and novel gene

isoforms using primary alignments of all reads

(f-i) Boxplots of (f) mean log2-transformed CPM, (g) percentage overlapping with repeat elements, (h) number of exons and (i) isoform length for annotated transcripts with at least 1 full-length read support across samples (Annotated ≥ 1 FL), novel isoforms, and novel gene isoforms using reads with unique alignments

Reviewer #1:

In fact, according to their Figure 6d, novel transcripts in annotated genes mostly overlap with “others”, which might be low-complexity repeats (which is expected), and LTRs, which is also expected, as LTRs often overlap with promoters and regulatory regions. All the LINE, SINE, and DNA elements might be intronic. This point should be clarified.

Q29 Response:

We now clarify in the revised text that the overlap only uses exons, introns are excluded.

Main text, section “(8) Novel transcripts are enriched in transposable elements”:

“Compared to annotated transcripts, we observed a significant enrichment of repetitive elements in exons from novel transcripts (two-sided t-test, $p < 0.001$, Figure 6c)[...]”

Reviewer #1:

--- fusion analysis ---

No details are given about which datasets were used to calculate the fusions. All long read datasets together? The manuscript is motivated on the advantages of having multiple technologies, but this crucial point is missed here again.

Not much information is provided in terms of reproducibility of the fusion discovery across replicates or technologies, what’s the read support? What are their expression values? These are questions where a dataset like SG-NEx could be useful for.

The reader is left with no information about whether there is anything in the SG-NEx data that is specially unique or useful for that. For instance, is it the number of reads? The combination of sequencing technologies? Etc. The authors should provide more details on how the SG-NEx data is particularly useful or unique for this type of analysis.

Q30 Response:

The SG-NEx data set several features that distinguish it from other long read RNA-Seq resources, among others the inclusion of multiple different long read RNA-Seq protocols and multiple cell lines (each with replicates).

In our original manuscript, we use the first aspect (multiple RNA-Seq protocols) for the first part of the manuscript, which specifically compares data from different library preparation and sequencing methods, and we used the second aspect (multiple cell lines) for the second

part of the manuscript to highlight the many different applications for which the SG-NEx data will be a useful resource. The variation of fusion transcripts across the different cell lines in particular highlights the importance of analysing data from multiple cell lines. Together with the experimental validation of selected fusion transcripts, the SG-NEx data provides the most comprehensive resource for analysing fusion transcripts with long read RNA-Seq.

However, we agree with Reviewer #1, that the availability of multiple different RNA-Seq protocols can also provide additional information for the analysis of fusion transcripts from long read RNA-Seq data. In the revised manuscript, we now show the transcript expression estimated by protocol for the fusion transcripts and the 5' and 3' genes (Figure R1.11, corresponding to Figure 7). Furthermore, we provide the expression estimates and the full-length read count estimates for each fusion candidate and cell line as a supplementary table (Supplementary Table 9). Finally, we have also updated the text and methods section to clarify how this analysis was done (see methods section).

The SG-NEx data was made publicly available as a community resource very early on, and it was already used for benchmarking of 3 different long read fusion detection methods (FLAIR fusion, JAFFAL, FusionSeekerLR), making it the most widely used resource for long read fusion detection. We believe that this further illustrates the value of the SG-NEx data for the development of computational methods for long read RNA-Seq data. We hope that the revised manuscript now more clearly describes the analysis of fusion transcripts, and the value of the SG-NEx data resource by providing multiple cell lines and multiple protocols.

Figure R1.11 Heatmap of fusion gene candidates detected using long read RNA-Seq data. Shown is the status of validations in this study and in the literature (top), number and class of break points (middle), and full-length read support for the 5' gene, 3' gene and the fusion gene (bottom).

Reviewer #1:

The authors mention “we additionally find full length read support for most of the 5' and 3' genes, suggesting that fusion genes are present together with their unfused wildtype gene.”.

This probably refers to the fact that fusions can be heterozygous or homozygous? The authors describe this as a remarkable result. Is that novel result? Or is it actually common? The authors should clarify.

Q31 Response:

This is not a surprising result as most of the cell lines are cancer cell lines with complex chromosome duplication, rearrangements and losses leading to fusions and other copy number mutations. It is too simple to say therefore, that the fusions are heterozygous but the observation of seeing both the wildtype and fusion genes implies that there is at least one wildtype chromosome and one rearranged chromosome at these positions. What we are able to see from the long reads is distinguishing the different isoforms (fusion or wildtype) arising from the different chromosomes which is difficult with short reads where only the breakpoint distinguishes the fusion and wildtype isoforms. In the revised manuscript we have rephrased this sentence accordingly.

Main text, section “(9) Detection and quantification of full-length fusion transcripts”

“Interestingly, we additionally find full-length read support for most of the 5’ and 3’ genes, showing that both the fusion genes and the unfused wildtype genes can be detected with long read RNA-Seq (Figure 7a).”

Reviewer #1:

The authors conclude this section with “... illustrating how long read RNA-Seq enables the discovery and quantification of full-length fusion transcripts in cancer.” The authors should clarify that these are not cancer samples, but cell lines. The material and sequencing efficiency might differ drastically.

Additionally, this statement might not be true in genera. Tthe authors may have identified those full-length fusion transcripts only because they sequenced so many replicates and used so many technologies. So the statement is misleading.

Q32 Response:

We have now rephrased this sentence, and we now refer to these cell lines as cancer cell lines throughout the manuscript (not “cancer”).

Main text, section “(9) Detection and quantification of full-length fusion transcripts”

“[...] illustrating how the SG-NEx data provides a unique resource for analysing full-length fusion transcripts with long read RNA sequencing.”

Reviewer #1:

Fusion discovery might be possible under certain conditions regarding sequencing depth and technologies. This is a question where the depth and breadth of SG-NEx data could provide useful insights, but this opportunity seems to be missed again.

Q33 Response:

The availability of a larger number of samples in the SG-NEx resource increases sensitivity to detect fusion transcripts. However the identification and quantification of fusion transcripts was done using individual samples, therefore each fusion transcript was identified in a single sample (not in combined samples). In the revised manuscript we now show expression estimates for fusion transcripts from all protocols for each cell line, which illustrates that most fusion transcripts are robustly identified (see response above). Here we did not attempt to systematically identify which sequencing depth would be required to identify fusion transcripts from a single sequencing run, as this depends on the expression level of fusion transcripts, which is highly variable across fusions and cell lines. Furthermore, different fusion detection and quantification methods might have different sensitivity, but we did not attempt to comprehensively evaluate computational methods. However, we believe that this is an interesting question that could possibly be answered using the SG-NEx data resource.

Reviewer #1:

--- m6A analysis ---

In the abstract, the authors mention various analysis indicating “..., quantify full-length fusion isoforms and simultaneously profile m6A RNA modifications when RNA is... “. This is misleading as the authors do not combine in the manuscript the m6A analysis with any of the other analyses.

Q34 Response:

We have now rephrased the Abstract accordingly:

“[...]quantify full-length fusion transcripts and profile m6A RNA modifications when RNA is sequenced directly.”

Reviewer #1:

The m6A analysis is carried out by aligning the read signals to the “reference nucleotide sequences”. The authors should be more specific about what they did. Are these genomic or transcriptomic sequences? The Methods section is not clear about this either.

Q35 Response:

The analysis of m6A is based on transcriptome alignment. We have included additional details in the Methods section and main text to clarify this.

Methods, section “m6A modification analysis”:

*“For the analysis of m6A RNA modifications, reads were aligned to the transcriptome (please refer to “Read Alignment” for details). For each direct RNA-Seq sample in the SG-NEx core dataset, the raw current signal from each read was aligned to the reference transcript sequence using nanopolish *eventalign* (Loman, Quick, and Simpson 2015). “*

Reviewer #1:

The authors describe that the m6A sites recovered follow an expected DRACH motif. But this is misleading. m6Anet only detects m6A on DRACH motifs. While they show that non-modified sites have a different logo, these are all DRACH motifs as well. They should rephrase or explain it better.

The authors also mention an enrichment in the 3' UTR. If the signals were aligned to the genome, the 3'UTR enrichment might be due to biases. 3' UTRs are generally longer, and read signals are easier to align to long sequences. In contrast, exons tend to be shorter than 150nt, and signals might be harder to align. The authors should control for length and for the efficiency of the signal alignments.

Q36 Response:

Reviewer #1 correctly points out that the analysis is limited to DRACH kmers, therefore even unmodified sites will resemble the DRACH motif. Furthermore, the enrichment at the 3' end might be due to specific biases that could also result in an enrichment for unmodified sites.

To address this, we have included the distribution along the transcript and the average kmer motif for modified and unmodified positions, thereby better illustrating the enrichment compared to the expected background (Figure R1.12, corresponding to Supplementary Figure 8b-c). Furthermore, we have now rephrased the text to emphasise the comparison against unmodified sites as a control.

Main text, section “(10) The landscape of m6A RNA modifications in the SG-NEx cell lines”

“As expected, m6A sites identified using direct RNA-Seq data show a shift in the current signal data, an enrichment at the 3' end of transcripts and resemble more closely the expected DRACH motif compared to unmodified positions (Supplementary Figures 8b-d).”

Reviewer #1:

Figure 8b. The figure caption indicates “Heatmap showing the two-sided Fisher’s test odd ratio (enrichment score) across sample replicates from the seven cell lines.”. However, it is not explained here or in the text what this test is about and what is meant for. The analysis described by this figure is not clear.

Q37 Response:

The heatmap in Figure 8b shows a clustering of all direct RNA-Seq samples based on the similarity of the transcriptome-wide m6A profile. To estimate this similarity for two samples, we have calculated an enrichment score that estimates if the set of m6A sites in one sample is significantly enriched in the set of m6A sites in the second sample. We have used the odds ratio from the Fisher's exact test to estimate this enrichment score. As a second method, we have also calculated the similarity using the Pearson correlation coefficient, which results in a similar clustering as shown in Supplementary Figure 8e.

To address this comment, we have modified the figure legend and included an additional section in the methods section of the manuscript.

Main text, Methods, section "m6A modification analysis"

"Comparison of m6A sites across samples"

To estimate the similarity of m6A sites across different direct RNA-Seq samples we calculated a pairwise enrichment score and the pairwise correlation. The enrichment score was obtained using a two-sided Fisher's test based on the number of common m6A sites ($p > 0.9$) among all candidate m6A sites that were included in each pairwise comparison. The odds ratio was then used as an enrichment score that only relied on the recommended threshold of $p = 0.9$ for m6Anet to estimate a pairwise similarity of the m6A profile across all samples from the seven cell lines (Figure 8b). We also calculated the Pearson correlation coefficient, which estimates the similarity using the p-value without a selected threshold (Supplementary Figure 8e). The similarity scores were visualised using the ComplexHeatmap R package (Gu, Eils, and Schlesner 2016) with default parameters."

Reviewer #1:

Figure 8c: the figure says genes, but the text mentions transcripts "While these transcripts also show cell type-specific modifications, the majority of cell type-specific m6A sites originate from cell type-specific transcripts (Figure 8c)." Cell specific genes or transcripts are two very different things. The authors should use consistently genes or transcripts in text and corresponding figures.

Q38 Response:

We thank Reviewer #1 for highlighting the inconsistency. The analysis of m6A in the SG-NEx core cell lines is based on alignment against the transcriptome reference sequence as required by m6Anet. We have rephrased the manuscript and the methods section, and updated the figures to clarify this (Figure R1.13).

Manuscript, main text, section "(10) The landscape of m6A RNA modifications in the SG-NEx cell lines"

*"Globally, we **note** that m6A sites can be either cell type-specific or modified across cell types (constitutive **sites**) (Figure 8b, c, Supplementary Figure 8e). While **constitutively***

expressed transcripts also show cell type-specific modifications, the majority of cell type-specific m6A sites originate from cell type-specific transcripts (Figure 8c).”

Reviewer #1:

The authors have included m6ACE-seq data in the manuscript, but this data (whether it is a new experiment or extracted from the literature) is not described. The authors should describe where this data comes from.

The authors claim that the SG-NEx data is useful as a benchmarking dataset, the authors should then compare the m6ACE-seq data with the nanopore-based m6A predictions to illustrate this aspect.

Q39 Response:

The m6ACE-Seq data from the Hct116 cell line was generated as an additional resource to provide an independent map of m6A for one of the core SG-NEx cell lines. In the revised manuscript, we now included a specific methods section to describe how the data was generated. Furthermore, as pointed out by Reviewer #1, these data were not used in the original manuscript. Following the suggestion, we now emphasise that the data was generated as an additional, independent map of m6A, and we included a comparison of the predictions from m6Anet and m6ACE-Seq on the Hct116 cell line (Figure R1.14, corresponding to Supplementary Figure 8a).

We thank Reviewer #1 for this comment, and hope that the revised manuscript now better utilises and highlights the SG-NEx data resource.

Main text, section “(10) The landscape of m6A RNA modifications in the SG-NEx cell lines”

“To provide an independent, transcriptome-wide map of m6A, we also generated m6ACE-Seq data for the Hct116 cell line. Across all 7 core cell lines, we find 6,337 positions that are predicted to be modified in at least 1 cell line (Supplementary Table 12), with 59% of m6A sites in the Hct116 cell line being confirmed by m6ACE-Seq (Supplementary Figure 8a).”

Reviewer #2:

Reviewer #2:

None

Reviewer #3:

Reviewer #3:

Remarks to the Author:

The authors addressed my questions and concerns in a thorough and satisfactory manner.

Response:

We are happy to hear that the revised manuscript has addressed the Reviewer’s comments.

Reviewer #4:

Remarks to the Author:

The updated manuscript by Chen et al. offers a thorough comparison of long-read RNA sequencing techniques utilizing various cell lines and replicates, enriched by a diverse collection of spike-ins. The state the superiority of long-read sequencing methods over Illumina for transcript detection and quantification. The data compilation is significant, and its availability via the Nextflow pipeline is beneficial. Although certain conclusions of the benchmarking may not be novel, some findings contradict previous works. Nonetheless, certain aspects of the analysis warrant further scrutiny. My detailed comments are as follows:

Reviewer #4: [point 1a]

1 The claim regarding the dataset's uniqueness for benchmarking purposes should be tempered. The dataset includes analyses and cell types potentially overlooked in other

studies, yet the diversity of the data is not unprecedented. For instance, LRGASP also incorporates multiple library preparation methods, sequencing platforms, cell lines, spike-ins, and it enables benchmarking for transcript models, quantification, RNA modifications, and fusion transcript detection. This acknowledgment does not diminish the value of the presented data collection, but the claim of uniqueness should be qualified. Specifically:

- a. The statement "The benchmark with the highest number of protocols" is inaccurate. LRGASP employed seven different protocols. This is wrong in Figure R.R1. (Supplementary Figure 2?) and should be corrected.
- b. The description "The most comprehensive resource with available ground-truth" should be rephrased in the context of spike-ins. LRGASP utilized 270 manually curated transcripts by GENCODE annotators as ground truth and also included orthogonal data sources where ground truth was established.

Response:

We thank Reviewer #4 for pointing out these details. We agree that the uniqueness of the SG-NEx data is not limited to the number of RNA-Seq protocols, but that the breadth of samples, protocols, replicates, and spike-in RNAs combined with easy access through AWS makes this a single, comprehensive resource for many different applications. We are also happy to see this being reflected in the growing numbers of publications that use the SG-NEx data resource.

In the revised manuscript, we carefully went through the main text and adjusted claims accordingly. Supplementary Figure 2 was initially based only on commercially available RNA-Seq protocols. Following this comment, we have updated Supplementary Figure 2 to include all RNA-Seq protocols (Figure R4.1). While we think that the SG-NEx data set is a comprehensive resource, we now avoid making statements that suggest that it is "the most comprehensive resource".

Figure R4.1 Revised Supplementary Figure 2 Data resource comparison
 Barplots of number of reads, number of long reads, number of protocols per samples, number of cell lines (or tissues), number of spike-in sets, number of median replicates per samples, and for different long read data resources. Note for protocol comparison, LRGASP also includes specialized RNA-seq protocols.

Reviewer #4: [point 2]

2. The manuscript should clarify the quality control steps in the nanoseq pipeline. Specifically, it should detail the measures taken to prevent artifacts like intrapriming and RT-switching events, which are relevant to the claim of enhanced detection of repeat-containing transcripts, predominantly found in intergenic regions. In my experience, such intergenic detections often diminish with stringent QC filters, so it is crucial to disclose which QC measures were employed to ensure the authenticity of these transcripts.

Response:

We agree with the reviewer that QC is important in data processing and will impact the analysis results. The nanoseq pipeline uses NanoPlot and Fastqc to generate read quality statistics, however, no automatic read filtering is applied as this might differ depending on the individual analysis. In this manuscript, we have filtered out samples with less than 400,000 reads (for Figures 3-6, see Supplementary Table 1 and Supplementary Table 13 for details) as these often had low read quality, however, no read filtering on the other samples was performed. Following the suggestion from Reviewer #4, we have tried to run SQANTI3, a computational tool that allows identification of intra-priming and RT-switching based on bam files. However, we were unable to run it successfully on our data. As we could not find

an alternative method to perform this analysis, we were unable to analyse the impact of RT switching and intrapriming on novel transcripts.

To evaluate the accuracy of our novel transcript predictions, we have performed a PCR validation for selected candidates, confirming 5 novel transcripts. Furthermore, we have compared our predictions with the most recently released genome annotation versions which further showed that 623 of the novel candidates have been confirmed (please refer to Reviewer #1, Q19 for additional details). We have included these results and the comparison with the recent genome annotations in the Supplementary Text Section 6.

We apologise that we couldn't find a way to perform the specific analysis that was proposed, but we hope that these alternative analyses confirm the quality of our analysis and findings.

Reviewer #4: [point 3]

3. To address the concerns by reviewer one on the possible role of mapping errors for novel transcript detections, the authors evaluate the differences in novel transcript calls when using mappers that allowed multiple or single mappings. They found that using reads that map only uniquely reduces the number of novel transcripts. As this reduces the number of novel transcripts, they claim this is controlling possible false mappings. However, to be sure this is specific for the novel transcripts, they should also evaluate the overall transcript detection when they discard an X number of reads, regardless of their mapping ambiguity.

Response:

Following the suggestion from Reviewer #4, we have performed random downsampling from aligned reads of one sample to obtain a similar number of reads after using only unique alignments (using "samtools -s "). We then compared the transcript discovery results when using 1) primary read alignments, 2) unique alignments, and 3) randomly downsampled alignments (Figure R4.2). We found that when a similar fraction of reads is sampled regardless of the mapping ambiguity, a lower reduction of novel transcripts was observed (unique vs downsampling: 6326 vs 6984 when a NDR threshold of 1 is used). However, as multi-aligned reads will be included, this downsampling strategy is unable to control for possible artefacts from such reads.

Figure R4.2 Impact of multiple-mapping reads on transcript discovery
 Barplots presenting the number of novel transcripts identified when primary alignments of all reads are used vs reads with unique alignments vs primary alignments of a subset of all reads are used with varying transcript discovery threshold ranging from 0.1 to 1, at an increasing step of 0.1 Blue bars represent the number of novel isoforms of annotated genes discovered, green represents the number of novel isoforms of novel genes discovered. The grey points represent the total number of novel transcripts.

Reviewer #4: [point 4]

4.The manuscript asserts that the benchmarking conclusions, particularly regarding library preparation methods, are not algorithm-dependent. However, this claim conflicts with some LRGASP findings. For instance, Figure R1.4.a shows significant differences in LongSIRV analysis outcomes between methods. This should be discussed. Also, for figure R1.3 a, Supplementary Figure 4?) why Long SRIVs have not been evaluated in terms of Spearman Correlation and R2? Moreover, R1.3 b) is hard to read. Possible other choice of colors or splitting in different figures will make it more readable.

Response:

In our manuscript we have compared different long and short read RNA-Seq protocols against each other, focusing on the transcriptome-wide similarities and differences. Our main claims are that long read and short read RNA-Seq expression estimates have a high correlation when gene expression estimates are used (for spike-in RNAs and for protein coding gene expression estimates from cell lines), that both technologies still show a good agreement for transcript expression estimates from spike-in RNAs, but that they show lower correlation when transcript expression estimates are compared from human cell lines. As pointed out by Reviewer #4, there is variation when different methods are used (Salmon, Bambu, RSEM, NanoCount), as reported in the LRGASP study that presents a much more detailed comparison of computational methods. However, this variation is less compared to the observed differences between technologies, therefore the main claims are all confirmed independently from the computational methods. Following the suggestion from Reviewer#4,

we have now updated the figure for more clarity with different colours, which now more clearly shows that differences between technologies are larger than differences between computational methods (Figure R4.3 and Figure R4.4).

The SIRV E0 and Long SIRV spike in RNAs have all identical concentrations. The Spearman correlation coefficient is calculated as the covariance of the two variable ranks divided by the product of the standard deviation of each variable rank. In the case of SIRV E0 and Long SIRV datasets, given the constant expected CPM, the standard deviation is 0 and thus the spearman correlation is not computable. For completeness, we have analysed all spike-in RNAs with alternative metrics such as the Mean Absolute Error (MAE) which can be computed for all spike-in RNAs (see Figure R4.3 and Figure R4.4). We thank Reviewer #4 for pointing out that the lack of a correlation coefficient for the Long SIRVs was not described in the text, we have therefore included a sentence in the figure legends with an explanation.

(a) Heatmaps of spearman correlation, mean absolute error (MAE), root mean squared error (RMSE), mean absolute relative difference (MARD), mean relative difference (MRD), and coefficient of determination (R^2), for ERCC, Sequin MixA V1, Sequin MixA V2, SIRV E2, SIRV E0 and Long SIRVs, when using Bambu, NanoCount and Salmon for cDNA, direct cDNA and PacBio data, and Salmon and RSEM for short read RNA-seq data

(b) Histograms of absolute errors (AE) for ERCC, Sequin MixA V1, Sequin MixA V2, SIRV E2, SIRV E0 and Long SIRVs, when using Bambu, NanoCount and Salmon for cDNA, direct cDNA and PacBio data, and Salmon and RSEM for short read RNA-seq data

Figure R4.4. Old Figure R1.4 revised with different color scheme for methods Comparing \log_2 -transformed estimated CPM against \log_2 -transformed expected CPM for spike-in transcripts

(a) Heatmaps of spearman correlation, mean absolute error (MAE), root mean squared error (RMSE), mean absolute relative difference (MARD), mean relative difference (MRD), and coefficient of determination (R^2), for ERCC, Sequin MixA V1, Sequin MixA V2, SIRV E2, SIRV E0 and Long SIRVs, when using Bambu, NanoCount and Salmon for cDNA, direct cDNA and PacBio data, and Salmon and RSEM for short read RNA-seq data

(b) Histograms of absolute errors (AE) for spike-in transcripts in ERCC, Sequin MixA V1, Sequin MixA V2, SIRV E2, SIRV E0 and Long SIRVs, when using Bambu, NanoCount and Salmon for cDNA, direct cDNA and PacBio data, and Salmon and RSEM for short read RNA-seq data

Reviewer #4: [point 5]

5. Certain terminology used in the paper, such as "full-length read," requires explicit definition. While "full-length read" has a specific connotation in PacBio technology, it seems the authors refer to "full-splice-junction", equivalent "full-splice-match" in the SQANTI terminology. A clear definition should be included to avoid ambiguity.

Response:

We thank Reviewer #4 for highlighting this point, which we agree is an important point. In our manuscript, we defined full-length reads as reads which span all exon junctions of a transcript (equivalent to full-splice-match" in SQUANTI). In the revised manuscript, we now specify this definition when we first mention full-length reads in the results, we included this definition in the figure legend, and we included the definition in the methods section as well.

Reviewer #4: [point 6]

6. Figure 3 illustrates the consistency of gene expression quantification, highlighting the superior performance of long-read RNA-seq methods compared to Illumina, especially with spike-ins. However, I feel this analysis needs further elaboration. Spike-ins are relatively highly expressed genes in comparison to a normal sample and correlation with expected values might not be as good at the lower expression range (results with lncRNA analysis reveals this), which may represent a large fraction of the transcriptome. How are correlation values, for example, at CPM < 2.5? Possibly a better breakdown of correlation by expression value with indication of the transcriptome fraction it encompasses, would be more informative. Moreover, the lack of correlation observed for the E0 SIRV dataset across all methods demands an explanation.

Response:

Following the suggestion from Reviewer #4, we have compared gene expression quantification among different RNA-Seq methods with spike-in RNAs specifically for those that are lowly expressed (CPM < 2.5, Figure R4.5). While the data show that gene expression estimates have higher variation for lowly expressed genes, the overall results remain similar, with both short read and long read achieving similar accuracy on spike-in RNAs for gene expression estimates (Figure R4.5).

Please refer to our response to point 4 with an explanation regarding the lack of correlation for the E0 SIRV data set, which is due to the spike-in RNAs having identical expected correlation, preventing the calculation of a correlation coefficient. However, we provide other metrics to estimate the similarity between the observed and expected expression that also quantify differences across RNA-Seq technologies for the E0 SIRVs.

We have included these additional results in the Supplementary Text Section 14, and we have included additional details in the respective figure legends to clarify the lack of correlation.

Reviewer #4: [point 7]

7. Figure 4 demonstrates the accuracy of long-read methods in isoform detection. Yet, it is unclear how isoforms were matched across different technologies, and whether newly detected isoforms were considered. If the analysis is limited to annotated transcripts, this limitation should be clearly stated, as it is not evident in the methods section.

Response:

The analysis in Figure 4 was done only using transcripts which are annotated, therefore isoforms were only matched by transcript ID across different technologies. We did not include novel transcripts in this analysis, which was instead done in a separate section in Figure 6.

The identification of novel transcripts was done using Bambu, which analyses all samples jointly, thereby returning a single, extended set of reference annotations and transcript quantification values for all samples in this analysis. In the revised manuscript, we now specify that the analysis in Figure 4 is purely based on annotated isoforms, and we emphasise that transcript discovery was performed jointly on all samples.

Reviewer #4: [point 8]

8. Figure 5 presents alternative splicing patterns uncovered by long-read RNA-seq. It should be specified whether these patterns are consistent across various methods and algorithms or if they stem from a single analysis approach. The LRGASP identified significant variability in isoform identification contingent on the analysis tool, which could influence the results. Furthermore, the data supporting the isoforms presented in panels a and b should be clarified—are these isoforms detected by a single read in at least one sample? This could represent transcriptional noise. Although the observation is still interesting, this should be clarified.

Response:

To evaluate if alternative splicing patterns are consistent when alternative methods are used, we have generated the same results using Bambu (Figure 5), Salmon (Supplementary Figure 6), and NanoCount (Supplementary Figure 6). This analysis shows that the overall patterns are consistent, with exon skipping and alternative promoters being most frequently used to generate alternative isoforms. As pointed out by Reviewer #4, there is still variation across methods, in particular for individual isoforms where different methods might disagree, however, the overall pattern is consistent with alternative analyses methods.

The analysis of alternative isoforms is based on alternative isoform pairs that satisfy the following expression condition per cell line:

1) the major isoform has an average full-length normalised read count ≥ 2 across replicates from the respective cell lines.

Here we estimate the normalised full-length read count for each transcript as the number of full-length reads divided by the total number of full-length reads in the sample, times 1 million (corresponding to the definition of CPM). This normalisation is only used in this specific analysis to ensure that thresholds are comparable across samples and cell lines with differences in sequencing depth.

2) both isoforms expressed with average expression ≥ 2 CPM across replicates from the respective cell lines.

3) the average expression for each isoforms contributes to at least 5% of the total average expression for this gene in the respective cell line

We thank Reviewer #4 for highlighting that these criteria were not specified in the original manuscript. We now included these details in the methods section of the revised manuscript (see “Analysis of alternative isoform expression”).

References

- Chen, Ying, Andre Sim, Yuk Kei Wan, Keith Yeo, Joseph Jing Xian Lee, Min Hao Ling, Michael I. Love, and Jonathan Göke. 2023. "Context-Aware Transcript Quantification from Long-Read RNA-Seq Data with Bambu." *Nature Methods* 20 (8): 1187–95.
- Gupta, Ishaan, Paul G. Collier, Bettina Haase, Ahmed Mahfouz, Anoushka Joglekar, Taylor Floyd, Frank Koopmans, et al. 2018. "Single-Cell Isoform RNA Sequencing Characterizes Isoforms in Thousands of Cerebellar Cells." *Nature Biotechnology*, October. <https://doi.org/10.1038/nbt.4259>.
- Gu, Zuguang, Roland Eils, and Matthias Schlesner. 2016. "Complex Heatmaps Reveal Patterns and Correlations in Multidimensional Genomic Data." *Bioinformatics* 32 (18): 2847–49.
- Joglekar, Anoushka, Andrey Prjibelski, Ahmed Mahfouz, Paul Collier, Susan Lin, Anna Katharina Schlusche, Jordan Marrocco, et al. 2021. "A Spatially Resolved Brain Region- and Cell Type-Specific Isoform Atlas of the Postnatal Mouse Brain." *Nature Communications* 12 (1): 463.
- Loman, Nicholas J., Joshua Quick, and Jared T. Simpson. 2015. "A Complete Bacterial Genome Assembled de Novo Using Only Nanopore Sequencing Data." *Nature Methods* 12 (8): 733–35.
- Love, Michael I., Simon Anders, Vladislav Kim, and Wolfgang Huber. 2015. "RNA-Seq Workflow: Gene-Level Exploratory Analysis and Differential Expression." *F1000Research* 4 (October): 1070.
- Love, Michael I., Charlotte Soneson, and Rob Patro. 2018. "Swimming Downstream: Statistical Analysis of Differential Transcript Usage Following Salmon Quantification." *F1000Research* 7 (June): 952.
- Tang, Alison D., Cameron M. Soulette, Marijke J. van Baren, Kevyn Hart, Eva Hrabeta-Robinson, Catherine J. Wu, and Angela N. Brooks. 2020. "Full-Length Transcript Characterization of SF3B1 Mutation in Chronic Lymphocytic Leukemia Reveals Downregulation of Retained Introns." *Nature Communications* 11 (1): 1438.
- Uapinyoying, Prech, Jeremy Goecks, Susan M. Knoblach, Karuna Panchapakesan, Carsten G. Bonnemann, Terence A. Partridge, Jyoti K. Jaiswal, and Eric P. Hoffman. 2020. "A Long-Read RNA-Seq Approach to Identify Novel Transcripts of Very Large Genes." *Genome Research* 30 (6): 885–97.
- Wang, Xi, Xintian You, Julian D. Langer, Jingyi Hou, Fiona Rupprecht, Irena Vlatkovic, Claudia Quedenau, et al. 2019. "Full-Length Transcriptome Reconstruction Reveals a Large Diversity of RNA and Protein Isoforms in Rat Hippocampus." *Nature Communications* 10 (1): 5009.

Summary of Response to Reviewers:

We thank the Reviewers very much for their detailed comments and suggestions related to the impact of analysis methodology on results, including the comparison between long read and short read RNA-Seq data, differential transcript usage (DTU) event detection, the analysis of m6A modification analysis, as well as the impact of read quality on highly repetitive novel transcripts. To address these comments, we have performed additional analyses and made several changes to the main text, which can be summarised as follows:

1. We reproduced the results for transcript expression estimates when comparing long read and short read RNA-Seq data using publicly available ENCODE data
2. We experimentally confirmed the long read specific major isoforms among the discordant major isoform pairs between long read and short read RNA-Seq data for 13 highly expressed candidate genes with both qPCR and dPCR experiments
3. We additionally used edgeR to perform DTU analysis and demonstrated the robustness of our findings
4. We added a new RNA004 sample and benchmarked the performance of m6Anet and Dorado
5. We added a SQANTI3 QC step to filter out potential RTS and intra-priming artefacts in the identification of highly repetitive novel transcripts
6. We evaluated the impact of Phred read quality filtering on the presence of artefact in novel transcript candidates
7. We further validated six novel transcripts, both repetitive and non-repetitive, using PCR Sanger sequencing

In addition, we have addressed all remaining minor comments and suggestions related to the main text. Here is a summarised list of items changed in this re-submission:

1. Figure 4: removed results of ERCC in Figure 4a, added qPCR and dPCR results
2. Figure 6: updated results after filtering RTS and intra-priming artefacts
3. Supplementary Figure 5: added missing panel j-p
4. Supplementary Figure 6: added results from edgeR for comparison with DEXSeq
5. Supplementary Figure 7: updated c for stringtie2 results (previous version underestimated number of novel transcripts due to corrupted annotation file used)
6. Supplementary Figure 8: legend text style edited a bit for better clarity
7. Supplementary Table 1: added a new sample "SGNex_Hek293T_directRNA_replicate5_run1"
8. Supplementary Table 5-14: added a new Supplementary Table 5 to list the 13 selected candidate genes for qPCR and dPCR experimental validation, original Supplementary Table 5-13 are renamed to Supplementary Table 6-14.
9. Supplementary Table 8 (Original Supplementary Table 7): added RTS status and Perc_A_downstream from SQANTI3 and status for whether kept for Figure 6c-d
10. Supplementary Table 9 (Original Supplementary Table 8): added a column Overlapping percentage with repeats and another 6 validated novel transcript candidates
11. Main text updated according to Reviewers' comments
12. Supplementary Text: updated Supplementary Section 6, added Section 15 to 23 for additional analysis mentioned in the rebuttal letter.

We hope that these new data and results further demonstrate the robustness of our analysis and that they have addressed the reviewers remaining comments and suggestions.

Please see below for our detailed point-by-point responses.

Reviewers' Comments:

Reviewer #1:

Remarks to the Author:

While the data is useful and interesting, the analyses performed are still not novel and significant enough and of sufficient quality for what I would consider to be an outstanding and exciting manuscript worth publishing in Nature Methods. The results are anecdotal or repeat what has already been published elsewhere. The manuscript reads as a collection of rather shallow results without any clear biological relevance. Additionally, the analyses still do not exploit the depth and breadth of the datasets fully. It is not clear how such analyses could be acceptable for publication in Nature Methods.

The datasets could be published and showcased in a more data-focused journal, such as Scientific Data. Some of the analyses, those more promising regarding the preliminary results, could be developed more thoroughly and with the appropriate controls as separate research article.

The introduction and sections (1)-(4) provide now a clearer description of the datasets and an interesting comparison between the different sequencing protocols.

Response:

We thank Reviewer #1 for the positive comments on sections (1) - (4). To further strengthen this manuscript, we have now included additional validation by qPCR and digital PCR for 13 genes where short and long read identified different major isoforms. These results confirm that long read RNA-Seq correctly identifies the major isoform (13 out of 13), whereas short read RNA-Seq identifies isoforms which were very lowly expressed as major. These results confirm our observations, and provide additional evidence that long read RNA-Seq results in more robust isoform expression estimates. Furthermore, we have addressed the remaining comments and suggestions related to the other sections in our detailed response below.

Reviewer #1:

From section (5) things become less clear. The authors report a difference of ~7000 genes per cell line, where the major isoform described by long or short reads is different. Some of the results related to this are a bit confusing, eg: "For genes where long and short read RNA-Seq identified different major isoforms, long read-specific major isoforms showed significantly higher correlation of expression estimates between both sequencing technologies". This result and its interpretation is not very clear.

The authors also indicate “major isoforms identified by short read RNA-Seq significantly more often used internal exons from the long read major isoform as first or last exons, typically corresponding to shorter versions...”. This should be clarified. It could be an issue with the mapping and reconstruction algorithm. In the Methods section, the authors indicate that they use STAR to map illumina reads to the genome. However, they only mention Salmon and RSEM in relation to transcript quantification, and they use a direct mapping against the annotation. They present an intriguing result but it may be related to a technical issue of the methodology rather than the data. There might be a methodology that could be used that would provide a better read alignment and transcript reconstruction strategy with the Illumina reads such that this result is not true anymore. This analysis is not of sufficient quality – it would require more in-depth analysis to more clearly show what is happening.

For section (6), the text does not describe the simulated paired-end read data that the authors claim to have added in the response to reviews. This is rather unusual.

It is also not very clear the relevance of the results in this section: “read fragmentation increases transcript expression estimates for isoforms that are only identified in Illumina short read RNA-Seq data as major isoforms”. It is not clear how this result is generalisable. This fails to grasp the opportunity of this rich dataset to provide a useful guide to the community.

Their final conclusion for this section could be all based on an artifact of their methodology to map/reconstruct/analyse short reads: “These observations suggest that the difference in transcript expression between short and long read RNA-Seq data may partially be the consequence of inflated expression estimates in short read RNA-Seq due to fragmentation and reduced read length.” Additionally, it remains unclear from the text how relevant and how widespread this issue is, whether it occurs in genes functionally relevant or genes of unknown function, genes that are key for cell identity, or house-keeping genes. The authors do not provide enough controls to support such a general claim.

Q1&2 Responses:

In this comment, Reviewer #1 suggests that our results that compare short read and long read RNA-Seq data are intriguing but suggest several additional analyses to confirm and strengthen this section. In particular, Reviewer #1 comments (1) that an alternative alignment or quantification methodology might lead to different results, (2) that the generalisability of these results are unclear and that additional controls are suggested, and (3) that the functional relevance was not investigated. To address Reviewer #1’s comments, we have now generated additional validation data and made the following new analyses:

- (1) We have now reproduced the results using short read RNA-Seq data downloaded from public resources and processed by a widely recognised RNA-Seq pipeline (ENCODE). This serves as an orthogonal confirmation to demonstrate that our findings are not due to the specific analysis method used in our manuscript
- (2) We performed quantitative PCR and digital PCR validation experiments on highly expressed long-read specific and short-read specific major isoforms. These results confirmed that for all genes, long read RNA-Seq data identified the correct major

- isoform. These results provide direct experimental evidence that long read RNA-Seq more robustly identifies major isoforms compared to short read RNA-Seq data
- (3) We performed a gene ontology (GO) enrichment analysis on genes with discordant major isoform between long and short read RNA-Seq data to investigate if they are biologically relevant

Please find the detailed response below:

(1) *Orthogonal analysis using ENCODE short read RNA-Seq data*

To demonstrate that our results are not an artefact of the specific analysis methodology used for data processing in this manuscript, we downloaded the already processed short read RNA-Seq data from ENCODE for the A549, HepG2, K562, MCF7, and Hct116 cell lines (Moore et al. 2020). For these datasets, transcript expression was quantified using two methods: Kallisto (Bray et al. 2016) and RSEM (Li and Dewey 2011). We then compared these estimates against the long read RNA-Seq data and the simulated short read data obtained from fragmentation of long read RNA-Seq data (fragmented long read RNA-Seq data). Similar to the SG-NEX short read RNA-Seq data, the ENCODE short read RNA-Seq datasets were more correlated with the fragmented long read RNA-Seq data than the original long read RNA-Seq data (see Figure R1.1). Across all datasets, we observed the strongest increase in correlation for short-read specific major isoforms (Two-sided Mann-Whitney U test $p < 0.0001$). Together, these analyses demonstrate that our results are similar when other short read RNA-Seq data sets and alternative analysis methodologies are used. This provides additional support for our hypothesis that read fragmentation partially explains the observed differences in transcript expression estimates between short read and long read RNA-Seq data. We have included these results in Supplementary Text Section 15.

Figure R1.1 ENCODE short-read RNA-Seq data and SG-NEx short-read RNA-Seq data showed similar correlations with both long-read original and fragmented data for transcript expression

(a-b) Scatterplots of log₂-transformed CPM for protein-coding gene isoforms obtained from (a) long read direct cDNA RNA-Seq (using Salmon) and (b) fragmentation simulated short read from direct cDNA RNA-Seq (using Salmon with bias correction) against that obtained from short read RNA-Seq (using Salmon with bias correction) in the A549 cell line are shown for major isoforms identified in both long and short read (Major isoform: long + short read), isoforms major only in long read (Major isoform: long read specific), isoforms major only in short read (Major isoform: short read specific) and isoforms that are not major isoforms (Minor isoforms).

(c-d) Scatterplots of log₂-transformed CPM for protein-coding gene isoforms obtained from (c) long read direct cDNA RNA-Seq (using Salmon), or (d) fragmentation simulated short read from direct cDNA RNA-Seq (using Salmon with bias correction) against that obtained from ENCODE short read RNA-Seq (using RSEM) in the A549 cell line are shown for are shown for major isoforms identified in both long and short read (Major isoform: long + short read), isoforms major only in long read (Major isoform: long read specific), isoforms major only in short read (Major isoform: short read specific) and isoforms that are not major isoforms (Minor isoforms).

(e-f) Scatterplots of log₂-transformed CPM for protein-coding gene isoforms obtained from (e) long read direct cDNA RNA-Seq (using Salmon), or (f) fragmentation simulated short read from direct cDNA RNA-Seq (using Salmon with bias correction) against that obtained from ENCODE short read RNA-Seq (using Kallisto) in the A549 cell line are shown for are shown for major isoforms identified in both long and short read (Major isoform: long + short read), isoforms major only in long read (Major isoform: long read specific), isoforms major only in short read (Major isoform: short read specific) and isoforms that are not major isoforms (Minor isoforms).

(g) Boxplots showing the median, upper and lower quartiles, and 1.5 x interquartile range of the mean absolute error (MAE) between short read (SR) and long read (LR), as well as in-silico simulated short read RNA-Seq data (simulated SR) for major isoforms identified in both long and

short read (Major isoform: long + short read), isoforms major only in long read (Major isoform: long read specific), isoforms major only in short read (Major isoform: short read specific) and isoforms that are not major isoforms (Minor isoforms). Light grey lines connect the same direct cDNA RNA-Seq sample

(h-i) Boxplots showing the median, upper and lower quartiles, and 1.5 x interquartile range of the mean absolute error (MAE) between (h) ENCODE short read (RSEM) or (i) ENCODE short read (Kallisto) and long read (LR) or in-silico simulated short read RNA-Seq data (simulated SR) for major isoforms identified in both long and short read (Major isoform: long + short read), isoforms major only in long read (Major isoform: long read specific), isoforms major only in short read (Major isoform: short read specific) and isoforms that are not major isoforms (Minor isoforms). Light grey lines connect the samples from ENCODE to direct cDNA RNA-Seq data in the same cell line.

(2) qPCR and dPCR validation experiments confirming long-read specific major isoforms

In our manuscript we report that short read and long read RNA-Seq identify different major isoforms for thousands of genes. By simulating read fragmentation from long read RNA-Seq data, we find that these results are likely an artefact of transcript expression quantification from short read RNA-Seq data, with long read RNA-Seq more robustly identifying major isoforms.

To provide independent experimental validation and control experiments for this claim, we have now selected 13 highly expressed genes from the MCF7 cell line where short read and long read RNA-Seq identified different major isoforms (discordant major isoforms), and quantified the expression of the long-read specific and short-read specific major isoforms using quantitative PCR (qPCR) and digital PCR (dPCR).

For each discordant major isoform pair, primers were designed to cover the unique sequences specific to each isoform. In some cases, where the short-read specific major isoform is a subset of the long-read specific major isoform (i.e., the splice junctions from the short-read specific major isoform are completely contained within the long-read specific major isoform), we compared the unique sequence from the long-read specific major isoform and the common sequence shared between the long-read and short-read specific major isoforms. We found that the concentration levels estimated (relative fluorescence units, RFU) from dPCR agree well with the abundance estimates from long-read RNA-Seq, but not those from short-read RNA-Seq (Pearson correlation coefficient (r) between the RFU and abundance estimates for tested isoforms across all candidate genes: 0.97 for abundance estimates obtained from long-read RNA-Seq vs -0.6 for those from short-read RNA-Seq; Figure R1.2a-b). Furthermore, qPCR results also confirmed that long-read specific major isoforms are detectable at high concentration levels, while short-read specific major isoforms were detected at much lower concentrations (Median ratio difference: 2417.6, Figure R1.2, Supplementary Text Figure 22).

These results show that for all tested genes (13/13), long read RNA-Seq more accurately quantified the major isoform expression compared to short read RNA-Seq data, confirming our claim that long read RNA-Seq more robustly identifies major isoforms.

We have now included the validation results in Figure 4 and described the findings in the main text, with a detailed description of the experiments provided in the Methods section. Detailed results for all selected candidate genes are included as a section in the Supplementary Text (see Supplementary Text Section 17).

(3) GO enrichment analysis on genes with discordant dominant isoforms between long and short read RNA-Seq data

To test if differences in major isoform estimates between short and long reads occur in biologically relevant genes, we performed a GO enrichment analysis and compared the results to those for genes with concordant dominant isoforms between long and short read RNA-Seq data. We used enrichGO in the clusterProfiler R package (Yu et al. 2012) for each of the three GO classes: Biological Process (BP), Cellular Component (CC), and Molecular Function (MF). We then shortlisted significantly enriched GO terms where both the q-value and Benjamini-Hochberg adjusted p-value were significant (significance level: 0.05), and the gene ratio was greater than 0.01 for Molecular Function (MF) and Cellular Component (CC), and greater than 0.025 for Biological Process (BP). The results showed genes with discordant major isoforms fell into a large number of GO categories, which were similar to genes with concordant dominant isoforms between long and short read RNA-Seq data (Figure R1.3). This suggests that genes with observed differences in transcript expression estimates between long and short read RNA-Seq data are of broad functional relevance, impacting similar processes as genes with concordant isoforms. We have included these results into Supplementary Text (see Supplementary Text Section 16).

In addition to these new results and experiments, we have also improved the clarity in text that described the methodologies employed for comparing transcript expression profiling between long and short read RNA-Seq data. We also apologise that one of our results that we referred to was missing in our last submission (as pointed out by Reviewer #1). This was due to the supplementary text not being correctly submitted. We have now included all results in this submission.

We hope that these new data and results address Reviewer #1's comments related to sections (5) and (6), confirming our observation of widespread differences between short read and long read RNA-Seq data, with long read RNA-Seq identifying more robustly major isoforms.

Figure R1.3 Genes with discordant or concordant dominant isoforms between long and short reads are enriched in similar GO categories

(a-c) Heatmaps showing the most significantly enriched GO terms in (a) Biological Processes categories with gene ratio > 0.025, (b) Cellular Components (CC) categories with gene ratio > 0.01, and (c) Molecular Functions (MF) categories with gene ratio > 0.01, for genes with discordant or concordant dominant isoforms between long and short reads

Reviewer #1:

Section (7) describes the analysis of alternative splicing events and isoform switches. Comparisons related to alternative splicing between long and short reads have been done before (eg <https://pubmed.ncbi.nlm.nih.gov/35012468/>), long-range coupling of transcript variations has also been described before (eg <https://pubmed.ncbi.nlm.nih.gov/29598823/>, <https://pubmed.ncbi.nlm.nih.gov/25985263/>), comparison between technologies has also been performed before, albeit at a lower scale (<https://pubmed.ncbi.nlm.nih.gov/35301264/>).

It is strange that the authors do not explore more thoroughly the differences in alternative splicing and isoform switches between nanopore and pacbio, or between nanopore methods.

The result of this section is underwhelming, they simply report the number of isoform switches detected across the cell lines. It is not clear the relevance of this result or whether it is simply a result of the method used, in this case DEXSeq. It could be that the results would be completely different using other analysis methods.

Q2 Response:

We thank Reviewer #1 for pointing out that the differential transcript usage (DTU) event detection results might vary depending on the DTU detection method used. To address Reviewer #1' comment, we have now made the following changes:

- 1) We repeated the alternative isoform switch analysis with another commonly used DTU detection method, edgeR (Robinson, McCarthy, and Smyth 2010; Chen et al. 2024) to compare the results obtained from DEXSeq followed by stageR.
- 2) We included the results comparing edgeR and DEXSeq in the Supplementary Figure 6 to demonstrate the robustness of our findings.

Alternative isoform switching analysis with edgeR

To check if different DTU detection methods give different results for alternative isoform switching analysis, we applied the same filtering steps before testing for DTU with DEXSeq and then repeated the analysis with edgeR. Here we performed two analyses: 1) to identify cell line specific DTU events by comparing one cell line against all other cell lines (cell line specific DTU events) as we have described in the main text, and 2) to identify cell line pairwise DTU events by comparing one cell line against another cell line (cell line pairwise DTU events) as an additional analysis.

As described in the main manuscript, we focused on major isoform switch events with at least 5 full-splice-match supporting reads in any of the samples involved in the comparisons. After these filtering steps, we looked at significant DTU events defined as events with significant differential transcript expression (DTE) (Benjamini-Hochberg corrected p-value <

0.05), absolute log₂ fold change being greater or equal to 2 and significant DTU (Benjamini-Hochberg corrected p-value < 0.05). Results were then compared with those obtained using our original approach described in the manuscript (DEXSeq followed by stageR).

In all 28 comparisons including both cell line-specific and cell line-pairwise comparisons, we found that edgeR reported significantly less number of events compared to DEXSeq (edgeR: median 10, IQR 7, 19.75 vs DEXSeq: median 46, IQR 37, 81; Mann-Whitney U test p < 0.0001), with on average 87% of the edgeR identified events also found by DEXSeq followed by stageR (median: 87%; IQR: 67%, 92%) (see Figure R1.4a). The average number of significant events identified by edgeR increased to 23 when we focused on significant DTE events instead, although still significantly less than the numbers reported by DEXSeq (edgeR: median 23, IQR 21, 37 vs DEXSeq: median 49, IQR 38, 81.75; Mann-Whitney U test p < 0.0001), with an average 85% of the edgeR identified events also found by DEXSeq (median: 85%; IQR: 66%, 92%) (see Figure R1.4b).

Overall, we found a large number of events are identified by both methods, along with some events uniquely identified by each method. However, we believe that the stringent criteria we imposed on the candidate set, such as requiring full-splice-match read support and dominant expression, has helped to ensure the robustness of the findings from these methods. We have now included these new results in Supplementary Figure 6.

Additionally, when comparing transcript expression estimates between different protocols, we observed that major isoform expression estimates were highly correlated between Nanopore protocols (Median spearman correlation coefficient (ρ): 0.92; IQR: 0.9 to 0.94) (see Figure 4b), suggesting that there is likely only minor variation in the major isoforms estimated by each Nanopore protocol. Hence, in our alternative isoform switching analysis, we focused on comparing different cell lines while adjusting for the protocol as a confounding variable, to emphasise isoform expression due to biological variation. Note PacBio IsoSeq protocol was not included in this analysis due to the uneven representation of identified transcripts (see Supplementary Text Figure 13).

We hope that the additional alternative isoform switching analysis with edgeR, along with our clarification on conducting alternative isoform switching analysis across cell lines while adjusting for protocol differences, addresses Reviewer #1's question in section (7) and further demonstrates the robustness of our analysis of alternative isoform switching events.

Figure R1.4 Significant cell line specific and cell line pairwise differential transcript usage (DTU) and differential transcript expression (DTE) events identified by DEXSeq (followed by stageR) and edgeR

(a) Venn diagrams showing the number of DTU events identified by both DEXSeq and edgeR, by DEXSeq only, and by edgeR only for each of the 7 cell lines when compared to other cell lines, and for pairwise cell line comparisons (b) Venn diagrams showing the number of DTE events identified by both DEXSeq and edgeR, by DEXSeq only, and by edgeR only for each of the 7 cell lines when compared to other cell lines, and for pairwise cell line comparisons

Reviewer #1:

Analysis of repeats

The authors explain that the NDR corresponds to the False Discovery Rate. However, they use a value of 0.1, which is a 10% false discovery rate. This is by no means a strict cutoff. From all the transcripts detected, 10% are expected to be false.

The authors have now clarified that they used the RepeatMasker annotations downloaded from the UCSC browser. However, this analysis remains problematic and shallow. First, the authors included the low-complexity repeats, whose implications are completely different from other repeat types. There seems to be also many overlaps identified with RNA pseudogene families. The implications of this is also quite different, but it is not investigated. The authors are misleading in their explanations:

“Compared to annotated transcripts, we observed a significant enrichment of repetitive elements in exons from novel transcripts (two-sided t-test, $p < 0.001$, Figure 6c), among others from LTRs, LINES, and SINEs (Figure 6d, Supplementary Text Figure 17, see methods). The enrichment of these highly repetitive elements was observed for both novel transcripts from annotated genes and for novel gene candidates, and was confirmed when only uniquely mapped reads are used for transcript discovery”.

It is important to note that “These highly repetitive elements” also include Low complexity repeats (no relation to mobile elements or repeat families whatsoever) and RNA pseudogenes (also not a clear relation with repeat families). The text is misleading as “These” may be interpreted as those mentioned explicitly: LTRs, LINES, SINEs. There is no clarity about what repeats are exactly overlapping the predicted and known transcripts.

In this regard, it is important to note that the bulk of the predicted novel transcripts have < 80% overlap (Fig 6c), so the Fig 6d may not be representative of this result. Here the

authors chain a series of results and build a narrative that does not seem well-supported upon scrutiny.

The authors tested the distribution of the % overlap (what % of the exons overlap a repeat) with a t-test (Fig. 6d). This test might not be appropriate given that the distributions present such skewness large number of outliers. Also, this does not test whether repeats are significantly associated with the predicted novel transcripts or not.

In Fig. 6d, there is no clear enrichment in the association of repeats with predicted novel transcripts. Relative to the number of novel or annotated shown, the proportions might be the same. The rate of overlap of known or novel transcripts with repeats could be the same. Since the predicted novel transcripts are shorter (they can be almost half the length), just by chance the fraction of their length covered by repeats, even if they overlap at the same rate, could be higher.

An additional problem with this analysis, is that all the plots show a large number of outliers, especially from the annotated transcripts. It could even be that there are many more annotated transcripts with a large overlap at a given cell line.

Figure 6d is also misleading, since the plots the information for all cell lines, and more annotated transcripts with repeat overlaps seem to be present across most of the cell lines, unlike the predicted novel transcripts, which often seem cell-type specific. This requires a more in-depth analysis, especially in relation to the association with repeats besides the overlap. It is well known that LTRs overlap frequently with promoter/first-exon regions, and that ALUs and other SINE, as well as LINE, elements overlap with alternative exons in annotated transcripts. The measured % overlap differences might be due to the data but not related to actual biology.

The authors tried to address the potential confounding factor due to multimapping reads. They indicate that they used two strategies, 1) using only the primary alignment, or 2) only keeping uniquely aligned reads without any secondary alignment. The authors indicate: “The results indicate that the majority of transcripts can be found with both alignment strategies (91.7%, Supplementary Figure 2a), suggesting that novel transcripts are not due to multiple alignments from reads due to repetitive genomic sequences.”

However, this analysis fails to deliver a convincing explanation. It is not clear whether “found” means that they would have sufficient expression and be considered reliable using just the unique reads. Also, if ~90% of transcripts are found by both strategies, wouldn't this mean that most of them may emerge from multimapping reads? Here the relative proportion of reads from one or the other type in each transcript, as well as their lengths, are important elements to provide reliable support to this claim.

In their response, the authors indicate Supp. Fig. 2, but this seems to be about something else.

The authors in the manuscript to Supp. Fig. 7 regarding the overlap with repeats “...was confirmed when only uniquely mapped reads are used for transcript discovery”. From this figure or their text, it is not clear whether the predicted novel transcripts were recalculated

using only uniquely mapping reads, or they just plotted the results for the subset of transcripts that had uniquely mapped reads assigned to them.

The claim that repeats are significantly associated with novel transcripts requires more thorough scrutiny. Also, it is not clear how that represents an important biological finding by itself. The work remains unclear and of insufficient quality to feel confident that it deserves publication in a journal like Nature Methods.

Q3 Response:

We thank Reviewer #1 for highlighting that (1) the 10% false discovery rate used in this analysis might not be stringent enough and could confound the results, (2) that the validity of these results might not hold when considering only retrotransposons, (3) that the t-test might have been inappropriately used, and (4) that the analysis of unique reads is not well described. To address Reviewer #1's comments, we have conducted additional analyses and revised the method description as follows:

- (1) We reproduced results using different NDR thresholds, namely, 0.05 and 0.15, demonstrating the stringency of the NDR threshold does not impact the results
- (2) We repeated the analysis using only retrotransposons, confirming our observations that retrotransposons are the dominant repeat type overlapping with novel transcripts in Figure 6d
- (3) We changed from t-test to Mann-Whitney U test due to the skewed distribution of repeat-overlapping percentages
- (4) We revised the text to clearly describe that results were reproduced after discarding multi-mapping reads, suggesting that multi-mapping reads did not significantly impact the results

Please find our detailed response below:

(1) Results reproduced using different NDR thresholds

To determine if the 10% false discovery rate used in this analysis (NDR = 0.1) is stringent enough, we now repeated the analysis using two other NDR thresholds: 0.05 and 0.15. We found consistent results across the different NDR thresholds, including significant enrichment of repeat overlapping and significantly lower expression of novel transcripts compared to annotated transcripts (see Figure R1.5a-b). We have included these new results in Supplementary Text Section 19.

(2) Similar results when using only retrotransposons

To check if results remain valid when using only retrotransposons, we repeated the analysis using retrotransposons—namely, LINE, SINE, and LTR—along with potential retrotransposons (those marked with “?”) for all transcripts detected at NDR = 0.1. We found that results were consistent whether using only retrotransposons or all repeats. Novel transcripts, along with novel gene isoforms, were consistently more enriched in repeats compared to annotated transcripts with at least one full-splice-match read support (see Figure R1.5c). This suggests that retrotransposons are the dominant repeat type overlapping with these highly repetitive novel transcripts, which confirms our observations in Figure 6d. The findings were similar when examining novel transcripts filtered at varying NDR thresholds (0.05 and 0.15). Together, these results show that our claims are valid for both all

repeats and when only retrotransposons are analysed. We included these results in Supplementary Text Section 19.

(3) Changed from t-test to Mann-Whitney U test

To account for the skewed distribution of repeat-overlapping percentages, we changed from the t-test to the Mann-Whitney U test. We found that the results remain unchanged. We also updated text in Method Section “*Overlap of transcripts with repetitive elements*”.

(4) Discarding multi-mapped reads for novel transcript discovery

To improve the clarity of how we analysed uniquely and multi-mapping reads, we updated the text in the Methods section to specify that we retained only uniquely aligned reads (Median: 68.8% of total reads; IQR: 48.8%-73% across all samples). We then re-processed these uniquely aligned reads with Bambu to identify novel transcripts and compared them with the novel transcripts identified previously using all reads. We found similar results compared to those obtained using all reads, suggesting that multi-mapping reads did not significantly impact the results. We updated the text in the Methods section as follows:

“To mitigate the impact of multiple alignments, we repeated the analysis with reads having only one alignment (unique alignment), i.e., only reads that have a single alignment were kept and reprocessed with Bambu for transcript discovery. By doing so, on average, 68.8% reads were kept for this analysis.”

Overall, we hope that the results obtained using additional NDR thresholds and only retrotransposons demonstrate the robustness and validity of our repeat findings. Additionally, we hope that our revised description of the unique reads analysis is clearer and that the results confirm the robustness of our findings.

Figure R1.5 Novel transcripts had lower expression values, and higher repeat overlapping percentages across varying NDR thresholds

(a) Boxplots showing the median, upper and lower quartile, and 1.5 x interquartile ranges of the mean CPM for annotated transcripts with at least 1 full-length read support across samples (Annotated ≥ 1 FL), all novel isoforms, and all novel gene isoforms, for 7 cell lines when NDR = 0.1, 0.05 and 0.15 (from left to right). **(b)** Boxplots showing the median, upper and lower quartile, and 1.5 x interquartile ranges of the percentage of exon sequence overlapping with repeat elements for annotated transcripts with at least 1 full-length read support across samples (Annotated ≥ 1 FL), all novel isoforms, and all novel gene isoforms when NDR = 0.1, 0.05 and 0.15 (from left to right), with Bonferroni corrected two-sided Mann-Whitney U test p-values reported for the pairwise mean differences, asterisks indicating $p < 0.0001$ **(c)** Boxplots showing the median, upper and lower quartile, and 1.5 x interquartile ranges of the percentage of exon sequence overlapping with retrotransposons only for annotated transcripts with at least 1 full-length read support across samples (Annotated ≥ 1 FL), all novel isoforms, and all novel gene isoforms when NDR = 0.1, 0.05 and 0.15 (from left to right), with Bonferroni corrected two-sided Mann-Whitney U test p-values reported for the pairwise mean differences, asterisks indicating $p < 0.0001$

Reviewer #1:

Fusion analysis

That authors explain that “The SG-NEx data was made publicly available as a community resource very early on, and it was already used for benchmarking of 3 different long read fusion detection methods (FLAIRfusion, JAFFAL, FusionSeekerLR), making it the most widely used resource for long read fusion detection”. Given that this is the case, it is the authors’ burden to provide an analysis that has not already been published elsewhere. It is not clear otherwise, what is the purpose of this section.

The authors indicate that fusion transcripts were identified using individual samples (not combined samples). But this is misleading, as they probably used all the replicates and sequencing runs for each condition. So again, the number of reads used in the detection of fusions is relevant and should be characterised.

The authors did not attempt to identify the sequencing depth required to identify fusion transcripts. However, they concede that this is “an interesting question that could possibly be answered using the SG-NEx data resource.” It is again not clear why this question, which is important for the problem of fusion detection and can be answered with this dataset and tools produced by the authors, is not addressed in the manuscript that the authors are putting forward to illustrate the advantage of the very same dataset for the study of fusions.

Q4 Response:

As pointed out by Reviewer #1, the SG-NEx data was used to develop new methods to detect fusion gene breakpoints from long read RNA-Seq data, one of which has been used in this manuscript as well (JAFFAL). In contrast to these methods, where the novelty is a new method to identifying fusion breakpoints, we demonstrate that long read RNA-Seq enables the identification and quantification of individual full-length fusion isoforms, and we illustrate this on the SG-NEx resource. To achieve this, we implement a novel approach to align reads to a genome that contains the fusion breakpoints, which allows us to perform transcript discovery and quantification with Bambu on individual samples. The number of reads that support each fusion transcript are reported in our results and supplementary table (see Supplementary Table 9). As described in our previous response, we did not evaluate the number of reads required for identifying fusion breakpoints as this depends on (1) the expression level of fusion transcripts which is highly variable and (2) on the thresholds and method that is used, with different methods having different sensitivity (we did not include multiple methods in this section). While this analysis is possible with the SG-NEx data set, it is a major effort that we believe is more related to the problem of fusion breakpoint discovery, which has been studied by others. We also provide new PCR data to validate novel fusion transcripts.

Reviewer #1:

M6A modification analysis

The prediction of m6A from nanopore sequencing and comparison with other methods has already been published before, eg <https://pubmed.ncbi.nlm.nih.gov/37019930/>. The landscape of m6A across different sample types has already been published before, eg across cell types <https://pubmed.ncbi.nlm.nih.gov/22575960/> or across tissues, including tissue specificity: <https://pubmed.ncbi.nlm.nih.gov/31676230/>. So it remains unclear what novelty this manuscript provides in relation to m6A.

The m6A analyses of the manuscript remain unconvincing and they do not provide any novel insights given what has already been published about m6A.

Supp. Fig 8a seems to indicate that m6Anet precision is low, and that high precision is only achieved at a very low recall, so it is not clear how relevant the results could be. The authors identified only ~6000 sites in all cell lines, ~2700 in the cell line where they performed m6ACE-seq, where only ~58% of them are validated by m6ACE-seq. This seems quite low given that they use a high probability and coverage cutoff. There might be now methods with better precision and recall than m6Anet that may provide a more accurate picture of m6A across the cell lines analysed.

Supp. Fig. 8b shows a metagene plot. This seems to indicate that no m6A sites are found in the 5' UTRs. However, m6A is also abundant in the 5'UTR as shown by others previously. The density shown in the y-axis is perhaps not properly calculated?

Supp. Fig. 8e: These correlations are actually quite low. Most of the correlations between the replicate samples appear to show low correlation values, very similar to the correlations between samples from different sources. The authors should indicate the correlation values and associated p-values for the pairs of replicates. Also, the authors do not describe what value is being used for the correlation. This might be due to a limitation of m6Anet rather than a property of the data.

Supp. Fig. 8g. The authors show a heatmap based on the m6A probability. It is not clear what this plot is supposed to illustrate or whether the finding has any biological relevance.

The m6A analysis remains anecdotal and suboptimal and does not showcase the possible advantages that such large dataset could provide. It is not clear how such analyses could be acceptable for publication in Nature Methods.

Q5 Response:

We thank Reviewer #1 for pointing out that the novelty of the m6A modification analysis using the SG-NEx data resource is insufficiently presented. To address Reviewer #1's comment, we have generated new data and conducted additional data analysis, summarised below:

- (1) We have now included new data that was generated using the most recent RNA sequencing kit (RNA004), which has not yet been analysed in other studies analysing m6A in direct RNA-Seq data
- (2) We additionally applied Dorado, the most recent basecaller that allows simultaneous basecalling and modification calling, as well as m6Anet, to this new sample to demonstrate the usefulness of the resource in evaluating sequencing kits and m6A modification calling methods

New Hek293T sample sequenced using RNA004 kit

To highlight the novelty of m6A modification analysis using SG-NEx data, we have now added a new HEK293T sample generated using RNA004, the most recent direct RNA sequencing kit. The new sample generated a total of 12,470,494 reads, of which 9,160,370 (73.5%) are mappable. The average read length is 1,076 bp, with a 7% error rate and an average read quality score of 25.5.

We used this new sample to benchmark m6Anet and the recent basecaller Dorado (Oxford Nanopore Technologies PLC. 2024), which allows for simultaneous basecalling and modification level estimation. We used two labelled datasets: 1) the m6ACE-seq labels in combination with miCLIP labels, which were previously used by Pratanwanich et al (Pratanwanich et al. 2021) and Hendra et al (Hendra et al. 2022), and 2) GLORI-seq labels that was recently published (Liu et al. 2023).

We first compared the total number of potential m6A sites profiled in the new RNA004 sample against one replicate of HEK293T generated previously using the RNA002 kit. We found that about 7.7 times as many m6A sites were profiled in the RNA004 sample, suggesting that the increased sequencing depth has greatly enhanced the detection of m6A sites (Number of m6A sites in RNA004 sample vs that in RNA002 sample identified by m6Anet: 647,455 vs 84,189, Figure R1.6a).

When comparing Dorado and m6Anet on the new RNA004 sample, we found that Dorado additionally profiled 2,338,875 sites. This increase is potentially due to Dorado's ability to screen all sites with modified adenine, including non-DRACH motifs, which are usually ignored by other m6A detection methods. After excluding non-DRACH motif sites, we find that m6Anet and Dorado profiled the same m6A candidate sites (see Figure R1.6a).

We then compared the two m6A datasets (GLORI-Seq and m6ACE-Seq+ miCLIP-Seq) and found that GLORI-seq contained labels for significantly more sites (labels in GLORI-Seq only: 157,260 vs. labels in m6ACE-Seq+miCLIP only: 4,796). This difference remained high even when we included only labels for sites with a coverage-weighted mean modification rate across the two replicates being greater than or equal to 0.5 (labels in GLORI-Seq (>=0.5) only: 59,144 vs. labels in m6ACE-Seq+miCLIP only: 4,796), see Figure R1.6b.

We then compared the performance of Dorado and m6Anet on the new RNA004 sample, as well as m6Anet on the selected RNA002 sample as a control. We found that m6Anet achieved better performance than Dorado when all candidate sites were used for evaluation (Figure R1.6c). However, when we evaluated both methods only on sites that were profiled by both methods and which were covered in the RNA002 sample, we saw an improved performance for Dorado, in particular for sites with low modification rates (see Figure R1.6c-d). Together these results show that m6Anet and Dorado both identify a large number of sites that are confirmed by other experimental methods.

Regarding the remaining points raised by Reviewer #1 related to Figure 8, we included additional control experiments that are described in more detail in our previous response.

Overall, we hope that the newly added RNA004 sample and the additional benchmark analysis of Dorado, along with the previously presented results (Figure 8, Supplementary Figure 8), sufficiently demonstrate the novelty of m6A modification analysis using the SG-NEx data resource. These new results are now included in Supplementary Text (see Supplementary Text Section 20).

Figure R1.6 Dorado and m6Anet performed similarly for sample generated using RNA004

(a) Venn diagram showing the m6A sites profiled by Dorado, m6Anet, and m6Anet in RNA002 sample. Note here for Dorado, sites profiled by Dorado only were removed. **(b)** Venn diagram showing the overlap between labels defined in GLORI-Seq, GLORI-Seq (with coverage weighted mean modification rate ≥ 0.5), and m6ACE-Seq + miCLIP-Seq **(c)** AUC curves showing the performance of Dorado, m6Anet in RNA004 sample and m6Anet in RNA002 sample, when using GLORI-Seq, GLORI-Seq (with coverage weighted mean modification rate ≥ 0.5), and m6ACE-Seq + miCLIP-Seq labels. Solid lines represent results on sites shared between Dorado (RNA004), m6Anet (RNA004), and m6Anet (RNA002), and dotted lines represent results on all sites for each method. **(d)** PR curves showing the performance of Dorado, m6Anet in RNA004 sample and m6Anet in RNA002 sample, when using GLORI-Seq, GLORI-Seq (with coverage-weighted mean modification rate ≥ 0.5), and m6ACE-Seq + miCLIP-Seq labels. Solid lines represent results on sites shared between Dorado (RNA004), m6Anet (RNA004), and m6Anet (RNA002), and dotted lines represent results on all sites for each method.

Reviewer #4:

Remarks to the Author:

I thank the authors for making an effort to address my points of review. While some of them have been satisfactorily resolved, there are still aspects that require attention.

1. My concerns about quality control of the data remain unaddressed. While the authors indicate that QC programs are included in their pipeline, they acknowledge that this leads to no quality filtering beyond removing samples with few reads. Therefore, library preparation artifacts such as RT and intra-priming may still bias their conclusions about the repeat-containing transcripts. The authors indicate that these biases can be removed by SQANTI but they were unable to run the tool, which is surprising as this is a widely used method applied in many long-read transcriptomics papers. As an alternative, they claim PCR amplification of 5 novel transcripts, but this does not seem to include those with repeated regions. Therefore, doubts about the soundness of these conclusions remain.

Q1 Response:

We thank Reviewer #4 for pointing out that the impact of read quality, such as reverse transcription switch (RTS) and intra-priming artefacts, on the identification of highly repetitive novel transcripts has not been fully addressed. Following this comment by Reviewer #4, we have performed additional analysis and included extra filters to enhance the quality control and robustness of our conclusions:

(1) We performed SQANTI3 on the set of novel transcripts to determine if RTS and intra-priming artefacts are impacting the identification of highly repetitive novel transcripts

(2) We performed read quality filtering to evaluate whether this filter reduces the number of RTS and intra-priming artefacts in the highly repetitive novel transcripts

(3) We included additional details on PCR validation of repetitive transcripts to demonstrate that we were able to validate both repetitive and non-repetitive novel transcripts

Filtering potential RTS and intra-priming artefacts with SQANTI3

To determine if RTS and intra-priming artefacts are impacting the identification of highly repetitive novel transcripts, we followed Reviewer #4's suggestion and successfully performed SQANTI3 (Pardo-Palacios et al. 2024) on the novel annotations generated by Bambu, using all SG-NEx samples with an NDR threshold of 0.1. We found that out of all 1,635 novel transcripts, 94 (5.7%) are classified as RTS artefacts (see Figure R4.1). Among these, only 2 overlap by more than 80% with repeat elements (2 out of 140, 1.4%), suggesting that most highly repetitive transcripts are not caused by RTS artefacts (Figure R4.1a). In terms of potential intra-priming artefacts, 12 (0.7%) of the novel transcripts have polyA tails comprising more than 60% of total A bases (following the default threshold as recommended in SQANTI3, Figure R4.1a). Similarly, 2 of these transcripts overlap by more than 80% with repeat elements (2 out of 140, 1.4%), showing that most highly repetitive transcripts are not caused by intra-priming artefacts (Figure R4.1a). Both results suggest that, at an NDR threshold of 0.1, most of the highly repetitive novel transcripts are neither RTS nor intra-priming artefacts. We have now included SQANTI3 QC in our analysis and removed novel transcripts identified as potential RTS and intra-priming artefacts. We have regenerated Figure 6 using these additional filtering steps (see Figure R4.2) and updated the

main text to mention that novel transcripts have been filtered by SQANTI3. Full details are also included in the Methods section.

“Using a 10% novel discovery rate threshold for transcript discovery in well annotated genomes (NDR=0.1), we identified 1531 novel multi-exon transcript candidates across all samples in the SG-NEx data after filtering potential reverse transcription switch (RTS) and intra-priming artefacts with SQANTI3⁶⁶, 622 (40.6%) of which belonged to genomic loci which were not annotated (Figure 6a, Supplementary Figure 7a-c, Supplementary Table 7).”

“For this analysis, we used the extended annotation set obtained at an NDR of 0.1 using all SG-NEx samples and filtered potential reverse transcription switch (RTS) and intra-priming artefacts identified by SQANTI3, version 5.2⁶⁶. Transcript candidates were classified as potential RTS artefacts if RTS_stage status is TRUE, and as intra-priming artefacts if perc_A_downstream_TTS is scored higher than 60, following the recommended threshold in SQANTI3.”

Impact of read quality filtering on highly repetitive novel transcripts

To investigate the impact of read quality filtering on the identification of repetitive transcripts, we also removed reads with Phred quality scores less than 7 (< Q7), which filtered out, on average, 5% (IQR: 2% to 5.4%) of reads. We then analysed RTS and intra-priming artefacts again with SQANTI3 on the novel transcripts identified at NDR = 0.1 using the reads that passed the Q7 filter. We found that a similar number of RTS or intra-priming artefacts were detected before and after Q7 filtering (number of potential RTS artefacts before vs. after Q7 filtering: 94 vs. 96; number of potential intra-priming artefacts before vs. after Q7 filtering: 12 vs. 9). All 4 highly repetitive RTS or intra-priming artefacts were still identified (see Figure R4.1b). These results demonstrate that our findings regarding highly repetitive novel transcripts are robust, even when read quality filtering is applied. Since the impact of read quality filtering is minimal and does not impact the main results, we have decided to continue using all reads without read quality filtering for the main analysis. However, we have included this analysis results in Supplementary Text (see Supplementary Text Section 22).

Experimental validation for additional novel transcript candidates

We thank Reviewer #4 for pointing out the missing information on repeat overlapping for the five novel transcript candidates validated through PCR Sanger sequencing in the previous revision. We have now added a new column in Supplementary Table 9 to describe the repeat percentage for these validated transcripts. Additionally, we have validated six novel transcript candidates that we identified, which have since been included in recent RefSeq or Ensembl annotation, meaning they are already validated by external resources. Together, we have validated five repetitive novel transcript candidates (repeat overlapping percentage \geq 10%, Figure R4.3). Among these, one highly repetitive novel transcript candidate (repeat overlapping percentage $>$ 80%) identified in the SG-NEx dataset was also validated by PCR Sanger sequencing (see Supplementary Table 9, Figure R4.3). We have updated Supplementary Text Section 6 to include these new results.

We hope that the added SQANTI3 QC filter, additional read quality filtering analysis, and the additional validation experiments, now address Reviewer #4's concerns, demonstrating the robustness of our findings on repetitive novel transcripts.

Figure R4.1 Few novel transcripts classified as potential RTS or intra-priming (polyA percentage > 60%) artefacts

(a) Boxplots showing the percentage of polyAs in the downstream of potentially RTS artefact novel transcripts vs non-RTS artefact novel transcripts, when applying SQANTI3 on the novel transcripts identified in the previous Figure 6a (NDR = 0.1). The point size indicated the percentage of overlapping with repeat elements, with dark blue representing repeat overlapping being greater than 80%. (b) Number of potential RTS, Intra-priming, or highly repetitive RTS/Intra-priming artefact transcripts found when applying SQANTI3 on novel transcripts set identified at NDR = 0.1 using all reads or only the reads passing the Q7 filter.

Figure R4.2 Updated Figure 6 excluding potential RTS and intra-priming artefact novel transcripts

Figure R4.3 PCR bands observed for selected novel transcript candidates in MCF7 cell line (a-b) PCR bands observed for selected novel transcript candidates in MCF7 cell line in two batches. Red arrows indicate the expected PCR band. The red bars indicate confirmed Sanger sequencing results. Single asterisk and double asterisks indicate inconclusive Sanger sequencing results due to multiple alignments existing and PCR product being too short to map respectively. Purple bars indicate novel transcripts identified with a NDR < 0.1 when using only reads with unique alignments. Grey bars indicate a NDR value between 0.1 and 1 when using only reads with unique alignments. Black bar indicate not identified when using only reads with unique alignments. The blue area in the pie charts represents the overlapping percentage with repeat elements.

Reviewer #4:

2. I am still puzzled by the results in Figures 3a and 4b. In Figure 3a, results for comparison to ground truth at the gene level are given, while Figure 4a reports results at the transcript level. However, for some of the spike-ins, such as the ERCC, transcript and gene levels are the same, as these transcripts contain only one transcript per gene. I am not sure about Sequins, as I could not find information about alternative splicing in this set. Moreover, SIRVs E0 indicate significant differences between expected and observed expression estimates, which are not commented on the manuscript. The authors indicate that no Spearman correlation can be computed for this dataset, but do not discuss the apparent difference between expected and observed expression values.

Q2 Response:

We thank Reviewer #4 for pointing out that the differences in spike-in RNAs were not well described and apologise that our previous description of these results was not sufficiently clear. To address this comment, we have made the following changes:

- (1) We have now included a table that describes each spike-in RNA set in detail

To clarify the metrics used for each set of spike-in RNAs, we have now included a table describing each set in terms of the number of genes and isoforms, transcript length, and

concentration variety (see Table R4.1), which is now included in Supplementary Text Section 23.

(2) We have removed the results for ERCC from the transcript expression evaluation

We thank Reviewer #4 for pointing out that ERCC gene and transcript results were identical. We have updated Figure 4a to remove the results for ERCC (see also Figure R4.4).

(3) We now comment on the observed differences for SIRV E0 and long SIRV based on mean absolute errors (MAE)

We also thank Reviewer #4 for pointing out the lack of discussion on the observed differences for the SIRV E0 and long SIRV sets. We have updated the main text to highlight the differences we observed for different spike-in RNA sets, including SIRV E0 and Long SIRVs, as detailed below:

“Similar to gene expression on spike-in RNAs, we observed that the nanopore long read sequencing protocols showed higher correlation between the estimated and expected transcript expression levels compared to short read RNA-Seq on the sequin RNAs and ERCCs (Figure 4a), and SIRV spike-ins showed higher absolute errors with short read RNA-Seq, including SIRV (E0) and Long SIRVs (Supplementary Figure 5a,b,d,e).”

We hope that these changes improve the clarity in our description of spike-in RNA results in the manuscript.

Figure R4.4 Updated Figure 4a with ERCC results removed

	Sequin	Sequin	SIRV	SIRV	ERCC	Long SIRV
--	--------	--------	------	------	------	-----------

	(MixA V1)	(MixA V2)	(E0)	(E2)		
Total number of genes	78		7		92	5
Total number of isoforms	164		69		92	15
Number of isoforms per gene*	2 (1,4)		8 (6,18)		1 (1,1)	3 (3,3)
Number of exons per isoform*	6 (1,36)		4(1,18)		1 (1,1)	1 (1,1)
Isoform length*	1045 (283, 6943)		783 (161, 2498)		978 (256,2014)	8000 (3997, 12029)
Intron length*	1793 (71, 544891)		283 (21, 110785)		0	0
Exon length*	126 (15,4414)		125 (9,2473)		978 (256, 2014)	8000 (3997, 12029)
Number of unique concentration values	81	117	1	4	22	1
Concentration values*	13.43 (0.004,30000)	4.23 (0.004,1691.89)	60 (60,60)	625 (31.25,4000)	1.1 (0.001,1500)	60 (60,60)

*: median (min, max)

Table R4.1 Characteristics of spike-in sets included in SG-NEx data resource

Reviewer #4:

3. Moreover, the statement in the last sentence of Page 7, indicating that library preparation introduces biases in read length, coverage, and transcript diversity, seems to be in contradiction with the next sentence stating that “Gene expression is robustly estimated across different RNA-seq protocols”. This contradiction is not well addressed in the manuscript and is not mentioned in the Discussion. If the conclusion is that gene expression estimates are robust but not transcript expression estimates, this needs to be discussed further, including how the mentioned biases affect estimates.

Q3 Response:

We thank Reviewer #4 for pointing out that there is a lack of discussion on how observed sample differences potentially caused by library preparation differences affects the gene and transcript expression profiling. Even though we included these analyses in the main figures, we did not reference them in the main text or discussion as the differences between long and short read RNA-Seq is substantially larger than differences between long read RNA-Seq protocols.

In our original manuscript, we first compared the different protocols to each other in terms of global quality metrics (section 3 - e.g. number of reads, average gene coverage, transcript diversity). As summarised by Reviewer #4, we observe that differences occur, even when long read RNA-Seq protocols are compared to each other, which suggests that this might impact gene and transcript expression estimates.

Following this protocol comparison section using global metrics, we compared how individual gene and transcript expression estimates differ between protocols (section 4 - gene

expression, section 5 to 7 - transcript expression). In these sections, we showed that the correlation of log2-transformed transcript expression differed between different protocols, including a comparison of all long-read RNA-Seq protocols (Figure 4b). Additionally, in Figure 4f-g, we compared all protocols in terms of the number of junctions covered per read and the number of transcripts to which reads align, to provide insights into how differences in the data due to the library preparation might impact transcript expression estimates (see Figure R4.5). However, as differences between short read and long read RNA-Seq were substantially higher compared to the differences between the different long read RNA-Seq protocols, we have emphasised this cross-technology comparison in the results section.

To address the Reviewer’s comment, we have now made the following changes:

- 1) We now reference the cross-protocol results in the main text to describe that gene expression profiling results were consistent between different RNA-Seq datasets
- 2) We highlight that transcript expression profiling differ between different protocols, including different long-read RNA-Seq protocols, and that differences between short read and long read RNA-Seq were strongest in this comparison
- 3) We now reference all protocols in the comparison of splice junctions and unique read assignments related to transcript expression, while still emphasising that long read vs short read RNA-Seq data showed the most profound differences

We hope that our analyses that investigate library preparation differences and their relation to gene and transcript expression profiling, are now more clearly highlighted in the revised manuscript.

Figure R4.5 Figure 4b,f,g
 (a) Figure 4b (b) Figure 4f (c) Figure 4g

Reviewer #4:

4. The authors indicate that “among all RNA-Seq protocols, the PacBio IsoSeq data showed the highest estimation error, largely due to the uneven representation of transcript lengths, with low coverage for short transcripts”). While this might be true, there is not a formal demonstration of this, therefore the statement should be rephrased as “possibly”, as other causes, such as sequencing depth may contribute. On the other hand, this statement is misleading as it focuses the problem on the poorer capture of PacBio of shorter transcripts and not on the limitation of Nanopore for quantifying longer transcripts, which might be an issue according to Figure 2b.

Q4 Response:

We thank Reviewer #4 for pointing out that the discussion on PacBio IsoSeq data is not clearly described and lacks sufficient references. To address Reviewer #4’s comments, we have revised the statement in the main text for clarity and added a reference to Supplementary Text Figure 13 in Supplementary Text Section 8. This figure shows that PacBio IsoSeq data does not include transcripts with length less than 1kb (see Figure R4.6). While we observe that this has a negative impact on the ability to quantify spike-in RNAs with PacBio data, we do not observe that nanopore data shows lower accuracy when quantifying longer transcripts. Following the suggestion from Reviewer #4, we have revised the text as follows:

“Among all RNA-Seq protocols, the largest difference was observed in the PacBio IsoSeq data, which was possibly due to an uneven representation of transcript lengths, with low coverage for short transcripts (<1kb, see Supplementary Figure 4a,d,e, Supplementary Figure 1h-i, Supplementary Text Figure 13).”

Figure R4.6 Transcription by gene length for different sequencing protocols

(a) Cumulative distribution plots for percentage of transcription when ranking genes by gene length for direct RNA-Seq (dRNA), direct cDNA (dcDNA), PCR cDNA (cDNA), PacBio long read RNA-Seq and Illumina short read RNA-Seq (b) Boxplots of fraction of transcription the top genes with gene length less than 1kb for direct RNA-Seq (dRNA), direct cDNA (dcDNA), PCR cDNA (cDNA), PacBio long read RNA-Seq and Illumina short read RNA-Seq

Reviewer #4:

5. I strongly suggest re-wording full-length-read to complete-junction-set reads or full-splice-match reads, as their definition does not correspond to the concept of full-length.

Q5 Response:

We thank Reviewer #4 for the suggestion. We have now reworded “full-length reads” to “full-splice-match reads” throughout the manuscript. When referring to full-length transcripts (not reads), we have kept this formulation.

Reviewer #4:

6. I could not find the Supplementary Text referred to in the rebuttal.

Q6 Response:

We apologise for not including the Supplementary Text in the previous submission. We have now added it, including the additional analysis results.

Reviewer #4:

7. A few improvements in the writing are suggested.

- a. Page 3. Line 4. Change “alternative gene isoforms” to “alternative isoforms”
- b. Page 3. Line 5. Change “can be independently regulated” to “can be differentially regulated”
- c. Page 3. Line 8. Change “their regulation and expression plays” to “their regulation and expression play”
- d. Page 3. The first sentence of paragraph 3 starting with “transcriptome profiling that uses” is awkward.
- e. Page 4. The sentence “our study provides a data-driven guide for experimental design of transcriptomics and epitranscriptomics experiments” should be removed. Experimental design questions (statistical power, number of replicates, sequencing depth, batch effects, etc) are not addressed in this work.
- f. Page 7. Line 3. Change “some genes generated” to “some genes had”

Q7 Response:

We thank Reviewer #4 for the valuable writing suggestions. We have revised the relevant texts accordingly:

a: “However, alternative promoters, exon skipping, intron retention, 3’ end sites and polyadenylation enable a single gene to generate a large number of *alternative isoforms*, each corresponding to a different RNA transcript.”

b: “Such alternative transcripts can be *differentially* regulated, leading to differences in transcript expression across samples and conditions even when the overall gene expression levels are stable¹⁻⁶.”

c: “Due to the differences in their RNA sequence, alternative transcripts can have different functionality, and their regulation and expression **play** a key role during early development^{7,8}, in defining cellular identity^{9,10}, and in human diseases^{11,12}.”

d: “Long read sequencing promises to overcome some of the main limitations of current short read RNA-Seq protocols **in transcriptome profiling**^{25–29} at a cost per gigabase comparable with current short read technologies (Supplementary Figure 1a), potentially enables the widespread adoption of long read RNA sequencing.”

e: We have removed the sentence now.

f: “Finally, we observed that **transcripts from some genes were** incompletely amplified and sequenced in the PCR cDNA protocol across all cell lines when compared to direct sequencing of the same RNA sample (Figure 2f, Supplementary Figure 1j-k, Supplementary Table 3).”

References

- Bray, Nicolas L., Harold Pimentel, Páll Melsted, and Lior Pachter. 2016. “Near-Optimal Probabilistic RNA-Seq Quantification.” *Nature Biotechnology* 34 (5): 525–27. <https://doi.org/10.1038/nbt.3519>.
- Chen, Yunshun, Lizhong Chen, Aaron T. L. Lun, Pedro L. Baldoni, and Gordon K. Smyth. 2024. “edgeR 4.0: Powerful Differential Analysis of Sequencing Data with Expanded Functionality and Improved Support for Small Counts and Larger Datasets.” bioRxiv. <https://doi.org/10.1101/2024.01.21.576131>.
- Hendra, Christopher, Ploy N. Pratanwanich, Yuk Kei Wan, W. S. Sho Goh, Alexandre Thiery, and Jonathan Göke. 2022. “Detection of m6A from Direct RNA Sequencing Using a Multiple Instance Learning Framework.” *Nature Methods* 19 (12): 1590–98. <https://doi.org/10.1038/s41592-022-01666-1>.
- Li, Bo, and Colin N. Dewey. 2011. “RSEM: Accurate Transcript Quantification from RNA-Seq Data with or without a Reference Genome.” *BMC Bioinformatics* 12 (1): 323. <https://doi.org/10.1186/1471-2105-12-323>.
- Liu, Cong, Hanxiao Sun, Yunpeng Yi, Weiguo Shen, Kai Li, Ye Xiao, Fei Li, et al. 2023. “Absolute Quantification of Single-Base m6A Methylation in the Mammalian Transcriptome Using GLORI.” *Nature Biotechnology* 41 (3): 355–66. <https://doi.org/10.1038/s41587-022-01487-9>.
- Oxford Nanopore Technologies PLC. 2024. “Dorado.” Oxford Nanopore Technologies PLC.
- Pardo-Palacios, Francisco J., Angeles Arzalluz-Luque, Liudmyla Kondratova, Pedro Salguero, Jorge Mestre-Tomás, Rocío Amorín, Eva Estevan-Morió, et al. 2024. “SQANTI3: Curation of Long-Read Transcriptomes for Accurate Identification of Known and Novel Isoforms.” *Nature Methods* 21 (5): 793–97. <https://doi.org/10.1038/s41592-024-02229-2>.
- Pratanwanich, Ploy N., Fei Yao, Ying Chen, Casslynn W. Q. Koh, Yuk Kei Wan, Christopher Hendra, Polly Poon, et al. 2021. “Identification of Differential RNA Modifications from Nanopore Direct RNA Sequencing with xPore.” *Nature Biotechnology* 39 (11): 1394–1402. <https://doi.org/10.1038/s41587-021-00949-w>.
- Robinson, Mark D., Davis J. McCarthy, and Gordon K. Smyth. 2010. “edgeR: A Bioconductor Package for Differential Expression Analysis of Digital Gene Expression Data.” *Bioinformatics* 26 (1): 139–40. <https://doi.org/10.1093/bioinformatics/btp616>.
- Yu, Guangchuang, Li-Gen Wang, Yanyan Han, and Qing-Yu He. 2012. “clusterProfiler: An R Package for Comparing Biological Themes Among Gene Clusters.” *OMICS: A Journal of Integrative Biology* 16 (5): 284–87. <https://doi.org/10.1089/omi.2011.0118>.

Response to Reviewers comments

Reviewer #1 (Remarks to the Author):

As previously mentioned, while the data is useful and the general analyses presented in the first part of the manuscript are interesting, the remaining analyses are still not novel for what one would consider to be an outstanding and exciting manuscript worth publishing in Nature Methods. Also, these additional analyses bear no relevance or connection with the presented datasets and the manuscript title “A systematic benchmark of Nanopore long read RNA sequencing for transcript level analysis in human cell lines”. No such systematic benchmarking is presented, but rather a collection of disconnected results that mostly repeat what has already been published elsewhere. They are also analyses that could have been done with any other datasets. In this sense, there is no connection with the comprehensive data presented. These additional analyses are preliminary and could be developed more thoroughly and with the appropriate controls as separate research articles.

Overlap with repeat elements

While the authors have added some additional details and filters, the results remain unclear. The initial set of novel transcripts includes ~40% of novel genes. However, from Fig. 6d, it seems that most of the largest overlaps are for novel transcripts that are within known genes and have one exon difference (first exon, internal exon, ...) with respect to the annotation. Some complex repeat families can be quite long, some LINEs can be >4kb. Figure 6 does not yet unequivocally show that these repeats are significantly associated with these novel transcripts. They may be the same repeats that also overlap the annotated transcripts. So the overlap is linked to the locus, not necessarily to the novel isoform. The authors do not describe how many isoforms overlap repeats in each subset and whether these are different repeats or the same repeats when the isoforms are in the same gene.

Also, the authors use annotated transcripts “with at least one full-splice-match read support”. It is not clear whether this may bias the comparison against long annotated genes, which may be more likely to overlap with repeats. Transcription initiation has been seen to overlap with retrotransposons (<https://pubmed.ncbi.nlm.nih.gov/19377475/>). It is possible that by restricting the annotated dataset, many overlaps are not considered, in particular the longest ones, which are more likely to overlap with repeats.

As described by the authors “New transcripts generally have lower expression, number of exons, and transcript length compared to reference transcripts”. It is possible that the enrichment observed is confounded by the differences in length. This could explain why after filtering for 80% overlap, figure 6d shows a low number of annotated transcripts, whereas thousands of annotated transcripts are known to overlap repeats (e.g. <https://www.ncbi.nlm.nih.gov/pmc/articles/PMC186627/>). So Figure 6d remains misleading, as it may be simply an artifact of the length difference.

The authors overlap the annotated and novel parts of the isoforms with repeats. But they talk about enrichment of novel isoforms. This is also misleading. Do they mean enrichment of novel parts? Considering that most of the overlaps are with isoforms that are within

annotated genes, are they actually finding an enrichment of repeats in novel “exons”? This links to the previous question. Are these mostly new alternative first exons? Figure 6d is just a subset of the cases and do not represent the global behaviour.

The authors also use the term “these highly repetitive novel transcripts” (supp text, page 25). This is misleading and “highly repetitive” should be quantified properly. The authors do not show that these transcripts are repetitive, or that the repeats that overlap with novel transcripts are multiple copies of the same repeat class and family.

Transposable elements have already been described to contribute to lncRNAs (<https://pubmed.ncbi.nlm.nih.gov/23637635/>), which have similar properties to the novel transcripts described in the manuscript. Is it still possible that those novel transcripts are lncRNAs? A clearer definition of novel transcript is necessary in this context. How did the authors confirm that these transcripts are novel?

As indicated before, the claim that repeats are significantly associated with novel transcripts requires unequivocal evidence, a clearer description, and a more thorough analysis. It is intriguing and interesting, but it may deserve a separate focused publication with more information. At the moment, it is unclear how these results represent an important biological finding by itself and whether the datasets used play any relevant role or not.

Fusion transcripts

The authors indicate “we implement a novel approach to align reads to a genome that contains the fusion breakpoints”. However, this approach is not novel. Other methods have used before the genomic breakpoints to identify the fusion transcripts, (e.g. <https://pubmed.ncbi.nlm.nih.gov/21478487/>, <https://pubmed.ncbi.nlm.nih.gov/31728961/>). The analysis produced by the authors is correct, but it is incremental compared with prior results accomplished in other papers. Also, as previously mentioned, the authors do not make any effort to show how their SG-NEx data resource was particularly unique or relevant to this analysis.

m6A analysis

The authors did not directly address the raised concerns. They have added additional datasets and comparisons that do not add much novelty to the paper. They simply illustrate the capacity of different methods to predict m6A from nanopore data and the general low precision of nanopore-based methods to recover m6A sites predicted by orthogonal methods. The results also show that in DRACH sites, dorado performs better than m6Anet. However, this is not mentioned in the main manuscript. It is not clear what this adds to the overall manuscript.

These additional results are correct but incremental compared with prior results reported in the literature, and do not add any novelty or significance to the main manuscript. As mentioned before, the prediction of m6A from nanopore sequencing and comparison with other methods has already been published before (e.g. <https://pubmed.ncbi.nlm.nih.gov/37019930/>), the landscape of m6A across different sample types has already been published before (e.g. across cell types

<https://pubmed.ncbi.nlm.nih.gov/22575960>, and across tissues, including tissue specificity: <https://pubmed.ncbi.nlm.nih.gov/31676230/>). So it remains unclear what novelty this manuscript provides in relation to m6A.

Response:

We thank Reviewer #1 for the detailed evaluation of our manuscript. We have further increased the emphasis on the data resource by moving results related to the points raised here to the supplementary material, where we also describe in detail all the control experiments. We hope that the manuscript and data will be a useful resource to the research community.

Reviewer #3:

None

Reviewer #4 (Remarks to the Author):

Concerns have been satisfactorily addressed. I thank the authors for their efforts.

Response:

We would like to thank Reviewer #4 for their helpful comments and the positive evaluation of this manuscript.